# FLUX: Efficient Descriptor-Driven Clustered Federated Learning under Arbitrary Distribution Shifts

**Dario Fenoglio**\*
Università della Svizzera italiana
Lugano, Switzerland
`dario.fenoglio@usi.ch`

**Mohan Li**\*
Università della Svizzera italiana
Lugano, Switzerland
`mohan.li@usi.ch`

**Pietro Barbiero**
IBM Research
Zurich, Switzerland
`pietro.barbiero@ibm.com`

**Nicholas D. Lane**
University of Cambridge
Cambridge, United Kingdom
`ndl32@cam.ac.uk`

**Marc Langheinrich**
Università della Svizzera italiana
Lugano, Switzerland
`marc.langheinrich@usi.ch`

**Martin Gjoreski**
Università della Svizzera italiana
Lugano, Switzerland
`martin.gjoreski@usi.ch`

## Abstract

Federated Learning (FL) enables collaborative model training across multiple clients while preserving data privacy. Traditional FL methods often use a global model to fit all clients, assuming that clients' data are independent and identically distributed (IID). However, when this assumption does not hold, the global model accuracy may drop significantly, limiting FL applicability in real-world scenarios. To address this gap, we propose FLUX, a novel clustering-based FL (CFL) framework that addresses the four most common types of distribution shifts during both training and test time. To this end, FLUX leverages privacy-preserving client-side descriptor extraction and unsupervised clustering to ensure robust performance and scalability across varying levels and types of distribution shifts. Unlike existing CFL methods addressing non-IID client distribution shifts, FLUX i) does not require any prior knowledge of the types of distribution shifts or the number of client clusters, and ii) supports test-time adaptation, enabling unseen and unlabeled clients to benefit from the most suitable cluster-specific models. Extensive experiments across four standard benchmarks, two real-world datasets and ten state-of-the-art baselines show that FLUX improves performance and stability under diverse distribution shifts—achieving an average accuracy gain of up to 23 percentage points over the best-performing baselines—while maintaining computational and communication overhead comparable to FedAvg.

## 1 Introduction

Federated Learning (FL) [1–4] is a distributed and privacy-preserving Machine Learning (ML) paradigm that enables multiple isolated clients to collaboratively train models without sharing their private local data. Traditional FL methods often use a global model to fit all clients' data [1, 5, 6], assuming that clients' data are independent and identically distributed (IID). However, this assumption

---

\*Equal contribution.

39th Conference on Neural Information Processing Systems (NeurIPS 2025).

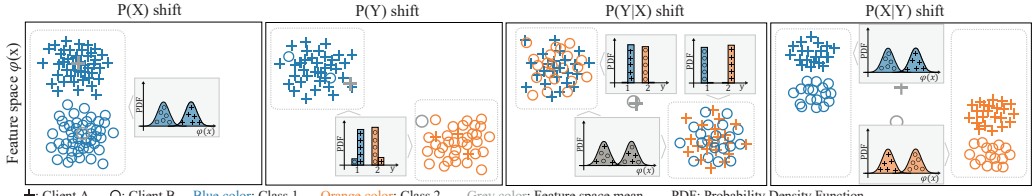

Figure 1: **Types of data distribution shifts. (a)** Feature distribution shift: two subsets differ in feature distributions, while label distributions are similar. **(b)** Label distribution shift: two subsets differ in label distributions, while feature distributions (for each class) are similar. **(c)** $P(Y|X)$ concept shift: two subsets share the same feature distributions but differ in label distributions. **(d)** $P(X|Y)$ concept shift: two subsets share the same label distributions but differ in feature distributions.

rarely holds in practical FL scenarios, where clients often exhibit *distribution shifts* due to holding non-IID data[†] [7, 8]. Such heterogeneity can significantly degrade the performance of the global model, limiting the effectiveness of FL in real-world applications [9].

To address this problem, both Clustered Federated Learning (CFL) [10, 11] and Personalized Federated Learning (PFL) [12, 13] have been proposed. These approaches relax the global model constraint by adapting models to subsets of clients with homogeneous data distributions—via clustering in CFL or client-specific personalization in PFL. Despite their promise, existing methods address only specific forms of heterogeneity (e.g., feature or label shift), and often fail when multiple or unforeseen shifts occur [12–17]. Many rely on impractical assumptions, such as prior knowledge of the optimal number of clusters [11, 15, 16, 18], and incur high communication and computational overhead [13, 14, 17, 18]. Crucially, most methods assume that test-time clients have already participated in training and contributed labeled data [11–19]. This assumption breaks down in realistic deployments, where unseen and unlabeled clients may appear post-training with unknown data distributions. To overcome this limitation, Test-Time Adaptive FL (TTA-FL) methods have emerged to adapt pre-trained models to the distribution of test-time clients without requiring labeled data [19, 20]. However, they typically assume a single starting global model and rely on online adaptation procedures or extra interactions with the client, limiting their practicality and robustness in deployment.

To address these challenges, we introduce FLUX (Federated Learning with Scalable Unsupervised Clustering and Test-Time Adaptation), a novel clustering-based approach that efficiently handles data heterogeneity in FL under minimal assumptions. **Our key contributions are:**

- We propose FLUX to address four common types of data distribution shifts in FL: feature shifts, label shifts, $P(Y|X)$-concept shifts, and $P(X|Y)$-concept shifts. Unlike most existing frameworks, FLUX does not require prior knowledge of unseen data distributions or the number of clusters.
- We empirically demonstrate that FLUX supports test-time adaptation by assigning previously unseen and unlabeled clients to the most suitable cluster models. FLUX consistently outperforms 10 state-of-the-art (SOTA) baselines—including CFL, PFL, and TTA-FL—across four standard FL benchmarks and two real-world datasets. Evaluations span a broad range of scenarios, covering the four most common types of distribution shift, their combinations, and eight levels of heterogeneity severity (from none to extremely high).
- We provide both theoretical bounds and empirical evidence that FLUX incurs minimal computational and communication overhead compared to baselines and verify its scalability, enabling deployment across a large number of clients while ensuring privacy guarantees.

To the best of our knowledge, FLUX is the first scalable and general-purpose FL framework explicitly designed to address the four most common distribution shifts during both training and test time.

## 2 Background

**Traditional FL under IID assumption.** Traditional FL systems [1] consist of $K \in \mathbb{N}$ clients, denoted by $\mathcal{K} = \{1, 2, \ldots, K\}$, coordinated by a central server to collaboratively train an ML model while preserving data privacy. Each client $k \in \mathcal{K}$ holds a private dataset $(x^{(k)}, y^{(k)})$, a realization

---

[†]We refer to these clients as *non-IID clients* in the next sections.

of the random variables $\left(X^{(k)}, Y^{(k)}\right)$ drawn from the data distribution $P(X^{(k)}, Y^{(k)})$. Under IID assumptions, FL assumes IID data across clients, i.e., $P(X^{(k)}, Y^{(k)}) = P(X, Y)$ for all $k$. Each client holds $s^{(k)} \in \mathbb{N}$ samples, with $x^{(k)} \in \mathbb{R}^{s^{(k)} \times z}$ denoting feature vectors and $y^{(k)} \in \{0, 1\}^{s^{(k)} \times u}$ the corresponding labels, where $z$ is the number of features and $u$ the number of classes. In each round, clients independently update local parameters $\theta^{(k)} \in \Theta^{(k)} \subseteq \mathbb{R}^p$ by minimizing a local loss on their private data, with $p$ the number of parameters. After local training, clients send their updated parameters to the server, which aggregates them using a permutation-invariant method, typically *FedAvg* [1]. The aggregated global model $\theta$ is broadcast back to clients to initialize the next round, and the process repeats until convergence to $\theta^*$, minimizing the overall risk across client distributions:

$$\theta^* = \arg \max_\theta \sum_{k=1}^{K} \sum_{(x,y) \in (x^{(k)}, y^{(k)})} \log P\big(y \mid x; \theta\big). \tag{1}$$

**Distribution shifts.** Real-world FL tasks involve clients with a combination of distribution shifts, driven by factors such as geographic and demographic variations, or even adversarial attacks. While distribution shift has been studied extensively in centralized settings [21–24], it remains a relatively new challenge in FL. The FL community has developed numerous approaches—such as FedNova [25], FedProx [5], and FedDyn [26]—to mitigate data heterogeneity, but these methods typically address only specific aspects of the non-IID challenge. This limitation is further exacerbated by the fact that the server has limited knowledge and control over client data distributions due to data isolation. Below, we outline four typical types of FL distribution shifts, as illustrated in Figure 1:

- *Feature distribution shift*: marginal distributions $P(X)$ vary across clients.
- *Label distribution shift*: marginal distributions $P(Y)$ vary across clients.
- *Concept shift* (same features, different label): conditional distributions $P(Y|X)$ vary across clients.
- *Concept shift* (same label, different features): conditional distributions $P(X|Y)$ vary across clients.

Unlike centralized training, FL clients, particularly in cross-device scenarios, often operate on devices with constrained computational resources. These limitations restrict both the computational capacity for training and the complexity of deployable models, reducing their ability to capture diverse and complex data distributions across clients [14, 16]. As a result, the trained models may lack sufficient expressiveness, adversely affecting both performance and generalizability [14].

**Clustered Federated Learning.** To address distribution shifts in federated environments, CFL extends traditional FL by partitioning the overall training data across clients into $M \in \mathbb{N}$ distinct clusters. Each cluster is associated with a unique data distribution $P(X_m, Y_m)$ such that all data within that cluster are drawn from the same distribution. Let $\mathcal{C} = \{c^{(1)}, c^{(2)}, \ldots, c^{(K)}\}$ denote the set of cluster-assignment vectors, where each $c^{(k)} \in \mathbb{R}^M$ represents the degree of membership of client $k$ to each of $M$ clusters, and satisfies $\sum_{m=1}^{M} c_m^{(k)} = 1$. Based on the nature of cluster assignments, we define two CFL frameworks: *soft-CFL* and *hard-CFL* (see Appendix A.1.1 for more details). In *soft-CFL*, the vectors of $c^{(k)}$ are fractional values in $[0, 1]$, reflecting probabilistic membership across clusters. In contrast, *hard-CFL* requires that each $c^{(k)}$ be a one-hot vector, i.e., $c^{(k)} \in \{0, 1\}^M$, meaning that client $k$ is assigned exclusively to a single cluster. Under hard-CFL, the client set $\mathcal{K}$ is partitioned into disjoint subsets corresponding to the different clusters, and within each cluster $m$, traditional FL is employed to optimize a dedicated model $\theta_m^*$, ensuring that the conventional FL assumptions hold locally for the data drawn from $P(X_m, Y_m)$. Formally, CFL seeks to jointly optimize the model parameters and the cluster assignments as follows:

$$\{\theta_m^*\}_{m=1}^{M}, \{c^{(k)*}\}_{k=1}^{K} = \arg \max_{\{\theta_m\}_{m=1}^{M}, \{c^{(k)}\}_{k=1}^{K}} \sum_{m=1}^{M} \sum_{k=1}^{K} \sum_{(x,y) \in (x^{(k)}, y^{(k)})} c_m^{(k)} \log P\big(y \mid x; \theta_m\big) \tag{2}$$

## 3 CFL Challenges and Problem Definition

**Limitations and challenges in current CFL approaches.** As summarized in Table 2, existing CFL methods fail to satisfy several key FL requirements. Most lack a robust mechanism for assigning clusters at test time—especially for unlabeled clients—and are not evaluated across all four types

of non-IID shifts. Moreover, methods such as [15, 11] require prior knowledge of the number of distributions or clusters before initiating FL training (i.e., $M^*$), which is often impractical in real-world applications. Additionally, most approaches impose significant computational burdens, either on the server side [14] or the client side [17], and demand increased communication costs if multiple models are transmitted [11]. These limitations hinder scalability, particularly as the number of clients or potential clusters grows, a common scenario in real FL deployments. For this reason, we opted for a hard-CFL approach: unlike soft-CFL methods, which require maintaining multiple models per client and learning personalized weight vectors (hindering efficiency and generalization to unseen clients), hard clustering provides a more scalable and deployable solution. A more detailed comparison among our work and baselines is provided in Appendix E.2.

**Problem statement.** We consider an FL setting with $K$ clients. Each client $k$ holds a private dataset $\left(x^{(k)}, y^{(k)}\right)$ drawn from one of $M$ unknown data distributions $\{P(X_m, Y_m)\}_{m=1}^M$, where $1 \le M \le K$. Our objective is to (i) determine the number of clusters $M$, (ii) identify the cluster assignment $\mathcal{C}$, so that all clients in cluster $m$ have (approximately) IID data from the same distribution $P(X_m, Y_m)$, and (iii) optimize the model $\theta_m$ for each cluster $m$. At test time, our goal is to assign any newly arriving, unlabeled client to the best-fitting cluster-specific model $\theta_m^*$. Crucially, no prior knowledge of the underlying data distributions or the number of distribution shifts is required; both $M$ and $\mathcal{C}$ are learned from the clients' data. The methodology must ensure that the solution is as scalable as traditional FedAvg and maintains the same privacy level.

# 4 FLUX

In this work, we propose FLUX, a robust, computationally efficient, and scalable CFL framework that handles all four types of data distribution shifts without requiring prior knowledge of client data. This section introduces the theoretical foundations of FLUX, using a probabilistic graphical model to represent the relationships among client distributions, descriptors, and cluster assignments, enabling decomposition into independently optimizable objectives (Section 4.1). We then present the operational pipeline of FLUX, designed for adaptability and real-world utility (Section 4.2).

## 4.1 Probabilistic Modeling and Optimization Objectives

FLUX aims to predict $Y^{(k)}$ from $X^{(k)}$ using a cluster-specific model parameterized by $\theta_m$, where $\theta_m \in \Theta_m$ depends on the cluster $m$ to which client $k$ is assigned. During training, each client learns a local model $\theta^{(k)} \in \Theta^{(k)}$ based on its private dataset. From these data, it then constructs a compact descriptor $d^{(k)} \in D^{(k)}$ via a feature extractor $\phi$ parameterized by $\psi \in \Psi$, which is designed to capture the essential information of the client's data distribution. After collecting the descriptors $\{d^{(k)}\}_{k=1}^K$

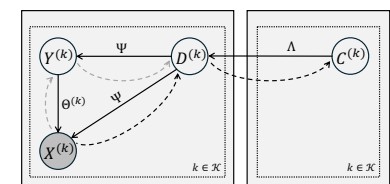

Figure 2: **FLUX's PGM.** *Solid line*: data generating mechanism. *Dashed line*: inference direction. *Gray line*: present only during training.

from all clients, the server applies an unsupervised clustering algorithm $\mathcal{U}$, parameterized by $\lambda \in \Lambda$, to obtain the cluster assignments $\{C^{(k)}\}_{k=1}^K$. These assignments allow the aggregation of local models into cluster-specific models $\theta_m$, each optimized to generalize well across clients in cluster $m$.

We formalize FLUX as a probabilistic graphical model (PGM) capturing dependencies among variables. Each client $k$ contributes to the joint distribution via its data $(X^{(k)}, Y^{(k)})$, descriptor $D^{(k)}$, and cluster assignment $C^{(k)}$. Figure 2 illustrates the global structure, which factorizes per client as:

$$P(C^{(k)}, D^{(k)}, Y^{(k)}, X^{(k)}; \Theta^{(k)}, \Lambda, \Psi) = \underbrace{P(D^{(k)} | C^{(k)}; \Lambda)}_{\text{clustering}} \underbrace{P(Y^{(k)}, X^{(k)} | D^{(k)}; \Psi)}_{\text{descriptor extractor}} \underbrace{P(Y^{(k)} | X^{(k)}; \Theta^{(k)})}_{\text{local classifier}}$$

Each element in the decomposition represents a distinct component of FLUX:

- $P(Y^{(k)} | X^{(k)}; \Theta^{(k)})$ (*local classifier*): Each client $k$ models its joint distribution, which factorizes as $P(Y^{(k)} | X^{(k)}; \Theta^{(k)}) P(X^{(k)})$. Since $P(X^{(k)})$ is independent of $\Theta^{(k)}$, each client locally learns the predictive model by minimizing the corresponding negative conditional log-likelihood.

- $P(Y^{(k)}, X^{(k)} \mid D^{(k)}; \Psi)$ (*descriptor extractor*): A feature extractor $\phi$ with parameters $\Psi$ maps the high-dimensional data $(Y^{(k)}, X^{(k)})$ to a compact descriptor $D^{(k)}$. At test time, only $X^{(k)}$ is available, i.e., $P(D^{(k)} \mid \mathbf{0}, X^{(k)}; \Psi)$. Parameters $\Psi$ are learned on the client side (Section 4.2.1).
- $P(D^{(k)} \mid C^{(k)}; \Lambda)$ (*unsupervised clustering*): This term models the probabilistic assignment of clients to clusters based on their extracted descriptors. It is parameterized by an unsupervised clustering algorithm $\mathcal{U}$ with parameters $\Lambda$ (see Section 4.2.1).

After clustering, the server aggregates local models into cluster-specific parameters $\Theta_m$ by deterministically combining assignments $\{C^{(k)}\}$ with $\{\Theta^{(k)}\}$. This hierarchical decomposition disentangles the overall optimization into distinct sub-problems—clustering, descriptor extraction, and local classification— each of which can be optimized independently as follows:

$$
\begin{aligned}
\{\theta^{(k),*}\}_{k=1}^K, \psi^*, \lambda^* = \arg \max_{\{\theta^{(k)}\}_{k=1}^K, \psi, \lambda} \sum_{k=1}^K \Big[ & \log P\big(d^{(k)} \mid c^{(k)}; \lambda\big) \\
& + \sum_{(x,y) \in (x^{(k)}, y^{(k)})} \log P\big(y, x \mid d^{(k)}; \psi\big) + \sum_{(x,y) \in (x^{(k)}, y^{(k)})} \log P\big(y \mid x; \theta^{(k)}\big) \Big].
\end{aligned}
\quad (3)
$$

## 4.2 FLUX Pipeline

Figure 3 provides an overview of the FLUX pipeline, illustrating how its components translate theoretical models into actionable strategies. This subsection details the pipeline's mechanisms during both training and inference phases. During training (Section 4.2.1), FLUX securely extracts representative descriptors from client data which are then used to cluster clients with similar distributions in an unsupervised manner to accommodate data heterogeneity. During the inference phase (Section 4.2.2), FLUX assigns trained models to unseen clients by matching their data distributions to the closest clusters without requiring labeled data, ensuring adaptability in real-world scenarios.

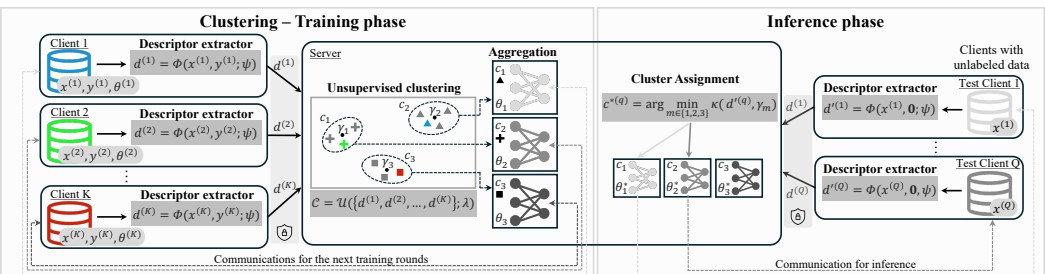

Figure 3: **Overview of the FLUX framework for efficient unsupervised CFL.** FLUX operates without prior knowledge of client data, handling distribution shifts and optimizing cluster-specific model assignment to unseen, unlabeled clients at inference.

### 4.2.1 Training Phase

The training phase is designed to address all four types of data distribution shifts—feature distribution shifts, label distribution shifts, $P(Y|X)$-concept shifts, and $P(X|Y)$-concept shifts. To identify these shifts, we extract representative descriptors that capture the key statistical characteristics of each client's labeled data, thus approximating the relevant data distributions for clustering.

**Descriptor extractor.** Consider a set of $M$ data distributions $\{P(X_m, Y_m)\}_{m=1}^M$, where each client holds data sampled uniformly from one of the $M$ distributions $P(X_m, Y_m)$. We define the descriptor extractor: $\phi_\psi : \mathbb{R}^{s^{(k)} \times (z+u)} \to \mathbb{R}^L$ $(L \ll s^{(k)} \times (z+u))$, parametrized by $\psi$, that maps a local dataset $(x^{(k)}, y^{(k)}) \sim P(X_m, Y_m)$ to a descriptor $d^{(k)} := \phi(x^{(k)}, y^{(k)}; \psi)$. For any two client pairs $k_1$ and $k_2$, the extractor must satisfy the requirements below:

(R1) *Distribution fidelity.* The function $\phi$ is designed to approximate a reference distance $D$ (e.g., Jensen–Shannon or Wasserstein distance) between the corresponding distributions:

$$
\big| \|d^{(k_1)} - d^{(k_2)}\|_2 - D\big(P(x^{(k_1)}, y^{(k_1)}), P(x^{(k_2)}, y^{(k_2)})\big) \big| \le \xi
\quad (4)
$$

In other words, the extractor $\phi$ maps similar joint distributions to nearby descriptors and dissimilar ones to distant descriptors.

(R2) *Label agnosticism.* A sub-vector $d'^{(k)} \subseteq d^{(k)}$ must be computable without any labels:

$$d'^{(k)} := \phi\big(x^{(k)}, \mathbf{0}; \psi\big) \in \mathbb{R}^p, \quad p \leq L \tag{5}$$

enabling descriptor extraction and similarity matching at test time. The sub-vector $d'^{(k)}$ encodes the marginal characteristics of the input distribution based solely on the features $x^{(k)}$.

(R3) *Compactness.* The function $\phi$ introduces minimal overhead relative to vanilla FL: its computational cost is negligible compared to a single local training epoch, and the descriptor dimensionality $L$ satisfies $L \ll p$ (typically $d/p \leq 10^{-2}$), ensuring that the additional communication cost remains marginal relative to the model update size $p$.

*Practical implementation.* To capture all four shift types, we factorize the joint distribution as $P(X,Y) = P(Y \mid X)P(X)$ and design $\phi$ to encode the marginal $P(X)$ and conditional $P(Y \mid X)$ separately. Appendix C.4 details how this design enables detection of feature, label, and both concept-shift variants, together with the step-by-step implementation. Concretely, each client's data $x^{(k)}$ is first mapped to a latent space by the shared encoder $f_e : \mathbb{R}^z \to \mathbb{R}^v$ with parameters $\theta$, then compressed by a client-invariant reduction $\xi_{v \to l} : \mathbb{R}^v \to \mathbb{R}^l$ ($v \gg l$), parametrized by $\psi$ (e.g., a shared PCA fitted on synthetic reference points, with $l = 10$):

$$z^{(k)} = \xi_{v \to l}\big(f_e(x^{(k)}; \theta); \psi\big). \tag{6}$$

To capture the marginal distribution $P(X)$, we compute the first two moments of the reduced latents—mean $\mu_x^{(k)}$ and covariance $\Sigma_x^{(k)}$. To capture the conditional distribution $P(Y \mid X)$, we compute the class-conditional moments $\{\mu_u^{(k)}, \Sigma_u^{(k)}\}_{u=1}^U$. The full descriptor $d^{(k)} \in \mathbb{R}^{2(U+1)l}$ is obtained by concatenating all marginal and conditional moments:

$$d^{(k)} = \big[\mu_x^{(k)}, \Sigma_x^{(k)}, \mu_1^{(k)}, \Sigma_1^{(k)}, \ldots, \mu_U^{(k)}, \Sigma_U^{(k)}\big]. \tag{7}$$

This instantiation meets every requirement: (R1) the mapping is provably Lipschitz-equivalent to the 2-Wasserstein metric, with $\xi < 1.1$ on MNIST and (and $< 0.54$ under Jensen–Shannon); (R2) the label-agnostic sub-vector $d'^{(k)}$ effectively captures $P(X)$; (R3) it adds negligible compute and a communication ratio $L/p \leq 3.5 \times 10^{-3}$. Theoretical justifications, motivation, and further details appear in Appendices C.1, C.3, and D.1. Additionally, differential privacy [27] can be seamlessly plugged into $d^{(k)}$ without affecting FLUX accuracy (see Appendix C.2). An ablation study and pseudocode are provided in Appendix F.6 and Algorithm 1.

**Unsupervised clustering.** In real-world FL settings, prior knowledge of client data distributions is rarely available. To address this, we propose using an unsupervised clustering method that determines the proper number of clusters $M$ and assigns each client to its corresponding cluster based on their descriptors. Formally, given the set of client descriptors $\{d^{(k)}\}_{k=1}^K$ and a clustering algorithm $\mathcal{U}$ parametrized by $\lambda$, the cluster assignments $\mathcal{C}$ are computed as:

$$\mathcal{C} = \mathcal{U}(d^{(1)}, d^{(2)}, \ldots, d^{(K)}; \lambda) \tag{8}$$

In our implementation, we design an adaptive density-based clustering method that automatically determines the number of clusters present. Specifically, we extend DBSCAN by estimating the $\epsilon$ parameter through elbow detection on the sorted second–nearest neighbour distance curve, calibrating it with a dataset-specific scaling factor, and reassigning noise points as singleton clusters to ensure every client is represented (see Appendix D.2 for details). Nonetheless, our formulation is agnostic to the choice of clustering method, provided it is unsupervised: in Appendix F.6.2, we report ablation studies with alternative clustering strategies, showing that our descriptors consistently enable robust client grouping regardless of the clustering algorithm, and in Appendix D.3, we provide illustrative examples of how the clustering process operates in practice.

### 4.2.2 Inference Phase

During inference, newly joining test clients require access to trained models. As these clients lack labeled data, we propose a mechanism that assigns test clients to the most suitable cluster-specific model based solely on their feature distributions. This approach optimizes inference process by leveraging the clustering structure learned during training.

**Test-time cluster assignment.** For each test client $q$, only feature-based descriptors can be extracted, as label information is unavailable. By fulfilling (R2), the descriptor extractor $\phi$ directly yields the label-agnostic descriptor as $d'^{(q)} = \phi(x^{(q)}, \mathbf{0}; \psi)$, which encodes the marginal distribution. Consequently, with information solely from the feature space $P(X)$, $P(Y|X)$-concept shifts cannot be solved during test time, as $P(X)$ across clients is identical (same input features, different labels).

Precisely, in our implementation, $d'^{(q)} = [\mu_x^{(q)}, \Sigma_x^{(q)}] \in \mathbb{R}^{2l}$. Each cluster $m$ identified during training is associated with a centroid $\gamma_m$, computed as the mean of the sub-vectors $d'^{(k)}$ of all clients $k$ in that cluster. To assign a test client $q$ to a cluster, we then compare its descriptor $d'^{(q)}$ with the cluster centroids $\gamma_m$ using a similarity metric $\kappa$ that measures proximity (e.g., Euclidean distance). The client is assigned to the cluster with the closest centroid according to the chosen metric:

$$c^{*(q)} = \arg \min_{m \in \{1, \ldots, M\}} \kappa(d'^{(q)} - \gamma_m) \tag{9}$$

This assignment ensures that the client $q$ uses the cluster-specific model $\theta_i^*$ that is most representative of its feature distribution. We provide the pseudo-code for our implementation in Algorithm 2.

# 5 Results

This section presents the experimental setup and results from two primary scaling experiments, analyzing FLUX performance under varying non-IID levels (Section 5.2) and an increasing number of clients (Section 5.3). These experiments evaluate and compare the scalability, robustness, and adaptability of FLUX against SOTA baselines across diverse distribution shifts and client configurations.

## 5.1 Experiment Settings

**Non-IID dataset generation.** We use six publicly available datasets in our experiments: MNIST [28], Fashion-MNIST (FMNIST) [29], CIFAR-10, CIFAR-100 [30]; and two real-world datasets, CheXpert [31] and Office-Home [32]. To simulate non-IID conditions in FL, we employ *ANDA*, a publicly available toolkit that enables data operations such as class isolation and label swapping. See Appendix B.1 for details on the *ANDA* and the datasets partitioning strategy.

**Baseline algorithms.** We evaluate our approach against ten baselines listed in Table 2, including FedAvg [1], CFL [14], IFCA [15], FeSEM [18], FedEM [16], FedRC [11], FedDrift [17], pFedMe [12], APFL [13], and ATP [20]. Full baseline descriptions are provided in Appendix E.1; experimental setup, hardware, models, and hyperparameter configurations are detailed in Appendices B.2 and B.3.

## 5.2 Robustness Across Different Types and Levels of Data Non-IID Heterogeneity

**Superior performance of FLUX across varying heterogeneity levels and datasets.** We benchmark FLUX against SOTA baselines across four shift types—$P(X)$, $P(Y)$, $P(Y|X)$, and $P(X|Y)$—each instantiated at eight increasing non-IID levels. Full setup and results are detailed in Appendix F.1. Figure 4 reports the average accuracy across shift types on MNIST (results for FMNIST, CIFAR-10, and CIFAR-100 are provided in the Appendix F.1), highlighting performance trends as heterogeneity severity increases.

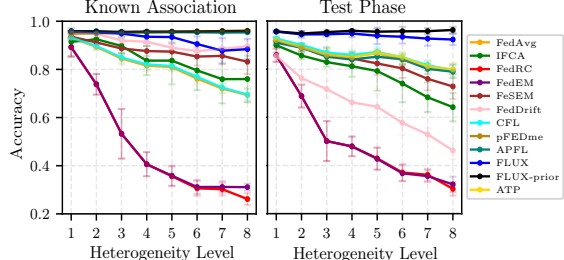

Figure 4: Mean accuracy and standard deviation across heterogeneity levels for MNIST.

FLUX consistently achieves robust performance across the heterogeneity spectrum, matching or exceeding PFL methods under the simplified *known association* condition—where test-time cluster assignments are assumed to be known—while maintaining high accuracy and stability even in the more realistic *test phase*, where such information is unavailable. Notably, FLUX yields absolute accuracy gains of up to 12.4, 23.0, 7.0, and 3.0 percentage points (pp) over the best-performing baselines on MNIST, FMNIST, CIFAR-10, and CIFAR-100, respectively. Table 1 summarizes the results for each dataset, averaging performance across all heterogeneity levels. In the test phase setting, FLUX consistently outperforms all baselines, improving accuracy by up to 7.84, 11.86, 2.17,

and 1.5 pp over the best-performing baselines—CFL on MNIST and FMNIST, IFCA on CIFAR-10, and FeSEM on CIFAR-100, respectively. Even under the simplified known association setting, FLUX matches or surpasses the best methods on MNIST, FMNIST, and CIFAR-100. On CIFAR-10, FLUX slightly trails PFL methods—expected, as they are fine-tuned on the same client distributions seen at test time—but their performance drops sharply on unseen clients. By contrast, ATP—an unsupervised test-time adaptation method—shows instability and underperforms compared to FLUX, particularly on complex datasets like CIFAR-100, where adaptation often leads to overconfident but incorrect predictions. Furthermore, FLUX-prior, which applies K-means clustering with the true number of clusters $M$ (oracle knowledge), sets a new CFL benchmark by achieving the highest accuracy across datasets and evaluation conditions. It is worth noting that most CFL baselines rely on this cluster information (Table 2), significantly simplifying their clustering process. Appendix F.3 further shows FLUX's robustness when multiple shift types occur simultaneously.

**Robust performance on real-world datasets.** To assess real-world applicability, we evaluate FLUX on the CheXpert dataset under three naturally occurring levels of non-IID heterogeneity (see Appendix F.4 for details), and the Office-Home dataset under four inherent domains. As CheXpert is a multi-label classification task, we report macro-averaged ROC AUC. Results summarized in Table 1 show that FLUX consistently outperforms all baselines in both the *known association* (up to 8.5 pp over the best-performing baseline, APFL) and *test phase* settings (up to 17.5 pp). On the Office-Home dataset, FLUX attains competitive performance, closely matching APFL in the *known association* setting and outperforming all baselines in the *test phase* by up to 1.3 pp. In contrast, most metric- and parameter-based CFL methods collapse to a single global model, failing to capture subtle real-world distribution shifts. FLUX performance closely matches that of FLUX-prior, underscoring the effectiveness of our learned descriptors and unsupervised clustering strategy—even without access to the true number of clusters.

| Dataset | MNIST | | FMNIST | | CIFAR-10 | | CIFAR-100 | | CheXpert | | Office-Home | |
|---|---|---|---|---|---|---|---|---|---|---|---|---|
| Algorithm | Known A. | Test Phase | Known A. | Test Phase | Known A. | Test Phase | Known A. | Test Phase | Known A. | Test Phase | Known A. | Test Phase |
| FedAvg | 80.9 ± 2.0 | 85.6 ± 2.1 | 65.5 ± 2.6 | 68.8 ± 2.7 | 30.9 ± 2.1 | 31.9 ± 2.3 | 33.8 ± 2.0 | 38.0 ± 2.3 | 56.1 ± 1.0 | 56.1 ± 1.0 | 37.1 ± 0.9 | 37.1 ± 0.9 |
| IFCA | 84.1 ± 7.5 | 78.2 ± 5.4 | 72.4 ± 5.2 | 63.5 ± 4.9 | 38.3 ± 2.5 | 36.6 ± 2.3 | 36.1 ± 4.1 | 38.6 ± 2.9 | 58.5 ± 0.4 | 58.5 ± 0.4 | 32.8 ± 8.1 | 29.6 ± 7.5 |
| FedRC | 47.5 ± 5.1 | 49.9 ± 4.5 | 54.3 ± 5.0 | 55.7 ± 5.3 | 17.3 ± 2.2 | 17.2 ± 2.4 | 33.8 ± 1.2 | 37.9 ± 1.1 | 58.8 ± 0.4 | 58.3 ± 0.4 | 22.2 ± 2.8 | 22.2 ± 2.8 |
| FedEM | 48.3 ± 5.0 | 50.0 ± 4.5 | 55.7 ± 5.2 | 56.0 ± 5.2 | 18.0 ± 2.3 | 17.5 ± 2.4 | 34.8 ± 1.1 | 38.0 ± 1.1 | 58.5 ± 0.4 | 58.5 ± 0.4 | 28.4 ± 2.3 | 28.8 ± 1.8 |
| FeSEM | 87.8 ± 3.1 | 82.8 ± 3.7 | 71.9 ± 3.7 | 66.2 ± 4.4 | 36.5 ± 2.7 | 35.3 ± 2.5 | 35.6 ± 2.4 | 39.8 ± 1.8 | 59.0 ± 0.4 | 58.3 ± 0.4 | 27.0 ± 2.9 | 25.8 ± 1.8 |
| FedDrift | 91.1 ± 2.9 | 65.1 ± 3.7 | 80.7 ± 2.1 | 51.1 ± 2.6 | 35.1 ± 2.6 | 32.1 ± 2.2 | 39.5 ± 3.1 | 26.5 ± 1.8 | 61.6 ± 0.6 | 61.6 ± 0.6 | 39.5 ± 4.0 | 34.6 ± 4.1 |
| CFL | 81.3 ± 1.8 | 86.1 ± 1.9 | 66.0 ± 2.8 | 69.4 ± 3.0 | 32.2 ± 2.1 | 33.2 ± 2.3 | 34.6 ± 1.5 | 38.6 ± 1.6 | 58.5 ± 0.4 | 58.5 ± 0.4 | 31.3 ± 3.3 | 21.0 ± 1.5 |
| pFedMe | 95.5 ± 0.3 | N/A | 81.9 ± 1.2 | N/A | 42.4 ± 1.1 | N/A | 36.1 ± 2.1 | N/A | 69.4 ± 0.2 | N/A | 30.9 ± 2.3 | N/A |
| APFL | 95.5 ± 0.3 | 84.7 ± 2.4 | 81.9 ± 1.6 | 69.2 ± 2.8 | **44.7 ± 1.1** | 36.6 ± 2.1 | **44.2 ± 1.7** | 37.3 ± 0.8 | 72.3 ± 0.2 | 64.0 ± 0.4 | **43.5 ± 2.1** | 36.7 ± 0.4 |
| ATP | N/A | 85.6 ± 1.8 | N/A | 68.4 ± 2.9 | N/A | 33.6 ± 2.2 | N/A | 37.5 ± 1.5 | N/A | N/A | N/A | 37.9 ± 0.9 |
| FLUX | 92.5 ± 2.9 | 94.0 ± 2.2 | 79.4 ± 2.8 | 81.2 ± 2.7 | 38.6 ± 2.3 | 38.7 ± 3.3 | 41.7 ± 2.0 | 41.3 ± 3.1 | **79.4 ± 0.3** | 78.6 ± 0.9 | 39.2 ± 0.2 | **39.2 ± 0.3** |
| FLUX-prior | **95.8 ± 0.3** | **95.7 ± 1.7** | **81.9 ± 1.2** | **83.3 ± 1.4** | 40.1 ± 1.6 | **39.3 ± 3.1** | 42.8 ± 1.2 | 41.3 ± 3.2 | 79.3 ± 0.3 | 78.5 ± 0.9 | 43.2 ± 3.1 | 38.9 ± 1.1 |

Í

Table 1: **Overall performance on MNIST, FMNIST, CIFAR-10, CIFAR-100, Office-Home (accuracy), and CheXpert (ROC AUC). Known A.:** Known Association. **N/A**: Not available.

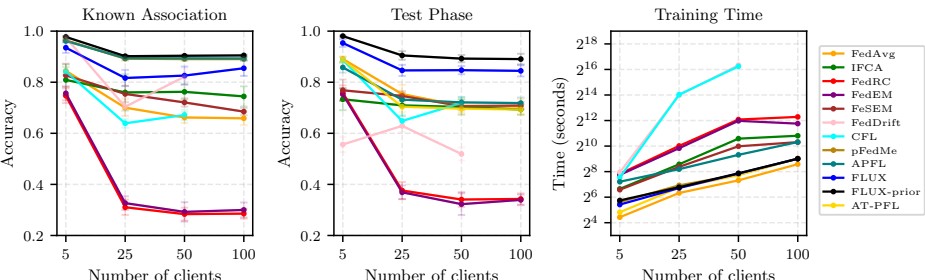

Figure 5: **Mean accuracy and standard deviation on MNIST dataset with varying numbers of clients.** Left: *known association* condition, where test-time cluster associations are available. Middle: *test phase* condition, where cluster associations are inferred. Right: training time per 10 rounds.

| ALGORITHM | CAT. | ALL NON-IID | TEST ADAPTATION | NO CLUSTER NUMBER PRIOR | LOW COMM. COST | LOW COMP. COST (SERVER) | LOW COMP. COST (CLIENT) | SCALABILITY |
|---|---|---|---|---|---|---|---|---|
| FEDAVG [1] | N/A | N/A | N/A | N/A | ✓ | ✓ | ✓ | ✓ |
| IFCA [15] | CFL | | | | | ✓ | | ○ |
| FESEM [18] | CFL | | | | ✓ | | ✓ | |
| FEDEM [16] | CFL | | | | | ✓ | | ○ |
| FEDRC [11] | CFL | | | | | ✓ | | ○ |
| FEDDRIFT [17] | CFL | | | ✓ | | ✓ | | |
| CFL [14] | CFL | | ✓ | ✓ | | ✓ | | |
| pFEDME [12] | PFL | | | ✓ | ✓ | ✓ | ○ | ✓ |
| APFL [13] | PFL | | | ✓ | ✓ | ✓ | | ✓ |
| ATP [20] | AFL | | ✓ | ✓ | ✓ | ✓ | ○ | ✓ |
| **FLUX** | CFL | ✓ | ✓ | ✓ | ✓ | ✓ | ✓ | ✓ |

Table 2: **Qualitative comparison of FLUX and baselines**. **Cat.**: FL category. **All non-IID**: designed to tackle all four non-IID types. **Test adaptation**: adapt to unseen, unlabeled clients. **No cluster number prior**: not required knowledge of distribution numbers. **Low comm. cost**: relatively low communication cost (comparable to FedAvg). **Low comp. cost (Server/Client)**: relatively low computational cost on server/client side (comparable to FedAvg). **Scalability**: scales efficiently with large client numbers. **N/A**: not applicable. ✓: property satisfied. ○: property conditionally satisfied.

### 5.3 Scalability and Efficiency Across Increasing Numbers of Clients

FL is designed to jointly train models across millions of clients with limited computational resources and memory. This necessitates efficient and lightweight methods capable of handling numerous data distribution shifts (i.e., clusters). To evaluate scalability, we measure the accuracy and training time of FLUX and the baseline methods by increasing the number of clients from 5 to 100 (Figure 5). Due to prohibitive memory and computational costs (see Appendix B.3), FedDrift and CFL could not be evaluated with 100 clients. Detailed experimental setup and results are provided in Appendix F.2.

**FLUX maintains high accuracy with increasing numbers of clients and clusters, demonstrating adaptability to large-scale FL.** Figure 5 shows that accuracy generally decreases for all methods, confirming that as the number of clients—and therefore clusters—increases, clustering becomes more challenging due to the presence of numerous diverse data distributions, reduced training data per cluster, and the higher aggregation divergence inherent to FL. Despite this, FLUX and FLUX-prior maintain consistent accuracy in both evaluation settings (*known association* and *test phase*), outperforming baselines, which exhibit significant accuracy degradation as the number of clients increases. During the *test phase*, FLUX sustains consistent accuracy above 84%, whereas the closest baseline, APFL, declines to over 70%.

**FLUX demonstrates scalability and efficiency with minimal computational overhead.** We further evaluated the efficiency of FLUX by measuring the overall training time as the number of clients increased. As illustrated in Figure 5, FLUX shows training times comparable to FedAvg, with only minor differences in execution time (on the order of seconds). The efficiency of FLUX stems from its minimal modifications to the traditional FedAvg framework. Specifically, the additional costs introduced by FLUX—primarily in the communication step and clustering process—are negligible relative to the total costs of model parameter transmission and aggregation. The communication overhead is proportional to the length of the descriptor ($L$), which is significantly smaller than the total model size ($L/p \leq 3.5 \times 10^{-3}$). Similarly, the clustering cost ($O(L \cdot \log(L))$) is substantially lower than the aggregation cost ($O(N_{\text{client}} \cdot \theta)$). Notably, the computational time of the fastest CFL baseline, FeSEM, is more than 4 times that of FLUX, while the slowest, FedDrift, requires over 300 times the computational time of FLUX. Further details are provided in Appendix E.2.

## 6 Discussion and Related Works

**Clustered Federated Learning.** CFL partitions the overall client data into clusters based on training behaviors or data distributions, serving as a middle ground between PFL and traditional FL. By aggregating within clusters, CFL can potentially accelerate model convergence and mitigate local overfitting. CFL methods are typically categorized as hard- [14, 15, 17, 18] or soft-CFL [11, 16], and further distinguished by their clustering principles: *metric-based* [15–17, 33, 34] or *parameter-based* [11, 14, 18, 35]. Metric-based CFL (e.g., loss-based) may miscluster clients with similar empirical risks but divergent loss distributions, showing that a single loss value is insufficient for reliable clustering. Likewise, feature-based metrics such as frequency coefficients [33] cannot

distinguish clients with different conditional distributions but identical marginals. Parameter-based CFL may misgroup models due to permutation invariance and overparameterization, assigning identical functions with different parameters to separate clusters or vice versa [36]. In contrast, FLUX introduces a descriptor-based clustering paradigm, where descriptors approximate the 2-Wasserstein distance and jointly capture both marginal and conditional properties of client data, a capability essential for handling all four types of distribution shift simultaneously.

**Personalized Federated Learning.** PFL [10, 13, 37–53] reframes FL as a client-centric optimization, learning a distinct model for each client to address data heterogeneity. PFL methods include fine-tuning a shared global model via additional local updates or meta-learning [10, 37, 38]; model-decoupling approaches that split networks into shared and client-specific components or adapt batch-norm layers [13, 39–42]; and regularization-based methods that add proximal or penalty terms linking personalized and global parameters [10, 54, 55]. While effective with ample labeled data, PFL demands hyperparameter tuning, adds on-device compute and memory overhead, and—being inherently supervised—cannot tackle unseen or unlabeled clients or distribution shifts at test time.

**Test-time Adaptive FL.** TTA-FL tackles post-deployment distribution shifts by adapting a pre-trained global model to each client's unlabeled test data using unsupervised objectives—e.g., entropy minimization [19, 20, 56] or contrastive losses [57, 58]. Without labels, these objectives are ill-conditioned: entropy minimization often yields overconfident mispredictions under concept shift, and contrastive losses can collapse on small or imbalanced batches. To enhance stability, recent works restrict adaptation to a few parameters (e.g., interpolation weights) [19]. However, they still depend on supervised pre-training of personalized components, rendering them unsuitable for truly test-only clients. A more detailed discussion of these methods is provided in Appendix A.1.2.

# 7   Limitations and Future Works

FLUX has two primary limitations. First, to obtain statistically robust descriptors of client data distributions, a sufficient amount of diverse training data is required, with the exact quantity depending on the variability of the underlying data distributions. Second, FLUX operates as a one-shot CFL framework, which does not account for dynamic scenarios where client data distributions evolve over time (i.e., distribution drift). Nonetheless, the efficiency and simplicity of FLUX make it well-suited for dynamic implementations. The clustering process can be repeated whenever a drift occurs within a subset of clients, with the new clusters being assigned to the closest existing cluster-specific models (see Appendix D.3). Building on this potential, we plan to extend FLUX to support dynamic scenarios, enabling adaptive clustering and model updates to address evolving client data distributions.

# 8   Conclusion

In this work, we proposed FLUX, a novel FL framework that addresses the four most common types of data distribution shifts ($P(X)$, $P(Y)$, $P(Y|X)$, and $P(X|Y)$) without requiring any prior knowledge of client data distributions, such as the number of clusters. By leveraging client-side extraction of representative descriptors and an unsupervised clustering approach, FLUX achieves superior performance and robustness across varying levels of heterogeneity and increasing numbers of clients. Unlike existing methods, FLUX enables testing-time association, allowing unseen and unlabeled clients to utilize the most suitable cluster-specific model obtained during training, addressing a critical gap in real-world FL applications. Moreover, FLUX incurs minimal computational and communication overhead, comparable to FedAvg, making it scalable and practical for large-scale FL deployments. Our results establish FLUX as a flexible, scalable, and robust solution for FL, paving the way for more practical and adaptable frameworks in decentralized and privacy-preserving ML.

## Acknowledgments and Disclosure of Funding

This research was funded by the Swiss National Science Foundation, the European Union's Horizon Europe programme, and the Slovenian Research and Innovation Agency through the projects SmartCHANGE (No. 101080965), TRUST-ME (No. 205121L_214991), and XAI-PAC (No. Z00P2_216405). PB acknowledges support from the Swiss National Science Foundation Postdoctoral Fellowships IMAGINE (No. 224226).

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

# APPENDIX CONTENTS

# A    Related Approaches to Distribution Shift

A wide range of methods has been proposed to address data heterogeneity in FL, which can be grouped into three main categories—clustered FL (CFL), personalized FL (PFL), and test-time adaptive FL (TTA-FL). Below, we summarize each category's key principles and limitations.

## A.1    Clustered Federated Learning

CFL assumes the presence of $M$ distinct data distributions within the federation and aims to learn one model per distribution. Existing CFL methods can be categorized by their *clustering strategy*—either as hard or soft clustering—and by the underlying *clustering principle*, namely metric-based or parameter-based (gradient-based) approaches.

### A.1.1    Clustering Strategy: Hard vs. Soft CFL

In Equation 2, we introduced a general definition of CFL that unifies hard- and soft CFL under a single framework. This formulation follows standard mixture-model derivations: soft-CFL defines client assignments $c^{(k)}$ as probability vectors, while hard-CFL is the special case where $c^{(k)}$ reduces to a one-hot vector. Hence, Equation 2 can be seen as a surrogate objective derived from the soft-CFL likelihood via the standard Jensen lower bound, subsuming both variants under a single expression. This appendix explicitly presents the optimization problems for hard-CFL and soft-CFL, as is standard in the literature, to provide greater clarity on their specific formulations.

- *Hard-CFL* assigns each client $k \in \mathcal{K}$ to exactly one cluster based on predefined criteria or similarity measures. Therefore, each cluster $c_m$ is a subset of $\mathcal{K}$, i.e., $c_m \subseteq \mathcal{K}$ for all $m \in \{1, \ldots, M\}$. Additionally, the clusters are disjoint and collectively exhaustive, satisfying $\bigcup_{m=1}^{M} c_m = \mathcal{K}$ and $c_m \cap c_n = \emptyset$ for all $m \neq n$. This means that each cluster $c_m$ includes a distinct subset of clients with similar data distributions, allowing the hard-CFL framework to train specialized models tailored to each cluster's specific data distribution. Formally, hard-CFL seeks to jointly optimize the model parameters and the cluster assignments as in Equation 10. This formulation ensures that each cluster-specific model $\theta_m^*$ minimizes the risk for all clients within cluster $c_m$, and generally suppose to know the number of clusters $M$.

$$\{\theta_m^*\}_{m=1}^{M^*}, \mathcal{C}^* = \arg \max_{\{\theta_m\}_{m=1}^{M}, \mathcal{C}} \sum_{m=1}^{M} \sum_{(x,y) \in (x^{(k)}, y^{(k)})} \log P(y|x; \theta_m), \tag{10}$$

- *Soft-CFL* allows clients $k \in \mathcal{K}$ to belong to multiple clusters with certain probabilities or weights, accommodating scenarios where client data may exhibit overlapping distributions that can be effectively modeled by a combination of specific models. Therefore, each client $k$ is associated with a weight vector $\pi^{(k)} = \{\pi_1^{(k)}, \ldots, \pi_M^{(k)}\}$, where $\pi_m^{(k)} \geq 0$ and $\sum_{m=1}^{M} \pi_m^{(k)} = 1$. This enables each client to contribute to multiple cluster-specific models based on weights. Mathematically, soft-CFL seeks to jointly optimize the cluster-specific model parameters $\{\theta_m\}_{m=1}^{M}$ and the assignment weights $\{\pi^{(k)}\}_{k=1}^{K}$ as follows:

$$\{\theta_m^*\}_{m=1}^{M^*}, \{\pi^{(k),*}\}_{k=1}^{K} = \arg \max_{\{\theta_m\}_{m=1}^{M}, \{\pi^{(k)}\}_{k=1}^{K}} \sum_{m=1}^{M} \sum_{k=1}^{K} \sum_{(x,y) \in (x^{(k)}, y^{(k)})} \pi_m^{(k)} \log P(y|x; \theta_m), \tag{11}$$

This formulation allows each client to be influenced by multiple cluster-specific models, thereby enhancing the flexibility of cluster assignment. However, optimizing both cluster-specific models and assignment weights $\pi^{(k)}$ simultaneously poses additional computational and convergence challenges, requiring alternative optimization techniques [59]. Furthermore, soft-CFL is generally not designed to handle unseen clients effectively, as determining assignment weights $\pi^{(k)}$ requires access to the training data of these clients [11, 16].

### A.1.2    Clustering Principles: Metric-Based vs. Parameter-Based CFL

CFL can be further classified into two categories based on their clustering principles: *metric-based CFL* and *parameter-based CFL*. Table 3 provides a categorization of some existing CFL frameworks according to these principles.

| CLUSTERING PRINCIPLE | FRAMEWORK |
|---|---|
| METRIC-BASED | IFCA [15] |
| | FEDEM [16] |
| | FEDDRIFT [17] |
| | FLIS [60] |
| | FEDCE [61] |
| | ACFL [62] |
| | FLSC [63] |
| | FEDGWC [34] |
| PARAMETER-BASED | CFL [14] |
| | FESEM [18] |
| | FEDRC [11] |
| | FL+HC [35] |
| | AUTOCFL [64] |
| | FLEXCFL [65] |

Table 3: **Summary of CFL frameworks categorized by clustering principles**.

- *Metric-Based CFL.* These algorithms assign cluster identities based on model losses or inference-phase accuracy. Typically, the server broadcasts $M$ models corresponding to $M$ clusters, and each client selects the model that minimizes the empirical risk on its local data—essentially identifying the closest model as its cluster. Alternatively, client-side loss or accuracy values can be directly utilized by the server for clustering. However, this approach relies on highly compressed information, which can lead to inaccuracies. For instance, two clients with distinct data distributions might achieve the same accuracy by making errors on different samples, causing them to be grouped in the same cluster.

- *Parameter-Based CFL.* These algorithms perform clustering based on the parameters or gradient updates of clients' models. When clients possess non-IID datasets, local training epochs result in diverging model parameter updates, which can be utilized for cluster identification. Two common approaches for parameter-based CFL are: i) the server broadcasts $M$ models, and each client selects the one closest to its updated model, and ii) the server directly clusters clients' updated model parameters or gradients. However, relying solely on distance metrics for clustering may also lead to inaccuracies, as such metrics provide overly condensed information, making it challenging to differentiate intrinsically non-IID clients.

FLUX explicitly employs a descriptor extractor to capture statistical characteristics of client data. It does not fall neatly into either category, as it clusters based on data-distribution descriptors rather than model parameters or compressed metrics. The closest related approach is HACCS [66], which also introduces descriptor-like summaries through label and pixel-value histograms. However, HACCS lacks semantic and structural awareness (pixel statistics may not reflect task-relevant features), requires labels at inference (preventing test-time adaptation), and produces descriptors whose size grows with the number of labels and bins. Moreover, its histograms are human-interpretable, raising stronger privacy concerns (e.g., revealing label distributions), and requiring strong differential privacy. In contrast, FLUX leverages compact latent representations aligned with the classification objective, which are non-invertible and task-relevant, enabling robust clustering across all four types of distribution shift, and generalization to unseen clients. Numerical results comparing FLUX with HACCS are provided in Section E.3.

## A.2 Personalized Federated Learning.

PFL addresses client heterogeneity by shifting from a global objective to a client-specific optimization framework, where each client $k$ learns a tailored model optimized for its local data distribution. Instead of enforcing a single shared model across all participants, PFL explicitly accounts for non-IID data by enabling model personalization. Existing methods can be broadly classified into three categories:

- *Fine-tuning.* These approaches begin with a shared global model that is subsequently adapted locally using additional gradient steps or meta-learning techniques to simplify personalization [37, 67, 38]. While conceptually simple, they often require careful tuning of hyperparameters and sufficient local data to avoid overfitting.

- *Model decoupling.* These methods separate the model into shared (global) and client-specific components, such as training a common backbone alongside global and personalized heads [39,

13, 40, 19, 42]. Other variants adapt only batch-norm statistics [41] or allow for heterogeneous encoder architectures [68]. These approaches enhance model expressiveness but incur higher on-device memory and computational overhead.

- *Regularization-based.* These approaches personalize models by adding a regularization term that encourages proximity between each client's local model $\phi^{(k)}$ and the global model $\theta$. A typical formulation augments the local objective with a penalty term: $\min_{\phi^{(k)}} \mathcal{L}^{(k)}\big(\phi^{(k)}\big) + \frac{\lambda}{2} \big\|\phi^{(k)} - \theta\big\|^2$, where $\lambda$ controls the trade-off between personalization and global consistency [67, 54]. While these methods offer smooth personalization, they often require tuning client-specific hyperparameters and solving nested or bi-level optimization problems.

Although PFL methods are effective when sufficient labeled data is available on each client, their performance degrades in low-data regimes. Moreover, they typically introduce additional computational and memory overhead on the client side, and—being inherently supervised—are not applicable to unseen, unlabeled clients that appear only at test time.

### A.3 Test-time Adaptive Federated Learning.

TTA-FL targets *post-deployment* distribution shifts by adapting a pre-trained global model to each client's unlabeled test data. Most approaches optimize *unsupervised* objectives—such as entropy minimization [19, 20, 56] or self-supervised contrastive losses [57, 58]—directly on the test set. However, in the absence of labels, the resulting optimization landscape is often unstable: entropy minimization can lead to overconfident mispredictions under concept shift, while contrastive losses may collapse on small or imbalanced test batches—a common constraint in FL deployment. To improve stability, recent methods constrain adaptation to a small set of parameters (e.g., batch-norm statistics or interpolation weights between global and personalized heads [19]). Nonetheless, these models still require labeled data during training to learn the personalized components, limiting their applicability to truly unseen and unlabeled clients at test time.

## B  Reproducibility & Implementation Overview

### B.1  Datasets and Non-IID Generation Protocol

**ANDA.** For MNIST, FMNIST, CIFAR-10, and CIFAR-100, we generate the four most common types of distribution shift using *ANDA* [‡] (A Non-IID Data generator supporting Any kind), a toolkit designed to create reproducible non-IID datasets for FL experimentation. It supports datasets MNIST, EMNIST, FMNIST, CIFAR-10, and CIFAR-100, and facilitates five types of data distribution shifts: feature distribution shift, label distribution shift, $P(Y|X)$ concept shift, $P(X|Y)$ concept shift, and quantity shift.

Figures 6 and 7 illustrate examples of MNIST and CIFAR-10 datasets under four distinct types of data distribution shifts generated using *ANDA*:

- In (a), the two clients experience different feature distribution shifts. Each image undergoes one of three color transformations (blue, green, or red) and one of four rotations ($0°$, $90°$, $180°$, or $270°$), with distinct distributions applied to each client. For instance, the first client has a higher proportion of blue images, while the second client has more red images.
- In (b), the clients experience label distribution shifts. Each client receives data from only three classes. For example, in the MNIST dataset, the first client has images of digits 0, 3, and 6, while the second client has images of digits 1, 2, and 4.
- In (c), the clients exhibit $P(Y|X)$-concept shifts. Given identical feature distributions (e.g., image pixels), labels differ between clients. For instance, in the MNIST dataset, the first client labels the digit '1' as '1' and '2' as '2', whereas the second client labels '1' as '2' and '2' as '1'.
- In (d), the clients demonstrate $P(X|Y)$-concept shifts. For the same label, different features are applied. For example, in the MNIST dataset, the first client applies a red hue to images labeled '2', while the second client applies a blue hue to the same label.

In subsequent sections, we detail the specific dataset generation settings employed with *ANDA*.

---

[‡]https://github.com/alfredoLimo/ANDA.git

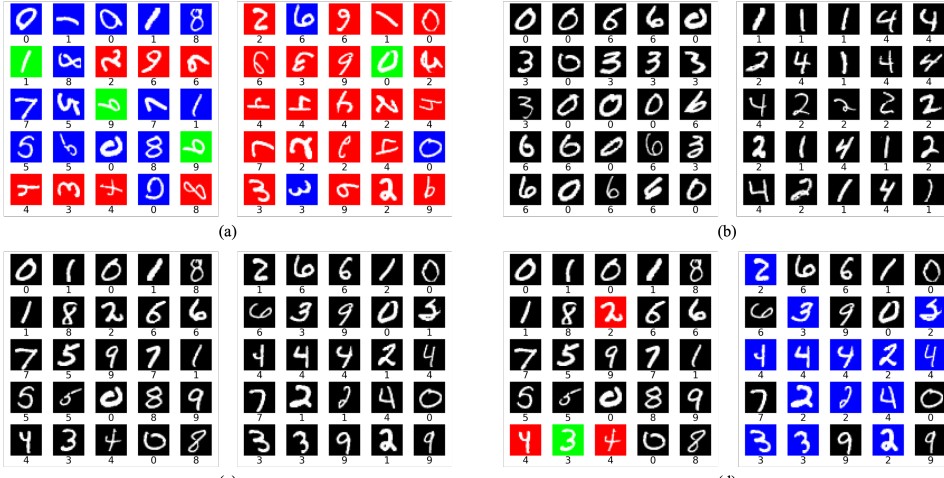

Figure 6: **MNIST datasets with four different types of data distribution shifts generated by** *ANDA*.

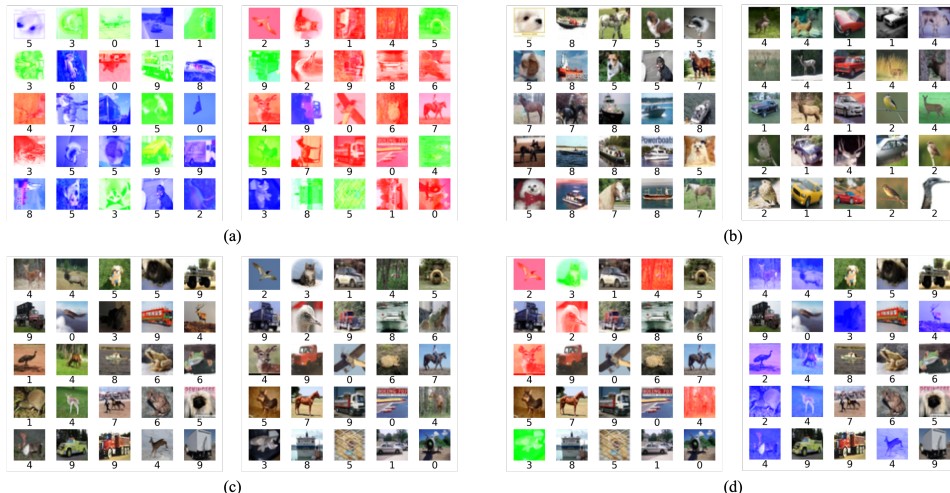

Figure 7: **CIFAR-10 datasets with four types of data distribution shifts generated by** *ANDA*.

**CheXpert.** For the real-world CheXpert dataset, we retain the original data without applying any image augmentations or label modifications to preserve the correctness and clinical integrity of the data. To simulate varying levels of client heterogeneity, we partition the dataset into multiple distributions using three available metadata attributes: *ViewPosition*, *Age*, and *Sex*. Specifically, we construct the following non-IID configurations:

- **Low heterogeneity:** 2 distributions based on *ViewPosition* (Frontal vs. Lateral).
- **Medium heterogeneity:** 4 distributions based on *ViewPosition* and *Age* ($<$50 vs. $\geq$50).
- **High heterogeneity:** 8 distributions based on *ViewPosition*, *Age*, and *Sex* (Male vs. Female).

**Office-Home.** For the real-world dataset Office-Home, we do not apply any image augmentation or label modification; instead, we directly use the dataset in its original form. For our experiments, we restrict the dataset to images with only 20 classes. To model different levels of data heterogeneity, we leverage the four inherent domains of the dataset: Art, Clipart, Product, and Real-World. The domain shift across these categories naturally induces heterogeneity in the data distribution.

### B.2 Models and Hyper-parameter Settings

We adopt a 5-fold cross-validation strategy to evaluate model performance, using fixed random seeds (42, 43, 44, 45, and 46) to ensure reproducibility. For the MNIST, FMNIST, and CIFAR-10

datasets, we use LeNet-5 [69] as the base model; for CIFAR-100, CheXpert, and Office-Home, we use ResNet-9 [70]. A batch size of 64 is used for both training and evaluation. Each client reserves 20% of its local data for validation. The FL process runs for 10 communication rounds on MNIST, FMNIST, and CIFAR-10, for 15 rounds on CIFAR-100, 20 rounds on CheXpert, 40 rounds on Office-Home, with each client performing 2 local training epochs per round. The learning rate is set to 0.005 with a momentum of 0.9. All models are trained using cross-entropy loss, except on CheXpert, where a binary cross-entropy loss is used due to the multi-label classification setting.

### B.3   Code, Licenses and Hardware

Our experiments were implemented using Python 3.12 and open-source libraries including PyTorch 2.4 [71] (BSD license), Scikit-learn 1.5 [72] (BSD license), and Flower 1.11 [73] (Apache License). For visualization, we utilized Matplotlib 3.9 [74] (BSD license) and Seaborn 0.13 [75] (BSD license), while data processing was performed using Pandas 2.2 [76] (BSD license). The datasets used in our experiments—MNIST (GNU license), FMNIST (MIT license), CIFAR-10, CIFAR-100, CheXpert, and Office-Home—are freely available online. To ensure reproducibility, our code, along with detailed instructions for reproducing the experiments, is publicly accessible on GitHub[§] under the MIT license. We used publicly available codes for our baselines (except FedAvg).

All experiments were conducted on a workstation equipped with four NVIDIA RTX A6000 GPUs (48 GB each), two AMD EPYC 7513 32-Core processors, and 512 GB of RAM.

### B.4   Algorithm Pseudo-code

We provide the pseudo-code for both the training phase (Algorithm 1) and inference phase (Algorithm 2) of our proposed implementation of FLUX. These include steps for extracting representative descriptors of client data distributions and performing unsupervised clustering.

## C   Descriptor Extractor Details and Theoretical Guarantees

### C.1   Descriptor Extractor

This section presents the implementation of the descriptor extractor $\phi$, which enables each client to encode its local data distribution in a compact and privacy-preserving way. The resulting descriptor $d^{(k)}$ supports consistent similarity comparisons across clients while satisfying all requirements in Definition 4.2.1. We outline the implementation in four federated steps:

### C.1.1   Implementation Details.

Let the final hidden layer of the global model produce, for client $k$, a matrix of latent representations $s^{(k)} \in \mathbb{R}^{s^{(k)} \times z}$. The descriptor extractor maps these latents to a descriptor $d^{(k)} \in \mathbb{R}^L$ through the following steps:

- S1 *Global alignment (one shot, no raw data).* Each client computes the element-wise minimum and maximum of its latents and sends only these two $z$-dimensional vectors (i.e., $2z$ floats) to the server. The server aggregates them by coordinate-wise min and max to obtain global bounds $[m^-, m^+]$, which are broadcast to all clients along with the model weights. This step ensures consistent alignment without transmitting raw data, labels, or gradients.
- S2 *Shared PCA on synthetic reference points.* Using a shared random seed, all clients generate 200 synthetic points uniformly sampled from $[m^-, m^+]$ and independently fit a PCA map $g : \mathbb{R}^z \to \mathbb{R}^l$ (with $l = 10$) to this synthetic set. Because both the sampling and data are identical, all clients derive the same linear projector $g$, ensuring that Euclidean geometry in the reduced space is aligned across the federation.
- S3 *Moment computation.* Each client projects its latent matrix using the shared PCA: $z^{(k)} = g(s^{(k)}) \in \mathbb{R}^{s^{(k)} \times l}$. It then computes: (i) global statistics $(\mu_x^{(k)}, \Sigma_x^{(k)})$, the mean and covariance of $z^{(k)}$, approximating the marginal distribution $P(X)$, and (ii) class-conditional statistics $\{(\mu_u^{(k)}, \Sigma_u^{(k)})\}_{u=1}^U$, approximating the conditional distribution $P(Y \mid X)$.

---

[§] https://github.com/dariofenoglio98/FLUX

---

**Algorithm 1:** FLUX Framework – Training Phase

---

**Input:** Model $f(\theta)$, client set $\mathcal{K} = \{1, \ldots, K\}$, rounds $R$, descriptor extractor $\phi(\cdot; \psi)$, clustering method $\mathcal{U}(\cdot; \lambda)$.

**Output:** Cluster-specific models $\{\theta_m\}_{m=1}^M$, cluster centroids $\{\gamma_m\}_{m=1}^M$

1 **Descriptor Extraction**;
2 $\mathcal{A} \leftarrow \text{RandomClientSelection}(\mathcal{K})$ ;          // $\mathcal{A}$ contains clients for initial clustering
3 **for** *each client* $k \in \mathcal{A}$ *in parallel* **do**
4     $\theta^{(k)} \leftarrow \theta^*$;
5     $d^{(k)} \leftarrow \phi\Big(x^{(k)}, y^{(k)}; \psi\Big)$ ;                    // Extract descriptor
6     **Send** $d^{(k)}, s^{(k)}$, and $\theta^{(k)}$ to server;

7 **Unsupervised Clustering**;
8 $\mathcal{C} \leftarrow \mathcal{U}\Big(\{d^{(k)}\}_{k \in \mathcal{A}}; \lambda\Big)$    with    $\mathcal{C} = \{c^{(k)}\}_{k \in \mathcal{A}}, \; c^{(k)} \in \{0,1\}^M$;
9 **for** $m = 1$ **to** $M$ **do**
10     $n_m \leftarrow \sum_{k \in \mathcal{A}} c_m^{(k)}$;
11     $\gamma_m \leftarrow \frac{1}{n_m} \sum_{k \in \mathcal{A}} c_m^{(k)} d^{(k)}$ ;                    // Cluster centroids

12 **Clustered Federated Learning**;
13 **for** $r = 1$ **to** $R$ **do**
14     **for** $m = 1$ **to** $M$ **do**
15        $S_m \leftarrow \sum_{k=1}^K c_m^{(k)} s^{(k)}$;
16        $\theta_m \leftarrow \frac{1}{S_m} \sum_{k=1}^K c_m^{(k)} s^{(k)} \theta^{(k)}$ ;                    // Weighted aggregation
17     $\mathcal{A} \leftarrow \text{RandomClientSelection}(\mathcal{K})$ ;          // New participants each round
18     **for** *each client* $k \in \mathcal{A}$ *without* $c^{(k)}$ *in parallel* **do**
19        $\theta^{(k)} \leftarrow \theta^*$;
20        $d^{(k)} \leftarrow \phi\Big(x^{(k)}, y^{(k)}; \psi\Big)$ ;                    // Descriptor for cluster assignment
21        **Send** $d^{(k)}, s^{(k)}$, and $\theta^{(k)}$ to server;
22     **for** *each client* $k \in \mathcal{A}$ *without* $c^{(k)}$ *in parallel* **do**
23        $c^{(k)} \leftarrow \arg\min_{m=1,\ldots,M} \kappa\Big(d^{(k)}, \gamma_m\Big)$ ;                    // Model/cluster assignment
24     **for** *each client* $k \in \mathcal{A}$ *with* $c_m^{(k)} = 1$ **do**
25        **Send** $\theta_m$ to client $k$;
26     **for** *each client* $k \in \mathcal{A}$ *in parallel* **do**
27        **Receive** $\theta_m$;
28        $\theta^{(k)} \leftarrow \theta_m$;
29        $\theta^{(k)} \leftarrow \text{LocalTrain}\Big(\theta^{(k)}, x^{(k)}, y^{(k)}\Big)$ ;                    // Local update
30        **Send** $\theta^{(k)}$ and $s^{(k)}$ to server;

---

*S4 Differential privacy (optional plug-in).* To enhance privacy guarantees, each statistic $g_i$ in the descriptor can be independently perturbed using the Laplace mechanism ($\delta = 0$) [77], as detailed in Appendix C.2:

$$\eta_i \sim \text{Laplace}(0, b_i), \qquad b_i = \Delta_{1,i}/\varepsilon,$$

where $\Delta_{1,i}$ denotes the $\ell_1$-sensitivity of the $i$-th statistic. For means and standard deviations, we conservatively bound $\Delta_{1,i} \leq \text{Range}(g_i)/v^{(k)}$. The resulting descriptor is

$$d^{(k)} = [g_1, \ldots, g_d]^\top + \eta, \quad \eta \sim \text{Laplace}(0, \text{diag}(b_1, \ldots, b_d)).$$

### C.1.2 Distribution Fidelity (R1).

To satisfy the condition in Equation 4, we design the descriptor extractor so that Euclidean distances in the descriptor space $\mathbb{R}^L$ closely approximate the 2-Wasserstein distance $W_2$ between client data distributions. This is achieved by globally aligning all clients using a consistent, privacy-preserving PCA, and constructing each descriptor from the first two moments of: (i) the marginal latent distribution $P(X)$, and (ii) the class-conditional latent distributions $\{P(X \mid Y = u)\}_{u=1}^U$.

**Algorithm 2:** FLUX Framework – Inference Phase

---

**Input:** set of test clients $\mathcal{Q} = \{1, \ldots, Q\}$, cluster-specific models $\{\theta_m\}_{m=1}^M$, cluster centroids $\{\gamma_m\}_{m=1}^M$, descriptor extractor $\phi(\cdot; \psi)$, distance function $\kappa(\cdot, \cdot)$.
**Output:** Predictions $\{\hat{y}^{(q)}\}_{q=1}^Q$

```
// Descriptor extraction
```
1 **for** *each client* $q \in \mathcal{Q}$ ***in parallel* do**
2     $d'^{(q)} \leftarrow \phi(x^{(q)}, \mathbf{0}; \psi)$;         `// Extract descriptor`
3     Send $d'^{(q)}$ to server;

```
// Unsupervised model assignment
```
4 **for** *each client* $q \in \mathcal{Q}$ ***in parallel* do**
5     $c^{*(q)} \leftarrow \arg\min_{m=1,\ldots,M} \kappa(d'^{(q)}, \gamma_m)$;         `// Model assignment`
6     Send $\theta_{c^{*(q)}}$ to client $q$;

```
// Local prediction
```
7 **for** *each client* $q \in \mathcal{Q}$ ***in parallel* do**
8     $\hat{y}^{(q)} \leftarrow f(x^{(q)}; \theta_{c^{*(q)}})$;         `// Inference`

---

We base our analysis on the following conditions:

(A1) (Gaussian latent approximation) Each latent distribution can be approximated by a Gaussian, i.e. $P_i \approx \mathcal{N}(\mu_i, \Sigma_i)$. This assumption is commonly adopted for deep feature representations.

(A2) (Spectral bounds) There exist constants $0 < \lambda_{\min} \le \lambda_{\max} < \infty$ such that all covariance eigenvalues remain in the interval $[\lambda_{\min}, \lambda_{\max}]$. In practice, this is enforced to Step S1 of the extractor, which restricts latent values to a shared bounding box $[m^-, m^+]$.

(A3) (Near-commutativity) After applying the shared PCA, covariances become nearly diagonal. This allows us to treat them as commuting matrices, which yields exact identities; otherwise, standard operator inequalities can be applied.

For clarity, we present the following proposition in the marginal setting; the class-conditional version follows by applying it to each class component.

**Proposition C.1** (Lipschitz-equivalence to $W_2$ for marginals). *Consider the distance between two client descriptors defined as*

$$\Delta^2 \; = \; \|\mu_1 - \mu_2\|_2^2 \; + \; \|\Sigma_1 - \Sigma_2\|_F^2. \tag{12}$$

*Under assumptions* (A1)–(A3)*, there exist constants*

$$c_- \; = \; \min\{1, (2\sqrt{\lambda_{\max}})^{-1}\}, \qquad c_+ \; = \; \max\{1, (2\sqrt{\lambda_{\min}})^{-1}\},$$

*such that the following inequality holds:*

$$c_-^2 \, \Delta^2 \; \le \; W_2^2\big(\mathcal{N}(\mu_1, \Sigma_1), \mathcal{N}(\mu_2, \Sigma_2)\big) \; \le \; c_+^2 \, \Delta^2.$$

*Hence, within the admissible covariance set, the squared 2-Wasserstein distance and the descriptor distance are equivalent up to constant factors.*

*Proof.* To streamline notation, let us denote each client's marginal descriptor by $d^{(k)} = [\mu_k, \text{vec}(\Sigma_k)]$. The squared Euclidean distance between two such descriptors is then

$$\|d^{(1)} - d^{(2)}\|_2^2 = \|\mu_1 - \mu_2\|_2^2 + \|\text{vec}(\Sigma_1 - \Sigma_2)\|_2^2.$$

Since for any matrix $A$ we have $\|\text{vec}(A)\|_2^2 = \|A\|_F^2$, taking $A = \Sigma_1 - \Sigma_2$ yields Equation 12:

$$\Delta^2 = \|d^{(1)} - d^{(2)}\|_2^2 = \|\mu_1 - \mu_2\|_2^2 + \|\Sigma_1 - \Sigma_2\|_F^2.$$

For Gaussian distributions, the squared 2-Wasserstein distance has the closed form $W_2^2 = \|\mu_1 - \mu_2\|_2^2 + B^2(\Sigma_1, \Sigma_2)$, where $B^2(\Sigma_1, \Sigma_2) = \text{Tr}\big(\Sigma_1 + \Sigma_2 - 2(\Sigma_1^{1/2}\Sigma_2\Sigma_1^{1/2})^{1/2}\big)$ is the squared Bures distance [78]. The mean terms in (12) and in $W_2^2$ coincide, so the comparison reduces to relating $B(\Sigma_1, \Sigma_2)$ with $\|\Sigma_1 - \Sigma_2\|_F$.

| Non-IID Level | Max | Min | Mean | Std |
|---|---|---|---|---|
| 3 | 1.060 | 0.194 | 0.526 | 0.106 |
| 5 | 1.028 | 0.159 | 0.479 | 0.113 |
| 7 | 1.016 | 0.136 | 0.458 | 0.108 |

(a) **MNIST**

| Non-IID Level | Max | Min | Mean | Std |
|---|---|---|---|---|
| 3 | 2.816 | 0.00017 | 1.328 | 0.522 |
| 5 | 3.464 | 0.00135 | 1.744 | 0.551 |
| 7 | 3.429 | 0.00441 | 1.734 | 0.590 |

(b) **CIFAR-10**

Table 4: Absolute error $\xi$ between descriptor distance (2-Wasserstein) and the true inter-client distance under three non-IID levels for MNIST and CIFAR-10 under $P(Y \mid X)$ concept shift.

Assumption (A3) ensures that the covariance matrices commute ($\Sigma_1\Sigma_2 = \Sigma_2\Sigma_1$), e.g., they are diagonal after the shared PCA. In this setting,

$$\Sigma_1^{1/2}\Sigma_2\Sigma_1^{1/2} = \Sigma_1^{1/2}\Sigma_2^{1/2}\Sigma_2^{1/2}\Sigma_1^{1/2} = \left(\Sigma_1^{1/2}\Sigma_2^{1/2}\right)^2 \quad \Rightarrow \quad \left(\Sigma_1^{1/2}\Sigma_2\Sigma_1^{1/2}\right)^{1/2} = \Sigma_1^{1/2}\Sigma_2^{1/2}.$$

Substituting into the Bures expression gives

$$B^2(\Sigma_1, \Sigma_2) = \mathrm{Tr}(\Sigma_1) + \mathrm{Tr}(\Sigma_2) - 2\,\mathrm{Tr}\left(\left(\Sigma_1^{1/2}\Sigma_2\Sigma_1^{1/2}\right)^{1/2}\right) = \mathrm{Tr}(\Sigma_1) + \mathrm{Tr}(\Sigma_2) - 2\,\mathrm{Tr}(\Sigma_1^{1/2}\Sigma_2^{1/2}).$$

But the right-hand side coincides with the expansion of the squared Frobenius norm

$$\|\Sigma_1^{1/2} - \Sigma_2^{1/2}\|_F^2 = \mathrm{Tr}(\Sigma_1) + \mathrm{Tr}(\Sigma_2) - 2\,\mathrm{Tr}(\Sigma_1^{1/2}\Sigma_2^{1/2}),$$

so we conclude that

$$B^2(\Sigma_1, \Sigma_2) = \|\Sigma_1^{1/2} - \Sigma_2^{1/2}\|_F^2.$$

Next, applying entrywise the mean value theorem for $f(x) = \sqrt{x}$ on $[\lambda_{\min}, \lambda_{\max}]$, yield the inequality

$$\frac{1}{2\sqrt{\lambda_{\max}}} \|\Sigma_1 - \Sigma_2\|_F \leq B(\Sigma_1, \Sigma_2) \leq \frac{1}{2\sqrt{\lambda_{\min}}} \|\Sigma_1 - \Sigma_2\|_F.$$

Finally, let $a = \|\mu_1 - \mu_2\|_2^2$ and $b = \|\Sigma_1 - \Sigma_2\|_F^2$, and write $k \in [k_{\min}, k_{\max}]$ with $k_{\min} = 1/(4\lambda_{\max})$ and $k_{\max} = 1/(4\lambda_{\min})$. Then the elementary inequality

$$\min\{1, k\}(a + b) \leq a + kb \leq \max\{1, k\}(a + b)$$

implies that $c_-^2\,\Delta^2 \leq W_2^2 \leq c_+^2\,\Delta^2$, with the constants $c_-$ and $c_+$ as stated.

$\square$

**Empirical validation.** We empirically validate this guarantee using 44,850 client–round pairs from MNIST and CIFAR-10 under three levels of non-IID concept shift affecting $P(Y \mid X)$ (see Appendix B.1 for protocol details). Table 4 reports the absolute deviation $\xi = |\|d^{(k_1)} - d^{(k_2)}\|_2 - W_2\big(P(x^{(k_1)}, y^{(k_1)}), P(x^{(k_2)}, y^{(k_2)})\big)|$. On MNIST, the worst-case error is below 1.1, with a mean of $0.49 \pm 0.11$. On CIFAR-10, the worst-case reaches 3.5, with a mean of $1.7 \pm 0.5$. These results confirm that the descriptor extractor satisfies Equation 4 across heterogeneous distribution shifts, while preserving client privacy.

### C.1.3 Label Agnosticism (R2)

In real-world deployments, test-time clients rarely possess reliable labels. To ensure applicability in this setting, Requirement (R2) mandates that each descriptor include a sub-vector computable using features only. Our implementation naturally satisfies this through the structure of the moments computed in Step *S3*.

- *Training phase.* Each client computes both the marginal moments $(\mu_x^{(k)}, \Sigma_x^{(k)})$ and class-conditional moments $\{(\mu_u^{(k)}, \Sigma_u^{(k)})\}_{u=1}^U$, resulting in a descriptor that splits as follows:

$$d^{(k)} = \big[\underbrace{\mu_x^{(k)}, \Sigma_x^{(k)}}_{d'^{(k)} \in \mathbb{R}^p}, \underbrace{\mu_1^{(k)}, \Sigma_1^{(k)}, \ldots, \mu_U^{(k)}, \Sigma_U^{(k)}}_{d''^{(k)} \in \mathbb{R}^{L-p}}\big],$$

where the sub-vector $d'^{(k)}$ encodes information from features only and serves as the label-free component.

- *Test phase (labels unavailable).* At test time, clients repeat Steps *S1–S2* and execute only the label-free portion of Step *S3*, computing $(\mu_x^{(k)}, \Sigma_x^{(k)})$ from features alone. Since $P(Y \mid X)$ cannot be estimated without labels, no class-conditional moments are computed. The resulting test-time descriptor is:

$$d_t'^{(k)} := \phi_\psi\big(x_t^{(k)}, \mathbf{0}\big) \in \mathbb{R}^p,$$

which satisfies the requirement for label-agnostic descriptor construction. When applying Step *S4*, the same coordinate-wise Laplace mechanism ensures that $d_t'^{(k)}$ enjoys the same $(\varepsilon, \delta=0)$-DP guarantee as the full descriptor.

Since the PCA projection is shared across clients (Step *S2*) and noise calibration in Step *S4* is data-independent, Euclidean distances between label-free descriptors—i.e., $\|d'^{(k_1)} - d'^{(k_2)}\|_2$—remain a valid proxy for the marginal Wasserstein distance between their respective $P(X)$ distributions. This enables our extractor to match unseen, unlabeled clients at test time to the most similar training distributions, thereby fully satisfying Requirement (R2).

### C.1.4 Compactness (R3)

Requirement (R3) stipulates that descriptor extraction should incur only marginal computational and communication overhead relative to standard FL. Our implementation satisfies this on both fronts.

- *Computation.* The only non-trivial operation is computing a rank-$l$ PCA on $s^{\text{PCA}} = 200$ synthetic latent vectors of dimension $z$ (Step *S2*). Using a standard SVD solver, the cost is $O\big(\min(s^{\text{PCA}} z^2, (s^{\text{PCA}})^2 z)\big)$. In practice, $z > s^{\text{PCA}}$, so the second term dominates, yielding roughly $4 \times 10^4 z$ floating-point operations. For typical latent sizes ($z \in [128, 2048]$), this results in at most $8.2 \times 10^7$ FLOPs—negligible compared to the cost of one local training epoch. For comparison, even a single forward pass with our smallest MNIST model exceeds $6.5 \times 10^8$ FLOPs, while training the largest model on CIFAR-100 requires over $6 \times 10^{11}$ FLOPs per epoch. Moment computation (Step *S3*) and noise addition (Step *S4*) are linear in descriptor size and thus negligible.

- *Communication.* Each descriptor transmits $d = 2(U + 1)l$ floats, corresponding to the mean and (diagonal) covariance for the marginal distribution and each of the $U$ classes. With $l = 10$, this amounts to $d = 220$ floats on MNIST ($U = 10$) and $d = 2020$ on CIFAR-100 ($U = 100$). In contrast, the corresponding model update sizes range from 62,006 parameters (MNIST) to 6,775,140 (CIFAR-100), yielding a descriptor-to-model ratio of at most $d/p \leq 3.5 \times 10^{-3}$. This remains well below the threshold of $10^{-2}$ imposed by the compactness requirement.

### C.2 Privacy Implications of Distribution Descriptors in FLUX

Because FLUX transmits client-side descriptors $\{d^{(k)}\}_{k \in \mathcal{K}} \subset \mathbb{R}^L$—which, by design, summarize each client's data distribution (Requirement (R1))—an adversary could, in principle, combine them with model updates to mount stronger reconstruction or membership-inference attacks [79–84]. Two properties mitigate this risk:

1. *Many-to-one mapping.* As defined in Section 4.2.1, the extractor $\phi \colon \mathbb{R}^{s^{(k)} \times (z+u)} \to \mathbb{R}^L$ performs a severe dimensionality reduction ($s^{(k)} \times (z + u) \gg L$) and aggregates client-level statistics. So from an information-theoretic perspective, infinitely many distinct datasets $\big(x^{(k)}, y^{(k)}\big)$ can yield the same descriptor $d^{(k)}$. Consequently, descriptors are non-invertible.

2. *Differential-privacy wrapper.* We can endow each descriptor with $(\varepsilon, \delta)$-differential privacy (DP) guarantee at the *sample*-level [27] by adding calibrated noise on the client-side. This ensure that the impact of any single sample remains bounded by the privacy budget $\epsilon$—without negatively affecting performance. Precisely, a mechanism $\mathcal{M}$ satisfies $(\epsilon, \delta)$-DP if, for any two neighbouring datasets $D$ and $D'$ differing by at most one entry, and for any measurable subset of outputs $S$, the following holds:

$$\Pr[\mathcal{M}(D) \in S] \leq e^\epsilon \Pr[\mathcal{M}(D') \in S] + \delta \tag{13}$$

where $\epsilon > 0$ is the privacy budget controlling the allowable difference between the probabilities, and $\delta \geq 0$ quantifies the probability of violating the bound.

**Differential privacy integration.** In our implementation, we perturb each coordinate of $d^{(k)}$ with independent Laplace noise ($\delta = 0$) [77]:

$$\eta_i \overset{\text{iid}}{\sim} \text{Laplace}(0, b_i), \qquad b_i = \Delta_{1,i}/\varepsilon,$$

where $\Delta_{1,i}$ is the $\ell_1$-sensitivity of statistic $g_i$. For a mean or standard-deviation coordinate we conservatively bound $\Delta_{1,i} \leq \frac{\text{Range}(g_i)}{s^{(k)}}$. The released descriptor is therefore

$$d^{(k)} = [\, g_1, \ldots, g_d \,]^\top + \eta, \quad \eta \sim \text{Laplace}(0, \text{diag}(b_1, \ldots, b_L)).$$

For example, setting a common $\varepsilon = 10.0$; with typical client sizes $v^{(k)} > 300$, the added variance $2(\Delta_{1,i}/\varepsilon)^2 \leq 2.2 \times 10^{-5}$ is negligible compared to natural data variability, so inter-descriptor distances remain reliable without affecting FLUX performance. Combined with the many-to-one nature of $\phi_\psi : \mathbb{R}^{s^{(k)} \times (z+u)} \to \mathbb{R}^L$—where infinitely many distinct datasets map to the same descriptor—this DP mechanism bounds any additional leakage to an $\varepsilon$-limited factor beyond what is already exposed by model parameters, thereby fully satisfying privacy guarantees.

| $\epsilon$ | Known Association | Test Phase |
|---|---|---|
| 0.01 | 90.06 ± 1.80 | 57.46 ± 13.26 |
| 0.1 | 92.19 ± 2.11 | 74.48 ± 12.25 |
| 1 | 91.80 ± 2.49 | 93.80 ± 1.42 |
| 10 | 92.49 ± 2.17 | 93.94 ± 1.40 |
| No DP | 92.86 ± 2.02 | 94.36 ± 1.33 |

Table 5: **Effect of Differential Privacy on FLUX based on varying $\epsilon$ values**. Results are presented with Known Association and Test Phase conditions, showing accuracy as mean ± standard deviation.

**Experimental validation.** As shown in Table 5, we evaluated FLUX under varying privacy budgets $\epsilon$ to assess the trade-off between privacy and performance. The results demonstrate that FLUX maintains robust performance across commonly used privacy budgets ($1 \leq \epsilon \leq 10$), achieving comparable accuracy to the non-DP version. Only under extremely stringent privacy settings (e.g., $\epsilon = 0.01$ and $\epsilon = 0.1$), which are uncommon in practical applications, does FLUX exhibit a noticeable decline in performance, particularly in the test phase. These results confirm that the proposed DP integration provides rigorous privacy guarantees while preserving the effectiveness of FLUX, making it suitable for deployment in scenarios requiring strict regulatory compliance.

### C.3 Advantages of Latent Space Descriptors in Neural Networks

We decided to extract data descriptors from the latent space of the trained model on the same dataset to leverage the inherent benefits of this representation. The latent space offers a reduced-dimensional representation, encapsulating only the most relevant features while eliminating redundant information. This dimensionality reduction enhances computational efficiency and facilitates downstream tasks without sacrificing the descriptive power of the features.

Furthermore, descriptors from the latent space reflect higher-level feature representations learned by the model. Neural networks are designed to prioritize task-relevant patterns, abstracting away low-level details in favor of semantically rich and domain-specific features. This makes latent space descriptors more informative and better aligned with the underlying classification task. Additionally, these representations are inherently resilient to noise and input variability, as the training process filters out irrelevant perturbations. This robustness ensures consistent and reliable descriptors, even under transformations or distortions in the input data. By aligning with the model's learned domain-specific knowledge, latent space descriptors also provide a task-informed perspective that enhances their interpretability and relevance. These advantages collectively establish the latent space as a superior choice for extracting meaningful and efficient data descriptors, especially in scenarios where robustness, efficiency, and task-specific alignment are critical.

### C.4 Identification of Distribution Shifts

Figure 1 offers an intuitive view of the four canonical distribution shifts introduced in Section 1. Below we explain how the statistics captured by our descriptor disentangle them.

- *Feature distribution shift ($P(X)$).* This shift arises when the marginal distribution of features varies across clients. In our case, it is detected via statistics computed in a reduced latent space $\varphi(x)$, whose geometry satisfies Requirement (R1) (see Appendix C.1 for implementation details and justifications). Shifts in $P(X)$ alter the global mean and covariance of these latent representations. Since our descriptor includes $(\mu_x^{(k)}, \Sigma_x^{(k)})$, clients with similar $P(X)$ naturally map to nearby points in the descriptor space (see Feature-space mean symbol in Figure 1).
- *Label distribution shift ($P(Y)$).* When label proportions differ across clients, the overall feature mean is skewed toward the predominant classes. These shifts are likewise reflected in $(\mu_x^{(k)}, \Sigma_x^{(k)})$, allowing label-imbalanced clients to cluster separately. For instance, in a client with few instances of label 1 and many instances of label 2, the mean of $\varphi(x^{(k)})$ will be predominantly influenced by label 2, resulting in proximity to the average representation of label 2 in the latent space. Figure 1 shows how these means in feature space reveal distinct client clusters.
- *Concept shift ($P(Y|X)$).* This shift arises when different clients assign different labels to identical inputs. Such discrepancies are invisible to feature-only statistics, as the latent representations may be indistinguishable. Figure 1 illustrates this with overlapping means between clients. To resolve this ambiguity, our descriptor includes class-conditional moments $\{(\mu_u^{(k)}, \Sigma_u^{(k)})\}_{u=1}^{U}$, which enable separation of clients with divergent conditional distributions.
- *Concept shift ($P(X|Y)$).* This shift occurs when inputs vary across clients for the same label—for example, the same digit rendered in different colors. In *ANDA* (Appendix B.1), MNIST digits may share the same label but differ in appearance between clients. These variations alter $(\mu_x^{(k)}, \Sigma_x^{(k)})$, and—except under rare conditions of perfect latent symmetry—can be detected without label information.

Because $P(X,Y) = P(Y \mid X)P(X)$, capturing both the marginal moments of $P(X)$ and the class-conditional moments of $P(Y \mid X)$ is sufficient to characterize *all* four shift types. At test time only the marginal component is available (labels are absent), so shifts that rely on $P(Y \mid X)$ cannot be detected online—a limitation inherent to any label-agnostic method, yet fully compatible with our clustering protocol.

# D   Clustering Mechanics

## D.1   Dynamic Clustering Initiation

In this study, we extract data descriptors from the initial trained global model in the federation. Due to the inherent high heterogeneity among client data, a global model trained using traditional FedAvg typically exhibits limited accuracy. To address this, the clustering process must be initiated early during training. However, the quality of descriptors improves with a more accurate global model, as additional training rounds refine its representation of the underlying data. To balance these competing factors and ensure the utility of FLUX in real-world applications, we dynamically determine the optimal time to initiate clustering during training.

To analyze the trade-offs involved, we analyze the impact of the training round at which clustering is initiated. Specifically, we tested FLUX by varying the clustering round from 1 to 9 on the MNIST, FMNIST, and CIFAR-10 datasets, with the non-IID levels fixed as follows: *Feature distribution shift* includes rotations ($0°$, $180°$) and three distinct colours; *Label distribution shift* restricts the number of classes to 3 with a bank size of 5; and $P(X|Y)$ *concept shift* applies augmentations limited to rotations ($0°$, $90°$, $180°$, $270°$) across 2 classes. As shown in Figure 8, initiating clustering too early results in inadequate representations, while delaying it excessively reduces the time available for cluster-specific models to converge. Based on this analysis, we establish that clustering should occur no earlier than three training rounds and no later than 80% of the total training rounds to allow sufficient convergence time for cluster-specific models.

To further optimize this process, we leverage the observation that the accuracy of the global model typically increases rapidly in early training rounds before plateauing with diminishing improvements. This behavior is evident in the derivative of performance metrics, such as accuracy, as shown in

Figure 9. Consequently, we dynamically initiate clustering when improvements in the global model's accuracy become negligible, formalized as:

$$C(r) = \begin{cases} 1 & \text{if } r \geq 3 \ \wedge \ \left(\min\left(\frac{dA}{dr}\big|_{3 \leq r' \leq r}\right) < T \ \vee \ r \geq 0.8R\right) \\ 0 & \text{otherwise} \end{cases} \tag{14}$$

Here, $A$ represents accuracy, $R$ is the total number of training rounds, and $T$ is a threshold determining negligible improvement. Through experimentation, we identified $T = 0.06$ as an effective threshold, as varying $T$ did not lead to significant differences in performance, as shown in Figure 10.

This dynamic approach ensures that clustering is initiated at a point where the global model is sufficiently accurate to extract meaningful descriptors while leaving ample time for the convergence of cluster-specific models.

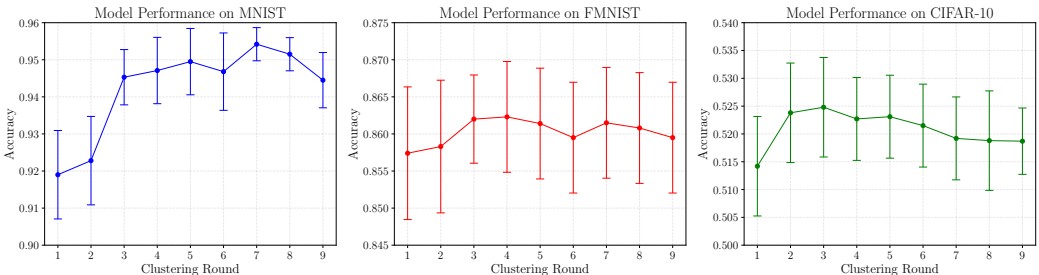

Figure 8: **Impact of clustering initialization round on model performance.** Mean accuracy and standard error are shown across $P(X)$, $P(Y)$, and $P(X|Y)$ shifts for MNIST, FMNIST, and CIFAR-10 datasets. Initiating clustering too early results in poor representations, while initiating too late limits convergence time for cluster-specific models.

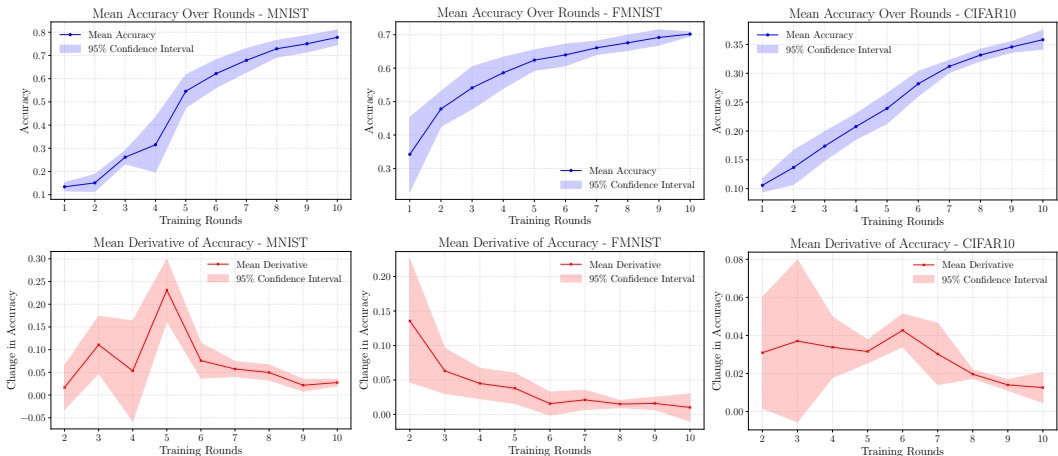

Figure 9: **Accuracy trends and their derivative during FedAvg training.** The figure shows the mean values and 95% confidence intervals across $P(X)$, $P(X|Y)$, and $P(Y)$ shifts for MNIST, FMNIST, and CIFAR-10 datasets. The derivative highlights the diminishing accuracy improvements after the initial training rounds.

### D.2 Unsupervised Clustering Setup

For the unsupervised clustering $\mathcal{U}$ in FLUX, we design an adaptive, density-based method that extends DBSCAN by automatically estimating $\epsilon$ from the data's density distribution and robustly handling outlier clients. To compute the radius parameter $\epsilon$ in DBSCAN, we first calculate the distances to the second-nearest neighbors for all data points. These distances are sorted in ascending order, forming a curve that reflects the density distribution of the data. To determine the optimal $\epsilon$, we identify the "elbow point" on this curve which marks the transition from dense to sparse regions in the data,

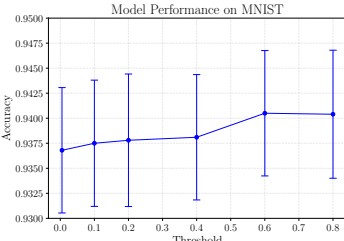 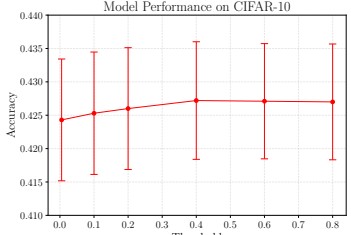

Figure 10: **Sensitivity analysis of threshold $T$ for clustering initiation.** The plot shows mean accuracy and standard error across $P(X)$, $P(Y)$, and $P(X|Y)$ shifts for MNIST and CIFAR-10 datasets. Results demonstrate that varying $T$ does not significantly affect performance.

providing a natural boundary for clustering[¶]. To further refine the $\epsilon$ value and adjust the sensitivity of the clustering, we scale the elbow point distance by a scaling factor. This scaling ensures that the resulting $\epsilon$ is appropriately calibrated to the characteristics of the dataset. In our experiments, we set the parameter $\epsilon$ to 1.0 (indicating no scaling) for the MNIST, CIFAR-10, CheXpert, and Office-Home datasets. For the FMNIST and CIFAR-100, $\epsilon$ was slightly reduced to 0.95 and 0.98, as we observed a lower variability in the latent representation. Using the dynamically computed $\epsilon$, DBSCAN is executed with a minimum sample threshold of 2, allowing the identification of clusters with as few as two points. Noise points, which are not part of any cluster, are reassigned to individual clusters to ensure that all clients are represented in the analysis. This approach preserves the integrity of the clustering while accommodating the unique characteristics of all clients' data.

For experiments with FLUX-prior, we use the K-means clustering algorithm, leveraging prior knowledge of the number of distributions $M$, which is used as the number of clusters $K$.

### D.3 Empirical Illustration of Clustering

To illustrate the clustering process in FLUX, we provide examples on two datasets (MNIST and CIFAR-100) under different levels of heterogeneity. Figure 11 visualizes client descriptors projected into a 2D space via PCA (for illustration only) across three heterogeneity levels ($M = 3, 4, 5$). In all cases, clients with similar data distributions cluster together in descriptor space, reflecting the latent distributional structure of the datasets and facilitating the grouping process. The figure also shows, with different colors, the cluster assignments produced by our unsupervised algorithm.

It is important to note that clustering is performed in the full descriptor space, not in two dimensions. This explains why, for instance, in CIFAR-100 with $M = 5$, clusters 1, 2, and 3 are not visually separable in the PCA projection, although clients are correctly assigned in the higher-dimensional space. Overall, these examples confirm that the descriptors capture essential distributional properties, enabling FLUX to form accurate clusters even under substantial heterogeneity.

**Extension to sparse soft assignments.** To demonstrate the flexibility of FLUX descriptors beyond hard clustering, we evaluate a sparse soft variant (FLUX-soft). In this setting, hard clustering is replaced with a soft-weighted aggregation scheme: each client receives a distribution-specific update by weighting contributions from other clients according to descriptor similarity with its own distribution. Table 6 reports averaged results on MNIST, covering all four types of distribution shifts and three levels of heterogeneity for each. The results show that FLUX-soft substantially outperforms all baselines—improving over the best-performing baseline (CFL) by nearly 11 percentage points—while approaching the accuracy of hard FLUX. As expected, in our experimental setting hard clustering (FLUX) excels, since the data exhibit clear distributional boundaries; however,

| Method | Accuracy (%) |
|---|---|
| FedAvg | $71.9 \pm 6.0$ |
| IFCA | $35.2 \pm 6.2$ |
| FedRC | $75.2 \pm 4.3$ |
| FedEM | $75.0 \pm 4.2$ |
| FedSEM | $66.0 \pm 6.0$ |
| FedDrift | $54.3 \pm 7.7$ |
| CFL | $75.5 \pm 4.0$ |
| pFedMe | $55.2 \pm 6.9$ |
| APFL | $72.8 \pm 5.2$ |
| ATP | $74.6 \pm 8.0$ |
| FLUX-soft | $\mathbf{86.5 \pm 1.3}$ |
| FLUX | $\mathbf{89.5 \pm 4.9}$ |

Table 6: Average accuracy on MNIST across shifts and heterogeneity levels.

---

[¶]https://github.com/arvkevi/kneed

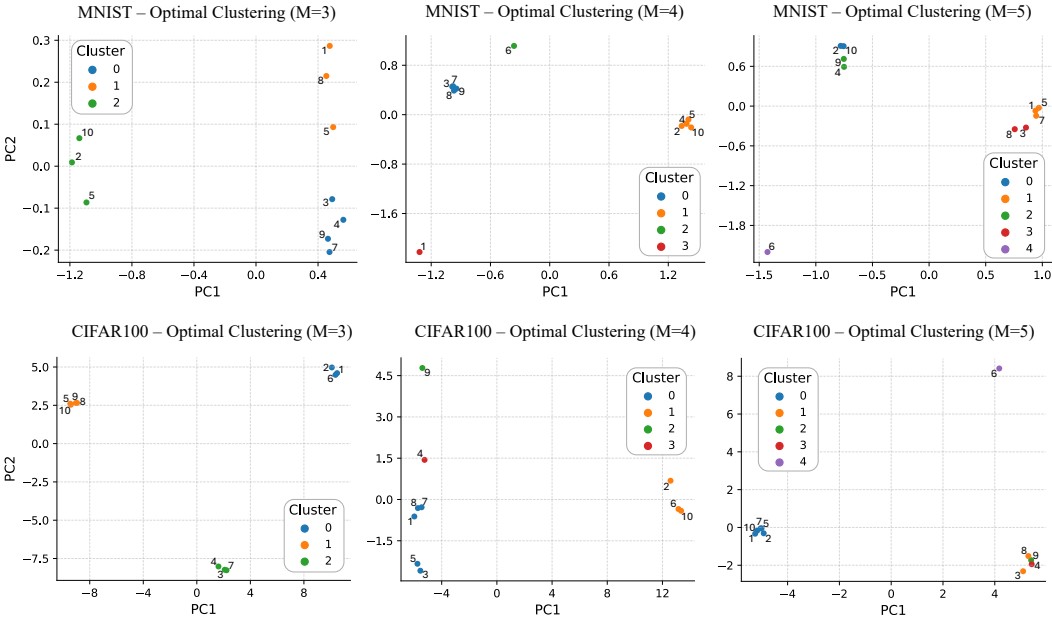

Figure 11: Visualization of client descriptors projected into 2D space for MNIST and CIFAR-100 under different numbers of underlying clusters ($M = 3, 4, 5$). Clients with similar data distributions are mapped close to each other, confirming the effectiveness of FLUX in capturing latent heterogeneity.

FLUX-soft may be advantageous in scenarios with overlapping or noisy distributions that require a more client-centric approach.

**Extension to dynamic scenarios.** Thanks to its modular design, FLUX can be seamlessly extended to non-stationary settings by periodically updating descriptors and reapplying the clustering process. In this dynamic variant, re-clustering is triggered whenever a distributional drift is detected or when clients join or leave the federation. To avoid retraining models from scratch after each re-clustering, each newly formed cluster is initialized with the model from the previously existing cluster whose centroid is closest in descriptor space. This association is obtained by matching the current cluster centroids to the most similar ones from the prior clustering phase. Such a warm-starting strategy enables efficient adaptation to changing distributions while preserving previously learned knowledge.

We validate this approach on MNIST under a dynamic setting with two distribution drifts during training and one during testing (Table 6), while all other settings match those in Table 1. As expected, FLUX achieves a substantially larger improvement over all baselines, since none of the existing methods are explicitly designed to handle non-stationary scenarios (except for FLUX-soft). These results demonstrate that FLUX remains highly effective under dynamic conditions, outperforming all baselines by over 14 percentage points compared to the best-performing method, thus showing strong resilience and robustness in realistic FL deployments.

# E  Baselines & Comparative Complexity

## E.1  Overview of Baseline Algorithms

In our experiments, we benchmarked nine recent baselines designed to address distribution shifts during training or test-time. We implemented six CFL-baed baselines, as our method falls in this category, that employ diverse clustering principles in CFL, spanning both hard-CFL approaches [15, 18, 17, 14], soft-CFL approaches [11, 16], metric-based clustering [15, 17, 16] and parameter-based clustering [18, 14, 11]. Additionally we adoted two recent state-of-the-art methods in PFL, and one TTA-FL method.

- *IFCA [15]:* Assuming the knowledge of the number of clusters, the server initializes and broadcasts $M$ models to all clients. Each client evaluates the models locally to identify the one

minimizing its empirical loss. Clients then train the selected model, and the server aggregates updated models from clients assigned to the same cluster.

- *FedRC [11]:* A soft-CFL method that assumes prior knowledge of the number of clusters $M$. The server broadcasts $M$ models to each client $k$, which first optimizes a weight vector $\pi^{(k)} \in \mathbb{R}^M$, determining the contribution of each cluster model, before refining all $M$ models. This iterative process alternates between optimizing the client-specific weights and the cluster models.

- *FedEM [16]:* FedEM assumes the number of distributions $M$ is known. It initializes $M$ cluster models, broadcasts them to all clients, and allows clients to learn a weighting vector $\pi^{(k)} \in \mathbb{R}^M$ to determine the relevance of each model for their data. Using alternating optimization, clients jointly update their weight vectors and the cluster models.

- *FeSEM [18]:* This parameter-based method assumes $M$ clusters are known and initializes $M$ cluster models randomly. After local training, clients are assigned to the cluster with the closest model parameters. The server aggregates updated models for each cluster and distributes only the assigned cluster model back to the corresponding clients.

- *FedDrift [17]:* A dynamic clustering framework that begins with one model per client. Based on loss similarities, client models are iteratively merged into clusters. The server continuously evaluates all models on all clients, assigning each to the closest model. New cluster models are created when a significant increase in client loss is detected, ensuring adaptive clustering throughout training.

- *CFL [14]:* This gradient-based hard-CFL method clusters clients based on the cosine similarity of their gradients. It operates under the assumption that at convergence, client gradients should approach zero. Clients with non-zero gradients beyond a threshold are split into separate clusters to reduce gradient differences. The process repeats iteratively until no further clusters are formed.

- *pFedMe [12]:* A personalized FL method based on Moreau envelopes that formulates training as a bilevel optimization problem. Each client solves the local subproblem $\min_{\phi^{(k)}} \mathcal{L}^{(k)}(\phi^{(k)}) + \frac{\lambda}{2}\|\phi^{(k)} - \theta\|^2$ via $C$ gradient steps, then updates its copy of the global model through a proximal step before sending it to the server. This decoupling enables adaptive personalization with controllable computation and strong convergence guarantees.

- *APFL [13]:* APFL augments each client with a global copy $\theta$ and a local model $\phi^{(k)}$, combined into a personalized model $\bar{\phi}^{(k)} = \alpha_k\,\phi^{(k)} + (1 - \alpha_k)\,\theta$. During each round, clients update both $\theta$ and $\phi^{(k)}$ via local SGD, adaptively tuning $\alpha_k$ to balance shared and local knowledge. Only $\theta$ is sent to the server for aggregation, while $\phi^{(k)}$ and $\bar{\phi}^{(k)}$ remain local. This design enables effective personalization under non-IID data with minimal additional overhead.

- *ATP [20]:* ATP learns a vector of module-wise adaptation rates $\alpha = \{\alpha^{[l]}\}_{l=1}^A$, where $A$ is the number of modules, by alternating unsupervised entropy minimization and supervised refinement on labeled source clients, then aggregates $\alpha$ via FedAvg. At test time, each unlabeled client downloads the global model $\theta$ and the full vector $\alpha$, performs unsupervised entropy-based adaptation on its batch (or online stream) using $\alpha$, and then predicts with the adapted model. Only $\theta$ and the low-dimensional $\alpha$ are communicated, enabling efficient personalization under distribution shifts.

### E.2 Comparative Analysis of Heterogeneity and Computational Costs

This section provides a comparative overview of the types of data heterogeneity addressed by the evaluated baselines, followed by a theoretical analysis of the communication and computational costs of FLUX and all implemented baselines.

**Supported types of heterogeneity.** Table 7 outlines the types of heterogeneity each method claims to support—or has been empirically evaluated on real dataset—including feature distribution shift $P(X)$, the label distribution shift $P(Y)$, or the concept distribution shift $P(X|Y)$ and $P(Y|X)$. The results indicate that most of the baselines were focused on solving only a single form of heterogeneity (e.g., IFCA, FeSEM, FedEM, FedDrift, pFedMe), limiting their applicability in real-world scenarios where the exact nature of heterogeneity is commonly unknown a priori. Notably, FedRC adopted feature distribution shifts $P(X)$, label distribution shifts $P(Y)$, and concept shift $P(Y|X)$, which closely aligns with the assumptions in this work.

| ALGORITHM | SUPPORTED TYPES OF HETEROGENEITY | COMMUNICATION COST | COMPUTATIONAL COST (SERVER) | COMPUTATIONAL COST (CLIENT) |
|---|---|---|---|---|
| FEDAVG | N/A | $pN_{\text{byte}}$ | $O(Kp)$ | $O(s^{(k)}F^{\text{train}})$ |
| IFCA | $P(X)$ | $MpN_{\text{byte}}$ | $O(Kp)$ | $O(s^{(k)}F^{\text{train}}+Ms^{(k)}F^{\text{inf}})$ |
| FESEM | NATURAL | $pN_{\text{byte}}$ | $O(MKp)$ | $O(s^{(k)}F^{\text{train}})$ |
| FEDEM | $P(Y)$ | $MpN_{\text{byte}}$ | $O(MKp)$ | $O(Ms^{(k)}F^{\text{train}}+Ms^{(k)}F^{\text{inf}})$ |
| FEDRC | $P(X),P(Y),P(Y|X)$ | $MpN_{\text{byte}}$ | $O(MKp)$ | $O(Ms^{(k)}F^{\text{train}}+Ms^{(k)}F^{\text{inf}})$ |
| FEDDRIFT | $P(Y|X)$ | $MpN_{\text{byte}}$ | $O(Kp)$ | $O(s^{(k)}F^{\text{train}}+Ms^{(k)}F^{\text{inf}})$ |
| CFL | $P(Y|X)$ | $pN_{\text{byte}}$ | $O(K^2p)$ | $O(s^{(k)}F^{\text{train}})$ |
| pFEDME | $P(Y)$ | $pN_{\text{byte}}$ | $O(Kp)$ | $O((C+1)s^{(k)}F^{\text{train}})$ |
| APFL | $P(X),P(Y)$ | $pN_{\text{byte}}$ | $O(Kp)$ | $O(2s^{(k)}F^{\text{train}})$ |
| ATP | $P(X),P(Y)$ | $(p+A)N_{\text{byte}}$ | $O(K(p+A))$ | $O(2s^{(k)}F^{\text{train}})$ |
| **FLUX** | $P(X),P(Y),P(X|Y),P(Y|X)$ | $(p+L)N_{\text{byte}}$ | $O(Kp)+O(\mathcal{U})$ | $O(s^{(k)}F^{\text{train}})+O(\xi_{v\to l})$ |

Table 7: **Extended comparison table for heterogeneity and cost among FLUX and FL baselines.** $P(X)$: Feature distribution shift. $P(Y)$: Label distribution shift. $P(X|Y)$: Concept shift (same features, different label). $P(Y|X)$: Concept shift (same label, different features). $N_{\text{byte}}$: Number of bytes per parameters. $M$: Number of clusters of the current round. $K$: Number of clients. $L$: Length of the descriptors. $C$: Number of gradient steps for local subproblem in pFedMe. $A$: Number of modules in ATP. $p$: Number of model parameters. $v$: Latent representation dimension ($f_e : \mathbb{R}^z \to \mathbb{R}^v$). $s^{(k)}$: Number of sample for client $k$. $F^{\text{train}}$: Computational cost forward pass and back-propagation. $F^{\text{inf}}$: Computational cost forward pass. $\mathcal{U}$: Clustering algorithm (Equation 8). $\xi_{v\to l}$: Dimensionality reduction (Equation 6).

**Computational costs.** Table 7 presents a theoretical analysis of the communication and computational overhead introduced by FLUX, in comparison to standard FL methods and the implemented baselines. We focus on the primary sources of system overhead in a single FL round, i.e., *communication cost per client* (in bytes) and the *computational costs* on both the server and the clients.

- *Communication cost:* FLUX introduces minimal additional overhead relative to FedAvg. The only modification during communication is the transmission of a client descriptor of length $L$, which is negligible compared to the model update size $p$. As shown in Section C.1.4, our implementation satisfies Requirement (R3) with a descriptor-to-model ratio of $L/p \leq 3.5 \times 10^{-3}$.

- *Computational cost (server):* the unsupervised clustering procedure (Appendix D.2) operates on client descriptors and introduces a computational complexity of $O(L \log L)$, which remains negligible compared to the $O(Kp)$ cost of aggregating model updates from $K$ clients.

- *Computational cost (client):* the descriptor extraction includes a dimensionality reduction step $\xi_{v\to l}$ (Equation 6) with complexity $O(s^2 v)$ (see Section C.1.4 for full justification), where $v$ is the latent dimension and $s$ is the number of local samples. For typical values of $v \in [128, 2048]$, this results in a computational cost of at most $8.2 \times 10^7$ FLOPs—negligible when compared to the cost of local training. For reference, even a single forward pass (no backward or optimization) using our smallest model on MNIST exceeds $6.5 \times 10^8$ FLOPs, while a full epoch with our largest model on CIFAR-100 exceeds $6 \times 10^{11}$ FLOPs.

In contrast, several baselines introduce substantial overhead. CFL methods such as IFCA, FedEM, FedRC, and FedDrift require each client to evaluate and/or train all $M$ cluster models locally, significantly increasing both communication ($\times M$) and computation costs. PFL and TTA-FL methods such as pFedMe, APFL, and ATP involve additional optimization steps—at minimum doubling the client-side compute. These overheads may render these methods impractical in cross-device FL scenarios. While methods like CFL and FeSEM avoid extra communication or client-side computation, they shift the burden to the server. These methods demand increased server-side computations, which can become a bottleneck in scenarios with large models or a substantial number of clients, potentially hindering scalability and performance.

Overall, this comparison highlights the efficiency of FLUX and its practical suitability for large-scale, cross-device FL deployments.

### E.3 Comparison with Descriptor-based Methods

We further compare FLUX with HACCS [66], the closest related CFL method that also employs descriptor-like representations. For fairness, HACCS is evaluated without differential privacy, even though its design requires it. Table 8 reports results on MNIST, FMNIST, and CIFAR-10 under both the *known association* and *test phase* conditions for FedAvg, FedDrift, HACCS, and FLUX, while the other baselines are reported in Tables 12–39.

The results highlight three key insights. First, HACCS matches FLUX only under the *known association* condition for label distribution shifts ($P(Y)$), where simple label histograms suffice and no semantic modelling is required. In these cases, even loss-based methods such as FedDrift achieve competitive results. However, HACCS's reliance on label and pixel-value histograms prevents it from capturing semantic and structural properties of the data: pixel statistics cannot discriminate subtle but task-relevant variations in the input distributions. As a result, HACCS underperforms FLUX by a wide margin on $P(X)$, $P(Y|X)$, and $P(X|Y)$ shifts. Second, HACCS requires label information at inference time to build histograms and assign clusters. This prevents test-time adaptation to unseen clients and explains the sharp performance drop in the *test phase*, where HACCS falls below FedAvg on CIFAR-10 and FMNIST, and far behind both FedDrift and FLUX. By contrast, FLUX produces label-agnostic descriptors at inference, enabling robust cluster assignments even in the absence of labels. Third, unlike HACCS, which generates large and interpretable histograms whose size grows with the number of labels and bins, FLUX uses compact, fixed-size, non-interpretable descriptors. This design ensures scalability and communication efficiency, while reducing privacy leakage risk since descriptors cannot directly reveal label distributions. Overall, these results confirm that while HACCS is the closest related work exploring descriptor-based clustering, its practical performance is poor—even compared to existing baselines such as FedDrift. In contrast, FLUX consistently provides a scalable, robust, privacy-preserving, and generalizable solution for clustered FL.

| Dataset | Method | P(X) | P(Y) | P(Y\|X) | P(X\|Y) |
|---|---|---|---|---|---|
| MNIST (Known A.) | FedAvg | 77.8 ± 2.1 | 91.9 ± 1.4 | 66.8 ± 1.4 | 87.0 ± 2.0 |
| | FedDrift | 94.1 ± 1.8 | 96.4 ± 0.4 | 81.1 ± 4.2 | **92.7 ± 1.7** |
| | HACCS | 77.5 ± 5.2 | 96.2 ± 0.2 | 79.0 ± 4.5 | 88.4 ± 1.4 |
| | FLUX | **95.5 ± 0.5** | **96.8 ± 0.4** | **85.1 ± 4.0** | 92.4 ± 2.4 |
| MNIST (Test Phase) | FedAvg | 77.8 ± 2.1 | 91.9 ± 1.4 | N/A | 87.0 ± 2.0 |
| | FedDrift | 47.7 ± 3.9 | 71.8 ± 3.0 | N/A | 75.9 ± 3.1 |
| | HACCS | 59.7 ± 9.5 | 69.3 ± 6.1 | N/A | 77.0 ± 4.8 |
| | FLUX | **95.0 ± 1.5** | **96.1 ± 1.2** | N/A | **90.8 ± 2.4** |
| FMNIST (Known A.) | FedAvg | 61.6 ± 2.4 | 72.8 ± 2.4 | 55.4 ± 2.3 | 72.0 ± 1.8 |
| | FedDrift | **79.9 ± 0.7** | **86.0 ± 1.8** | **74.5 ± 2.7** | **82.5 ± 1.6** |
| | HACCS | 68.5 ± 6.1 | 85.2 ± 2.0 | 66.0 ± 3.0 | 77.8 ± 3.2 |
| | FLUX | 77.6 ± 1.8 | 85.9 ± 1.9 | 72.2 ± 3.1 | 81.9 ± 2.3 |
| FMNIST (Test Phase) | FedAvg | 61.6 ± 2.4 | 72.8 ± 2.4 | N/A | 72.0 ± 1.8 |
| | FedDrift | 30.0 ± 1.5 | 61.4 ± 3.1 | N/A | 61.9 ± 2.3 |
| | HACCS | 41.6 ± 6.2 | 62.5 ± 3.5 | N/A | 60.8 ± 2.8 |
| | FLUX | **77.0 ± 2.2** | **85.7 ± 1.8** | N/A | **81.0 ± 2.8** |
| CIFAR-10 (Known A.) | FedAvg | 22.1 ± 1.2 | 37.4 ± 2.9 | 28.0 ± 1.0 | 36.0 ± 1.1 |
| | FedDrift | 25.2 ± 1.9 | 49.0 ± 3.4 | 29.0 ± 1.0 | 37.0 ± 1.1 |
| | HACCS | 23.2 ± 1.6 | 45.4 ± 1.3 | 24.8 ± 1.3 | 33.4 ± 2.0 |
| | FLUX | **33.3 ± 0.9** | **50.4 ± 2.9** | **31.7 ± 1.3** | **39.1 ± 1.8** |
| CIFAR-10 (Test Phase) | FedAvg | 22.1 ± 1.2 | 37.4 ± 2.9 | N/A | 36.0 ± 1.1 |
| | FedDrift | 24.0 ± 1.3 | 35.4 ± 2.8 | N/A | **36.8 ± 1.3** |
| | HACCS | 18.9 ± 1.5 | 32.3 ± 3.1 | N/A | 30.1 ± 1.1 |
| | FLUX | **33.3 ± 1.0** | **46.2 ± 4.0** | N/A | 36.7 ± 2.1 |

Table 8: **Per-shift comparison of FLUX with HACCS, FedAvg, and FedDrift on MNIST, CIFAR-10, and FMNIST.** Results are shown for each type of distribution shift ($P(X)$, $P(Y)$, $P(Y|X)$, $P(X|Y)$) under both *known association* and *test phase* conditions.

# F  Additional Experiment Results

## F.1  Scaling Data Heterogeneity Types and Levels

We evaluate the effects of data heterogeneity using four non-IID dataset types across 10 clients, generated with *ANDA* at eight distinct levels of heterogeneity. The detailed configurations for each type of data distribution shift are summarized in Table 9 and elaborated below:

- **Feature distribution shift** ($P(X)$)**:** Datasets are augmented with rotations and color transformations. Each client is assigned a specific augmentation pattern applied to all local data points (images). The non-IID level determines the number of augmentation choices per client. For instance, at Level 5, a client selects one rotation from $\{0°, 180°\}$ and one color from {Red, Blue, Green} for all images. At Levels 1 to 4, the "Original" color indicates no color transformation, preserving the original image channels.
- **Label distribution shift** ($P(Y)$)**:** Datasets retain data points only for specific classes. For example, at Level 8, each client keeps data points from three classes while discarding the rest. Starting from Level 1, with 10 total classes (e.g., MNIST, CIFAR-10, FMNIST), we define a class selection bank of size 5 (e.g., {[0,2,4], [1,3,9], [3,4,5], [5,6,7], [6,8,9]} at Level 8) to limit the clustering complexity. Clients select subsets from this bank to decide which classes to keep.
- **Concept shift** ($P(Y|X)$)**:** Datasets swap labels among specific classes. A swapping pool is constructed by selecting a subset of classes, which are permuted to assign new labels. For instance, at Level 4, the pool {2,3,5,8} might be permuted to {5,8,3,2}, mapping images originally labeled as '2' to the new label '5'. The size of the swapping pool increases with the heterogeneity level, and each client permutes the classes independently and randomly.
- **Concept shift** ($P(X|Y)$)**:** Clients apply distinct augmentations to data points of the same class. For example, Client A applies a $0°$ rotation to class '5', while Client B applies a $180°$ rotation. Augmentation options are limited to rotations ($\{0°, 90°, 180°, 270°\}$), and the number of classes subjected to augmentation increases with the heterogeneity level.

| non-IID type | $P(X)$ | $P(Y)$ | $P(Y|X)$ | $P(X|Y)$ |
|---|---|---|---|---|
| Level 1 | Rotation $\{0°, 180°\}$, Color {Original} | #Class = 10 | #Swapped Class = 1 | #Class Under Augmentation = 1 |
| Level 2 | Rotation $\{0°, 120°, 240°\}$, Color {Original} | #Class = 9 | #Swapped Class = 2 | #Class Under Augmentation = 2 |
| Level 3 | Rotation $\{0°, 90°, 180°, 270°\}$, Color {Original} | #Class = 8 | #Swapped Class = 3 | #Class Under Augmentation = 3 |
| Level 4 | Rotation $\{0°, 72°, 144°, 216°, 288°\}$, Color {Original} | #Class = 7 | #Swapped Class = 4 | #Class Under Augmentation = 4 |
| Level 5 | Rotation $\{0°, 180°\}$, Color {Red, Blue, Green} | #Class = 6 | #Swapped Class = 5 | #Class Under Augmentation = 5 |
| Level 6 | Rotation $\{0°, 120°, 240°\}$, Color {Red, Blue, Green} | #Class = 5 | #Swapped Class = 6 | #Class Under Augmentation = 6 |
| Level 7 | Rotation $\{0°, 90°, 180°, 270°\}$, Color {Red, Blue, Green} | #Class = 4 | #Swapped Class = 7 | #Class Under Augmentation = 7 |
| Level 8 | Rotation $\{0°, 72°, 144°, 216°, 288°\}$, Color {Red, Blue, Green} | #Class = 3 | #Swapped Class = 8 | #Class Under Augmentation = 8 |

Table 9: **Summary of non-IID data heterogeneity configurations across levels. Each type of heterogeneity corresponds to a specific distribution shift.**

Tables 10 to Table 49 present detailed results of FLUX and all baseline methods across various types and levels of distribution shifts. We did not evaluate the performance of *test phase* on $P(Y|X)$ concept shift, as this problem is unsolvable without access to labels (refer to the Section 4.2.2 and Appendix C.4 for more details). For the baselines that do not provide a solution for real *test phase*, we weight all models by the number of clients in the cluster, and use the expectation that weights all model outputs as an estimation of the predicted labels.

We also provide Figure 12, which summarizes of the averaged accuracy across the distribution shifts (i.e., across tables), showing trends under both the *known association* condition and the real *test phase* condition. It shows that FLUX consistently outperforms baselines during real test phase, particularly at high heterogeneity, which demand accurate identification and clustering of client data distributions to avoid model degradation. The only cases where baselines surpass FLUX are PFL methods on CIFAR datasets. This is expected in scenarios involving $P(Y|X)$ shifts (e.g., Tables 46–47), where each client may have different label preferences for the same input. In such cases, collaboration can be harmful, and methods fine-tuned on the same client distributions seen at test time (as in PFL) naturally gain an advantage. However, these methods fail to generalize to unseen clients (see Table 1 and Figure 12, Test Phase), limiting their applicability in realistic deployments. On CIFAR-10, the relatively shallow backbone can also constrain descriptor quality, highlighting that limited model

expressiveness may degrade clustering and final accuracy; indeed, the gap with PFL methods narrows when using deeper models (e.g., ResNet9 instead of LeNet5) on CIFAR-100. Notably, FLUX achieves its maximum improvement on FMNIST, surpassing the best baselines by over 23 pp. Furthermore, FLUX achieves performance comparable to FLUX-prior where the prior knowledge of the number of clusters ($M$) is known. This result highlights the challenge of identifying clusters in real-world settings and validates the coherence of our unsupervised clustering approach. It is worth noting that most baselines rely on this cluster information (see Table 2), significantly simplifying their clustering tasks. Even under the known association setting, FLUX remains the most robust method across all heterogeneity levels and datasets, demonstrating its utility and generalizability.

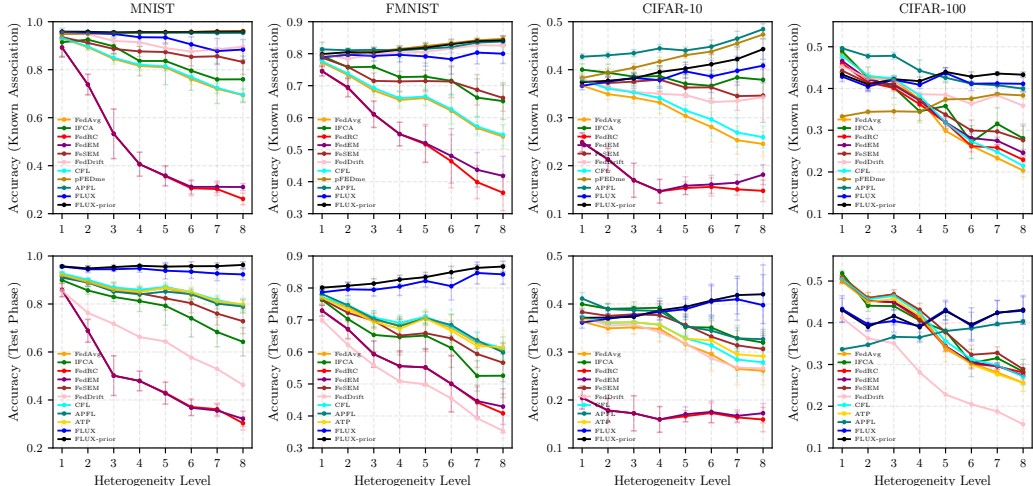

Figure 12: **Mean accuracy and standard deviation across heterogeneity levels for MNIST, FMNIST, CIFAR-10, and CIFAR-100, averaged over P(X), P(Y), P(Y|X), and P(X|Y) shifts.** First row: *known association* condition, where test-time cluster associations are available. Second row: *test phase* condition, where cluster associations are inferred.

| non-IID Level | Level 1 | | Level 2 | | Level 3 | | Level 4 | |
|---|---|---|---|---|---|---|---|---|
| **Algorithm** | **Known A.** | **Test Phase** | **Known A.** | **Test Phase** | **Known A.** | **Test Phase** | **Known A.** | **Test Phase** |
| FedAvg | 93.07 ± 0.50 | 92.15 ± 0.58 | 89.38 ± 1.42 | 89.80 ± 0.88 | 84.49 ± 2.43 | 86.31 ± 2.77 | 81.56 ± 2.22 | 85.32 ± 2.17 |
| IFCA | 91.45 ± 5.81 | 89.93 ± 3.07 | 92.52 ± 2.22 | 85.65 ± 1.55 | 89.74 ± 3.27 | 82.94 ± 1.95 | 83.63 ± 6.09 | 81.22 ± 4.36 |
| FedRC | 89.27 ± 4.03 | 85.83 ± 2.77 | 73.82 ± 4.29 | 68.84 ± 4.78 | 53.28 ± 10.31 | 50.14 ± 8.28 | 40.64 ± 4.99 | 47.95 ± 4.08 |
| FedEM | 89.24 ± 3.98 | 85.80 ± 2.78 | 73.81 ± 4.28 | 68.83 ± 4.74 | 53.27 ± 10.32 | 50.13 ± 8.28 | 40.61 ± 5.01 | 47.92 ± 4.09 |
| FeSEM | 93.58 ± 1.31 | 91.69 ± 1.76 | 91.19 ± 1.83 | 88.99 ± 1.60 | 88.67 ± 2.00 | 85.63 ± 3.09 | 87.62 ± 2.12 | 84.46 ± 2.37 |
| FedDrift | 95.75 ± 0.44 | 85.50 ± 1.88 | 94.77 ± 1.27 | 76.28 ± 3.01 | 91.99 ± 2.97 | 71.72 ± 3.46 | 91.48 ± 3.79 | 66.23 ± 3.41 |
| CFL | 93.22 ± 0.43 | 92.74 ± 1.13 | 89.68 ± 1.40 | 90.10 ± 0.77 | 84.95 ± 2.13 | 86.91 ± 2.42 | 82.09 ± 1.96 | 85.95 ± 1.92 |
| pFedMe | 94.84 ± 0.22 | N/A | 94.97 ± 0.28 | N/A | 95.24 ± 0.31 | N/A | 95.41 ± 0.30 | N/A |
| APFL | 95.72 ± 0.15 | 91.00 ± 0.96 | 95.56 ± 0.24 | 88.91 ± 1.48 | 95.42 ± 0.29 | 85.20 ± 2.16 | 95.43 ± 0.37 | 84.01 ± 3.09 |
| ATP | N/A | 92.07 ± 0.69 | N/A | 89.11 ± 0.87 | N/A | 85.91 ± 2.34 | N/A | 84.92 ± 1.99 |
| FLUX | 95.72 ± 0.38 | 95.61 ± 0.57 | 95.34 ± 0.62 | 94.45 ± 1.42 | 94.88 ± 1.16 | 94.49 ± 0.88 | 93.54 ± 2.68 | 94.81 ± 1.55 |
| FLUX-prior | 95.89 ± 0.20 | 95.69 ± 0.33 | 95.82 ± 0.09 | 94.83 ± 1.02 | 95.55 ± 0.56 | 95.39 ± 0.80 | 95.68 ± 0.22 | 95.93 ± 0.32 |

Table 10: Performance comparison across non-IID Levels 1–4, summarizing all four types of heterogeneity ($P(X)$, $P(Y)$, $P(Y|X)$, $P(X|Y)$) on the MNIST dataset. **Known A.**: Known Association.

### F.2  Scaling the Number of Clients

We evaluate the scalability and efficiency of the proposed approach across varying numbers of clients on MNIST dataset, with the non-IID levels fixed as follows: **Feature distribution shift** involves rotations ($\{0°, 90°, 180°, 270°\}$). **Label distribution shift** sets the number of classes to 4, with a bank size of 5. $P(Y|X)$ **concept shift** uses a swapping pool size of 5, while $P(X|Y)$ **concept shift**

| non-IID Level | Level 5 | | Level 6 | | Level 7 | | Level 8 | |
|---|---|---|---|---|---|---|---|---|
| **Algorithm** | **Known A.** | **Test Phase** | **Known A.** | **Test Phase** | **Known A.** | **Test Phase** | **Known A.** | **Test Phase** |
| FedAvg | 80.94 ± 1.53 | 86.62 ± 1.75 | 76.16 ± 2.38 | 83.92 ± 2.58 | 72.01 ± 1.96 | 81.12 ± 2.09 | 69.36 ± 2.71 | 79.36 ± 2.90 |
| IFCA | 83.64 ± 9.88 | 79.23 ± 8.15 | 79.58 ± 8.22 | 74.04 ± 7.79 | 75.95 ± 10.07 | 68.32 ± 6.13 | 76.00 ± 9.68 | 64.22 ± 5.89 |
| FedRC | 35.61 ± 4.08 | 42.96 ± 4.58 | 30.59 ± 3.09 | 37.10 ± 3.50 | 30.25 ± 2.61 | 36.15 ± 2.50 | 26.15 ± 2.41 | 30.32 ± 2.84 |
| FedEM | 35.85 ± 4.15 | 42.75 ± 4.55 | 31.16 ± 2.94 | 36.75 ± 3.33 | 31.18 ± 2.39 | 35.60 ± 2.53 | 31.10 ± 1.49 | 32.15 ± 3.21 |
| FeSEM | 87.37 ± 3.28 | 82.37 ± 4.89 | 85.33 ± 3.72 | 80.32 ± 3.96 | 85.60 ± 3.50 | 75.99 ± 4.77 | 83.20 ± 5.17 | 72.83 ± 5.17 |
| FedDrift | 88.98 ± 2.34 | 64.39 ± 5.30 | 87.59 ± 4.00 | 57.70 ± 5.00 | 88.53 ± 3.62 | 52.91 ± 2.69 | 89.56 ± 2.42 | 46.28 ± 3.66 |
| CFL | 81.39 ± 1.73 | 87.14 ± 1.93 | 76.94 ± 1.87 | 84.94 ± 1.95 | 72.45 ± 2.06 | 81.66 ± 2.14 | 69.51 ± 2.37 | 79.52 ± 2.54 |
| pFedMe | 95.56 ± 0.28 | N/A | 95.71 ± 0.29 | N/A | 95.89 ± 0.16 | N/A | **96.09 ± 0.19** | N/A |
| APFL | 95.45 ± 0.26 | 85.19 ± 2.24 | 95.41 ± 0.30 | 84.08 ± 1.95 | 95.30 ± 0.26 | 80.14 ± 2.19 | 95.37 ± 0.39 | 78.96 ± 2.72 |
| ATP | N/A | 86.80 ± 1.41 | N/A | 84.99 ± 2.36 | N/A | 81.14 ± 1.92 | N/A | 79.92 ± 2.58 |
| FLUX | 93.44 ± 2.28 | 93.88 ± 3.43 | 90.53 ± 3.14 | 93.46 ± 2.82 | 87.77 ± 5.23 | 92.70 ± 2.96 | 88.40 ± 4.22 | 92.32 ± 2.25 |
| FLUX-prior | **95.67 ± 0.24** | **95.58 ± 1.12** | **95.73 ± 0.28** | **95.73 ± 1.88** | **91.91 ± 0.18** | **95.78 ± 1.36** | 95.93 ± 0.32 | **96.28 ± 1.33** |

Table 11: Performance comparison across non-IID Levels 5–8, summarizing all four types of heterogeneity ($P(X)$, $P(Y)$, $P(Y|X)$, $P(X|Y)$) on the MNIST dataset. **Known A.**: Known Association.

| non-IID Level | Level 1 | | Level 2 | | Level 3 | | Level 4 | |
|---|---|---|---|---|---|---|---|---|
| **Algorithm** | **Known A.** | **Test Phase** | **Known A.** | **Test Phase** | **Known A.** | **Test Phase** | **Known A.** | **Test Phase** |
| FedAvg | 86.29 ± 0.45 | 86.29 ± 0.45 | 81.69 ± 1.07 | 81.69 ± 1.07 | 74.76 ± 4.20 | 74.76 ± 4.20 | 73.18 ± 2.94 | 73.18 ± 2.94 |
| IFCA | 79.43 ± 11.60 | 78.79 ± 4.93 | 92.17 ± 3.43 | 70.38 ± 1.91 | 88.47 ± 4.78 | 64.70 ± 1.75 | 77.68 ± 8.61 | 64.00 ± 6.44 |
| FedRC | 81.75 ± 5.98 | 77.68 ± 2.20 | 58.30 ± 4.40 | 56.56 ± 4.63 | 43.07 ± 10.11 | 42.45 ± 9.55 | 41.07 ± 6.77 | 40.82 ± 6.57 |
| FedEM | 81.66 ± 5.88 | 77.62 ± 2.24 | 58.30 ± 4.33 | 56.54 ± 4.47 | 43.05 ± 10.18 | 42.40 ± 9.54 | 41.09 ± 6.80 | 40.87 ± 6.55 |
| FeSEM | 88.51 ± 2.54 | 85.01 ± 2.30 | 86.48 ± 2.23 | 80.42 ± 2.44 | 82.25 ± 2.89 | 74.25 ± 3.90 | 82.04 ± 3.24 | 74.23 ± 2.73 |
| FedDrift | 96.36 ± 0.12 | 64.87 ± 2.15 | 95.33 ± 1.59 | 44.27 ± 4.58 | 95.42 ± 1.41 | 43.18 ± 3.00 | 95.50 ± 1.02 | 40.01 ± 2.61 |
| CFL | 86.98 ± 0.56 | 86.98 ± 0.56 | 82.68 ± 0.81 | 82.68 ± 0.81 | 75.95 ± 3.63 | 75.95 ± 3.63 | 74.09 ± 2.68 | 74.09 ± 2.68 |
| pFedMe | 95.04 ± 0.10 | N/A | 94.97 ± 0.09 | N/A | 95.08 ± 0.26 | N/A | 94.90 ± 0.17 | N/A |
| APFL | 95.11 ± 0.15 | 82.97 ± 1.23 | 94.74 ± 0.35 | 79.44 ± 2.60 | 94.76 ± 0.33 | 73.21 ± 3.74 | 94.32 ± 0.29 | 70.36 ± 4.81 |
| ATP | N/A | 89.87 ± 0.39 | N/A | 83.69 ± 1.06 | N/A | 79.71 ± 2.60 | N/A | 78.77 ± 2.46 |
| FLUX | 95.96 ± 0.16 | 95.48 ± 0.76 | 96.09 ± 0.19 | 96.03 ± 0.26 | 96.20 ± 0.23 | 96.20 ± 0.23 | 95.86 ± 0.24 | 95.85 ± 0.24 |
| FLUX-prior | **96.40 ± 0.14** | **96.40 ± 0.14** | **96.24 ± 0.07** | **96.24 ± 0.07** | **96.26 ± 0.23** | **96.26 ± 0.23** | **95.88 ± 0.19** | **95.88 ± 0.19** |

Table 12: Performance comparison across non-IID Levels 1–4 of $P(X)$ on the MNIST dataset. **Known A.**: Known Association.

| non-IID Level | Level 5 | | Level 6 | | Level 7 | | Level 8 | |
|---|---|---|---|---|---|---|---|---|
| **Algorithm** | **Known A.** | **Test Phase** | **Known A.** | **Test Phase** | **Known A.** | **Test Phase** | **Known A.** | **Test Phase** |
| FedAvg | 83.16 ± 0.65 | 83.16 ± 0.65 | 78.67 ± 2.14 | 78.67 ± 2.14 | 73.25 ± 2.15 | 73.25 ± 2.15 | 71.35 ± 3.52 | 71.35 ± 3.52 |
| IFCA | 86.07 ± 18.86 | 67.56 ± 13.55 | 82.46 ± 9.90 | 64.44 ± 10.15 | 79.63 ± 13.11 | 58.81 ± 8.85 | 86.99 ± 6.96 | 58.70 ± 8.04 |
| FedRC | 30.51 ± 7.26 | 30.51 ± 7.35 | 14.78 ± 4.88 | 14.75 ± 4.82 | 11.40 ± 0.10 | 11.40 ± 0.10 | 11.35 ± 0.00 | 11.35 ± 0.00 |
| FedEM | 30.47 ± 7.26 | 30.47 ± 7.35 | 14.79 ± 4.89 | 14.76 ± 4.83 | 11.40 ± 0.10 | 11.40 ± 0.10 | 11.35 ± 0.00 | 11.35 ± 0.00 |
| FeSEM | 86.82 ± 3.88 | 76.47 ± 7.83 | 88.19 ± 1.38 | 74.07 ± 3.48 | 89.19 ± 2.31 | 65.76 ± 5.29 | 89.21 ± 2.27 | 64.79 ± 5.49 |
| FedDrift | 92.86 ± 3.39 | 51.55 ± 6.53 | 92.51 ± 2.39 | 46.83 ± 5.48 | 91.50 ± 2.62 | 48.02 ± 2.13 | 93.64 ± 1.64 | 42.80 ± 5.34 |
| CFL | 84.06 ± 0.52 | 84.06 ± 0.52 | 79.60 ± 1.90 | 79.60 ± 1.90 | 73.40 ± 2.23 | 73.40 ± 2.23 | 72.02 ± 3.44 | 72.02 ± 3.44 |
| pFedMe | 94.74 ± 0.21 | N/A | 94.86 ± 0.17 | N/A | 94.81 ± 0.09 | N/A | 94.87 ± 0.03 | N/A |
| APFL | 94.78 ± 0.27 | 79.86 ± 2.29 | 94.58 ± 0.06 | 78.67 ± 1.08 | 94.29 ± 0.27 | 71.38 ± 2.05 | 94.16 ± 0.28 | 70.68 ± 3.54 |
| ATP | N/A | 87.47 ± 0.79 | N/A | 83.81 ± 1.27 | N/A | 78.16 ± 2.28 | N/A | 78.39 ± 2.66 |
| FLUX | **95.50 ± 0.10** | 92.95 ± 5.01 | **95.35 ± 0.17** | **95.34 ± 0.18** | 93.99 ± 2.46 | **93.98 ± 2.45** | **95.26 ± 0.10** | 93.96 ± 2.60 |
| FLUX-prior | 95.35 ± 0.26 | **95.35 ± 0.26** | 95.12 ± 0.28 | 95.13 ± 0.25 | **95.07 ± 0.18** | 93.38 ± 2.09 | 94.90 ± 0.38 | **94.92 ± 0.37** |

Table 13: Performance comparison across non-IID Levels 5–8 of $P(X)$ on the MNIST dataset. **Known A.**: Known Association.

| non-IID Level | Level 1 | | Level 2 | | Level 3 | | Level 4 | |
|---|---|---|---|---|---|---|---|---|
| Algorithm | Known A. | Test Phase | Known A. | Test Phase | Known A. | Test Phase | Known A. | Test Phase |
| FedAvg | 95.85 ± 0.03 | 95.85 ± 0.03 | 94.80 ± 0.36 | 94.80 ± 0.36 | 93.98 ± 0.62 | 93.98 ± 0.62 | 92.32 ± 1.18 | 92.32 ± 1.18 |
| IFCA | 96.01 ± 0.14 | 96.01 ± 0.14 | 94.99 ± 0.69 | 94.33 ± 0.80 | 95.23 ± 0.39 | 93.77 ± 1.77 | 88.51 ± 7.92 | 90.18 ± 3.13 |
| FedRC | 90.21 ± 3.28 | 90.21 ± 3.29 | 87.75 ± 1.85 | 87.76 ± 1.87 | 86.96 ± 3.13 | 86.95 ± 3.13 | 86.78 ± 1.32 | 86.78 ± 1.33 |
| FedEM | 90.20 ± 3.28 | 90.20 ± 3.28 | 87.73 ± 1.86 | 87.73 ± 1.86 | 86.96 ± 3.15 | 86.96 ± 3.16 | 86.65 ± 1.46 | 86.63 ± 1.50 |
| FeSEM | 95.18 ± 0.32 | 95.94 ± 0.26 | 94.75 ± 0.93 | 94.59 ± 0.70 | 94.54 ± 0.91 | 93.21 ± 2.18 | 94.27 ± 1.73 | 90.93 ± 2.37 |
| FedDrift | 96.05 ± 0.04 | **96.13 ± 0.04** | 94.81 ± 0.31 | 94.81 ± 0.31 | 94.32 ± 0.42 | 88.16 ± 3.10 | 95.72 ± 0.70 | 77.39 ± 4.24 |
| CFL | **96.12 ± 0.04** | 96.12 ± 0.04 | 94.87 ± 0.25 | 94.87 ± 0.25 | 94.32 ± 0.58 | 94.32 ± 0.58 | 92.92 ± 1.07 | 92.92 ± 1.07 |
| pFedMe | 94.62 ± 0.33 | N/A | 94.95 ± 0.34 | N/A | 95.63 ± 0.33 | N/A | 96.34 ± 0.25 | N/A |
| APFL | 95.82 ± 0.14 | 96.10 ± 0.22 | **95.81 ± 0.22** | 94.55 ± 0.45 | 96.08 ± 0.44 | 94.64 ± 0.29 | 96.41 ± 0.45 | 92.04 ± 1.44 |
| ATP | N/A | 95.69 ± 0.08 | N/A | **95.09 ± 0.30** | N/A | 94.62 ± 0.36 | N/A | 94.21 ± 0.82 |
| FLUX | 95.52 ± 0.47 | 95.41 ± 0.55 | 95.50 ± 0.43 | 93.81 ± 1.29 | 96.09 ± 0.21 | 94.94 ± 1.09 | 96.27 ± 0.63 | 96.10 ± 1.07 |
| FLUX-prior | 95.42 ± 0.11 | 94.95 ± 0.33 | 95.67 ± 0.10 | 93.95 ± 1.12 | **96.16 ± 0.15** | **94.98 ± 1.10** | **96.64 ± 0.30** | **96.64 ± 0.30** |

Table 14: Performance comparison across non-IID Levels 1–4 of $P(Y)$ on the MNIST dataset. **Known A.**: Known Association.

| non-IID Level | Level 5 | | Level 6 | | Level 7 | | Level 8 | |
|---|---|---|---|---|---|---|---|---|
| Algorithm | Known A. | Test Phase | Known A. | Test Phase | Known A. | Test Phase | Known A. | Test Phase |
| FedAvg | 91.72 ± 1.15 | 91.72 ± 1.15 | 90.74 ± 2.71 | 90.74 ± 2.71 | 89.22 ± 1.87 | 89.22 ± 1.87 | 86.83 ± 3.09 | 86.83 ± 3.09 |
| IFCA | 91.65 ± 3.91 | 86.66 ± 1.85 | 85.30 ± 8.96 | 75.18 ± 8.42 | 80.98 ± 12.90 | 66.70 ± 4.98 | 72.75 ± 15.69 | 54.54 ± 5.97 |
| FedRC | 83.78 ± 1.36 | 83.68 ± 1.40 | 83.78 ± 3.56 | 83.60 ± 3.49 | 84.11 ± 3.90 | 84.09 ± 3.90 | 68.42 ± 4.09 | 67.61 ± 4.80 |
| FedEM | 84.79 ± 2.07 | 83.08 ± 1.16 | 86.05 ± 3.01 | 82.54 ± 2.93 | 87.84 ± 3.30 | 82.41 ± 3.95 | 88.24 ± 1.55 | 73.10 ± 5.47 |
| FeSEM | 94.80 ± 1.60 | 88.14 ± 2.21 | 93.55 ± 1.89 | 84.99 ± 5.73 | 91.65 ± 2.37 | 82.92 ± 5.98 | 87.03 ± 3.85 | 76.17 ± 6.61 |
| FedDrift | 95.85 ± 0.73 | 69.48 ± 5.39 | 97.62 ± 0.36 | 57.36 ± 4.59 | **98.13 ± 0.28** | 49.37 ± 3.25 | 98.45 ± 0.35 | 41.41 ± 2.95 |
| CFL | 92.00 ± 1.18 | 92.00 ± 1.18 | 92.00 ± 1.09 | 92.00 ± 1.09 | 90.11 ± 1.62 | 90.11 ± 1.62 | 85.58 ± 1.88 | 85.58 ± 1.88 |
| pFedMe | 96.88 ± 0.21 | N/A | 97.08 ± 0.22 | N/A | 97.68 ± 0.28 | N/A | 98.10 ± 0.56 | N/A |
| APFL | 96.69 ± 0.26 | 91.95 ± 0.81 | 97.02 ± 0.18 | 91.39 ± 1.38 | 97.49 ± 0.28 | 89.24 ± 1.83 | 97.94 ± 0.37 | 86.99 ± 2.56 |
| ATP | N/A | 94.02 ± 0.68 | N/A | 94.06 ± 1.06 | N/A | 93.99 ± 0.47 | N/A | 92.76 ± 1.02 |
| FLUX | 97.02 ± 0.40 | 95.95 ± 1.70 | 97.53 ± 0.38 | 96.11 ± 3.28 | 98.00 ± 0.17 | **98.00 ± 0.17** | 98.35 ± 0.41 | 98.35 ± 0.42 |
| FLUX-prior | **97.03 ± 0.33** | **95.96 ± 1.64** | **97.68 ± 0.27** | **96.19 ± 3.15** | 97.96 ± 0.15 | 97.95 ± 0.15 | **98.46 ± 0.35** | **98.46 ± 0.35** |

Table 15: Performance comparison across non-IID Levels 5–8 of $P(Y)$ on the MNIST dataset. **Known A.**: Known Association.

| non-IID Level | Level 1 | | Level 2 | | Level 3 | | Level 4 | |
|---|---|---|---|---|---|---|---|---|
| Algorithm | Known A. | Test Phase | Known A. | Test Phase | Known A. | Test Phase | Known A. | Test Phase |
| FedAvg | 95.85 ± 0.03 | N/A | 88.11 ± 2.41 | N/A | 79.01 ± 0.87 | N/A | 70.30 ± 2.35 | N/A |
| IFCA | 95.86 ± 0.14 | N/A | 88.33 ± 2.52 | N/A | 83.27 ± 3.62 | N/A | 75.44 ± 2.88 | N/A |
| FedRC | 95.53 ± 3.28 | N/A | 86.96 ± 2.68 | N/A | 62.06 ± 14.42 | N/A | 18.46 ± 6.84 | N/A |
| FedEM | 95.52 ± 3.28 | N/A | 86.94 ± 2.68 | N/A | 62.03 ± 14.40 | N/A | 18.46 ± 6.84 | N/A |
| FeSEM | 95.92 ± 0.14 | N/A | 89.20 ± 2.72 | N/A | 84.10 ± 2.35 | N/A | 80.26 ± 1.46 | N/A |
| FedDrift | 95.67 ± 0.09 | N/A | 95.06 ± 1.68 | N/A | 85.41 ± 5.67 | N/A | 80.55 ± 7.40 | N/A |
| CFL | 95.13 ± 0.12 | N/A | 88.43 ± 2.47 | N/A | 79.06 ± 0.74 | N/A | 70.52 ± 2.09 | N/A |
| pFedMe | 94.62 ± 0.33 | N/A | 94.56 ± 0.37 | N/A | 94.58 ± 0.40 | N/A | 94.41 ± 0.35 | N/A |
| APFL | 95.83 ± 0.22 | N/A | 95.19 ± 0.20 | N/A | **94.69 ± 0.20** | N/A | 94.32 ± 0.37 | N/A |
| ATP | 95.69 ± 0.08 | N/A | 90.50 ± 0.84 | N/A | 82.09 ± 1.50 | N/A | 76.27 ± 3.01 | N/A |
| FLUX | 95.36 ± 0.49 | N/A | 95.21 ± 0.16 | N/A | 94.27 ± 1.70 | N/A | 88.64 ± 4.99 | N/A |
| FLUX-prior | **95.93 ± 0.05** | N/A | **95.72 ± 0.08** | N/A | 94.31 ± 1.06 | N/A | **94.53 ± 0.18** | N/A |

Table 16: Performance comparison across non-IID Levels 1–4 of $P(Y|X)$ on the MNIST dataset. **Known A.**: Known Association.

| non-IID Level | Level 5 | | Level 6 | | Level 7 | | Level 8 | |
|---|---|---|---|---|---|---|---|---|
| Algorithm | Known A. | Test Phase | Known A. | Test Phase | Known A. | Test Phase | Known A. | Test Phase |
| FedAvg | 63.87 ± 0.36 | N/A | 52.90 ± 1.65 | N/A | 44.70 ± 1.53 | N/A | 39.38 ± 2.04 | N/A |
| IFCA | 68.15 ± 3.42 | N/A | 63.80 ± 8.91 | N/A | 58.20 ± 7.93 | N/A | 56.59 ± 8.67 | N/A |
| FedRC | 13.46 ± 2.27 | N/A | 10.86 ± 0.54 | N/A | 12.53 ± 2.90 | N/A | 12.82 ± 2.33 | N/A |
| FedEM | 13.45 ± 2.27 | N/A | 10.86 ± 0.54 | N/A | 12.53 ± 2.90 | N/A | 12.82 ± 2.32 | N/A |
| FeSEM | 76.16 ± 4.81 | N/A | 68.62 ± 6.90 | N/A | 70.48 ± 5.73 | N/A | 66.21 ± 9.16 | N/A |
| FedDrift | 75.46 ± 1.26 | N/A | 69.98 ± 6.47 | N/A | 72.42 ± 6.65 | N/A | 74.11 ± 4.31 | N/A |
| CFL | 64.16 ± 0.90 | N/A | 52.92 ± 1.60 | N/A | 44.81 ± 1.80 | N/A | 39.49 ± 1.75 | N/A |
| pFedMe | **94.41 ± 0.41** | N/A | **94.54 ± 0.36** | N/A | **94.46 ± 0.12** | N/A | **94.50 ± 0.04** | N/A |
| APFL | 94.20 ± 0.32 | N/A | 93.64 ± 0.54 | N/A | 93.28 ± 0.32 | N/A | 93.20 ± 0.46 | N/A |
| ATP | 70.45 ± 0.53 | N/A | 61.45 ± 0.47 | N/A | 54.87 ± 1.57 | N/A | 48.92 ± 1.72 | N/A |
| FLUX | 85.62 ± 4.26 | N/A | 79.74 ± 4.59 | N/A | 70.09 ± 8.83 | N/A | 70.02 ± 7.70 | N/A |
| FLUX-prior | 94.18 ± 0.16 | N/A | 94.01 ± 0.21 | N/A | 94.07 ± 0.14 | N/A | 93.94 ± 0.20 | N/A |

Table 17: Performance comparison across non-IID Levels 5–8 of $P(Y|X)$ on the MNIST dataset. **Known A.**: Known Association.

| non-IID Level | Level 1 | | Level 2 | | Level 3 | | Level 4 | |
|---|---|---|---|---|---|---|---|---|
| Algorithm | Known A. | Test Phase | Known A. | Test Phase | Known A. | Test Phase | Known A. | Test Phase |
| FedAvg | 94.32 ± 0.89 | 94.32 ± 0.89 | 92.92 ± 1.02 | 92.92 ± 1.02 | 90.21 ± 2.21 | 90.21 ± 2.21 | 90.47 ± 2.04 | 90.47 ± 2.04 |
| IFCA | 94.49 ± 0.58 | 95.00 ± 2.00 | 94.57 ± 1.08 | 92.25 ± 1.71 | 92.00 ± 2.59 | 90.34 ± 2.29 | 92.91 ± 1.73 | 89.48 ± 2.43 |
| FedRC | 89.57 ± 2.74 | 89.60 ± 2.72 | 62.25 ± 6.60 | 62.19 ± 6.61 | 21.03 ± 10.25 | 21.03 ± 10.24 | 16.24 ± 2.25 | 16.25 ± 2.25 |
| FedEM | 89.59 ± 2.72 | 89.58 ± 2.72 | 62.26 ± 6.62 | 62.21 ± 6.64 | 21.03 ± 10.26 | 21.02 ± 10.24 | 16.24 ± 2.25 | 16.25 ± 2.25 |
| FeSEM | 94.71 ± 0.52 | 94.12 ± 1.99 | 94.33 ± 0.37 | 91.95 ± 1.13 | 93.79 ± 1.15 | 89.42 ± 2.96 | 93.92 ± 1.53 | 88.20 ± 1.93 |
| FedDrift | 94.93 ± 0.87 | 95.51 ± 2.45 | 93.88 ± 1.00 | 89.75 ± 2.48 | 92.82 ± 1.07 | 83.81 ± 4.15 | 94.14 ± 1.16 | 81.27 ± 3.19 |
| CFL | 94.65 ± 0.64 | 95.11 ± 1.87 | 92.75 ± 1.03 | 92.75 ± 1.03 | 90.47 ± 2.03 | 90.47 ± 2.03 | 90.86 ± 1.65 | 90.86 ± 1.65 |
| pFedMe | 95.08 ± 0.10 | N/A | 95.39 ± 0.33 | N/A | 95.69 ± 0.25 | N/A | 95.97 ± 0.45 | N/A |
| APFL | 95.80 ± 0.08 | 93.87 ± 1.43 | **96.30 ± 0.19** | 92.73 ± 1.40 | **96.09 ± 0.18** | 87.75 ± 2.45 | **96.68 ± 0.37** | 89.63 ± 3.02 |
| ATP | N/A | 95.68 ± 0.46 | N/A | **95.07 ± 0.47** | N/A | 93.68 ± 1.17 | N/A | 93.46 ± 0.99 |
| FLUX | **96.03 ± 0.31** | **95.93 ± 0.32** | 94.58 ± 1.15 | 93.50 ± 2.08 | 92.94 ± 1.55 | 92.33 ± 1.05 | 93.38 ± 1.81 | 92.46 ± 2.45 |
| FLUX-prior | 95.82 ± 0.36 | 95.71 ± 0.45 | 95.64 ± 0.11 | 94.30 ± 1.37 | 95.45 ± 0.22 | **94.92 ± 0.80** | 95.69 ± 0.17 | **95.28 ± 0.42** |

Table 18: Performance comparison across non-IID Levels 1–4 of $P(X|Y)$ on the MNIST dataset. **Known A.**: Known Association.

| non-IID Level | Level 5 | | Level 6 | | Level 7 | | Level 8 | |
|---|---|---|---|---|---|---|---|---|
| Algorithm | Known A. | Test Phase | Known A. | Test Phase | Known A. | Test Phase | Known A. | Test Phase |
| FedAvg | 84.99 ± 2.74 | 84.99 ± 2.74 | 82.34 ± 2.82 | 82.34 ± 2.82 | 80.89 ± 2.23 | 80.89 ± 2.23 | 79.91 ± 1.85 | 79.91 ± 1.85 |
| IFCA | 88.68 ± 2.72 | 83.47 ± 3.46 | 86.76 ± 3.56 | 82.51 ± 2.83 | 84.99 ± 2.18 | 79.45 ± 3.12 | 87.64 ± 2.28 | 79.43 ± 1.92 |
| FedRC | 14.69 ± 2.61 | 14.70 ± 2.61 | 12.96 ± 1.16 | 12.94 ± 1.15 | 12.97 ± 1.88 | 12.98 ± 1.89 | 12.00 ± 1.04 | 12.00 ± 1.03 |
| FedEM | 14.69 ± 2.60 | 14.70 ± 2.61 | 12.96 ± 1.16 | 12.94 ± 1.15 | 12.96 ± 1.88 | 12.98 ± 1.89 | 12.00 ± 1.04 | 12.00 ± 1.03 |
| FeSEM | 91.72 ± 1.46 | 82.50 ± 2.35 | 90.98 ± 1.49 | 81.91 ± 1.47 | 91.08 ± 2.30 | 79.28 ± 2.11 | 90.36 ± 1.75 | 77.52 ± 2.53 |
| FedDrift | 91.76 ± 2.88 | 72.13 ± 3.53 | 90.26 ± 4.02 | 68.90 ± 4.89 | 92.06 ± 1.08 | 61.35 ± 2.56 | 92.06 ± 1.45 | 54.63 ± 1.72 |
| CFL | 85.36 ± 3.09 | 85.36 ± 3.09 | 83.23 ± 2.57 | 83.23 ± 2.57 | 81.47 ± 2.46 | 81.47 ± 2.46 | 80.94 ± 2.01 | 80.94 ± 2.01 |
| pFedMe | 96.19 ± 0.28 | N/A | **96.52 ± 0.39** | N/A | **96.61 ± 0.13** | N/A | **96.90 ± 0.13** | N/A |
| APFL | 96.14 ± 0.21 | 83.78 ± 3.63 | 96.39 ± 0.41 | 82.18 ± 3.38 | 96.14 ± 0.18 | 79.78 ± 2.70 | 96.20 ± 0.46 | 79.20 ± 2.05 |
| ATP | N/A | 90.22 ± 1.51 | N/A | 89.18 ± 1.50 | N/A | 87.43 ± 1.40 | N/A | 86.89 ± 1.13 |
| FLUX | 93.50 ± 1.59 | 92.73 ± 2.71 | 89.52 ± 4.27 | 88.93 ± 3.61 | 88.99 ± 5.04 | 86.13 ± 4.50 | 89.95 ± 3.46 | 84.67 ± 2.87 |
| FLUX-prior | **96.21 ± 0.17** | **95.41 ± 1.00** | 96.13 ± 0.34 | **95.88 ± 0.78** | 96.54 ± 0.23 | **95.99 ± 1.09** | 96.43 ± 0.32 | **95.47 ± 2.25** |

Table 19: Performance comparison across non-IID Levels 5–8 of $P(X|Y)$ on the MNIST dataset. **Known A.**: Known Association.

| non-IID Level | Level 1 | | Level 2 | | Level 3 | | Level 4 | |
|---|---|---|---|---|---|---|---|---|
| Algorithm | Known A. | Test Phase | Known A. | Test Phase | Known A. | Test Phase | Known A. | Test Phase |
| FedAvg | 77.23 ± 0.71 | 76.74 ± 0.80 | 73.34 ± 1.88 | 74.01 ± 1.52 | 68.52 ± 2.27 | 69.94 ± 2.23 | 65.53 ± 2.46 | 68.45 ± 2.74 |
| IFCA | 79.29 ± 4.26 | 76.29 ± 1.71 | 75.76 ± 5.71 | 70.30 ± 2.88 | 75.91 ± 4.51 | 65.34 ± 5.23 | 72.70 ± 6.00 | 64.66 ± 5.60 |
| FedRC | 74.54 ± 1.50 | 72.92 ± 1.74 | 69.29 ± 2.87 | 67.10 ± 2.93 | 61.03 ± 4.07 | 59.32 ± 4.05 | 54.87 ± 3.71 | 55.59 ± 4.78 |
| FedEM | 74.53 ± 1.47 | 72.92 ± 1.67 | 69.49 ± 2.87 | 67.14 ± 3.06 | 61.11 ± 4.15 | 59.35 ± 4.13 | 54.90 ± 3.67 | 55.58 ± 4.75 |
| FeSEM | 78.74 ± 2.53 | 76.48 ± 1.33 | 75.87 ± 3.39 | 72.25 ± 2.82 | 71.51 ± 2.45 | 69.77 ± 2.40 | 71.34 ± 3.12 | 65.12 ± 5.32 |
| FedDrift | 79.77 ± 0.65 | 69.91 ± 0.74 | 78.39 ± 1.84 | 62.19 ± 2.41 | 79.81 ± 2.16 | 55.85 ± 1.97 | 80.42 ± 1.01 | 50.84 ± 1.97 |
| CFL | 77.62 ± 0.81 | 77.15 ± 0.94 | 73.79 ± 1.85 | 74.50 ± 1.42 | 69.27 ± 2.10 | 70.80 ± 2.08 | 66.11 ± 2.32 | 69.09 ± 2.61 |
| pFedMe | 78.86 ± 0.84 | N/A | 79.74 ± 1.02 | N/A | 80.79 ± 0.97 | N/A | **81.45 ± 1.10** | N/A |
| APFL | **81.36 ± 0.51** | 78.13 ± 1.12 | **81.08 ± 0.73** | 74.57 ± 1.92 | **81.21 ± 0.99** | 70.36 ± 2.53 | 81.04 ± 1.06 | 68.08 ± 2.91 |
| ATP | N/A | 76.91 ± 0.79 | N/A | 73.34 ± 1.50 | N/A | 69.43 ± 2.58 | N/A | 67.22 ± 2.33 |
| FLUX | 78.89 ± 0.67 | 78.65 ± 0.68 | 79.68 ± 1.40 | 79.86 ± 1.47 | 79.54 ± 1.61 | 79.68 ± 1.96 | 80.28 ± 1.93 | 81.43 ± 2.30 |
| FLUX-prior | 79.88 ± 0.56 | **80.10 ± 0.47** | 80.49 ± 1.03 | **80.70 ± 1.17** | 80.36 ± 1.47 | **81.39 ± 1.53** | 81.27 ± 1.16 | **82.60 ± 1.41** |

Table 20: Performance comparison across non-IID Levels 1–4, summarizing all four types of heterogeneity ($P(X)$, $P(Y)$, $P(Y|X)$, $P(X|Y)$) on the FMNIST dataset. **Known A.**: Known Association.

| non-IID Level | Level 5 | | Level 6 | | Level 7 | | Level 8 | |
|---|---|---|---|---|---|---|---|---|
| Algorithm | Known A. | Test Phase | Known A. | Test Phase | Known A. | Test Phase | Known A. | Test Phase |
| FedAvg | 66.14 ± 2.50 | 70.42 ± 2.12 | 62.15 ± 3.99 | 67.53 ± 4.37 | 56.77 ± 1.92 | 62.81 ± 1.64 | 54.08 ± 3.72 | 60.61 ± 3.98 |
| IFCA | 72.84 ± 4.25 | 65.12 ± 4.03 | 71.52 ± 4.36 | 61.24 ± 6.06 | 66.27 ± 5.93 | 52.53 ± 4.75 | 65.18 ± 5.71 | 52.58 ± 6.80 |
| FedRC | 51.69 ± 5.69 | 55.22 ± 5.76 | 46.37 ± 8.32 | 50.08 ± 8.89 | 39.86 ± 5.17 | 44.29 ± 4.77 | 36.54 ± 5.61 | 40.83 ± 6.07 |
| FedEM | 52.03 ± 5.90 | 55.15 ± 5.59 | 48.04 ± 8.51 | 50.00 ± 8.65 | 43.71 ± 5.39 | 44.67 ± 4.92 | 41.81 ± 6.11 | 42.93 ± 5.59 |
| FeSEM | 71.46 ± 3.38 | 65.88 ± 5.69 | 71.31 ± 4.95 | 64.23 ± 5.32 | 68.72 ± 4.76 | 59.38 ± 4.23 | 66.16 ± 4.13 | 56.64 ± 5.95 |
| FedDrift | 80.56 ± 1.84 | 49.91 ± 4.04 | 81.60 ± 2.31 | 45.48 ± 3.40 | 82.64 ± 2.86 | 39.30 ± 2.88 | 82.55 ± 3.00 | 35.37 ± 2.36 |
| CFL | 66.63 ± 2.59 | 71.02 ± 2.22 | 62.63 ± 4.76 | 67.96 ± 5.31 | 57.23 ± 2.98 | 63.31 ± 3.13 | 54.62 ± 3.34 | 61.21 ± 3.55 |
| pFedMe | **82.38 ± 1.18** | N/A | **83.25 ± 0.94** | N/A | **84.20 ± 0.74** | N/A | **84.63 ± 0.90** | N/A |
| APFL | 81.54 ± 1.53 | 70.55 ± 1.97 | 82.06 ± 1.64 | 68.41 ± 3.32 | 83.47 ± 0.92 | 63.61 ± 2.50 | 83.82 ± 1.58 | 59.90 ± 3.09 |
| ATP | N/A | 70.51 ± 2.41 | N/A | 66.65 ± 4.77 | N/A | 61.87 ± 2.86 | N/A | 61.49 ± 5.07 |
| FLUX | 78.90 ± 3.06 | 81.72 ± 2.35 | 78.98 ± 4.03 | 82.26 ± 2.98 | 79.01 ± 4.38 | 82.05 ± 4.90 | 80.00 ± 3.06 | 84.26 ± 3.08 |
| FLUX-prior | 81.88 ± 1.42 | **83.40 ± 1.49** | 82.96 ± 1.60 | **84.92 ± 1.83** | 83.95 ± 0.79 | **86.29 ± 0.93** | 84.16 ± 1.47 | **86.71 ± 1.69** |

Table 21: Performance comparison across non-IID Levels 5–8, summarizing all four types of heterogeneity ($P(X)$, $P(Y)$, $P(Y|X)$, $P(X|Y)$) on the FMNIST dataset. **Known A.**: Known Association.

| non-IID Level | Level 1 | | Level 2 | | Level 3 | | Level 4 | |
|---|---|---|---|---|---|---|---|---|
| Algorithm | Known A. | Test Phase | Known A. | Test Phase | Known A. | Test Phase | Known A. | Test Phase |
| FedAvg | 72.07 ± 0.93 | 72.07 ± 0.93 | 67.35 ± 1.20 | 67.35 ± 1.20 | 61.17 ± 2.66 | 61.17 ± 2.66 | 57.95 ± 2.57 | 57.95 ± 2.57 |
| IFCA | 77.29 ± 8.46 | 68.53 ± 2.72 | 67.37 ± 10.47 | 53.81 ± 4.65 | 74.36 ± 7.58 | 44.84 ± 8.54 | 69.11 ± 8.72 | 48.37 ± 6.79 |
| FedRC | 69.16 ± 1.38 | 68.47 ± 1.46 | 65.72 ± 1.38 | 60.29 ± 2.22 | 53.64 ± 6.35 | 47.41 ± 5.79 | 50.00 ± 5.26 | 43.32 ± 6.58 |
| FedEM | 69.24 ± 1.22 | 68.56 ± 1.18 | 65.87 ± 1.45 | 60.33 ± 2.34 | 53.98 ± 6.37 | 47.46 ± 5.78 | 49.87 ± 5.08 | 43.34 ± 6.51 |
| FeSEM | 77.45 ± 4.83 | 69.96 ± 1.83 | 74.26 ± 5.03 | 63.05 ± 4.13 | 64.98 ± 3.91 | 60.19 ± 3.34 | 68.82 ± 3.90 | 52.94 ± 7.97 |
| FedDrift | **81.21 ± 0.27** | 50.90 ± 0.19 | **80.94 ± 0.31** | 34.10 ± 0.24 | 80.21 ± 0.29 | 28.18 ± 2.40 | **80.47 ± 0.34** | 27.27 ± 1.58 |
| CFL | 72.65 ± 1.08 | 72.65 ± 1.08 | 68.18 ± 1.22 | 68.18 ± 1.22 | 62.64 ± 2.24 | 62.64 ± 2.24 | 58.59 ± 2.76 | 58.59 ± 2.76 |
| pFedMe | 79.43 ± 0.58 | N/A | 78.96 ± 0.28 | N/A | 79.51 ± 0.20 | N/A | 78.95 ± 0.07 | N/A |
| APFL | 80.86 ± 0.12 | 73.19 ± 1.07 | 80.01 ± 0.08 | 66.92 ± 2.74 | 80.11 ± 0.48 | 59.59 ± 4.47 | 79.26 ± 0.50 | 54.71 ± 5.10 |
| ATP | N/A | 75.69 ± 0.17 | N/A | 70.96 ± 0.59 | N/A | 67.70 ± 1.32 | N/A | 66.29 ± 1.75 |
| FLUX | 79.58 ± 0.81 | 79.35 ± 0.56 | 78.72 ± 0.68 | 78.48 ± 0.66 | 79.63 ± 0.47 | 79.21 ± 0.66 | 79.28 ± 0.39 | 79.23 ± 0.34 |
| FLUX-prior | 80.92 ± 0.23 | **80.92 ± 0.23** | 80.02 ± 0.15 | **80.02 ± 0.15** | 80.91 ± 0.41 | **80.21 ± 0.41** | 79.77 ± 0.15 | **79.77 ± 0.15** |

Table 22: Performance comparison across non-IID Levels 1–4 of $P(X)$ on the FMNIST dataset. **Known A.**: Known Association.

| non-IID Level | Level 5 | | Level 6 | | Level 7 | | Level 8 | |
|---|---|---|---|---|---|---|---|---|
| **Algorithm** | **Known A.** | **Test Phase** | **Known A.** | **Test Phase** | **Known A.** | **Test Phase** | **Known A.** | **Test Phase** |
| FedAvg | 67.63 ± 0.62 | 67.63 ± 0.62 | 61.41 ± 3.49 | 61.41 ± 3.49 | 53.65 ± 1.51 | 53.65 ± 1.51 | 51.73 ± 6.16 | 51.73 ± 6.16 |
| IFCA | 78.56 ± 1.23 | 53.01 ± 5.08 | 74.12 ± 3.20 | 45.70 ± 8.00 | 65.53 ± 6.55 | 28.82 ± 6.38 | 71.93 ± 5.07 | 36.54 ± 9.91 |
| FedRC | 46.85 ± 8.46 | 44.72 ± 7.86 | 30.74 ± 15.17 | 29.49 ± 14.50 | 20.83 ± 6.89 | 20.35 ± 6.78 | 18.21 ± 8.04 | 17.95 ± 7.77 |
| FedEM | 46.58 ± 8.59 | 44.51 ± 7.70 | 30.80 ± 15.17 | 29.46 ± 14.43 | 20.94 ± 7.14 | 20.40 ± 6.94 | 18.26 ± 8.11 | 17.98 ± 7.87 |
| FeSEM | 72.19 ± 3.81 | 57.96 ± 8.19 | 71.27 ± 3.53 | 52.52 ± 7.98 | 67.47 ± 2.64 | 44.47 ± 5.41 | 67.39 ± 3.57 | 39.63 ± 8.77 |
| FedDrift | **79.78 ± 0.57** | 30.69 ± 2.86 | **79.59 ± 0.33** | 24.53 ± 1.47 | 78.07 ± 1.31 | 22.05 ± 1.64 | 77.49 ± 2.55 | 22.41 ± 1.59 |
| CFL | 68.41 ± 0.98 | 68.41 ± 0.98 | 62.63 ± 3.20 | 62.63 ± 3.20 | 53.67 ± 5.01 | 53.67 ± 5.01 | 53.38 ± 5.13 | 53.38 ± 5.13 |
| pFedMe | 79.21 ± 0.40 | N/A | 79.11 ± 0.49 | N/A | 79.00 ± 0.30 | N/A | **79.08 ± 0.21** | N/A |
| APFL | 79.65 ± 0.14 | 65.90 ± 1.53 | 79.47 ± 0.45 | 60.77 ± 5.49 | **79.52 ± 0.41** | 52.10 ± 2.84 | 78.26 ± 0.86 | 52.53 ± 5.01 |
| ATP | N/A | 70.91 ± 1.40 | N/A | 66.86 ± 2.77 | N/A | 61.93 ± 3.03 | N/A | 64.13 ± 3.73 |
| FLUX | 78.63 ± 0.67 | 77.66 ± 2.40 | 76.78 ± 2.29 | 76.38 ± 2.77 | 73.92 ± 5.15 | 71.30 ± 6.07 | 74.37 ± 3.94 | 74.32 ± 3.89 |
| FLUX-prior | 78.75 ± 1.00 | **78.72 ± 1.04** | 78.23 ± 0.53 | **78.19 ± 0.51** | 78.49 ± 0.11 | **78.09 ± 0.11** | 77.66 ± 0.74 | **77.66 ± 0.74** |

Table 23: Performance comparison across non-IID Levels 5–8 of $P(X)$ on the FMNIST dataset. **Known A.**: Known Association.

| non-IID Level | Level 1 | | Level 2 | | Level 3 | | Level 4 | |
|---|---|---|---|---|---|---|---|---|
| **Algorithm** | **Known A.** | **Test Phase** | **Known A.** | **Test Phase** | **Known A.** | **Test Phase** | **Known A.** | **Test Phase** |
| FedAvg | 78.76 ± 0.23 | 78.76 ± 0.23 | 77.78 ± 1.96 | 77.78 ± 1.96 | 74.81 ± 2.45 | 74.81 ± 2.45 | 75.07 ± 1.38 | 75.07 ± 1.38 |
| IFCA | 80.64 ± 0.57 | 80.64 ± 0.57 | 80.77 ± 2.00 | 79.96 ± 1.52 | 81.40 ± 1.85 | 78.13 ± 2.22 | 77.20 ± 7.54 | 74.26 ± 4.67 |
| FedRC | 75.03 ± 0.28 | 75.03 ± 0.26 | 73.54 ± 2.11 | 73.53 ± 2.11 | 70.28 ± 2.38 | 70.25 ± 2.37 | 68.21 ± 1.73 | 68.19 ± 1.79 |
| FedEM | 74.98 ± 0.29 | 74.98 ± 0.28 | 73.46 ± 2.11 | 73.44 ± 2.11 | 70.22 ± 2.37 | 70.20 ± 2.40 | 68.26 ± 1.68 | 68.10 ± 1.81 |
| FeSEM | 79.71 ± 0.72 | 80.33 ± 0.43 | 78.20 ± 2.00 | 78.69 ± 2.11 | 77.02 ± 2.08 | 76.12 ± 2.15 | 76.61 ± 2.93 | 71.85 ± 2.03 |
| FedDrift | 79.14 ± 0.17 | 79.14 ± 0.17 | 77.89 ± 1.51 | 76.24 ± 2.94 | 81.90 ± 1.63 | 69.68 ± 1.71 | 83.25 ± 1.26 | 62.63 ± 2.14 |
| CFL | 79.05 ± 0.13 | 79.05 ± 0.13 | 78.21 ± 1.68 | 78.21 ± 1.68 | 75.24 ± 2.66 | 75.24 ± 2.66 | 75.29 ± 1.63 | 75.29 ± 1.63 |
| pFedMe | 77.75 ± 0.68 | N/A | 80.85 ± 2.03 | N/A | **83.10 ± 0.90** | N/A | **84.72 ± 1.93** | N/A |
| APFL | **80.81 ± 0.32** | **81.01 ± 0.65** | 81.82 ± 1.16 | 79.08 ± 1.55 | 83.08 ± 1.62 | 77.53 ± 2.28 | 84.50 ± 1.52 | 74.66 ± 1.79 |
| ATP | N/A | 79.13 ± 0.27 | N/A | 79.06 ± 1.16 | N/A | 78.21 ± 2.27 | N/A | 78.76 ± 1.22 |
| FLUX | 77.37 ± 0.46 | 76.91 ± 0.46 | 80.69 ± 1.41 | 79.70 ± 1.08 | 81.05 ± 1.94 | 80.77 ± 2.01 | 84.09 ± 1.55 | 83.83 ± 1.74 |
| FLUX-prior | 78.01 ± 0.39 | 78.00 ± 0.46 | 80.88 ± 1.45 | **80.06 ± 1.59** | 81.75 ± 1.89 | **81.15 ± 1.91** | 84.48 ± 1.36 | **84.30 ± 1.57** |

Table 24: Performance comparison across non-IID Levels 1–4 of $P(Y)$ on the FMNIST dataset. **Known A.**: Known Association.

| non-IID Level | Level 5 | | Level 6 | | Level 7 | | Level 8 | |
|---|---|---|---|---|---|---|---|---|
| **Algorithm** | **Known A.** | **Test Phase** | **Known A.** | **Test Phase** | **Known A.** | **Test Phase** | **Known A.** | **Test Phase** |
| FedAvg | 73.83 ± 3.48 | 73.83 ± 3.48 | 71.89 ± 6.42 | 71.89 ± 6.42 | 67.33 ± 1.87 | 67.33 ± 1.87 | 63.11 ± 1.67 | 63.11 ± 1.67 |
| IFCA | 81.74 ± 5.61 | 73.58 ± 4.37 | 81.17 ± 7.46 | 70.92 ± 6.33 | 75.47 ± 7.03 | 64.04 ± 4.52 | 71.61 ± 4.57 | 57.42 ± 5.50 |
| FedRC | 69.06 ± 3.78 | 68.97 ± 3.72 | 69.97 ± 4.48 | 69.85 ± 4.53 | 65.84 ± 3.30 | 65.08 ± 3.97 | 59.28 ± 5.51 | 58.30 ± 6.23 |
| FedEM | 70.50 ± 4.80 | 68.77 ± 3.53 | 76.97 ± 5.63 | 69.91 ± 3.00 | 81.22 ± 3.90 | 66.26 ± 4.04 | 80.67 ± 6.89 | 64.93 ± 3.91 |
| FeSEM | 77.32 ± 4.70 | 71.02 ± 5.37 | 80.89 ± 7.50 | 72.31 ± 4.48 | 78.50 ± 8.17 | 67.89 ± 4.81 | 72.53 ± 5.58 | 65.51 ± 2.93 |
| FedDrift | 86.93 ± 2.51 | 60.42 ± 4.92 | 89.74 ± 3.64 | 55.94 ± 5.54 | 93.63 ± 0.90 | 47.12 ± 3.98 | **95.13 ± 2.50** | 39.98 ± 3.03 |
| CFL | 74.18 ± 3.48 | 74.18 ± 3.48 | 70.99 ± 8.39 | 70.99 ± 8.39 | 67.47 ± 1.50 | 67.47 ± 1.50 | 62.69 ± 2.38 | 62.69 ± 2.38 |
| pFedMe | **87.44 ± 1.75** | N/A | 88.42 ± 2.48 | N/A | 92.84 ± 0.95 | N/A | 93.55 ± 2.81 | N/A |
| APFL | 85.65 ± 3.13 | 75.08 ± 3.32 | 87.55 ± 4.80 | 74.69 ± 3.33 | 91.40 ± 1.10 | 69.98 ± 2.14 | 93.04 ± 4.41 | 59.33 ± 3.45 |
| ATP | N/A | 79.06 ± 3.67 | N/A | 77.94 ± 5.93 | N/A | 78.76 ± 3.65 | N/A | 78.47 ± 6.03 |
| FLUX | 86.38 ± 2.64 | 86.48 ± 2.53 | 89.63 ± 2.96 | 89.65 ± 3.06 | 93.54 ± 1.04 | 93.55 ± 1.00 | 94.72 ± 2.82 | 94.74 ± 2.79 |
| FLUX-prior | 86.95 ± 2.12 | **86.93 ± 2.12** | 89.97 ± 3.01 | **89.97 ± 3.01** | 93.71 ± 1.03 | **93.61 ± 1.03** | 94.80 ± 2.73 | **94.80 ± 2.73** |

Table 25: Performance comparison across non-IID Levels 5–8 of $P(Y)$ on the FMNIST dataset. **Known A.**: Known Association.

| non-IID Level | Level 1 | | Level 2 | | Level 3 | | Level 4 | |
|---|---|---|---|---|---|---|---|---|
| Algorithm | Known A. | Test Phase | Known A. | Test Phase | Known A. | Test Phase | Known A. | Test Phase |
| FedAvg | 78.69 ± 0.32 | N/A | 71.32 ± 2.69 | N/A | 64.28 ± 2.39 | N/A | 56.78 ± 1.30 | N/A |
| IFCA | 78.70 ± 0.27 | N/A | 74.47 ± 4.00 | N/A | 67.91 ± 4.42 | N/A | 64.90 ± 2.57 | N/A |
| FedRC | 78.70 ± 0.27 | N/A | 70.45 ± 3.23 | N/A | 59.87 ± 3.20 | N/A | 46.03 ± 1.56 | N/A |
| FedEM | 78.69 ± 0.27 | N/A | 70.96 ± 2.87 | N/A | 59.83 ± 3.28 | N/A | 46.11 ± 1.70 | N/A |
| FeSEM | 78.71 ± 0.27 | N/A | 74.09 ± 3.88 | N/A | 66.84 ± 1.91 | N/A | 63.26 ± 1.14 | N/A |
| FedDrift | 79.06 ± 0.14 | N/A | 76.86 ± 2.99 | N/A | 74.52 ± 3.63 | N/A | 74.70 ± 0.90 | N/A |
| CFL | 79.03 ± 0.14 | N/A | 71.66 ± 2.78 | N/A | 64.69 ± 2.16 | N/A | 57.17 ± 1.05 | N/A |
| pFedMe | 77.75 ± 0.68 | N/A | 77.94 ± 0.39 | N/A | 78.10 ± 0.54 | N/A | **77.84 ± 0.47** | N/A |
| APFL | **80.81 ± 0.32** | N/A | **79.41 ± 0.89** | N/A | **78.61 ± 0.52** | N/A | 77.05 ± 0.89 | N/A |
| ATP | 79.13 ± 0.27 | N/A | 73.53 ± 1.48 | N/A | 67.17 ± 1.99 | N/A | 61.58 ± 0.65 | N/A |
| FLUX | 77.43 ± 0.37 | N/A | 77.85 ± 0.56 | N/A | 77.85 ± 0.44 | N/A | 75.33 ± 2.13 | N/A |
| FLUX-prior | 78.86 ± 0.19 | N/A | 78.80 ± 0.30 | N/A | 76.98 ± 1.28 | N/A | 77.10 ± 0.32 | N/A |

Table 26: Performance comparison across non-IID Levels 1–4 of $P(Y|X)$ on the FMNIST dataset. **Known A.**: Known Association.

| non-IID Level | Level 5 | | Level 6 | | Level 7 | | Level 8 | |
|---|---|---|---|---|---|---|---|---|
| Algorithm | Known A. | Test Phase | Known A. | Test Phase | Known A. | Test Phase | Known A. | Test Phase |
| FedAvg | 53.31 ± 3.40 | N/A | 46.01 ± 2.53 | N/A | 38.65 ± 2.60 | N/A | 34.50 ± 2.82 | N/A |
| IFCA | 51.57 ± 5.47 | N/A | 50.23 ± 2.26 | N/A | 45.19 ± 4.88 | N/A | 44.26 ± 5.60 | N/A |
| FedRC | 38.85 ± 4.49 | N/A | 33.77 ± 4.56 | N/A | 25.31 ± 6.46 | N/A | 22.40 ± 4.42 | N/A |
| FedEM | 38.84 ± 4.50 | N/A | 33.78 ± 4.55 | N/A | 25.32 ± 6.46 | N/A | 22.40 ± 4.42 | N/A |
| FeSEM | 61.14 ± 2.62 | N/A | 57.73 ± 4.96 | N/A | 55.22 ± 3.65 | N/A | 52.56 ± 3.71 | N/A |
| FedDrift | 72.43 ± 2.34 | N/A | 73.31 ± 2.02 | N/A | 72.71 ± 5.20 | N/A | 72.80 ± 4.16 | N/A |
| CFL | 53.46 ± 3.48 | N/A | 46.62 ± 2.50 | N/A | 39.00 ± 2.49 | N/A | 34.83 ± 2.58 | N/A |
| pFedMe | **77.61 ± 0.35** | N/A | **77.63 ± 0.20** | N/A | **77.29 ± 0.27** | N/A | **77.77 ± 0.23** | N/A |
| APFL | 76.26 ± 0.60 | N/A | 76.13 ± 0.28 | N/A | 76.39 ± 0.35 | N/A | 77.41 ± 0.68 | N/A |
| ATP | 58.71 ± 3.03 | N/A | 52.65 ± 2.79 | N/A | 48.16 ± 1.91 | N/A | 43.68 ± 2.50 | N/A |
| FLUX | 69.37 ± 5.00 | N/A | 68.26 ± 6.41 | N/A | 64.61 ± 6.32 | N/A | 66.84 ± 3.27 | N/A |
| FLUX-prior | 76.96 ± 0.25 | N/A | 77.03 ± 0.46 | N/A | 76.77 ± 0.31 | N/A | 76.53 ± 0.33 | N/A |

Table 27: Performance comparison across non-IID Levels 5–8 of $P(Y|X)$ on the FMNIST dataset. **Known A.**: Known Association.

| non-IID Level | Level 1 | | Level 2 | | Level 3 | | Level 4 | |
|---|---|---|---|---|---|---|---|---|
| Algorithm | Known A. | Test Phase | Known A. | Test Phase | Known A. | Test Phase | Known A. | Test Phase |
| FedAvg | 79.39 ± 1.00 | 79.39 ± 1.00 | 76.90 ± 1.29 | 76.90 ± 1.29 | 73.84 ± 1.34 | 73.84 ± 1.34 | 72.32 ± 3.74 | 72.32 ± 3.74 |
| IFCA | 80.53 ± 0.84 | 79.60 ± 1.03 | 80.43 ± 0.92 | 77.11 ± 1.00 | 79.67 ± 0.85 | 73.04 ± 2.05 | 79.59 ± 2.06 | 71.35 ± 5.11 |
| FedRC | 75.26 ± 2.65 | 75.27 ± 2.62 | 67.46 ± 4.03 | 67.47 ± 4.05 | 60.33 ± 3.18 | 60.31 ± 3.17 | 55.25 ± 4.68 | 55.25 ± 4.69 |
| FedEM | 75.22 ± 2.65 | 75.21 ± 2.62 | 67.69 ± 4.26 | 67.65 ± 4.27 | 60.40 ± 3.45 | 60.40 ± 3.46 | 55.36 ± 4.72 | 55.31 ± 4.69 |
| FeSEM | 79.10 ± 1.33 | 79.14 ± 1.34 | 76.95 ± 1.23 | 75.00 ± 1.53 | 77.19 ± 0.89 | 72.99 ± 1.25 | 76.68 ± 3.70 | 70.56 ± 4.18 |
| FedDrift | 79.69 ± 1.25 | 79.69 ± 1.25 | 77.89 ± 1.51 | 76.24 ± 2.94 | 81.90 ± 1.63 | 69.68 ± 1.71 | 83.25 ± 1.26 | 62.63 ± 2.14 |
| CFL | 79.74 ± 1.21 | 79.70 ± 1.21 | 77.12 ± 1.31 | 77.12 ± 1.31 | 74.51 ± 0.97 | 74.51 ± 0.97 | 73.38 ± 3.19 | 73.38 ± 3.19 |
| pFedMe | 80.49 ± 1.42 | N/A | 81.14 ± 1.38 | N/A | 82.44 ± 2.25 | N/A | **84.30 ± 1.91** | N/A |
| APFL | **82.97 ± 1.28** | 80.18 ± 1.64 | **82.89 ± 0.77** | 77.72 ± 1.47 | **83.03 ± 1.32** | 73.95 ± 0.84 | 83.32 ± 1.34 | 74.88 ± 1.86 |
| ATP | N/A | 80.37 ± 0.87 | N/A | 79.27 ± 0.90 | N/A | 77.57 ± 1.79 | N/A | 77.18 ± 1.81 |
| FLUX | 81.16 ± 0.89 | 79.74 ± 0.94 | 81.45 ± 2.26 | 81.40 ± 2.20 | 79.63 ± 2.49 | 79.07 ± 2.65 | 82.44 ± 2.78 | 81.21 ± 3.58 |
| FLUX-prior | 81.74 ± 1.01 | **81.39 ± 0.62** | 82.26 ± 1.41 | **82.02 ± 1.24** | 82.80 ± 1.79 | **82.80 ± 1.79** | 83.72 ± 1.86 | **83.72 ± 1.86** |

Table 28: Performance comparison across non-IID Levels 1–4 of $P(X|Y)$ on the FMNIST dataset. **Known A.**: Known Association.

| non-IID Level | Level 5 | | Level 6 | | Level 7 | | Level 8 | |
|---|---|---|---|---|---|---|---|---|
| **Algorithm** | **Known A.** | **Test Phase** | **Known A.** | **Test Phase** | **Known A.** | **Test Phase** | **Known A.** | **Test Phase** |
| FedAvg | 69.80 ± 0.97 | 69.80 ± 0.97 | 69.28 ± 1.99 | 69.28 ± 1.99 | 67.44 ± 1.50 | 67.44 ± 1.50 | 66.99 ± 2.57 | 66.99 ± 2.57 |
| IFCA | 79.48 ± 3.05 | 68.78 ± 1.97 | 80.55 ± 2.29 | 67.10 ± 2.47 | 78.89 ± 4.93 | 64.73 ± 2.59 | 72.93 ± 7.25 | 63.79 ± 3.25 |
| FedRC | 52.00 ± 4.87 | 51.97 ± 4.89 | 50.97 ± 2.48 | 50.91 ± 2.49 | 47.46 ± 2.58 | 47.44 ± 2.55 | 46.27 ± 3.37 | 46.22 ± 3.39 |
| FedEM | 52.20 ± 4.70 | 52.17 ± 4.71 | 50.64 ± 2.65 | 50.63 ± 2.66 | 47.36 ± 2.86 | 47.36 ± 2.83 | 45.90 ± 4.07 | 45.88 ± 4.06 |
| FeSEM | 75.18 ± 1.45 | 68.66 ± 1.14 | 75.35 ± 2.17 | 67.85 ± 1.06 | 73.67 ± 1.84 | 65.76 ± 1.14 | 72.15 ± 3.27 | 64.79 ± 4.55 |
| FedDrift | 83.08 ± 1.19 | 58.60 ± 4.06 | 83.75 ± 1.99 | 55.96 ± 1.34 | 85.65 ± 1.74 | 48.73 ± 2.51 | 84.79 ± 2.42 | 43.72 ± 2.23 |
| CFL | 70.47 ± 1.28 | 70.47 ± 1.28 | 70.26 ± 1.96 | 70.26 ± 1.96 | 68.79 ± 1.39 | 68.79 ± 1.39 | 67.57 ± 2.43 | 67.57 ± 2.43 |
| pFedMe | **85.24 ± 2.22** | N/A | **87.82 ± 0.62** | N/A | **87.67 ± 1.45** | N/A | **88.10 ± 0.36** | N/A |
| APFL | 84.62 ± 2.26 | 70.66 ± 1.07 | 85.09 ± 1.03 | 69.78 ± 1.15 | 86.58 ± 1.83 | 68.75 ± 2.51 | 86.57 ± 0.35 | 67.83 ± 0.80 |
| ATP | N/A | 76.20 ± 1.97 | N/A | 75.99 ± 1.24 | N/A | 75.09 ± 1.36 | N/A | 74.25 ± 0.58 |
| FLUX | 81.24 ± 2.23 | 81.01 ± 2.09 | 81.26 ± 3.16 | 80.76 ± 3.11 | 83.98 ± 3.02 | 81.30 ± 5.84 | 84.06 ± 1.82 | 83.73 ± 2.37 |
| FLUX-prior | 84.90 ± 1.59 | **84.54 ± 1.05** | 86.62 ± 0.83 | **86.62 ± 0.83** | 87.34 ± 1.16 | **87.18 ± 1.24** | 87.67 ± 0.74 | **87.67 ± 0.74** |

Table 29: Performance comparison across non-IID Levels 5–8 of $P(X|Y)$ on the FMNIST dataset. **Known A.**: Known Association.

| non-IID Level | Level 1 | | Level 2 | | Level 3 | | Level 4 | |
|---|---|---|---|---|---|---|---|---|
| **Algorithm** | **Known A.** | **Test Phase** | **Known A.** | **Test Phase** | **Known A.** | **Test Phase** | **Known A.** | **Test Phase** |
| FedAvg | 36.69 ± 0.19 | 36.33 ± 0.19 | 34.90 ± 1.08 | 34.92 ± 0.97 | 34.19 ± 1.22 | 35.13 ± 1.37 | 33.16 ± 2.28 | 34.81 ± 2.60 |
| IFCA | 40.01 ± 1.20 | 39.96 ± 1.29 | 39.37 ± 1.78 | 38.99 ± 1.21 | 38.56 ± 1.91 | 39.09 ± 1.41 | 38.66 ± 2.13 | 39.22 ± 2.15 |
| FedRC | 24.88 ± 1.97 | 20.52 ± 2.44 | 21.29 ± 2.33 | 17.82 ± 2.52 | 16.95 ± 3.54 | 17.23 ± 3.66 | 14.66 ± 2.45 | 15.94 ± 2.67 |
| FedEM | 24.84 ± 2.01 | 20.44 ± 2.30 | 21.29 ± 2.31 | 17.82 ± 2.48 | 16.97 ± 3.56 | 17.22 ± 3.65 | 14.68 ± 2.50 | 15.96 ± 2.69 |
| FeSEM | 37.65 ± 0.83 | 38.33 ± 0.79 | 37.23 ± 1.23 | 37.52 ± 1.16 | 37.52 ± 1.47 | 37.85 ± 1.46 | 37.93 ± 2.43 | 37.68 ± 3.00 |
| FedDrift | 38.08 ± 0.88 | 37.27 ± 0.93 | 36.28 ± 1.18 | 35.51 ± 1.42 | 35.40 ± 1.07 | 35.70 ± 2.01 | 34.98 ± 1.74 | 34.16 ± 2.15 |
| CFL | 37.83 ± 0.34 | 37.39 ± 0.61 | 36.04 ± 1.11 | 36.04 ± 0.98 | 35.27 ± 1.11 | 36.20 ± 1.20 | 34.10 ± 2.29 | 35.68 ± 2.62 |
| pFedMe | 38.28 ± 1.07 | N/A | 39.36 ± 0.88 | N/A | 40.41 ± 0.83 | N/A | 41.71 ± 0.87 | N/A |
| APFL | **42.71 ± 0.62** | **41.13 ± 1.23** | **43.01 ± 0.66** | 38.93 ± 1.08 | **43.45 ± 0.71** | 38.70 ± 0.85 | **44.44 ± 0.81** | 38.55 ± 1.18 |
| ATP | N/A | 36.85 ± 0.65 | N/A | 36.20 ± 1.08 | N/A | 36.17 ± 1.19 | N/A | 35.71 ± 2.06 |
| FLUX | 36.65 ± 0.85 | 36.19 ± 1.17 | 37.49 ± 1.02 | 37.06 ± 1.11 | 38.29 ± 1.05 | 37.69 ± 1.79 | 37.82 ± 1.69 | 38.35 ± 1.76 |
| FLUX-prior | 37.39 ± 0.82 | 37.12 ± 0.95 | 37.69 ± 1.12 | 37.17 ± 1.40 | 38.19 ± 1.07 | 37.32 ± 1.60 | 39.52 ± 1.29 | 38.68 ± 1.89 |

Table 30: Performance comparison across non-IID Levels 1–4, summarizing all four types of heterogeneity ($P(X)$, $P(Y)$, $P(Y|X)$, $P(X|Y)$) on the CIFAR-10 dataset. **Known A.**: Known Association.

| non-IID Level | Level 5 | | Level 6 | | Level 7 | | Level 8 | |
|---|---|---|---|---|---|---|---|---|
| **Algorithm** | **Known A.** | **Test Phase** | **Known A.** | **Test Phase** | **Known A.** | **Test Phase** | **Known A.** | **Test Phase** |
| FedAvg | 30.37 ± 1.16 | 31.61 ± 1.14 | 28.07 ± 3.16 | 29.50 ± 3.56 | 25.30 ± 2.93 | 26.47 ± 3.31 | 24.53 ± 2.59 | 26.09 ± 2.90 |
| IFCA | 37.08 ± 2.05 | 35.20 ± 2.38 | 36.64 ± 3.12 | 35.08 ± 2.84 | 38.38 ± 3.46 | 32.87 ± 2.69 | 37.90 ± 3.51 | 31.96 ± 3.13 |
| FedRC | 15.35 ± 1.31 | 16.62 ± 1.18 | 15.58 ± 1.89 | 17.29 ± 2.05 | 15.07 ± 1.21 | 16.37 ± 1.14 | 14.76 ± 2.25 | 15.92 ± 2.54 |
| FedEM | 15.83 ± 1.89 | 17.02 ± 1.56 | 16.10 ± 1.87 | 17.51 ± 2.24 | 16.47 ± 1.65 | 16.69 ± 1.25 | 18.13 ± 2.07 | 17.25 ± 2.02 |
| FeSEM | 36.28 ± 2.06 | 35.64 ± 2.30 | 36.32 ± 2.30 | 33.20 ± 3.33 | 34.50 ± 2.98 | 31.37 ± 3.10 | 34.61 ± 5.60 | 30.62 ± 3.23 |
| FedDrift | 34.71 ± 2.06 | 31.77 ± 1.72 | 33.25 ± 2.69 | 28.68 ± 2.44 | 33.48 ± 3.58 | 26.75 ± 3.31 | 34.27 ± 4.68 | 26.58 ± 2.51 |
| CFL | 31.51 ± 1.11 | 32.87 ± 1.00 | 29.65 ± 3.44 | 31.36 ± 3.85 | 26.91 ± 3.08 | 28.42 ± 3.48 | 25.95 ± 2.29 | 27.83 ± 2.52 |
| pFedMe | 43.03 ± 0.69 | N/A | 43.78 ± 1.36 | N/A | 45.49 ± 1.23 | N/A | 47.34 ± 0.77 | N/A |
| APFL | **44.00 ± 0.72** | 35.34 ± 2.39 | **44.80 ± 1.40** | 34.44 ± 2.85 | **46.47 ± 1.54** | 32.84 ± 2.68 | **48.45 ± 1.14** | 32.66 ± 1.61 |
| ATP | N/A | 32.78 ± 1.39 | N/A | 32.41 ± 2.71 | N/A | 29.50 ± 2.76 | N/A | 29.11 ± 2.73 |
| FLUX | 39.65 ± 1.41 | 39.13 ± 2.06 | 38.60 ± 3.33 | 40.56 ± 3.60 | 39.78 ± 3.00 | 40.98 ± 4.35 | 40.84 ± 3.55 | 39.83 ± 6.39 |
| FLUX-prior | 40.25 ± 1.30 | **39.40 ± 2.13** | 41.13 ± 2.01 | **40.72 ± 3.04** | 42.21 ± 2.49 | **41.85 ± 3.93** | 44.28 ± 2.14 | **42.03 ± 6.11** |

Table 31: Performance comparison across non-IID Levels 5–8, summarizing all four types of heterogeneity ($P(X)$, $P(Y)$, $P(Y|X)$, $P(X|Y)$) on the CIFAR-10 dataset. **Known A.**: Known Association.

| non-IID Level | Level 1 | | Level 2 | | Level 3 | | Level 4 | |
|---|---|---|---|---|---|---|---|---|
| Algorithm | Known A. | Test Phase | Known A. | Test Phase | Known A. | Test Phase | Known A. | Test Phase |
| FedAvg | 33.26 ± 0.21 | 33.26 ± 0.21 | 28.68 ± 1.03 | 28.68 ± 1.03 | 29.97 ± 0.45 | 29.97 ± 0.45 | 28.72 ± 0.69 | 28.72 ± 0.69 |
| IFCA | 39.49 ± 1.30 | 37.27 ± 0.27 | 35.18 ± 2.26 | 32.23 ± 1.24 | 34.82 ± 2.35 | 33.08 ± 1.47 | 34.45 ± 1.62 | 32.31 ± 1.02 |
| FedRC | 28.35 ± 0.67 | 28.33 ± 0.66 | 18.86 ± 3.83 | 18.87 ± 3.83 | 15.64 ± 4.32 | 15.65 ± 4.34 | 11.69 ± 1.16 | 11.68 ± 1.15 |
| FedEM | 28.43 ± 0.79 | 28.38 ± 0.76 | 18.87 ± 3.81 | 18.86 ± 3.81 | 15.66 ± 4.32 | 15.65 ± 4.33 | 11.68 ± 1.16 | 11.69 ± 1.15 |
| FeSEM | 35.07 ± 1.24 | 35.48 ± 0.53 | 31.34 ± 1.70 | 31.57 ± 1.31 | 33.80 ± 1.69 | 31.78 ± 1.86 | 32.04 ± 1.93 | 30.48 ± 1.42 |
| FedDrift | 35.20 ± 1.72 | 33.73 ± 1.24 | 30.54 ± 1.42 | 28.00 ± 2.05 | 30.88 ± 0.29 | 30.88 ± 0.29 | 29.19 ± 0.82 | 29.19 ± 0.82 |
| CFL | 34.11 ± 0.33 | 34.11 ± 0.33 | 29.25 ± 1.03 | 29.25 ± 1.03 | 30.74 ± 0.39 | 30.74 ± 0.39 | 29.21 ± 0.74 | 29.21 ± 0.74 |
| pFedMe | 37.75 ± 0.93 | N/A | 36.88 ± 0.99 | N/A | 37.96 ± 0.43 | N/A | 36.71 ± 0.73 | N/A |
| APFL | **41.02 ± 0.87** | 37.20 ± 1.33 | **39.06 ± 1.02** | 31.51 ± 0.22 | **40.15 ± 1.05** | 33.25 ± 1.60 | **38.85 ± 1.33** | 31.42 ± 0.46 |
| ATP | N/A | 34.07 ± 0.49 | N/A | 30.47 ± 0.95 | N/A | 31.26 ± 1.36 | N/A | 30.06 ± 0.47 |
| FLUX | 36.73 ± 0.56 | 36.32 ± 0.91 | 34.49 ± 0.68 | 34.31 ± 0.72 | 35.74 ± 0.22 | **35.64 ± 0.23** | 34.25 ± 0.34 | 34.48 ± 0.38 |
| FLUX-prior | 37.20 ± 0.29 | 37.20 ± 0.29 | 34.52 ± 1.64 | **34.73 ± 1.33** | 35.51 ± 0.41 | 35.46 ± 0.49 | 34.74 ± 0.39 | **34.74 ± 0.39** |

Table 32: Performance comparison across non-IID Levels 1–4 of $P(X)$ on the CIFAR-10 dataset.
**Known A.**: Known Association.

| non-IID Level | Level 5 | | Level 6 | | Level 7 | | Level 8 | |
|---|---|---|---|---|---|---|---|---|
| Algorithm | Known A. | Test Phase | Known A. | Test Phase | Known A. | Test Phase | Known A. | Test Phase |
| FedAvg | 19.53 ± 0.60 | 19.53 ± 0.60 | 11.76 ± 2.02 | 11.76 ± 2.02 | 12.38 ± 1.80 | 12.38 ± 1.80 | 12.47 ± 2.56 | 12.47 ± 2.56 |
| IFCA | 28.67 ± 0.56 | 20.71 ± 3.27 | 25.62 ± 2.32 | 19.77 ± 1.62 | 30.04 ± 2.36 | 18.74 ± 1.35 | 27.46 ± 1.85 | 17.47 ± 3.23 |
| FedRC | 11.90 ± 1.40 | 11.89 ± 1.38 | 11.32 ± 1.13 | 11.33 ± 1.14 | 10.70 ± 0.89 | 10.70 ± 0.88 | 10.66 ± 0.70 | 10.65 ± 0.69 |
| FedEM | 11.90 ± 1.40 | 11.89 ± 1.39 | 11.32 ± 1.14 | 11.33 ± 1.15 | 10.70 ± 0.89 | 10.70 ± 0.88 | 10.66 ± 0.70 | 10.65 ± 0.69 |
| FeSEM | 23.31 ± 2.51 | 19.84 ± 2.24 | 20.50 ± 1.93 | 15.51 ± 2.43 | 23.72 ± 1.73 | 18.28 ± 1.99 | 21.77 ± 2.52 | 15.97 ± 2.03 |
| FedDrift | 22.44 ± 1.89 | 21.16 ± 1.51 | 16.79 ± 3.44 | 15.57 ± 1.53 | 18.08 ± 3.26 | 16.69 ± 1.88 | 18.23 ± 2.28 | 16.97 ± 1.26 |
| CFL | 20.21 ± 0.95 | 20.21 ± 0.95 | 15.66 ± 1.36 | 15.66 ± 1.36 | 16.43 ± 1.32 | 16.43 ± 1.32 | 16.08 ± 1.06 | 16.08 ± 1.06 |
| pFedMe | **35.53 ± 0.73** | N/A | **34.03 ± 0.80** | N/A | **34.84 ± 0.21** | N/A | **33.96 ± 0.53** | N/A |
| APFL | 33.84 ± 0.91 | 21.43 ± 2.45 | 33.45 ± 1.28 | 18.75 ± 2.38 | 34.08 ± 1.18 | 18.28 ± 3.16 | 33.82 ± 0.52 | 18.89 ± 2.96 |
| ATP | N/A | 22.95 ± 1.56 | N/A | 21.66 ± 0.61 | N/A | 22.28 ± 0.83 | N/A | 21.71 ± 0.98 |
| FLUX | 33.51 ± 1.07 | 33.51 ± 1.07 | 32.52 ± 0.56 | **32.52 ± 0.56** | 30.34 ± 1.81 | 30.34 ± 1.81 | 29.11 ± 2.23 | 29.11 ± 2.23 |
| FLUX-prior | 33.64 ± 0.75 | **33.71 ± 0.64** | 32.32 ± 0.68 | 32.32 ± 0.67 | 32.70 ± 0.98 | **32.66 ± 1.05** | 31.60 ± 0.31 | **31.60 ± 0.31** |

Table 33: Performance comparison across non-IID Levels 5–8 of $P(X)$ on the CIFAR-10 dataset.
**Known A.**: Known Association.

| non-IID Level | Level 1 | | Level 2 | | Level 3 | | Level 4 | |
|---|---|---|---|---|---|---|---|---|
| Algorithm | Known A. | Test Phase | Known A. | Test Phase | Known A. | Test Phase | Known A. | Test Phase |
| FedAvg | 37.74 ± 0.18 | 37.74 ± 0.18 | 38.41 ± 1.12 | 38.41 ± 1.12 | 38.35 ± 1.52 | 38.35 ± 1.52 | 38.81 ± 4.02 | 38.81 ± 4.02 |
| IFCA | 41.31 ± 0.99 | 41.31 ± 0.99 | 43.15 ± 1.43 | 43.29 ± 1.59 | 44.60 ± 2.59 | 43.66 ± 1.74 | 46.69 ± 3.29 | **45.08 ± 3.13** |
| FedRC | 21.25 ± 3.69 | 21.24 ± 3.68 | 22.68 ± 1.62 | 22.69 ± 1.60 | 24.83 ± 4.32 | 24.77 ± 4.36 | 24.83 ± 4.21 | 24.82 ± 4.18 |
| FedEM | 21.26 ± 3.73 | 21.25 ± 3.72 | 22.71 ± 1.47 | 22.70 ± 1.46 | 24.86 ± 4.33 | 24.74 ± 4.37 | 24.93 ± 4.32 | 24.87 ± 4.21 |
| FeSEM | 38.68 ± 0.76 | 40.60 ± 0.78 | 40.55 ± 1.17 | 40.72 ± 1.42 | 41.91 ± 2.07 | 41.36 ± 1.61 | 44.80 ± 4.22 | 42.39 ± 4.76 |
| FedDrift | 39.06 ± 0.24 | 39.06 ± 0.24 | 39.65 ± 1.10 | 39.65 ± 1.10 | 40.01 ± 1.34 | 38.14 ± 3.14 | 43.41 ± 2.76 | 35.75 ± 3.19 |
| CFL | 39.07 ± 0.29 | 39.07 ± 0.29 | 39.91 ± 1.09 | 39.91 ± 1.09 | 39.70 ± 1.32 | 39.70 ± 1.32 | 39.92 ± 4.06 | 39.92 ± 4.06 |
| pFedMe | 37.89 ± 1.17 | N/A | 41.56 ± 1.12 | N/A | 44.27 ± 0.36 | N/A | 48.74 ± 0.49 | N/A |
| APFL | **42.95 ± 0.59** | **42.79 ± 1.55** | **45.93 ± 0.85** | **43.44 ± 1.85** | **47.39 ± 0.40** | 41.27 ± 0.53 | **51.27 ± 0.79** | 43.09 ± 1.32 |
| ATP | N/A | 38.94 ± 0.18 | N/A | 40.86 ± 1.10 | N/A | 40.86 ± 1.08 | N/A | 42.35 ± 3.77 |
| FLUX | 36.24 ± 0.64 | 35.72 ± 0.00 | 40.54 ± 1.66 | 39.27 ± 1.28 | 42.68 ± 1.61 | 39.84 ± 2.56 | 46.76 ± 1.86 | 43.58 ± 2.76 |
| FLUX-prior | 36.66 ± 0.49 | 36.17 ± 0.62 | 40.81 ± 0.61 | 39.46 ± 1.03 | 43.53 ± 0.66 | 39.90 ± 2.26 | 47.59 ± 2.20 | 44.41 ± 2.69 |

Table 34: Performance comparison across non-IID Levels 1–4 of $P(Y)$ on the CIFAR-10 dataset.
**Known A.**: Known Association.

| non-IID Level | Level 5 | | Level 6 | | Level 7 | | Level 8 | |
|---|---|---|---|---|---|---|---|---|
| **Algorithm** | **Known A.** | **Test Phase** | **Known A.** | **Test Phase** | **Known A.** | **Test Phase** | **Known A.** | **Test Phase** |
| FedAvg | 39.30 ± 1.47 | 39.30 ± 1.47 | 41.17 ± 5.75 | 41.17 ± 5.75 | 33.02 ± 5.37 | 33.02 ± 5.37 | 32.69 ± 4.12 | 32.69 ± 4.12 |
| IFCA | 49.60 ± 3.01 | 45.26 ± 2.26 | 52.71 ± 5.05 | 46.73 ± 4.58 | 58.26 ± 5.94 | 42.46 ± 4.40 | 60.29 ± 6.64 | 41.98 ± 4.29 |
| FedRC | 27.15 ± 1.23 | 27.06 ± 1.18 | 29.40 ± 3.23 | 29.38 ± 3.10 | 27.10 ± 1.01 | 27.09 ± 0.99 | 25.83 ± 4.06 | 25.69 ± 4.05 |
| FedEM | 29.07 ± 2.97 | 28.28 ± 2.10 | 31.46 ± 3.19 | 30.05 ± 3.47 | 32.70 ± 2.47 | 28.05 ± 1.31 | 39.32 ± 3.66 | 29.66 ± 3.04 |
| FeSEM | 47.61 ± 2.92 | 47.61 ± 2.92 | 50.81 ± 3.83 | 45.31 ± 5.17 | 44.86 ± 5.15 | 38.43 ± 4.95 | 47.72 ± 10.78 | 39.51 ± 5.13 |
| FedDrift | 51.52 ± 3.19 | 36.94 ± 2.27 | 55.03 ± 3.63 | 34.30 ± 3.87 | 58.81 ± 6.19 | 29.52 ± 5.11 | 64.49 ± 8.93 | 29.60 ± 3.70 |
| CFL | 41.19 ± 0.84 | 41.19 ± 0.84 | 42.07 ± 6.49 | 42.07 ± 6.49 | 34.22 ± 5.80 | 34.22 ± 5.80 | 34.02 ± 4.03 | 34.02 ± 4.03 |
| pFedMe | 53.36 ± 0.06 | N/A | 58.07 ± 1.90 | N/A | 63.35 ± 2.52 | N/A | 72.70 ± 0.40 | N/A |
| APFL | **54.94 ± 0.37** | 44.97 ± 2.37 | **59.43 ± 2.37** | 45.32 ± 4.94 | **65.07 ± 2.72** | 42.75 ± 3.09 | **74.35 ± 1.51** | 44.27 ± 1.71 |
| ATP | N/A | 44.65 ± 1.35 | N/A | 45.32 ± 5.44 | N/A | 39.28 ± 4.88 | N/A | 44.13 ± 6.05 |
| FLUX | 51.59 ± 2.23 | 47.79 ± 2.31 | 54.88 ± 6.30 | **51.72 ± 5.90** | 60.67 ± 4.75 | **55.12 ± 6.71** | 69.75 ± 4.18 | 56.54 ± 10.14 |
| FLUX-prior | 51.65 ± 2.31 | 47.89 ± 2.66 | 56.97 ± 3.82 | 51.39 ± 5.09 | 60.78 ± 4.77 | **55.22 ± 6.65** | 69.84 ± 4.14 | **57.90 ± 10.37** |

Table 35: Performance comparison across non-IID Levels 5–8 of $P(Y)$ on the CIFAR-10 dataset. **Known A.**: Known Association.

| non-IID Level | Level 1 | | Level 2 | | Level 3 | | Level 4 | |
|---|---|---|---|---|---|---|---|---|
| **Algorithm** | **Known A.** | **Test Phase** | **Known A.** | **Test Phase** | **Known A.** | **Test Phase** | **Known A.** | **Test Phase** |
| FedAvg | 37.74 ± 0.18 | N/A | 34.86 ± 1.36 | N/A | 31.39 ± 0.59 | N/A | 28.19 ± 0.70 | N/A |
| IFCA | 37.74 ± 1.17 | N/A | 37.72 ± 2.23 | N/A | 34.29 ± 0.94 | N/A | 33.02 ± 0.96 | N/A |
| FedRC | 37.83 ± 0.18 | N/A | 31.73 ± 1.58 | N/A | 16.08 ± 3.27 | N/A | 10.78 ± 1.53 | N/A |
| FedEM | 37.70 ± 0.18 | N/A | 31.70 ± 1.66 | N/A | 16.11 ± 3.31 | N/A | 10.77 ± 1.52 | N/A |
| FeSEM | 37.77 ± 0.22 | N/A | 37.81 ± 1.03 | N/A | 34.67 ± 0.72 | N/A | 33.81 ± 1.10 | N/A |
| FedDrift | 39.03 ± 0.22 | N/A | 36.04 ± 1.31 | N/A | 32.60 ± 0.76 | N/A | 29.40 ± 0.63 | N/A |
| CFL | 39.04 ± 0.10 | N/A | 36.04 ± 1.42 | N/A | 32.48 ± 0.75 | N/A | 29.38 ± 0.65 | N/A |
| pFedMe | 37.89 ± 1.17 | N/A | 38.05 ± 1.04 | N/A | 37.78 ± 1.05 | N/A | 37.71 ± 1.01 | N/A |
| APFL | **42.95 ± 0.59** | N/A | **42.23 ± 0.39** | N/A | **40.87 ± 0.63** | N/A | **40.51 ± 0.83** | N/A |
| ATP | 38.94 ± 0.18 | N/A | 36.47 ± 1.44 | N/A | 33.03 ± 0.89 | N/A | 29.69 ± 0.97 | N/A |
| FLUX | 36.24 ± 0.73 | N/A | 36.11 ± 0.14 | N/A | 35.13 ± 0.87 | N/A | 31.64 ± 1.01 | N/A |
| FLUX-prior | 37.56 ± 0.35 | N/A | 36.75 ± 0.59 | N/A | 34.78 ± 0.73 | N/A | 34.60 ± 0.70 | N/A |

Table 36: Performance comparison across non-IID Levels 1–4 of $P(Y|X)$ on the CIFAR-10 dataset. **Known A.**: Known Association.

| non-IID Level | Level 5 | | Level 6 | | Level 7 | | Level 8 | |
|---|---|---|---|---|---|---|---|---|
| **Algorithm** | **Known A.** | **Test Phase** | **Known A.** | **Test Phase** | **Known A.** | **Test Phase** | **Known A.** | **Test Phase** |
| FedAvg | 26.65 ± 1.24 | N/A | 23.77 ± 1.45 | N/A | 21.81 ± 1.17 | N/A | 19.87 ± 1.28 | N/A |
| IFCA | 30.02 ± 2.64 | N/A | 29.23 ± 2.34 | N/A | 27.12 ± 2.26 | N/A | 26.50 ± 0.86 | N/A |
| FedRC | 11.43 ± 1.57 | N/A | 10.42 ± 0.86 | N/A | 11.17 ± 1.38 | N/A | 11.15 ± 0.96 | N/A |
| FedEM | 11.43 ± 1.57 | N/A | 10.42 ± 0.87 | N/A | 11.17 ± 1.38 | N/A | 11.16 ± 0.95 | N/A |
| FeSEM | 33.05 ± 1.02 | N/A | 32.62 ± 1.04 | N/A | 29.40 ± 2.26 | N/A | 28.75 ± 1.34 | N/A |
| FedDrift | 27.61 ± 1.29 | N/A | 24.60 ± 1.57 | N/A | 22.40 ± 1.18 | N/A | 20.35 ± 1.41 | N/A |
| CFL | 27.45 ± 1.39 | N/A | 24.50 ± 1.68 | N/A | 22.36 ± 1.23 | N/A | 20.29 ± 1.39 | N/A |
| pFedMe | 38.57 ± 0.94 | N/A | 38.36 ± 1.24 | N/A | 38.33 ± 1.19 | N/A | 37.67 ± 1.40 | N/A |
| APFL | **39.76 ± 1.43** | N/A | **39.09 ± 1.45** | N/A | **39.08 ± 1.55** | N/A | **38.81 ± 1.64** | N/A |
| ATP | 27.53 ± 1.06 | N/A | 25.06 ± 1.68 | N/A | 22.97 ± 1.09 | N/A | 21.26 ± 1.45 | N/A |
| FLUX | 32.55 ± 0.67 | N/A | 26.55 ± 1.05 | N/A | 28.72 ± 2.27 | N/A | 26.86 ± 3.86 | N/A |
| FLUX-prior | 34.07 ± 0.53 | N/A | 33.78 ± 0.73 | N/A | 33.71 ± 0.78 | N/A | 33.35 ± 0.58 | N/A |

Table 37: Performance comparison across non-IID Levels 5–8 of $P(X)$ on the CIFAR-10 dataset. **Known A.**: Known Association.

| non-IID Level | Level 1 | | Level 2 | | Level 3 | | Level 4 | |
|---|---|---|---|---|---|---|---|---|
| Algorithm | Known A. | Test Phase | Known A. | Test Phase | Known A. | Test Phase | Known A. | Test Phase |
| FedAvg | 38.00 ± 0.16 | 38.00 ± 0.16 | 37.67 ± 0.73 | 37.67 ± 0.73 | 37.06 ± 1.77 | 37.06 ± 1.77 | 36.90 ± 1.90 | 36.90 ± 1.90 |
| IFCA | 41.50 ± 1.33 | 41.31 ± 1.99 | 41.43 ± 0.76 | 41.46 ± 0.57 | 40.54 ± 1.22 | 40.53 ± 0.88 | 40.46 ± 1.93 | 40.26 ± 1.73 |
| FedRC | 12.10 ± 1.21 | 11.99 ± 1.99 | 11.90 ± 1.36 | 11.91 ± 1.36 | 11.26 ± 1.49 | 11.26 ± 1.50 | 11.33 ± 1.62 | 11.32 ± 1.61 |
| FedEM | 11.98 ± 1.28 | 11.68 ± 1.21 | 11.90 ± 1.37 | 11.89 ± 1.37 | 11.26 ± 1.49 | 11.26 ± 1.49 | 11.33 ± 1.62 | 11.32 ± 1.61 |
| FeSEM | 39.07 ± 0.78 | 38.91 ± 0.99 | 39.21 ± 0.85 | 40.26 ± 0.56 | 39.59 ± 0.99 | 40.40 ± 0.58 | 41.07 ± 0.91 | 40.17 ± 1.52 |
| FedDrift | 39.05 ± 0.13 | 39.02 ± 0.99 | 38.89 ± 0.79 | 38.89 ± 0.79 | 38.09 ± 1.46 | 38.09 ± 1.46 | 37.91 ± 1.85 | 37.52 ± 1.75 |
| CFL | 39.10 ± 0.50 | 39.00 ± 0.96 | 38.96 ± 0.79 | 38.96 ± 0.79 | 38.15 ± 1.56 | 38.15 ± 1.56 | 37.90 ± 1.89 | 37.90 ± 1.89 |
| pFedMe | 39.58 ± 1.01 | N/A | 40.95 ± 0.38 | N/A | 41.62 ± 1.50 | N/A | 43.68 ± 1.24 | N/A |
| APFL | **43.93 ± 0.42** | **43.41 ± 0.82** | **44.81 ± 0.36** | **41.86 ± 1.18** | **45.38 ± 0.74** | **40.99 ± 0.41** | **47.13 ± 0.29** | **41.12 ± 1.75** |
| ATP | N/A | 39.19 ± 1.21 | N/A | 39.28 ± 0.73 | N/A | 38.66 ± 1.03 | N/A | 38.47 ± 1.36 |
| FLUX | 37.40 ± 1.28 | 36.53 ± 1.81 | 38.81 ± 0.98 | 37.58 ± 1.24 | 39.70 ± 1.03 | 37.59 ± 1.72 | 38.65 ± 2.62 | 37.00 ± 1.23 |
| FLUX-prior | 38.15 ± 1.51 | 37.99 ± 1.50 | 38.68 ± 1.27 | 37.32 ± 1.74 | 38.95 ± 1.85 | 36.60 ± 1.53 | 41.16 ± 1.09 | 36.88 ± 1.83 |

Table 38: Performance comparison across non-IID Levels 1–4 of $P(X|Y)$ on the CIFAR-10 dataset. **Known A.**: Known Association.

| non-IID Level | Level 5 | | Level 6 | | Level 7 | | Level 8 | |
|---|---|---|---|---|---|---|---|---|
| Algorithm | Known A. | Test Phase | Known A. | Test Phase | Known A. | Test Phase | Known A. | Test Phase |
| FedAvg | 36.01 ± 1.17 | 36.01 ± 1.17 | 35.58 ± 0.89 | 35.58 ± 0.89 | 34.00 ± 0.87 | 34.00 ± 0.87 | 33.10 ± 1.29 | 33.10 ± 1.29 |
| IFCA | 40.02 ± 0.72 | 39.61 ± 1.11 | 38.99 ± 1.66 | 38.74 ± 0.80 | 38.08 ± 1.35 | 37.41 ± 0.73 | 37.33 ± 1.01 | 36.45 ± 0.68 |
| FedRC | 10.92 ± 0.98 | 10.90 ± 0.95 | 11.17 ± 1.32 | 11.16 ± 1.31 | 11.31 ± 1.45 | 11.33 ± 1.47 | 11.40 ± 1.55 | 11.43 ± 1.58 |
| FedEM | 10.92 ± 0.99 | 10.90 ± 0.96 | 11.18 ± 1.33 | 11.16 ± 1.31 | 11.31 ± 1.46 | 11.33 ± 1.48 | 11.39 ± 1.55 | 11.43 ± 1.58 |
| FeSEM | 41.07 ± 1.04 | 39.48 ± 1.51 | 40.37 ± 1.32 | 38.44 ± 0.75 | 40.00 ± 1.01 | 37.42 ± 0.71 | 40.21 ± 1.16 | 36.39 ± 0.94 |
| FedDrift | 37.22 ± 1.21 | 37.22 ± 1.21 | 36.59 ± 1.25 | 36.17 ± 0.75 | 34.62 ± 1.01 | 34.05 ± 1.76 | 34.02 ± 0.78 | 33.16 ± 1.93 |
| CFL | 37.20 ± 1.19 | 37.20 ± 1.19 | 36.35 ± 0.79 | 36.35 ± 0.79 | 34.62 ± 1.05 | 34.62 ± 1.05 | 33.39 ± 1.33 | 33.39 ± 1.33 |
| pFedMe | 44.64 ± 1.03 | N/A | 44.65 ± 1.52 | N/A | 45.44 ± 1.00 | N/A | 45.02 ± 0.76 | N/A |
| APFL | **47.48 ± 0.17** | **39.63 ± 2.36** | **47.25 ± 0.51** | **39.25 ± 1.23** | **47.66 ± 0.71** | 37.50 ± 1.79 | **46.81 ± 0.90** | 34.82 ± 0.16 |
| ATP | N/A | 37.29 ± 1.02 | N/A | 36.89 ± 0.68 | N/A | 35.39 ± 1.04 | N/A | 34.63 ± 1.40 |
| FLUX | 41.14 ± 1.17 | 36.09 ± 2.50 | 41.43 ± 1.81 | 37.45 ± 1.92 | 39.38 ± 2.26 | 37.50 ± 2.90 | 37.65 ± 3.63 | 33.83 ± 3.81 |
| FLUX-prior | 41.65 ± 0.77 | 36.59 ± 2.48 | 41.46 ± 0.77 | 38.77 ± 1.17 | 41.66 ± 0.67 | **37.67 ± 1.02** | 42.35 ± 0.82 | **36.59 ± 2.08** |

Table 39: Performance comparison across non-IID Levels 5–8 of $P(X|Y)$ on the CIFAR-10 dataset. **Known A.**: Known Association.

| non-IID Level | Level 1 | | Level 2 | | Level 3 | | Level 4 | |
|---|---|---|---|---|---|---|---|---|
| Algorithm | Known A. | Test Phase | Known A. | Test Phase | Known A. | Test Phase | Known A. | Test Phase |
| FedAvg | 48.30 ± 0.74 | 51.06 ± 0.76 | 42.82 ± 1.24 | 46.02 ± 1.31 | 42.01 ± 1.00 | 46.69 ± 1.06 | 37.40 ± 1.63 | 42.48 ± 1.39 |
| IFCA | 48.93 ± 0.87 | 51.85 ± 0.58 | 42.58 ± 2.63 | 44.08 ± 1.41 | 40.19 ± 3.24 | 43.99 ± 1.07 | 34.52 ± 2.73 | 40.54 ± 0.90 |
| FedRC | 45.96 ± 0.85 | 49.98 ± 0.52 | 41.47 ± 0.97 | 45.26 ± 1.03 | 40.13 ± 0.93 | 44.90 ± 0.94 | 36.21 ± 1.44 | 40.80 ± 1.34 |
| FedEM | 46.60 ± 0.65 | 50.04 ± 0.63 | 42.15 ± 0.91 | 45.30 ± 1.00 | 40.81 ± 1.11 | 44.96 ± 1.12 | 37.11 ± 1.21 | 41.42 ± 1.15 |
| FeSEM | 44.27 ± 0.88 | 50.86 ± 0.44 | 41.34 ± 1.90 | 45.97 ± 1.54 | 41.05 ± 1.63 | 46.85 ± 1.12 | 37.27 ± 1.74 | 43.08 ± 1.78 |
| FedDrift | 45.33 ± 2.90 | 41.16 ± 2.48 | 42.42 ± 2.79 | 36.28 ± 2.10 | 42.00 ± 0.83 | 35.08 ± 1.55 | 38.62 ± 1.99 | 28.14 ± 1.20 |
| CFL | 47.79 ± 0.42 | 50.47 ± 0.40 | 43.02 ± 0.87 | 45.61 ± 0.87 | 42.41 ± 0.88 | 46.42 ± 0.80 | 38.30 ± 1.08 | 42.40 ± 0.82 |
| pFedMe | 33.30 ± 0.33 | N/A | 34.41 ± 0.45 | N/A | 34.53 ± 2.95 | N/A | 34.43 ± 4.81 | N/A |
| APFL | **49.64 ± 0.33** | 33.65 ± 0.32 | **47.73 ± 0.68** | 34.72 ± 0.37 | **47.83 ± 1.05** | 36.63 ± 0.77 | **44.28 ± 4.50** | 36.50 ± 1.74 |
| ATP | N/A | **55.88 ± 0.43** | N/A | **51.19 ± 0.82** | N/A | **52.47 ± 0.91** | N/A | **48.42 ± 1.01** |
| FLUX | 42.85 ± 1.33 | 43.29 ± 3.59 | 40.46 ± 1.38 | 39.63 ± 1.60 | 42.07 ± 0.81 | 40.43 ± 2.72 | 40.88 ± 0.99 | 39.24 ± 2.87 |
| FLUX-prior | 43.36 ± 0.64 | 43.02 ± 3.40 | 40.99 ± 2.00 | 39.05 ± 1.28 | 42.23 ± 0.89 | 41.60 ± 2.14 | 41.76 ± 1.29 | 38.95 ± 3.39 |

Table 40: Performance comparison across non-IID Levels 1–4, summarizing all four types of heterogeneity ($P(X)$, $P(Y)$, $P(Y|X)$, $P(X|Y)$) on the CIFAR-100 dataset. **Known A.**: Known Association.

| non-IID Level | Level 5 | | Level 6 | | Level 7 | | Level 8 | |
|---|---|---|---|---|---|---|---|---|
| Algorithm | Known A. | Test Phase | Known A. | Test Phase | Known A. | Test Phase | Known A. | Test Phase |
| FedAvg | 29.86 ± 4.39 | 33.76 ± 5.05 | 26.24 ± 0.62 | 30.32 ± 0.70 | 23.34 ± 0.87 | 28.09 ± 0.90 | 20.36 ± 2.69 | 25.63 ± 3.10 |
| IFCA | 35.80 ± 7.25 | 38.00 ± 5.32 | 27.15 ± 4.28 | 30.36 ± 4.15 | 31.52 ± 4.92 | 31.53 ± 1.81 | 28.06 ± 3.91 | 28.30 ± 3.88 |
| FedRC | 31.85 ± 1.76 | 34.28 ± 1.52 | 26.19 ± 1.57 | 30.17 ± 1.47 | 25.82 ± 0.87 | 29.57 ± 0.98 | 22.92 ± 0.81 | 28.08 ± 1.05 |
| FedEM | 31.93 ± 1.29 | 34.42 ± 1.45 | 28.12 ± 1.26 | 30.49 ± 1.46 | 27.48 ± 1.01 | 29.68 ± 0.87 | 24.60 ± 1.02 | 27.33 ± 0.86 |
| FeSEM | 33.74 ± 2.72 | 37.77 ± 2.72 | 29.94 ± 2.74 | 32.34 ± 2.61 | 29.68 ± 2.54 | 32.78 ± 1.53 | 27.62 ± 3.79 | 28.88 ± 1.33 |
| FedDrift | 38.51 ± 3.28 | 22.80 ± 1.47 | 36.28 ± 2.87 | 20.49 ± 2.81 | 38.33 ± 2.41 | 18.76 ± 0.94 | 35.88 ± 5.69 | 15.69 ± 1.15 |
| CFL | 31.85 ± 2.26 | 35.61 ± 2.56 | 27.31 ± 0.73 | 31.28 ± 0.81 | 24.80 ± 0.40 | 29.81 ± 0.38 | 21.47 ± 3.02 | 27.12 ± 3.48 |
| pFedMe | 37.38 ± 0.78 | N/A | 37.54 ± 0.76 | N/A | 38.67 ± 0.50 | N/A | 38.34 ± 0.95 | N/A |
| APFL | 42.59 ± 0.98 | 38.03 ± 0.73 | 41.15 ± 0.66 | 38.67 ± 0.45 | 40.78 ± 0.39 | 39.71 ± 0.49 | 39.95 ± 0.57 | 40.32 ± 0.52 |
| ATP | N/A | 43.99 ± 2.33 | N/A | 38.14 ± 1.15 | N/A | 38.34 ± 0.59 | N/A | 35.63 ± 1.86 |
| FLUX | 43.71 ± 1.56 | 43.09 ± 2.43 | 41.15 ± 1.78 | 39.26 ± 3.74 | 41.21 ± 3.57 | **42.42 ± 3.28** | 41.12 ± 2.70 | 42.89 ± 3.77 |
| FLUX-prior | **43.97 ± 1.49** | 42.80 ± 2.35 | **42.84 ± 1.33** | **39.59 ± 4.60** | **43.58 ± 0.53** | 42.37 ± 3.56 | **43.31 ± 0.76** | **43.11 ± 3.89** |

Table 41: Performance comparison across non-IID Levels 5–8, summarizing all four types of heterogeneity ($P(X)$, $P(Y)$, $P(Y|X)$, $P(X|Y)$) on the CIFAR-100 dataset. **Known A.**: Known Association.

| non-IID Level | Level 1 | | Level 2 | | Level 3 | | Level 4 | |
|---|---|---|---|---|---|---|---|---|
| Algorithm | Known A. | Test Phase | Known A. | Test Phase | Known A. | Test Phase | Known A. | Test Phase |
| FedAvg | 45.55 ± 0.59 | 45.55 ± 0.59 | 31.99 ± 2.09 | 31.99 ± 2.09 | 38.48 ± 1.34 | 38.48 ± 1.34 | 30.03 ± 2.19 | 30.03 ± 2.19 |
| IFCA | 46.16 ± 0.63 | 46.16 ± 0.63 | 35.24 ± 3.06 | 33.63 ± 1.44 | 39.95 ± 1.87 | 39.04 ± 1.47 | 30.69 ± 1.22 | 30.69 ± 1.22 |
| FedRC | 47.51 ± 0.59 | 47.45 ± 0.49 | 36.61 ± 1.43 | 36.25 ± 1.66 | 39.85 ± 1.49 | 39.78 ± 1.53 | 32.88 ± 1.58 | 32.76 ± 1.54 |
| FedEM | 47.34 ± 0.77 | 47.37 ± 0.70 | 36.71 ± 1.30 | 36.33 ± 1.48 | 39.85 ± 1.83 | 39.71 ± 1.83 | 32.89 ± 1.73 | 32.82 ± 1.73 |
| FeSEM | 42.94 ± 0.35 | 46.39 ± 0.56 | 34.24 ± 2.42 | 34.24 ± 2.42 | 38.31 ± 1.07 | 39.35 ± 1.36 | 31.12 ± 0.73 | 32.79 ± 1.63 |
| FedDrift | 44.27 ± 4.37 | 31.58 ± 1.37 | 36.34 ± 4.91 | 22.65 ± 1.58 | 39.60 ± 0.78 | 28.65 ± 1.61 | 32.51 ± 3.72 | 22.61 ± 1.18 |
| CFL | 44.75 ± 0.50 | 44.75 ± 0.50 | 32.38 ± 1.34 | 32.38 ± 1.34 | 38.17 ± 0.94 | 38.17 ± 0.94 | 30.01 ± 0.85 | 30.01 ± 0.85 |
| pFedMe | 31.32 ± 0.06 | N/A | 29.85 ± 0.23 | N/A | 31.06 ± 0.05 | N/A | 28.68 ± 0.34 | N/A |
| APFL | 46.94 ± 0.17 | 30.81 ± 0.23 | **38.81 ± 0.86** | 29.24 ± 0.14 | **42.44 ± 0.59** | 30.74 ± 0.08 | **37.04 ± 0.73** | 28.70 ± 0.43 |
| ATP | N/A | 49.52 ± 0.31 | N/A | 36.84 ± 1.21 | N/A | 43.41 ± 1.16 | N/A | 34.37 ± 1.41 |
| FLUX | 48.56 ± 2.00 | 45.16 ± 5.20 | 35.64 ± 2.13 | 35.39 ± 2.17 | 40.54 ± 1.07 | 36.18 ± 3.32 | 34.42 ± 0.83 | 30.06 ± 2.71 |
| FLUX-prior | **49.60 ± 0.47** | 45.48 ± 5.43 | 37.92 ± 3.72 | 33.31 ± 1.76 | 40.50 ± 1.55 | 39.23 ± 1.81 | 34.80 ± 2.25 | 30.14 ± 3.40 |

Table 42: Performance comparison across non-IID Levels 1–4 of $P(X)$ on the CIFAR-100 dataset. **Known A.**: Known Association.

| non-IID Level | Level 5 | | Level 6 | | Level 7 | | Level 8 | |
|---|---|---|---|---|---|---|---|---|
| Algorithm | Known A. | Test Phase | Known A. | Test Phase | Known A. | Test Phase | Known A. | Test Phase |
| FedAvg | 7.49 ± 8.71 | 7.49 ± 8.71 | 1.27 ± 0.13 | 1.27 ± 0.13 | 1.26 ± 0.10 | 1.26 ± 0.10 | 1.21 ± 0.13 | 1.21 ± 0.13 |
| IFCA | 25.80 ± 13.11 | 24.35 ± 9.08 | 5.43 ± 6.26 | 5.65 ± 6.58 | 19.64 ± 8.22 | 15.51 ± 2.96 | 9.71 ± 6.18 | 9.83 ± 6.26 |
| FedRC | 17.68 ± 2.97 | 14.12 ± 2.41 | 7.92 ± 1.22 | 5.67 ± 1.00 | 12.11 ± 1.57 | 8.22 ± 1.54 | 7.02 ± 0.17 | 5.36 ± 0.85 |
| FedEM | 17.50 ± 2.25 | 14.11 ± 2.29 | 8.08 ± 1.39 | 5.82 ± 1.07 | 11.95 ± 1.46 | 7.94 ± 1.38 | 7.44 ± 0.34 | 5.46 ± 0.78 |
| FeSEM | 18.90 ± 4.68 | 18.46 ± 4.59 | 13.02 ± 3.33 | 9.89 ± 2.76 | 18.62 ± 4.03 | 13.92 ± 1.83 | 12.24 ± 1.19 | 7.70 ± 0.82 |
| FedDrift | 34.61 ± 2.33 | 11.90 ± 1.55 | 23.45 ± 3.07 | 8.35 ± 0.28 | 30.33 ± 0.34 | 10.18 ± 1.02 | 19.17 ± 9.52 | 7.66 ± 1.45 |
| CFL | 13.22 ± 4.39 | 13.22 ± 4.39 | 4.00 ± 0.11 | 4.00 ± 0.11 | 6.35 ± 0.20 | 6.35 ± 0.20 | 3.63 ± 0.27 | 3.63 ± 0.27 |
| pFedMe | 27.98 ± 0.31 | N/A | 26.11 ± 0.56 | N/A | 28.07 ± 0.24 | N/A | 25.77 ± 0.44 | N/A |
| APFL | 27.50 ± 1.21 | 27.38 ± 0.32 | 22.99 ± 0.31 | 26.02 ± 0.21 | 24.55 ± 0.52 | 27.43 ± 0.28 | 22.55 ± 0.51 | 25.73 ± 0.27 |
| ATP | N/A | 23.75 ± 3.90 | N/A | 7.86 ± 1.61 | N/A | 12.86 ± 0.54 | N/A | 8.00 ± 1.08 |
| FLUX | **36.68 ± 2.72** | **35.52 ± 3.19** | 29.50 ± 2.33 | 28.80 ± 2.74 | **31.23 ± 0.93** | 31.24 ± 0.89 | 27.98 ± 0.97 | 27.87 ± 1.10 |
| FLUX-prior | 36.61 ± 2.82 | 34.56 ± 3.33 | **32.15 ± 1.73** | **30.51 ± 4.04** | 31.22 ± 0.60 | **32.27 ± 0.51** | **28.43 ± 0.91** | **28.33 ± 1.09** |

Table 43: Performance comparison across non-IID Levels 5–8 of $P(X)$ on the CIFAR-100 dataset. **Known A.**: Known Association.

| non-IID Level | Level 1 | | Level 2 | | Level 3 | | Level 4 | |
|---|---|---|---|---|---|---|---|---|
| Algorithm | Known A. | Test Phase | Known A. | Test Phase | Known A. | Test Phase | Known A. | Test Phase |
| FedAvg | 53.31 ± 0.14 | 53.31 ± 0.14 | 52.09 ± 0.60 | 52.09 ± 0.60 | **50.39 ± 0.51** | 50.39 ± 0.51 | **49.14 ± 0.87** | 49.14 ± 0.87 |
| IFCA | **53.76 ± 0.56** | **53.76 ± 0.56** | 46.94 ± 2.08 | 46.84 ± 0.93 | 39.83 ± 6.00 | 42.60 ± 1.05 | 42.40 ± 2.72 | 44.11 ± 0.94 |
| FedRC | 52.72 ± 0.62 | 52.72 ± 0.58 | 50.25 ± 0.29 | 50.31 ± 0.33 | 47.55 ± 0.25 | 47.50 ± 0.26 | 44.52 ± 1.75 | 44.48 ± 1.71 |
| FedEM | 52.92 ± 0.66 | 52.99 ± 0.68 | 50.22 ± 0.34 | 50.32 ± 0.35 | 47.59 ± 0.19 | 47.56 ± 0.20 | 46.21 ± 0.85 | 46.27 ± 0.95 |
| FeSEM | 46.59 ± 0.96 | 53.54 ± 0.15 | 46.32 ± 2.47 | 51.86 ± 0.97 | 45.39 ± 2.78 | 49.97 ± 1.20 | 41.89 ± 3.23 | 47.28 ± 2.57 |
| FedDrift | 53.31 ± 0.14 | 53.31 ± 0.14 | 51.57 ± 0.45 | 50.16 ± 1.85 | 49.91 ± 0.90 | 46.67 ± 1.94 | 45.28 ± 0.72 | 34.31 ± 0.95 |
| CFL | 52.65 ± 0.15 | 52.65 ± 0.15 | 51.10 ± 0.61 | 51.10 ± 0.61 | 49.81 ± 0.51 | 49.80 ± 0.51 | 49.11 ± 1.12 | 49.11 ± 1.12 |
| pFedMe | 31.27 ± 0.16 | N/A | 31.93 ± 0.45 | N/A | 28.29 ± 5.88 | N/A | 26.90 ± 9.59 | N/A |
| APFL | 52.75 ± 0.21 | 30.82 ± 0.18 | **52.11 ± 0.46** | 31.70 ± 0.29 | 49.59 ± 1.94 | 32.25 ± 1.28 | 42.83 ± 8.95 | 31.21 ± 2.96 |
| ATP | N/A | 58.08 ± 0.33 | N/A | **57.70 ± 0.55** | N/A | **57.24 ± 0.41** | N/A | **57.38 ± 0.98** |
| FLUX | 44.90 ± 1.25 | 42.92 ± 1.17 | 45.07 ± 0.84 | 39.60 ± 0.29 | 44.85 ± 0.76 | 37.48 ± 2.37 | 46.03 ± 1.32 | 37.73 ± 3.86 |
| FLUX-prior | 45.01 ± 1.11 | 41.27 ± 1.07 | 44.80 ± 0.98 | 39.51 ± 0.29 | 44.82 ± 0.71 | 37.83 ± 2.24 | 46.00 ± 1.19 | 37.80 ± 3.83 |

Table 44: Performance comparison across non-IID Levels 1–4 of $P(Y)$ on the CIFAR-100 dataset.
**Known A.**: Known Association.

| non-IID Level | Level 5 | | Level 6 | | Level 7 | | Level 8 | |
|---|---|---|---|---|---|---|---|---|
| Algorithm | Known A. | Test Phase | Known A. | Test Phase | Known A. | Test Phase | Known A. | Test Phase |
| FedAvg | 49.23 ± 0.66 | 49.23 ± 0.66 | 48.18 ± 1.07 | 48.18 ± 1.07 | 45.94 ± 1.49 | 45.94 ± 1.49 | 42.70 ± 5.32 | 42.70 ± 5.32 |
| IFCA | 47.82 ± 3.10 | 46.22 ± 1.53 | 45.42 ± 4.75 | 45.01 ± 2.86 | 46.62 ± 4.91 | 42.35 ± 0.56 | 46.93 ± 4.12 | 41.92 ± 2.45 |
| FedRC | 46.34 ± 0.87 | 46.25 ± 0.98 | 44.98 ± 2.09 | 44.72 ± 2.31 | 44.44 ± 0.52 | 44.11 ± 0.71 | 45.99 ± 1.53 | 45.95 ± 1.58 |
| FedEM | 48.06 ± 0.46 | 46.70 ± 0.97 | 49.23 ± 2.05 | 45.46 ± 2.26 | 51.28 ± 1.32 | 44.72 ± 0.56 | 52.31 ± 1.92 | 43.61 ± 1.24 |
| FeSEM | 46.46 ± 2.57 | 49.37 ± 0.95 | 43.38 ± 3.85 | 44.93 ± 3.53 | 46.11 ± 2.21 | 46.62 ± 1.74 | 44.62 ± 2.29 | 45.84 ± 2.00 |
| FedDrift | 48.30 ± 0.93 | 32.16 ± 1.87 | 48.77 ± 3.04 | 30.10 ± 4.66 | 49.57 ± 2.19 | 26.50 ± 1.25 | 51.81 ± 3.56 | 22.06 ± 1.02 |
| CFL | 48.65 ± 0.55 | 48.65 ± 0.55 | 48.27 ± 1.15 | 48.27 ± 1.15 | 46.77 ± 0.50 | 46.77 ± 0.50 | 45.02 ± 6.01 | 45.02 ± 6.01 |
| pFedMe | 37.10 ± 1.52 | N/A | 38.13 ± 1.31 | N/A | 40.12 ± 0.94 | N/A | 40.97 ± 1.76 | N/A |
| APFL | 49.24 ± 1.42 | 36.50 ± 1.20 | 49.08 ± 1.18 | 38.27 ± 0.73 | 49.01 ± 0.45 | 41.47 ± 0.73 | 49.17 ± 0.96 | 43.32 ± 0.75 |
| ATP | N/A | **58.28 ± 1.00** | N/A | **60.24 ± 1.15** | N/A | **61.15 ± 0.63** | N/A | **61.68 ± 2.97** |
| FLUX | **51.87 ± 1.06** | 39.45 ± 2.27 | 50.53 ± 2.10 | 37.64 ± 5.72 | 53.69 ± 1.02 | 43.22 ± 4.69 | **56.33 ± 0.92** | 48.31 ± 3.34 |
| FLUX-prior | 51.65 ± 0.92 | 39.65 ± 1.86 | **50.86 ± 1.89** | 36.96 ± 6.76 | **54.73 ± 0.85** | 43.58 ± 5.04 | 56.32 ± 1.12 | 48.36 ± 3.73 |

Table 45: Performance comparison across non-IID Levels 5–8 of $P(Y)$ on the CIFAR-100 dataset.
**Known A.**: Known Association.

| non-IID Level | Level 1 | | Level 2 | | Level 3 | | Level 4 | |
|---|---|---|---|---|---|---|---|---|
| Algorithm | Known A. | Test Phase | Known A. | Test Phase | Known A. | Test Phase | Known A. | Test Phase |
| FedAvg | 40.04 ± 0.71 | N/A | 33.21 ± 1.04 | N/A | 27.97 ± 0.77 | N/A | 22.16 ± 2.18 | N/A |
| IFCA | 40.16 ± 1.43 | N/A | 35.36 ± 2.06 | N/A | 30.39 ± 1.44 | N/A | 15.68 ± 4.56 | N/A |
| FedRC | 33.83 ± 1.35 | N/A | 29.79 ± 1.15 | N/A | 25.57 ± 0.93 | N/A | 22.12 ± 1.62 | N/A |
| FedEM | 36.32 ± 0.65 | N/A | 32.44 ± 0.92 | N/A | 28.21 ± 1.16 | N/A | 24.03 ± 1.43 | N/A |
| FeSEM | 37.29 ± 1.42 | N/A | 34.73 ± 1.27 | N/A | 30.33 ± 1.27 | N/A | 26.53 ± 0.96 | N/A |
| FedDrift | 37.73 ± 0.40 | N/A | 34.76 ± 1.30 | N/A | 31.43 ± 0.55 | N/A | 27.01 ± 0.69 | N/A |
| CFL | 39.77 ± 0.47 | N/A | 35.25 ± 0.84 | N/A | 30.38 ± 1.09 | N/A | 26.00 ± 1.62 | N/A |
| pFedMe | 30.94 ± 0.07 | N/A | 31.15 ± 0.31 | N/A | 30.58 ± 0.33 | N/A | 30.87 ± 0.13 | N/A |
| APFL | **43.34 ± 0.36** | N/A | **42.58 ± 0.54** | N/A | **39.84 ± 0.33** | N/A | **37.29 ± 0.33** | N/A |
| ATP | 43.21 ± 0.73 | N/A | 38.10 ± 0.91 | N/A | 32.98 ± 1.21 | N/A | 28.20 ± 1.24 | N/A |
| FLUX | 34.05 ± 0.25 | N/A | 33.28 ± 0.48 | N/A | 32.37 ± 0.88 | N/A | 29.58 ± 1.17 | N/A |
| FLUX-prior | 34.31 ± 0.09 | N/A | 33.04 ± 0.11 | N/A | 33.01 ± 0.24 | N/A | 32.88 ± 0.16 | N/A |

Table 46: Performance comparison across non-IID Levels 1–4 of $P(Y|X)$ on the CIFAR-100 dataset.
**Known A.**: Known Association.

| non-IID Level | Level 5 | | Level 6 | | Level 7 | | Level 8 | |
|---|---|---|---|---|---|---|---|---|
| Algorithm | Known A. | Test Phase | Known A. | Test Phase | Known A. | Test Phase | Known A. | Test Phase |
| FedAvg | 18.16 ± 0.86 | N/A | 14.01 ± 0.24 | N/A | 9.06 ± 0.74 | N/A | 4.53 ± 0.09 | N/A |
| IFCA | 18.89 ± 5.27 | N/A | 7.62 ± 2.78 | N/A | 10.68 ± 0.10 | N/A | 9.05 ± 1.40 | N/A |
| FedRC | 20.72 ± 1.63 | N/A | 11.22 ± 2.01 | N/A | 9.54 ± 0.42 | N/A | 4.72 ± 0.17 | N/A |
| FedEM | 19.46 ± 1.14 | N/A | 14.56 ± 0.50 | N/A | 9.51 ± 0.43 | N/A | 4.75 ± 0.23 | N/A |
| FeSEM | 22.99 ± 0.93 | N/A | 18.27 ± 1.20 | N/A | 14.44 ± 1.93 | N/A | 12.96 ± 4.63 | N/A |
| FedDrift | 22.28 ± 3.39 | N/A | 23.28 ± 2.74 | N/A | 23.16 ± 2.92 | N/A | 24.23 ± 1.58 | N/A |
| CFL | 20.56 ± 0.95 | N/A | 15.40 ± 0.38 | N/A | 9.78 ± 0.46 | N/A | 4.51 ± 0.27 | N/A |
| pFedMe | 30.67 ± 0.06 | N/A | 31.03 ± 0.21 | N/A | 30.95 ± 0.19 | N/A | 30.94 ± 0.24 | N/A |
| APFL | **33.04 ± 0.21** | N/A | **32.60 ± 0.24** | N/A | 31.08 ± 0.05 | N/A | 30.08 ± 0.22 | N/A |
| ATP | 22.85 ± 1.07 | N/A | 17.16 ± 0.60 | N/A | 12.04 ± 0.52 | N/A | 7.11 ± 0.36 | N/A |
| FLUX | 31.01 ± 1.07 | N/A | 28.39 ± 1.54 | N/A | 23.37 ± 7.00 | N/A | 23.54 ± 5.21 | N/A |
| FLUX-prior | 32.50 ± 0.30 | N/A | 32.31 ± 0.36 | N/A | **32.01 ± 0.12** | N/A | **31.73 ± 0.33** | N/A |

Table 47: Performance comparison across non-IID Levels 5–8 of $P(Y|X)$ on the CIFAR-100 dataset. **Known A.**: Known Association.

| non-IID Level | Level 1 | | Level 2 | | Level 3 | | Level 4 | |
|---|---|---|---|---|---|---|---|---|
| Algorithm | Known A. | Test Phase | Known A. | Test Phase | Known A. | Test Phase | Known A. | Test Phase |
| FedAvg | 54.31 ± 1.16 | 54.31 ± 1.16 | 53.96 ± 0.62 | 53.96 ± 0.62 | 51.21 ± 1.15 | 51.21 ± 1.15 | 48.27 ± 0.53 | 48.27 ± 0.53 |
| IFCA | **55.62 ± 0.53** | 55.62 ± 0.53 | 52.78 ± 3.13 | 51.77 ± 1.75 | 50.60 ± 0.56 | 50.35 ± 0.40 | 49.31 ± 0.35 | 46.82 ± 0.26 |
| FedRC | 49.78 ± 0.56 | 49.78 ± 0.46 | 49.21 ± 0.56 | 49.22 ± 0.46 | 47.55 ± 0.52 | 47.43 ± 0.53 | 45.34 ± 0.32 | 45.16 ± 0.25 |
| FedEM | 49.84 ± 0.50 | 49.76 ± 0.48 | 49.22 ± 0.81 | 49.26 ± 0.83 | 47.57 ± 0.49 | 47.51 ± 0.51 | 45.30 ± 0.38 | 45.17 ± 0.27 |
| FeSEM | 50.26 ± 0.29 | 52.65 ± 0.48 | 50.07 ± 0.95 | 51.80 ± 0.56 | 50.19 ± 0.29 | 51.22 ± 0.69 | 49.55 ± 0.42 | 49.17 ± 0.48 |
| FedDrift | 46.01 ± 3.81 | 38.59 ± 4.06 | 47.00 ± 2.27 | 36.03 ± 2.71 | 47.08 ± 1.03 | 29.94 ± 0.94 | 49.69 ± 1.04 | 27.50 ± 1.41 |
| CFL | 53.99 ± 0.46 | 53.99 ± 0.46 | 53.34 ± 0.37 | 53.34 ± 0.37 | 51.28 ± 0.87 | 51.28 ± 0.87 | 48.07 ± 0.16 | 48.07 ± 0.16 |
| pFedMe | 39.66 ± 0.63 | N/A | 44.71 ± 0.68 | N/A | 48.20 ± 0.31 | N/A | 51.26 ± 0.49 | N/A |
| APFL | 55.53 ± 0.47 | 39.33 ± 0.47 | **57.42 ± 0.77** | 43.22 ± 0.56 | **59.46 ± 0.45** | 46.90 ± 0.37 | **59.95 ± 0.41** | 49.58 ± 0.30 |
| ATP | N/A | **60.04 ± 0.60** | N/A | **59.02 ± 0.50** | N/A | **56.75 ± 0.99** | N/A | **53.52 ± 0.28** |
| FLUX | 43.87 ± 1.20 | 41.80 ± 3.19 | 47.83 ± 1.47 | 43.89 ± 1.69 | 50.51 ± 0.40 | 47.65 ± 2.36 | 53.50 ± 0.34 | 49.92 ± 1.61 |
| FLUX-prior | 44.51 ± 0.42 | 42.30 ± 2.02 | 48.21 ± 1.10 | 44.33 ± 1.32 | 50.57 ± 0.47 | 47.75 ± 2.32 | 53.34 ± 0.48 | 48.90 ± 2.86 |

Table 48: Performance comparison across non-IID Levels 1–4 of $P(X|Y)$ on the CIFAR-100 dataset. **Known A.**: Known Association.

| non-IID Level | Level 5 | | Level 6 | | Level 7 | | Level 8 | |
|---|---|---|---|---|---|---|---|---|
| Algorithm | Known A. | Test Phase | Known A. | Test Phase | Known A. | Test Phase | Known A. | Test Phase |
| FedAvg | 44.57 ± 0.12 | 44.57 ± 0.12 | 41.50 ± 0.54 | 41.50 ± 0.54 | 37.09 ± 0.45 | 37.09 ± 0.45 | 32.98 ± 0.72 | 32.98 ± 0.72 |
| IFCA | 50.67 ± 0.94 | 43.43 ± 0.37 | 50.12 ± 1.91 | 40.42 ± 0.45 | 49.13 ± 2.29 | 36.74 ± 0.87 | 46.56 ± 1.97 | 33.14 ± 0.21 |
| FedRC | 42.64 ± 0.33 | 42.46 ± 0.36 | 40.63 ± 0.20 | 40.13 ± 0.36 | 37.18 ± 0.34 | 36.38 ± 0.19 | 33.95 ± 0.45 | 32.94 ± 0.31 |
| FedEM | 42.69 ± 0.29 | 42.46 ± 0.38 | 40.60 ± 0.15 | 40.20 ± 0.34 | 37.16 ± 0.19 | 36.39 ± 0.25 | 33.92 ± 0.50 | 32.92 ± 0.31 |
| FeSEM | 46.60 ± 0.48 | 45.47 ± 0.53 | 45.11 ± 1.63 | 42.19 ± 0.65 | 39.56 ± 0.92 | 37.80 ± 0.82 | 40.67 ± 5.43 | 33.12 ± 0.82 |
| FedDrift | 48.84 ± 5.03 | 24.36 ± 0.77 | 49.63 ± 2.61 | 23.01 ± 1.40 | 50.25 ± 3.13 | 19.60 ± 0.22 | 48.32 ± 4.87 | 17.35 ± 0.92 |
| CFL | 44.96 ± 0.18 | 44.96 ± 0.18 | 41.58 ± 0.79 | 41.58 ± 0.79 | 36.30 ± 0.35 | 36.30 ± 0.36 | 32.70 ± 0.37 | 32.70 ± 0.37 |
| pFedMe | 53.77 ± 0.10 | N/A | 54.88 ± 0.46 | N/A | 55.53 ± 0.13 | N/A | 55.68 ± 0.53 | N/A |
| APFL | **60.56 ± 0.55** | 50.20 ± 0.24 | **59.91 ± 0.43** | **51.73 ± 0.14** | **58.47 ± 0.35** | 50.24 ± 0.35 | **58.01 ± 0.26** | 51.92 ± 0.44 |
| ATP | N/A | 49.95 ± 0.32 | N/A | 46.31 ± 0.24 | N/A | 41.00 ± 0.61 | N/A | 37.21 ± 0.61 |
| FLUX | 55.27 ± 0.15 | **54.30 ± 1.53** | 56.17 ± 0.63 | 51.32 ± 1.30 | 56.53 ± 0.06 | **52.80 ± 3.08** | 56.62 ± 0.45 | 52.48 ± 5.49 |
| FLUX-prior | 55.14 ± 0.14 | 54.21 ± 1.41 | 56.02 ± 0.59 | 51.31 ± 1.24 | 56.36 ± 0.09 | 51.26 ± 3.53 | 56.73 ± 0.27 | **52.66 ± 5.51** |

Table 49: Performance comparison across non-IID Levels 5–8 of $P(X|Y)$ on the CIFAR-100 dataset. **Known A.**: Known Association.

limits augmentations to rotations ($\{0°, 90°, 180°, 270°\}$), with 6 classes subject to augmentation. Appendix F.1 describes more details about the non-IID levels setup.

To ensure sufficient training data per client, local datasets are duplicated based on the number of clients. For 5 clients, no duplication is applied (original size after partition). For 25 clients, datasets are duplicated once (size after partition $\times 2$). For 50 clients, duplication is applied twice (size after partition $\times 4$). For 100 clients, datasets are duplicated four times (size after partition $\times 8$).

Tables 51 to Table 55 provide detailed results for FLUX and baseline methods, evaluated across increasing numbers of clients and various distribution shift types and levels. We did not evaluate the performance of *test phase* on $P(Y|X)$ concept shift, as this problem is unsolvable without access to labels (refer to the Section 4.2.2 and Appendix C.4 for more details). We could not evaluate the performance of FedDrift and CFL under 100 clients due to prohibitive memory and computational costs (see Appendix B.3 for details). For the baselines that do not provide a solution for real *test phase*, we weight all models by the number of clients in the cluster, and use the expectation that weights all model outputs as an estimation of the predicted labels.

**Large-scale experiment with 1,200 clients.**   To further strengthen our evaluation and demonstrate practical scalability, we extended the experiments to a real-world large-scale FL setting with 1,200 clients under partial participation. Specifically, we adopted a participation rate of 0.2 (i.e., on average, 240 clients per round), which is both computationally practical and reflective of realistic deployments where device availability, network conditions, and battery constraints prevent full client participation. This setup follows the same protocol as Figure 5 for known association setting, but reports results separately for each distribution shift type. Table 50 reports results for FLUX and FedAvg. The findings confirm that FLUX not only scales effectively to 1,200 clients but also maintains strong robustness under extreme heterogeneity and limited participation. Notably, FLUX outperforms FedAvg by large margins—up to 29.3 percentage points under the most challenging $P(Y|X)$ shift—while exhibiting consistently low variance, highlighting its stability and scalability in realistic FL deployments.

| Method | P(X) | P(Y) | P(Y|X) | P(X|Y) |
|---|---|---|---|---|
| FedAvg | $70.3 \pm 0.1$ | $84.2 \pm 2.5$ | $65.5 \pm 0.0$ | $86.5 \pm 4.2$ |
| FLUX | $\mathbf{96.1 \pm 0.0}$ | $\mathbf{98.2 \pm 0.2}$ | $\mathbf{94.8 \pm 0.3}$ | $\mathbf{96.3 \pm 0.1}$ |

Table 50: **Large-scale scalability with 1,200 clients (partial participation rate = 0.2).**

| # Clients | 5 Clients | | | 25 Clients | | | 50 Clients | | | 100 Clients | | |
|---|---|---|---|---|---|---|---|---|---|---|---|---|
| Algorithm | Known A. | Test Phase | Time | Known A. | Test Phase | Time | Known A. | Test Phase | Time | Known A. | Test Phase | Time |
| FedAvg | $84.44 \pm 1.55$ | $89.09 \pm 0.94$ | **4.443** | $70.03 \pm 3.24$ | $75.34 \pm 3.64$ | **6.662** | $66.20 \pm 2.19$ | $70.75 \pm 2.45$ | **7.471** | $65.87 \pm 2.69$ | $70.49 \pm 3.04$ | **8.964** |
| IFCA | $80.91 \pm 6.08$ | $73.33 \pm 4.27$ | 6.654 | $76.01 \pm 7.10$ | $70.93 \pm 4.22$ | 8.636 | $76.24 \pm 2.99$ | $70.32 \pm 3.01$ | 10.963 | $74.45 \pm 3.95$ | $69.32 \pm 2.10$ | 10.884 |
| FedRC | $74.71 \pm 2.87$ | $75.85 \pm 2.92$ | 7.855 | $31.05 \pm 2.99$ | $37.60 \pm 3.36$ | 10.111 | $28.36 \pm 2.52$ | $34.08 \pm 2.89$ | 12.527 | $28.55 \pm 1.95$ | $34.30 \pm 2.16$ | 12.457 |
| FedEM | $75.66 \pm 2.78$ | $75.27 \pm 2.41$ | 7.784 | $32.67 \pm 2.68$ | $36.90 \pm 2.68$ | 9.900 | $29.24 \pm 3.79$ | $32.27 \pm 4.24$ | 12.377 | $29.99 \pm 2.92$ | $33.93 \pm 2.08$ | 11.810 |
| FeSEM | $82.69 \pm 4.59$ | $76.84 \pm 4.23$ | 6.603 | $75.42 \pm 3.24$ | $74.63 \pm 1.89$ | 8.384 | $72.08 \pm 1.63$ | $70.64 \pm 1.53$ | 10.252 | $68.49 \pm 1.38$ | $70.87 \pm 1.46$ | 10.337 |
| FedDrift | $97.48 \pm 0.20$ | $55.62 \pm 3.00$ | 8.072 | $70.37 \pm 3.51$ | $62.91 \pm 2.46$ | 14.162 | $82.17 \pm 3.28$ | $51.91 \pm 2.55$ | 16.481 | N/A | N/A | N/A |
| CFL | $84.18 \pm 1.65$ | $88.71 \pm 1.18$ | 7.650 | $63.92 \pm 1.76$ | $64.84 \pm 2.45$ | 14.231 | $67.16 \pm 2.04$ | $71.87 \pm 2.28$ | 16.395 | N/A | N/A | N/A |
| pFedMe | $96.41 \pm 0.19$ | N/A | 5.619 | $89.21 \pm 0.81$ | N/A | 6.923 | $89.06 \pm 0.77$ | N/A | 7.760 | $89.06 \pm 0.63$ | N/A | 9.031 |
| APFL | $96.16 \pm 0.30$ | $85.78 \pm 2.23$ | 7.210 | $89.41 \pm 0.76$ | $73.18 \pm 2.66$ | 8.192 | $89.35 \pm 0.67$ | $72.10 \pm 2.16$ | 9.313 | $89.29 \pm 0.74$ | $71.83 \pm 2.31$ | 10.313 |
| ATP | N/A | $88.82 \pm 1.25$ | 5.862 | N/A | $70.41 \pm 2.12$ | 6.950 | N/A | $69.65 \pm 2.35$ | 8.037 | N/A | $69.31 \pm 2.34$ | 9.321 |
| FLUX | $93.51 \pm 2.02$ | $95.35 \pm 1.54$ | 5.436 | $81.65 \pm 3.18$ | $84.62 \pm 2.18$ | 6.738 | $82.59 \pm 3.55$ | $84.72 \pm 1.60$ | 7.962 | $85.47 \pm 3.00$ | $84.45 \pm 2.18$ | 9.263 |
| FLUX-prior | $\mathbf{97.73 \pm 0.22}$ | $\mathbf{98.06 \pm 0.25}$ | 5.753 | $\mathbf{90.20 \pm 0.47}$ | $\mathbf{90.48 \pm 1.67}$ | 6.796 | $\mathbf{90.36 \pm 0.40}$ | $\mathbf{89.24 \pm 1.42}$ | 7.960 | $\mathbf{90.50 \pm 0.33}$ | $\mathbf{89.06 \pm 2.08}$ | 9.232 |

Table 51: Performance comparison across the number of clients in 5, 25, 50, and 100, summarizing all four types of heterogeneity ($P(X)$, $P(Y)$, $P(Y|X)$, $P(X|Y)$) on the MNIST dataset. **Known A.**: Known Association.

## F.3   Robustness Under Mixed Distribution Shifts

To further assess the robustness of FLUX, we conducted additional experiments involving combinations of multiple distribution shifts occurring simultaneously. Specifically, we evaluated performance under three mixed-shift scenarios: (i) a mixture of $P(X)$ and $P(Y)$ (Table 56); (ii) a mixture of $P(X|Y)$ and $P(Y)$ (Table 57); and (iii) a mixture of $P(Y|X)$ and $P(Y)$ (Table 58), across four non-IID levels (see Table 9 for detailed setting of heterogeneity levels).

| non-IID type | $P(X)$ | | $P(Y)$ | | $P(Y|X)$ | | $P(X|Y)$ | |
|---|---|---|---|---|---|---|---|---|
| **Algorithm** | **Known A.** | **Test Phase** | **Known A.** | **Test Phase** | **Known A.** | **Test Phase** | **Known A.** | **Test Phase** |
| FedAvg | 85.59 ± 0.35 | 85.59 ± 0.35 | 92.90 ± 1.00 | 92.90 ± 1.00 | 70.50 ± 2.62 | N/A | 88.77 ± 1.25 | 88.77 ± 1.25 |
| IFCA | 87.97 ± 6.71 | 76.44 ± 4.17 | 81.64 ± 5.56 | 74.91 ± 6.10 | 77.01 ± 5.99 | N/A | 77.01 ± 5.99 | 69.48 ± 4.01 |
| FedRC | 78.29 ± 1.26 | 74.30 ± 2.09 | 85.66 ± 2.82 | 83.83 ± 2.90 | 67.44 ± 3.42 | N/A | 67.44 ± 3.42 | 65.64 ± 3.52 |
| FedEM | 78.44 ± 1.16 | 74.38 ± 1.98 | 89.50 ± 2.59 | 82.28 ± 0.91 | 67.35 ± 3.37 | N/A | 67.35 ± 3.37 | 65.66 ± 3.47 |
| FeSEM | 88.38 ± 3.03 | 74.48 ± 9.85 | 94.03 ± 4.09 | 83.54 ± 4.63 | 74.17 ± 5.40 | N/A | 74.17 ± 5.40 | 67.68 ± 1.15 |
| FedDrift | 97.29 ± 0.12 | 31.69 ± 0.51 | 98.68 ± 0.28 | 40.56 ± 6.13 | 96.49 ± 0.15 | N/A | 97.46 ± 0.21 | 65.89 ± 1.04 |
| CFL | 85.14 ± 0.78 | 25.09 ± 0.24 | 92.86 ± 1.25 | 26.41 ± 3.54 | 70.58 ± 2.58 | N/A | 88.12 ± 1.42 | 25.81 ± 0.40 |
| pFedMe | 95.84 ± 0.29 | N/A | 97.53 ± 0.07 | N/A | 95.63 ± 0.15 | N/A | 96.65 ± 0.17 | N/A |
| APFL | 95.67 ± 0.20 | 81.23 ± 1.62 | 97.13 ± 0.18 | 89.31 ± 3.28 | 95.17 ± 0.49 | N/A | 96.66 ± 0.20 | 86.81 ± 1.21 |
| ATP | N/A | 83.84 ± 0.86 | N/A | 93.13 ± 1.50 | N/A | N/A | N/A | 89.50 ± 1.30 |
| FLUX | 97.59 ± 0.21 | 97.63 ± 0.18 | 98.69 ± 0.56 | 97.84 ± 2.23 | 85.71 ± 2.54 | N/A | 92.04 ± 3.08 | 90.57 ± 1.44 |
| FLUX-prior | **97.67 ± 0.15** | **97.67 ± 0.15** | **98.79 ± 0.32** | **98.80 ± 0.32** | **96.75 ± 0.14** | N/A | **97.71 ± 0.24** | **97.71 ± 0.24** |

Table 52: Performance comparison across 5 clients with all four types of heterogeneity ($P(X)$, $P(Y)$, $P(Y|X)$, $P(X|Y)$) on the MNIST dataset.

| non-IID type | $P(X)$ | | $P(Y)$ | | $P(Y|X)$ | | $P(X|Y)$ | |
|---|---|---|---|---|---|---|---|---|
| **Algorithm** | **Known A.** | **Test Phase** | **Known A.** | **Test Phase** | **Known A.** | **Test Phase** | **Known A.** | **Test Phase** |
| FedAvg | 67.53 ± 1.61 | 67.53 ± 1.61 | 78.56 ± 5.75 | 78.56 ± 5.75 | 54.12 ± 1.50 | N/A | 79.92 ± 2.04 | 79.92 ± 2.04 |
| IFCA | 82.49 ± 3.07 | 65.23 ± 2.34 | 79.68 ± 13.31 | 68.95 ± 6.36 | 61.00 ± 3.30 | N/A | 80.88 ± 2.06 | 78.61 ± 2.71 |
| FedRC | 26.70 ± 2.79 | 26.69 ± 2.81 | 74.88 ± 5.21 | 74.75 ± 5.09 | 11.25 ± 0.93 | N/A | 11.35 ± 0.00 | 11.35 ± 0.00 |
| FedEM | 26.70 ± 3.00 | 26.69 ± 2.92 | 81.36 ± 4.34 | 72.66 ± 3.61 | 11.27 ± 0.95 | N/A | 11.35 ± 0.00 | 11.35 ± 0.00 |
| FeSEM | 65.53 ± 2.07 | 65.86 ± 1.46 | 87.67 ± 5.64 | 79.03 ± 2.75 | 65.67 ± 2.30 | N/A | 82.79 ± 0.73 | 79.02 ± 1.01 |
| FedDrift | 63.69 ± 3.71 | 54.72 ± 1.26 | 77.33 ± 2.12 | 66.82 ± 3.44 | 64.32 ± 5.28 | N/A | 76.14 ± 1.77 | 67.19 ± 2.17 |
| CFL | 54.81 ± 1.52 | 54.81 ± 1.52 | 73.42 ± 2.07 | 66.82 ± 3.44 | 54.56 ± 1.37 | N/A | 72.88 ± 1.97 | 72.88 ± 1.97 |
| pFedMe | 88.06 ± 0.22 | N/A | 93.91 ± 1.29 | N/A | **86.68 ± 0.74** | N/A | 88.19 ± 0.60 | N/A |
| APFL | 87.96 ± 0.19 | 61.14 ± 1.45 | 94.10 ± 0.67 | 84.74 ± 3.30 | 86.63 ± 1.31 | N/A | **88.96 ± 0.32** | 73.66 ± 2.86 |
| ATP | N/A | 57.91 ± 1.65 | N/A | 82.18 ± 2.36 | N/A | N/A | N/A | 71.14 ± 2.28 |
| FLUX | 90.05 ± 0.41 | 89.31 ± 0.66 | 95.91 ± 0.89 | 93.28 ± 3.01 | 65.68 ± 6.01 | N/A | 74.96 ± 1.80 | 71.29 ± 2.19 |
| FLUX-prior | **91.10 ± 0.13** | **91.10 ± 0.13** | **96.22 ± 0.64** | **94.70 ± 2.70** | 85.38 ± 0.34 | N/A | 88.09 ± 0.59 | **85.65 ± 0.99** |

Table 53: Performance comparison across 25 clients with all four types of heterogeneity ($P(X)$, $P(Y)$, $P(Y|X)$, $P(X|Y)$) on the MNIST dataset.

| non-IID type | $P(X)$ | | $P(Y)$ | | $P(Y|X)$ | | $P(X|Y)$ | |
|---|---|---|---|---|---|---|---|---|
| **Algorithm** | **Known A.** | **Test Phase** | **Known A.** | **Test Phase** | **Known A.** | **Test Phase** | **Known A.** | **Test Phase** |
| FedAvg | 59.70 ± 0.58 | 59.70 ± 0.58 | 80.47 ± 3.94 | 80.47 ± 3.94 | 52.53 ± 1.08 | N/A | 72.08 ± 1.45 | 72.08 ± 1.45 |
| IFCA | 82.36 ± 3.83 | 64.14 ± 1.41 | 83.27 ± 4.24 | 70.18 ± 4.96 | 62.25 ± 1.31 | N/A | 77.07 ± 1.16 | 76.62 ± 0.74 |
| FedRC | 26.36 ± 1.85 | 26.35 ± 1.85 | 63.25 ± 3.66 | 63.08 ± 3.63 | 11.07 ± 0.75 | N/A | 12.76 ± 2.81 | 12.80 ± 2.91 |
| FedEM | 26.41 ± 1.75 | 26.39 ± 1.74 | 68.11 ± 7.35 | 59.06 ± 7.14 | 11.09 ± 0.76 | N/A | 11.35 ± 0.00 | 11.35 ± 0.00 |
| FeSEM | 60.52 ± 1.50 | 62.59 ± 1.14 | 85.10 ± 2.21 | 74.85 ± 2.31 | 67.67 ± 1.51 | N/A | 75.04 ± 1.06 | 74.46 ± 0.65 |
| FedDrift | 90.80 ± 2.77 | 40.04 ± 2.45 | 96.48 ± 0.41 | 46.98 ± 2.72 | 65.48 ± 5.30 | N/A | 75.74 ± 2.65 | 68.69 ± 2.48 |
| CFL | 61.23 ± 0.65 | 61.23 ± 0.65 | 81.41 ± 3.42 | 81.41 ± 3.42 | 53.04 ± 1.03 | N/A | 72.97 ± 1.87 | 72.97 ± 1.87 |
| pFedMe | 87.93 ± 0.39 | N/A | 93.36 ± 0.82 | N/A | **86.62 ± 1.06** | N/A | 88.34 ± 0.64 | N/A |
| APFL | 88.06 ± 0.17 | 58.29 ± 1.29 | 93.93 ± 0.90 | 84.42 ± 2.78 | 86.45 ± 0.97 | N/A | **88.95 ± 0.23** | 73.60 ± 2.16 |
| ATP | N/A | 56.21 ± 1.03 | N/A | 81.84 ± 3.42 | N/A | N/A | N/A | 70.89 ± 1.94 |
| FLUX | 90.83 ± 0.34 | 90.13 ± 0.75 | 96.04 ± 0.42 | **93.69 ± 1.52** | 69.40 ± 6.76 | N/A | 74.10 ± 2.14 | 70.35 ± 2.19 |
| FLUX-prior | **91.43 ± 0.21** | **91.43 ± 0.21** | **96.68 ± 0.49** | 93.36 ± 1.63 | 85.78 ± 0.24 | N/A | 87.98 ± 0.53 | **82.95 ± 1.83** |

Table 54: Performance comparison across 50 clients with all four types of heterogeneity ($P(X)$, $P(Y)$, $P(Y|X)$, $P(X|Y)$) on the MNIST dataset.

| non-IID type | $P(X)$ | | $P(Y)$ | | $P(Y|X)$ | | $P(X|Y)$ | |
|---|---|---|---|---|---|---|---|---|
| Algorithm | Known A. | Test Phase | Known A. | Test Phase | Known A. | Test Phase | Known A. | Test Phase |
| FedAvg | 59.48 ± 0.51 | 59.48 ± 0.51 | 79.48 ± 4.72 | 79.48 ± 4.72 | 52.02 ± 1.16 | N/A | 72.52 ± 2.26 | 72.52 ± 2.26 |
| IFCA | 85.71 ± 4.83 | 65.75 ± 0.67 | 80.10 ± 5.11 | 66.93 ± 3.03 | 56.55 ± 2.84 | N/A | 75.42 ± 2.19 | 75.26 ± 1.88 |
| FedRC | 26.39 ± 1.69 | 26.39 ± 1.69 | 65.29 ± 3.44 | 65.17 ± 3.34 | 11.16 ± 0.78 | N/A | 11.35 ± 0.00 | 11.35 ± 0.00 |
| FedEM | 26.46 ± 1.74 | 26.46 ± 1.74 | 71.00 ± 5.52 | 63.97 ± 3.16 | 11.16 ± 0.77 | N/A | 11.35 ± 0.00 | 11.35 ± 0.00 |
| FeSEM | 59.87 ± 0.41 | 61.80 ± 0.75 | 83.18 ± 1.95 | 76.71 ± 2.08 | 57.04 ± 1.13 | N/A | 73.87 ± 1.54 | 74.10 ± 1.24 |
| FedDrift | N/A | N/A | N/A | N/A | N/A | N/A | N/A | N/A |
| CFL | N/A | N/A | N/A | N/A | N/A | N/A | N/A | N/A |
| pFedMe | 87.99 ± 0.08 | N/A | 93.34 ± 0.99 | N/A | **86.64 ± 0.66** | N/A | 88.26 ± 0.40 | N/A |
| APFL | 87.99 ± 0.17 | 57.30 ± 1.18 | 93.82 ± 0.62 | 84.39 ± 2.91 | 86.34 ± 1.32 | N/A | **88.99 ± 0.23** | 73.80 ± 2.48 |
| ATP | N/A | 55.57 ± 0.15 | N/A | 81.18 ± 3.50 | N/A | N/A | N/A | 71.19 ± 2.04 |
| FLUX | 91.55 ± 0.13 | 90.32 ± 0.71 | 96.30 ± 0.45 | 91.48 ± 3.27 | 77.96 ± 5.44 | N/A | 76.07 ± 2.47 | 71.55 ± 1.77 |
| FLUX-prior | **91.57 ± 0.10** | **91.57 ± 0.10** | **96.58 ± 0.34** | **93.32 ± 3.28** | 85.73 ± 0.27 | N/A | 88.11 ± 0.49 | **82.28 ± 1.47** |

Table 55: Performance comparison across 100 clients with all four types of heterogeneity ($P(X)$, $P(Y)$, $P(Y|X)$, $P(X|Y)$) on the MNIST dataset.

Across all settings, FLUX consistently maintains high accuracy, demonstrating strong robustness under compound shift conditions. In contrast, existing CFL methods exhibit substantial performance degradation when faced with mixed-shift scenarios. As expected, personalized FL methods (e.g., APFL, pFedMe) perform well under the *known association* condition, where test-time distributions match those where the models were fine-tuned on, achieving accuracy comparable to FLUX-prior. However, their performance drops sharply on unseen clients, where no labeled data is available for fine-tuning. ATP, which performs unsupervised test-time adaptation, remains relatively stable on unseen clients but fails to generalize across all distribution types, performing notably worse than FLUX. For example, under the $P(X)+P(Y)$ mixture (Table 56), FLUX sustains a test-phase accuracy of 95.9% at Level 3—outperforming the best-performing methods by a wide margin, e.g., ATP (85.4%), APFL (80.6%), and CFL (75.2%)—and remains above 82.9% even at Level 6, where many baselines fall below 60%. Similar trends hold for the $P(X|Y)+P(Y)$ and $P(Y|X)+P(Y)$ mixtures (Tables 57, 58), confirming FLUX's consistent robustness to compound shifts.

| non-IID Level | Level 3 | | Level 4 | | Level 5 | | Level 6 | |
|---|---|---|---|---|---|---|---|---|
| Algorithm | Known A. | Test Phase | Known A. | Test Phase | Known A. | Test Phase | Known A. | Test Phase |
| FedAvg | 77.88 ± 8.21 | 77.88 ± 8.21 | 82.48 ± 3.99 | 82.48 ± 3.99 | 71.53 ± 11.15 | 71.53 ± 11.15 | 67.68 ± 7.80 | 67.68 ± 7.80 |
| IFCA | 86.35 ± 7.45 | 69.44 ± 15.54 | 95.78 ± 0.44 | 81.85 ± 2.31 | 90.31 ± 1.39 | 69.30 ± 12.37 | 85.97 ± 7.14 | 66.00 ± 11.00 |
| FedRC | 42.10 ± 10.75 | 41.75 ± 10.93 | 31.75 ± 4.95 | 31.52 ± 4.96 | 34.77 ± 4.73 | 34.15 ± 4.34 | 20.53 ± 3.73 | 20.23 ± 3.78 |
| FedEM | 66.02 ± 4.55 | 42.09 ± 10.75 | 48.74 ± 4.31 | 30.69 ± 6.52 | 59.60 ± 21.72 | 35.07 ± 6.03 | 50.71 ± 4.32 | 19.96 ± 3.69 |
| FeSEM | 78.67 ± 4.83 | 67.74 ± 9.51 | 75.93 ± 11.99 | 66.20 ± 20.17 | 78.87 ± 7.18 | 68.59 ± 13.27 | 73.24 ± 8.99 | 59.38 ± 11.38 |
| FedDrift | 97.20 ± 0.42 | 37.51 ± 5.97 | 95.23 ± 0.75 | 38.62 ± 5.32 | 95.39 ± 1.82 | 34.82 ± 10.98 | 95.14 ± 0.53 | 37.00 ± 5.31 |
| CFL | 75.18 ± 9.56 | 75.18 ± 9.56 | 81.92 ± 4.88 | 81.92 ± 4.88 | 67.54 ± 13.22 | 67.54 ± 13.22 | 72.49 ± 5.84 | 72.49 ± 5.84 |
| pFedMe | 97.26 ± 0.63 | N/A | 96.08 ± 0.35 | N/A | 95.58 ± 1.84 | N/A | 95.04 ± 0.60 | N/A |
| APFL | 97.14 ± 0.59 | 80.58 ± 6.58 | **96.25 ± 0.20** | 80.67 ± 3.92 | **95.82 ± 1.29** | 69.30 ± 13.45 | 94.71 ± 0.95 | 63.18 ± 17.98 |
| ATP | N/A | 85.39 ± 3.64 | N/A | 77.30 ± 15.52 | N/A | 68.12 ± 12.86 | N/A | 72.07 ± 9.56 |
| FLUX | 96.14 ± 1.74 | 95.91 ± 2.06 | 88.89 ± 6.08 | 86.78 ± 4.10 | 87.16 ± 2.27 | 81.73 ± 4.64 | 83.44 ± 3.80 | 82.94 ± 3.64 |
| FLUX-prior | **97.33 ± 0.72** | **97.33 ± 0.72** | 96.13 ± 0.45 | **96.13 ± 0.45** | 95.77 ± 1.66 | **95.77 ± 1.66** | **95.17 ± 0.49** | **94.97 ± 0.49** |

Table 56: Performance comparison across mixture of $P(X)$ and $P(Y)$ on the MNIST dataset. **Known A.**: Known Association.

## F.4 Results on Real-world Datasets

To evaluate the real-world applicability of FLUX, we follow the setup described in Appendix B.1 and compare its performance against baseline methods on the CheXpert dataset across three levels of non-IID heterogeneity (low, medium, high), using 20 clients. Since CheXpert is a multi-label classification task, we use macro-averaged ROC AUC as the evaluation metric instead of accuracy.

| non-IID Level | Level 3 | | Level 4 | | Level 5 | | Level 6 | |
|---|---|---|---|---|---|---|---|---|
| Algorithm | Known A. | Test Phase | Known A. | Test Phase | Known A. | Test Phase | Known A. | Test Phase |
| FedAvg | 69.61 ± 5.58 | 69.61 ± 5.58 | 78.94 ± 4.02 | 78.94 ± 4.02 | 80.06 ± 5.49 | 80.06 ± 5.49 | 68.93 ± 0.71 | 68.93 ± 0.71 |
| IFCA | 80.38 ± 8.16 | 58.52 ± 18.12 | 93.06 ± 4.19 | 69.82 ± 5.52 | 87.71 ± 6.44 | 65.42 ± 6.32 | 88.70 ± 9.01 | 50.92 ± 10.26 |
| FedRC | 59.14 ± 12.52 | 55.28 ± 8.47 | 67.16 ± 7.36 | 64.02 ± 6.84 | 62.89 ± 3.87 | 60.12 ± 3.50 | 64.29 ± 11.31 | 60.12 ± 10.74 |
| FedEM | 88.79 ± 3.80 | 57.00 ± 6.93 | 77.97 ± 3.08 | 63.29 ± 3.94 | 82.40 ± 3.07 | 59.86 ± 3.42 | 85.96 ± 5.73 | 58.91 ± 10.21 |
| FeSEM | 83.02 ± 4.49 | 61.57 ± 3.60 | 88.46 ± 2.56 | 66.18 ± 7.15 | 81.51 ± 7.07 | 68.67 ± 7.33 | 81.18 ± 4.27 | 67.83 ± 4.96 |
| FedDrift | 98.80 ± 0.32 | 35.31 ± 5.58 | 98.57 ± 0.24 | 39.59 ± 3.56 | 98.02 ± 0.15 | 42.26 ± 5.02 | 95.80 ± 4.49 | 33.60 ± 2.26 |
| CFL | 67.20 ± 4.42 | 67.20 ± 4.42 | 76.88 ± 1.42 | 76.88 ± 1.42 | 78.76 ± 8.34 | 78.76 ± 8.34 | 62.96 ± 6.35 | 62.96 ± 6.35 |
| pFedMe | 98.35 ± 0.22 | N/A | **98.96 ± 0.31** | N/A | **98.97 ± 0.40** | N/A | **99.34 ± 0.07** | N/A |
| APFL | 99.03 ± 0.17 | 64.09 ± 6.38 | 98.77 ± 0.03 | 75.87 ± 4.28 | 98.66 ± 0.32 | 75.33 ± 6.18 | 98.91 ± 0.20 | 69.58 ± 5.40 |
| ATP | N/A | 70.27 ± 7.99 | N/A | 84.00 ± 3.98 | N/A | 82.10 ± 7.02 | N/A | 63.84 ± 9.93 |
| FLUX | 94.22 ± 4.25 | 95.05 ± 3.27 | 94.21 ± 5.52 | 89.59 ± 8.65 | 89.91 ± 2.86 | 79.25 ± 5.17 | 92.06 ± 6.69 | 88.36 ± 11.12 |
| FLUX-prior | **99.10 ± 0.18** | **99.10 ± 0.18** | 98.92 ± 0.23 | **98.92 ± 0.23** | 98.91 ± 0.41 | **98.91 ± 0.41** | 99.27 ± 0.14 | **99.27 ± 0.14** |

Table 57: Performance comparison across mixture of $P(X|Y)$ and $P(Y)$ on the MNIST dataset. **Known A.**: Known Association.

| non-IID Level | Level 3 | | Level 4 | | Level 5 | | Level 6 | |
|---|---|---|---|---|---|---|---|---|
| Algorithm | Known A. | Test Phase | Known A. | Test Phase | Known A. | Test Phase | Known A. | Test Phase |
| FedAvg | 79.81 ± 10.96 | 79.81 ± 10.96 | 65.10 ± 6.34 | 65.10 ± 6.34 | 59.41 ± 5.36 | 59.41 ± 5.36 | 54.25 ± 3.16 | 54.25 ± 3.16 |
| IFCA | 95.96 ± 0.71 | 75.32 ± 12.02 | 83.89 ± 3.83 | 56.60 ± 13.64 | 83.64 ± 2.74 | 52.82 ± 7.76 | 82.38 ± 5.75 | 46.16 ± 3.56 |
| FedRC | 60.63 ± 11.64 | 60.30 ± 11.95 | 42.56 ± 15.53 | 41.84 ± 15.96 | 43.89 ± 6.29 | 43.54 ± 7.47 | 42.13 ± 3.04 | 40.54 ± 3.96 |
| FedEM | 73.77 ± 5.83 | 60.42 ± 10.92 | 59.59 ± 4.70 | 40.04 ± 16.06 | 51.61 ± 10.32 | 42.63 ± 8.09 | 55.88 ± 10.08 | 37.99 ± 6.75 |
| FeSEM | 86.58 ± 5.31 | 74.19 ± 8.91 | 83.83 ± 1.89 | 60.10 ± 6.56 | 78.47 ± 6.07 | 54.62 ± 4.51 | 81.21 ± 1.36 | 52.37 ± 1.93 |
| FedDrift | 97.33 ± 0.68 | 39.96 ± 8.82 | 97.33 ± 1.11 | 34.68 ± 11.11 | 95.90 ± 2.09 | 29.84 ± 4.57 | 96.87 ± 0.47 | 28.38 ± 2.51 |
| CFL | 81.14 ± 8.47 | 81.14 ± 8.47 | 67.05 ± 5.84 | 67.05 ± 5.84 | 61.90 ± 6.90 | 61.90 ± 6.90 | 51.66 ± 3.06 | 51.66 ± 3.06 |
| pFedMe | 97.76 ± 0.30 | N/A | 97.95 ± 0.43 | N/A | **97.54 ± 0.21** | N/A | **97.39 ± 0.42** | N/A |
| APFL | **99.03 ± 0.17** | 81.75 ± 5.69 | 97.50 ± 0.48 | 66.61 ± 5.13 | 96.95 ± 0.58 | 59.71 ± 7.75 | 97.01 ± 0.23 | 51.26 ± 4.52 |
| ATP | N/A | 81.94 ± 10.74 | N/A | 70.22 ± 6.05 | N/A | 60.23 ± 3.90 | N/A | 53.47 ± 3.08 |
| FLUX | 92.24 ± 4.30 | 87.89 ± 8.87 | 95.73 ± 1.31 | 90.85 ± 3.48 | 92.98 ± 3.90 | 87.13 ± 6.17 | 94.17 ± 4.64 | 91.25 ± 8.77 |
| FLUX-prior | 97.47 ± 0.49 | **97.47 ± 0.49** | **98.00 ± 0.39** | **98.00 ± 0.39** | 97.47 ± 0.25 | **97.47 ± 0.25** | 97.05 ± 0.54 | **97.05 ± 0.54** |

Table 58: Performance comparison across mixture of $P(Y|X)$ and $P(Y)$ on the MNIST dataset. **Known A.**: Known Association.

The ATP baseline is omitted, as its unsupervised optimization procedure is designed for multi-class classification and is not directly applicable to the multi-label setting.

As shown in Table 59, FLUX consistently and significantly outperforms baseline methods across all heterogeneity levels, in both the *known association* and *test phase* settings. In the known association setting, the best-performing baseline (APFL) degrades under increasing heterogeneity—dropping from 74.6 pp to 70.6 pp—whereas FLUX maintains strong and stable accuracy, achieving up to 80.3 pp under low heterogeneity and remaining above 79.2 pp even under the most challenging configuration. In the test phase, where clients are previously unseen and unlabeled, APFL's accuracy drops to 59.3 pp, while FLUX stays above 76.7 pp. Notably, most metric-based and parameter-based CFL baselines collapse to a single global model during the training, demonstrating limited capacity to capture real-world distribution shifts. By contrast, FLUX matches the performance of its oracle variant, FLUX-prior, which has access to the true number of clusters $M$, highlighting the discriminative strength of our learned descriptors and the effectiveness of the unsupervised clustering strategy. Finally, the difference between known association and test phase is negligible across most settings—except under high heterogeneity ($M = 8$), where a slight gap appears—indicating that FLUX can reliably assign previously unseen, unlabeled test-time clients to the appropriate clusters without supervision. These results highlight the practical viability of FLUX in federated medical imaging tasks, where substantial distribution shifts between training and deployment are common.

| non-IID Level | Low | | Medium | | High | |
|---|---|---|---|---|---|---|
| Algorithm | Known A. | Test Phase | Known A. | Test Phase | Known A. | Test Phase |
| FedAvg | 57.95 ± 2.83 | 57.95 ± 2.83 | 55.81 ± 0.69 | 55.81 ± 0.69 | 54.53 ± 0.78 | 54.53 ± 0.78 |
| IFCA | 62.68 ± 1.05 | 62.68 ± 1.05 | 57.79 ± 0.48 | 57.79 ± 0.48 | 55.19 ± 0.05 | 55.19 ± 0.05 |
| FedRC | 62.84 ± 1.06 | 62.84 ± 1.06 | 58.08 ± 0.48 | 56.78 ± 0.48 | 55.42 ± 0.06 | 55.31 ± 0.06 |
| FedEM | 62.69 ± 1.05 | 62.69 ± 1.05 | 57.78 ± 0.49 | 57.78 ± 0.49 | 55.19 ± 0.06 | 55.19 ± 0.06 |
| FeSEM | 62.54 ± 1.15 | 62.51 ± 1.16 | 58.04 ± 0.17 | 57.48 ± 0.45 | 56.45 ± 0.26 | 54.96 ± 0.06 |
| FedDrift | 67.46 ± 1.08 | 67.46 ± 1.08 | 60.60 ± 1.28 | 60.60 ± 1.28 | 56.87 ± 0.13 | 56.87 ± 0.13 |
| CFL | 62.67 ± 1.06 | 62.67 ± 1.06 | 57.79 ± 0.48 | 57.79 ± 0.48 | 55.20 ± 0.05 | 55.20 ± 0.05 |
| pFedMe | 70.93 ± 0.62 | N/A | 68.75 ± 0.05 | N/A | 68.56 ± 0.09 | N/A |
| APFL | 74.62 ± 0.60 | 69.66 ± 0.70 | 71.64 ± 0.24 | 63.02 ± 0.77 | 70.59 ± 0.13 | 59.23 ± 0.23 |
| FLUX | **80.29 ± 0.74** | **80.29 ± 0.74** | 78.65 ± 0.48 | 78.65 ± 0.48 | **79.17 ± 0.17** | 76.74 ± 2.63 |
| FLUX-prior | 80.22 ± 0.69 | 80.22 ± 0.69 | **78.68 ± 0.37** | **78.68 ± 0.37** | 79.16 ± 0.15 | **76.76 ± 2.65** |

Table 59: Performance across three non-IID levels on CheXpert. **Known A.**: Known Association.

## F.5 Robustness to Partial Participation

We further evaluate FLUX under partial client participation, reflecting realistic FL deployments where device availability, network conditions, or energy constraints prevent full participation in every round. Table 60 reports average accuracies on MNIST across all distribution-shift types, using the same experimental protocol as the scalability experiments in Figure 5. Results are shown for different client participation rates, ranging from full participation (1.0) to as low as 0.2.

These results show that FLUX maintains strong performance even under significant dropout, consistently outperforming FedAvg. The most notable degradation occurs at a 0.2 participation rate, where only ~4 clients participate in clustering. In highly heterogeneous settings (e.g., $M = 8$ clusters), such limited participation can exclude entire distributions, naturally reducing performance. However, in large-scale FL, it is reasonable to assume that the number of participating clients per round exceeds the number of underlying clusters, ensuring sufficient distributional coverage. Importantly, unlike CFL/PFL baselines which generally assume full participation to isolate clustering effectiveness, FLUX extends naturally to partial participation. This opens the door to combining FLUX with existing client-selection strategies to further improve efficiency in resource-constrained settings.

| Participation Rate | FLUX Accuracy (%) | FedAvg Accuracy (%) |
|---|---|---|
| 1.0 | 91.3 ± 0.4 | 76.7 ± 2.0 |
| 0.8 | 91.2 ± 0.3 | 75.9 ± 1.4 |
| 0.6 | 90.2 ± 0.8 | 75.0 ± 4.6 |
| 0.4 | 87.9 ± 2.5 | 72.3 ± 1.0 |
| 0.2 | 82.0 ± 2.2 | 70.8 ± 4.0 |

Table 60: **Accuracy of FLUX and FedAvg under different participation rates** on MNIST.

## F.6 Ablation Studies

### F.6.1 Effect of Client Data Size on Descriptor Quality

We further studied the robustness of descriptor estimation under varying local data sizes, simulating realistic FL scenarios where clients may contribute only a few samples. Experiments were conducted on MNIST across four non-IID levels, with client data sizes scaled to 4%, 8%, 16%, 32%, and 100% of the original. To isolate the effect of data size, other parameters were kept fixed. Table 61 summarizes results under the *Known Association* and *Test Phase* settings. As expected, overall accuracy decreases with smaller client datasets. Nevertheless, FLUX maintains substantially higher accuracy than *FedAvg* across all data sizes and heterogeneity levels. Even at only 4% of the data, FLUX descriptors remain sufficiently informative to capture client heterogeneity, highlighting their suitability in practical FL deployments where clients often hold very limited data.

| Method\Data size | Known Association | | | | Test Phase | | | |
|---|---|---|---|---|---|---|---|---|
| | **Low** | **Mid-low** | **Mid-high** | **High** | **Low** | **Mid-low** | **Mid-high** | **High** |
| FLUX 4% | $82.3 \pm 3.6$ | $80.1 \pm 4.3$ | $72.7 \pm 7.3$ | $65.4 \pm 9.5$ | $81.7 \pm 2.1$ | $76.9 \pm 2.3$ | $68.5 \pm 5.7$ | $60.8 \pm 7.2$ |
| FLUX 8% | $83.7 \pm 2.3$ | $80.5 \pm 3.9$ | $70.3 \pm 8.3$ | $66.9 \pm 9.7$ | $81.7 \pm 1.9$ | $77.7 \pm 2.1$ | $66.4 \pm 7.1$ | $62.2 \pm 8.6$ |
| FLUX 16% | $87.3 \pm 1.8$ | $84.6 \pm 2.4$ | $77.8 \pm 6.0$ | $70.0 \pm 5.7$ | $84.0 \pm 1.7$ | $80.9 \pm 1.5$ | $73.3 \pm 5.2$ | $66.0 \pm 5.8$ |
| FLUX 32% | $88.1 \pm 1.7$ | $85.0 \pm 1.7$ | $79.7 \pm 4.8$ | $66.8 \pm 7.3$ | $85.2 \pm 1.9$ | $81.5 \pm 1.3$ | $75.2 \pm 4.7$ | $62.6 \pm 6.4$ |
| FLUX 100% | $96.4 \pm 2.8$ | $93.4 \pm 1.4$ | $94.9 \pm 0.9$ | $87.1 \pm 4.2$ | $96.2 \pm 0.4$ | $90.0 \pm 2.8$ | $86.4 \pm 1.9$ | $79.4 \pm 3.0$ |
| FedAvg 4% | $70.4 \pm 2.8$ | $59.8 \pm 4.0$ | $47.6 \pm 5.9$ | $37.6 \pm 7.5$ | $70.4 \pm 2.8$ | $59.8 \pm 4.0$ | $47.6 \pm 5.9$ | $37.6 \pm 7.5$ |
| FedAvg 8% | $71.8 \pm 2.4$ | $60.5 \pm 3.1$ | $50.2 \pm 4.7$ | $42.8 \pm 5.7$ | $71.8 \pm 2.4$ | $60.5 \pm 3.1$ | $50.2 \pm 4.7$ | $42.8 \pm 5.7$ |
| FedAvg 16% | $72.3 \pm 3.6$ | $63.8 \pm 4.1$ | $55.6 \pm 5.2$ | $46.9 \pm 7.0$ | $72.3 \pm 3.6$ | $63.8 \pm 4.1$ | $55.6 \pm 5.2$ | $46.9 \pm 7.0$ |
| FedAvg 32% | $72.1 \pm 2.7$ | $64.8 \pm 3.3$ | $62.0 \pm 3.3$ | $54.4 \pm 2.9$ | $72.1 \pm 2.7$ | $64.8 \pm 3.3$ | $62.0 \pm 3.3$ | $54.4 \pm 2.9$ |
| FedAvg 100% | $93.1 \pm 0.4$ | $86.0 \pm 1.2$ | $79.4 \pm 2.0$ | $72.0 \pm 1.9$ | $93.1 \pm 0.4$ | $86.0 \pm 1.2$ | $79.4 \pm 2.0$ | $72.0 \pm 1.9$ |

Table 61: **Descriptor quality vs. client data size.** Results on MNIST, averaged across four shift types and four non-IID levels, comparing performance under *Known Association* and *Test Phase*.

To further test robustness under extreme sparsity, we designed a setting where clients hold only $\sim$1% of the total data on average (Table 62). In this case, many clients contribute fewer than 1% of samples, leading to very small local datasets. To enable convergence, we increased the number of clients to 20 and extended training to 200 rounds. Despite severe data scarcity, FLUX descriptors remain informative and significantly outperform *FedAvg*. This robustness arises from the design of our descriptors, which rely on low-order statistics (means and covariances) after dimensionality reduction, making them less sensitive to sparsity. While our current implementation does not use additional regularization, future work could further stabilize descriptor estimation using Bayesian shrinkage priors.

| Method | Known Association | | | | Test Phase | | | |
|---|---|---|---|---|---|---|---|---|
| | **P(X)** | **P(Y)** | **P(Y\|X)** | **P(X\|Y)** | **P(X)** | **P(Y)** | **P(Y\|X)** | **P(X\|Y)** |
| FedAvg | $36.3 \pm 7.7$ | $84.8 \pm 1.4$ | $46.1 \pm 5.2$ | $31.6 \pm 11.5$ | $36.3 \pm 7.7$ | $84.8 \pm 1.4$ | N/A | $31.6 \pm 11.5$ |
| FLUX | $74.3 \pm 0.5$ | $97.5 \pm 0.5$ | $49.2 \pm 6.9$ | $45.5 \pm 6.5$ | $78.2 \pm 0.1$ | $95.6 \pm 0.2$ | N/A | $42.0 \pm 9.3$ |

Table 62: **Descriptor robustness in extreme low-data regime ($\sim$1%).** Results under *Known Association* and *Test Phase*.

### F.6.2 Effect of Clustering Algorithm Choice

The primary contribution of FLUX lies in the design of informative, privacy-preserving descriptors that approximate the Wasserstein distance between client distributions. These descriptors enable meaningful unsupervised clustering while preserving privacy, unlike alternatives that rely on direct metrics or gradient similarity. Consequently, the framework is agnostic to the specific clustering algorithm, requiring only that clustering operates solely on descriptors without relying on prior knowledge such as the number of clusters (see Equation 6).

To validate this agnosticism, we conducted ablation studies comparing different clustering strategies. Table 63 reports results on MNIST, averaged across shift types, heterogeneity levels, and seeds. We compared Unsupervised $k$-Means, HDBSCAN, Agglomerative Clustering, and our density-based method. Except for unsupervised $k$-Means, all algorithms achieve strong performance, confirming that FLUX descriptors enable robust client grouping regardless of the specific clustering algorithm. Importantly, all methods converge to comparable accuracy in the test phase.

| Method | Known Association (%) | Test Phase (%) |
|---|---|---|
| Unsupervised $k$-Means | $86.8 \pm 14.6$ | $93.5 \pm 3.8$ |
| HDBSCAN | $90.5 \pm 7.7$ | $92.9 \pm 2.6$ |
| Agglomerative | $93.9 \pm 3.2$ | $94.2 \pm 0.8$ |
| FLUX (ours) | $92.5 \pm 5.2$ | $94.0 \pm 2.8$ |

Table 63: **Comparison of clustering algorithms on MNIST.** Except for unsupervised $k$-Means, all methods achieve strong accuracy, confirming that FLUX is not tied to a specific clustering algorithm.

To further assess generality, we compared the two best-performing methods (Agglomerative and ours) on CIFAR-100. Results in Table 64 show similar conclusions: both methods converge to comparable accuracy, with differences within variance margins. This confirms that the robustness of FLUX stems from its descriptors rather than the specific clustering algorithm employed.

| Method | Known Association (%) | Test Phase (%) |
|---|---|---|
| Agglomerative | $42.0 \pm 17.8$ | $36.0 \pm 5.4$ |
| FLUX (ours) | $41.7 \pm 11.0$ | $41.3 \pm 7.8$ |

Table 64: **Comparison of clustering algorithms on CIFAR-100.** Both methods achieve similar performance, demonstrating that FLUX remains robust across clustering strategies.

### F.6.3 Effect of Test-Time Cluster Assignment Strategy

To evaluate the importance of our test-time assignment mechanism, we performed an ablation study where we replaced FLUX's descriptor-based assignment with a random cluster assignment at inference. Following the same evaluation protocol as in Table 1, we compared both strategies across all heterogeneity levels and distribution shift types on MNIST, FMNIST, CIFAR-10, and CIFAR-100.

As summarized in Table 65, FLUX consistently outperforms random assignment by a large margin—up to 53 pp on MNIST and 25 pp on CIFAR-100. These results highlight the crucial role of our label-agnostic descriptors in reliably matching test clients to the most suitable cluster-specific models, even under severe non-IID conditions. Without this mechanism, performance degrades sharply, particularly in feature- and label-shift scenarios where random assignment fails to capture distributional similarity.

| Dataset | Test-Assignment | $P(X)$ | $P(Y)$ | $P(X|Y)$ |
|---|---|---|---|---|
| MNIST | FLUX | $95.0 \pm 1.2$ | $96.1 \pm 1.5$ | $90.8 \pm 3.9$ |
| | Random | $41.9 \pm 11.4$ | $69.2 \pm 17.2$ | $73.2 \pm 13.7$ |
| FMNIST | FLUX | $77.0 \pm 2.9$ | $85.7 \pm 6.6$ | $81.0 \pm 1.4$ |
| | Random | $34.0 \pm 10.5$ | $61.1 \pm 13.5$ | $63.1 \pm 11.1$ |
| CIFAR-10 | FLUX | $33.3 \pm 2.5$ | $46.2 \pm 7.8$ | $36.7 \pm 1.3$ |
| | Random | $20.4 \pm 5.1$ | $31.8 \pm 5.4$ | $32.3 \pm 2.7$ |
| CIFAR-100 | FLUX | $33.8 \pm 5.6$ | $40.8 \pm 3.8$ | $49.3 \pm 4.5$ |
| | Random | $17.0 \pm 7.8$ | $15.9 \pm 3.9$ | $34.1 \pm 0.1$ |

Table 65: **Ablation study on test-time cluster assignment.** Accuracy under different distribution shifts when using FLUX's descriptor-based assignment versus random assignment. Results averaged across all heterogeneity levels.

### F.6.4 Effect of Descriptor Configurations and Length

We performed an ablation study to evaluate the impact of different descriptor configurations in FLUX. Specifically, we tested various combinations of descriptors, including those approximating the marginal distribution $P(X)$ (i.e., $(\mu_x^{(k)}, \Sigma_x^{(k)})$) and label-wise descriptors capturing the conditional distribution $P(Y|X)$ (i.e., $\{(\mu_u^{(k)}, \Sigma_u^{(k)})\}_{u=1}^{U}$). These experiments were conducted on the MNIST dataset under four distinct distribution shifts, with non-IID levels fixed as described in Appendix F.2. Furthermore, we analyzed the effect of the descriptor length $l$, varying it by scaling the output dimensionality of the reduction transformation (Equation 6) by factors of 1, 2, and 5.

Table 66 summarizes the average accuracy and standard deviation across all distribution shifts. The results in the *known association* setting underscore the clear utility of incorporating label-wise descriptors for $P(Y|X)$, which enables FLUX to accurately identify clusters of clients with similar distributions under label-conditional shifts. While the inclusion of standard deviation descriptors ($\sigma$) for the latent distribution also leads to improved accuracy, the performance gains are more modest. Interestingly, increasing the descriptor length $l$ (i.e., scaling it by 2 or 5) did not yield improvements, likely due to the reduced compactness of the client distribution representation, which complicates the clustering process. In the *test phase*, a flatter performance trend is observed. This behavior is primarily attributed to two factors. First, the results for label-conditional shifts are not averaged during the test phase, as test-time association without access to label data is inherently infeasible.

Second, label-wise descriptors cannot be used during testing as they rely on label information for computation. Consequently, FLUX relies solely on descriptors of $P(X)$ in this phase, limiting the variability between different configurations.

| Descriptors | | | | | | Experiments | |
|---|---|---|---|---|---|---|---|
| $\mu_x^{(k)}$ | $\Sigma_x^{(k)}$ | $\{\mu_u^{(k)}\}$ | $\{\Sigma_u^{(k)}\}$ | Length | | Known Association | Test Phase |
| ✓ | | | | $1l$ | | 90.96 ± 2.09 | 95.87 ± 1.50 |
| ✓ | | | | $2l$ | | 91.75 ± 2.18 | 95.35 ± 1.62 |
| ✓ | | | | $5l$ | | 91.03 ± 1.65 | 95.63 ± 0.78 |
| ✓ | ✓ | | | $1l$ | | 92.62 ± 1.19 | 95.21 ± 1.39 |
| ✓ | ✓ | | | $2l$ | | 90.32 ± 2.75 | 95.51 ± 1.43 |
| ✓ | ✓ | | | $5l$ | | 89.83 ± 2.55 | 94.72 ± 1.33 |
| ✓ | | ✓ | | $1l$ | | 93.07 ± 0.75 | 95.55 ± 2.09 |
| ✓ | | ✓ | | $2l$ | | 91.58 ± 2.55 | 95.02 ± 1.14 |
| ✓ | | ✓ | | $5l$ | | 90.25 ± 2.48 | 94.92 ± 1.26 |
| ✓ | ✓ | ✓ | ✓ | $1l$ | * | **93.86 ± 1.35** | 95.64 ± 1.63 |
| ✓ | ✓ | ✓ | ✓ | $2l$ | | 91.82 ± 1.34 | **96.06 ± 1.11** |
| ✓ | ✓ | ✓ | ✓ | $5l$ | | 90.76 ± 3.57 | 94.95 ± 2.13 |

Table 66: **Results of the ablation study evaluating the impact of different descriptor configurations in FLUX under the *Known Association* and *Test Phase* settings**. The configuration marked with * indicates the setting used in this paper. The table shows average accuracy and standard deviation across four distribution shifts, highlighting the contribution of various descriptors approximating $P(X)$ and $P(Y|X)$ and analyzing the effect of descriptor length ($l$) on performance.

