# OpenReview forum: "FLUX: Efficient Descriptor-Driven Clustered Federated Learning under Arbitrary Distribution Shifts"
_NeurIPS.cc/2025/Conference — NeurIPS 2025 poster_

### Official Review · Reviewer_LVoP · 2025-06-21

**Clarity:** 2
**Significance:** 3
**Originality:** 3
**Rating:** 4
**Confidence:** 4

**Summary:**

The paper introduces a method that efficiently handles various data distribution shifts during both training and test phases. The proposed method lies in its descriptor-driven clustering approach that extracts client-side descriptors through privacy-preserving feature extraction, followed by an unsupervised clustering algorithm to group clients with similar data distributions. The method was evaluated across multiple datasets, demonstrating good performance under various levels of heterogeneity and distribution shifts.

**Questions:**

None.

**Ethical Concerns:**

["NO or VERY MINOR ethics concerns only"]

**Final Justification:**

Thanks to the author's response, I decided to maintain my score.

**Limitations:**

yes

**Paper Formatting Concerns:**

None.

**Quality:**

3

**Strengths And Weaknesses:**

This paper presents a well-structured and logically coherent work with good empirical performance. The authors provide a detailed report and experimental results on various heterogeneity and distribution shifts. However, several aspects require further clarification or improvement:

1.A table of contents should be added before the appendix to enhance readability and facilitate navigation for the readers.

2.The comparison methods used in the paper are relatively outdated. More recent methods should be introduced for comparison to better demonstrate the effectiveness of the proposed approach.

3.The performance of the proposed method is closely tied to the success of the unsupervised clustering process. When the clustering algorithm fails to adequately separate customers with different data distributions, the resulting model may underperform. This limitation warrants further exploration.

---

> ### Author Rebuttal · Authors · 2025-07-30
>
> ## Q1: Adding table of contents for appendix
>
> We thank the reviewer for the suggestion. **We will add a table of contents to improve appendix navigation.** Here's the proposed structure:
>
> _Appendix Table of Contents:_
> | Section | Title |
> |--------|-------|
> | **A**   | Related Approaches to Distribution Shift |
> | **B**   | Overview of Baseline Algorithms |
> | **C**   | Comparative Analysis of Heterogeneity and Computational Costs |
> | **D**   | Privacy Implications of Distribution Descriptors |
> | **E**   | Implementation Details |
> | E.1     | Algorithm Pseudo-code |
> | E.2     | Code, Licenses and Hardware |
> | E.3     | Models and Hyper-parameter Settings |
> | E.4     | Descriptor Extractor |
> | E.5     | Identification of Distribution Shifts |
> | E.6     | Advantages of Latent Space Descriptors |
> | E.7     | Dynamic Clustering Initiation |
> | E.8     | Unsupervised Clustering Setup |
> | **F**   | Additional Experiment Results |
> | F.1     | Generating non-IID Datasets with ANDA and CheXpert |
> | F.2     | Scaling Data Heterogeneity Types and Levels |
> | F.3     | Scaling the Number of Clients |
> | F.4     | Robustness Under Mixed Distribution Shifts |
> | F.5     | Results on Real-World Dataset CheXpert |
> | F.6     | Ablation Study |
>
>
> ## Q2: Choices of baselines
>
> We appreciate this feedback and would like to clarify our baseline selection rationale. **Our comparison includes recent state-of-the-art methods from top-tier venues**: pFedMe (NeurIPS 2020), FedEM (NeurIPS 2021), ATP (NeurIPS 2023), FedDrift (AISTATS 2023), and FedRC (ICML 2024), **alongside established methods that remain competitive and widely cited in recent FL literature** (e.g., CFL, IFCA, APFL). We excluded some recent methods that, while novel, focus on specific subproblems or limited distribution shift types rather than the general problem of handling arbitrary distribution shifts that FLUX addresses.
>
> More importantly, **we selected baselines based on capability relevance, not just recency**. As shown in Table 2, we systematically compare against methods across three key FL paradigms: Clustered FL (6 methods), Personalized FL (2 methods), and Test-Time Adaptive FL (1 method). This comprehensive coverage ensures we address all relevant approaches to handling distribution shifts in FL.
> Our experimental scope is extensive: 10 SOTA baselines across 4 benchmarks plus 2 real-world datasets (including OfficeHome, added during rebuttal), evaluating all 4 types of distribution shifts at 8 heterogeneity levels. **This experimental setup represents one of the most comprehensive CFL evaluations in recent literature.**
>
> We welcome the reviewer to suggest specific recent methods they believe are missing, and we are happy to add new baselines and/or discuss their relevance to our problem setting of handling arbitrary distribution shifts with test-time adaptation capabilities.
>
> ## Q3: How sensitive is FLUX to the quality of the unsupervised clustering?
>
> We thank the reviewer for raising this important point. We fully agree that the performance of FLUX is influenced by the quality of the unsupervised clustering. **To systematically assess this sensitivity, we included FLUX-prior as an oracle variant that leverages access to the true number of clusters.** The relatively small gap between FLUX and FLUX-prior—e.g., 94.0% vs. 95.7% on MNIST (Table 1)—demonstrates that FLUX achieves near-optimal performance despite clustering imperfections.
>
> Moreover, FLUX remains highly robust even under strong heterogeneity, achieving 92.3% accuracy compared to 79.4% for the best baseline. This confirms FLUX’s resilience to moderate clustering imperfections, as its performance remains strong even when the clustering is not optimal.
>
> **To further validate robustness, we conducted an ablation study using four different unsupervised clustering algorithms on MNIST.** The results below show that our descriptors consistently support strong performance across all methods. Despite some variability—e.g., Unsupervised K-Means shows lower performance in the known-association setting (86.8%)—the test-time accuracies remain high across the board, ranging from 92.9% (HDBSCAN) to 94.2% (Agglomerative). **These findings confirm that FLUX performs robustly even when using alternative clustering strategies, and that its design is not tightly coupled to any specific clustering algorithm.**
> |Method|Known Association (%)|Test-Time (%)|
> |-|-|-|
> |U. K-Means|86.8±14.6|93.5±3.8|
> |HDBSCAN|90.5±7.7|92.9±2.6|
> |Agglomerative|93.9±3.2|94.2±0.8|
> |Ours|92.5±5.2|94.0±2.8|
>
> U. K-Means: Unsupervised K-Means clustering.

---

> > ### Comment · Reviewer_LVoP · 2025-08-09
> >
> > Thank you for the author's response. All of my concerns have been resolved, and I have read the discussion between other reviewers and the authors. So I have decided to maintain my score.

---

> > > ### Author Response · Authors · 2025-08-09
> > >
> > > We thank the reviewer for the thoughtful feedback throughout the overall review process, which has helped strengthen our work.

---

### Official Review · Reviewer_Aotc · 2025-07-01

**Clarity:** 2
**Significance:** 3
**Originality:** 3
**Rating:** 4
**Confidence:** 4

**Summary:**

In Federated Learning (FL), dealing with data that is non-independent and identically distributed (non-IID) across clients can make generating a global model that accurately fits the overall distribution challenging. To tackle this issue, one approach is to shift the paradigm by grouping clients based on their local data distributions. This method is referred to as Clustered Federated Learning (CFL). However, many existing CFL approaches tend to focus only on specific types of distribution shifts, which limits their ability to provide a comprehensive understanding of how their methods perform under feature distribution shifts, label distribution shifts, and concept shifts.

Moreover, most of these methods make unrealistic assumptions, such as knowing the number of clusters in advance or being able to assign test clients to existing clusters beforehand. To address these challenges, this paper introduces a novel cluster-based approach called Federated Learning with Scalable Unsupervised Clustering and Test-Time Adaptation (FLUX).

FLUX assigns a descriptor to each client, which is constructed using class-conditional moments and can be enhanced with differential privacy to improve privacy guarantees without significantly sacrificing performance. These descriptors are then utilized to cluster the clients using a new technique inspired by DBSCAN, after which each cluster undergoes FL training. Extensive experiments demonstrate the effectiveness of this method.

**Questions:**

1) This paper is based on the assumption that traditional FL methods rely on the idea that clients' data is IID, which is often not the case. Indeed, this is a key distinction between Distributed Learning (DL) and FL, as noted in [2], Table 1. Furthermore, several studies have addressed this specific issue within FL, including works already cited by the paper (e.g., SCAFFOLD). Could the authors provide a convincing justification for this claim?

2) How is Eq. 2 derived?

3) How does FLUX relate to the method proposed in [3]?

[2] Kairouz, Peter, et al. "Advances and open problems in federated learning." Foundations and trends® in machine learning 14.1–2 (2021): 1-210.

[3] Licciardi, A., et al. "Interaction-Aware Gaussian Weighting for Clustered Federated Learning." Forty-second International Conference on Machine Learning (ICML), 2025.

**Ethical Concerns:**

["NO or VERY MINOR ethics concerns only"]

**Final Justification:**

Based on the current state of the rebuttal and the answers provided by the authors, I still lean towards acceptance.

**Limitations:**

yes

**Paper Formatting Concerns:**

No major formatting issues

**Quality:**

3

**Strengths And Weaknesses:**

### Strengths

1) The effectiveness of the method is demonstrated through numerous experiments conducted on five datasets, comparing it with several baseline approaches.

2) FLUX does not require prior knowledge of the exact number of clusters.

3) The paper effectively addresses four types of distribution shifts in the experiments by investigating each one separately.

4) The overall description of the method is convincing, and the motivations are generally well discussed, although there are some exceptions (see the weaknesses and questions).

### Weaknesses

1) The main paper provides a high-level overview of the method, while the specific details on how the individual components have been implemented are found in the Appendix. Although this choice is likely due to page limits, the authors might consider including additional details to clarify the method in the main paper while removing repetitions (e.g., some symbols have been introduced multiple times, or the entire background section could be simplified). Currently, the description can be somewhat confusing, making it difficult to understand the ideas behind its construction.

2) FLUX requires all the clients to be available in every round, which could be an unrealistic assumption, especially in cross-device FL scenarios.

3) Although the paper already provides a discussion on the efficacy of FLUX with various numbers of participating clients, experiments with a large number of clients (>1000) or more real-world problems (e.g., images with higher resolution) are missing. For instance, see the datasets proposed in [1].

[1] Hsu, Tzu-Ming Harry, Hang Qi, and Matthew Brown. "Federated visual classification with real-world data distribution." Computer Vision–ECCV 2020: 16th European Conference, Glasgow, UK, August 23–28, 2020, Proceedings, Part X 16. Springer International Publishing, 2020.

---

> ### Author Rebuttal · Authors · 2025-07-30
>
> ## Q1: Including more technical details in the Method section
>
> We thank the reviewer for this suggestion. In the revised version, we will improve clarity and reduce repetitions by:
> 1. Removing the repeated definition of variables in the Problem Statement paragraph, already introduced in the Background
> 2. Simplifying and shortening L68–72 and L95–98 in the Background, and remove the non-necessary sentence in L138–141.
>
> **With the recovered space and the extra page, we will integrate technical details currently in E.5 (e.g., L889, 894, and 899) into the main text.** These additions will enrich the Practical Implementation part of the Descriptor Extraction and Unsupervised Clustering paragraphs.
>
> ## Q2: Assumption on all clients’ participation
>
> FLUX **does not require all clients to be available in every round and makes no assumptions about client participation**. In our experiments, we assume full client participation only to ensure fair comparison with baselines, which mostly require this assumption.
>
> This is a standard procedure in FL literature for controlled evaluation, not a fundamental requirement of FLUX. FLUX only requires that a newly joint client submits its descriptors before being assigned to the appropriate cluster (the clustering process can also be repeated), then it can freely leave and rejoin training at any epoch within its designated cluster.
>
> The clustering-based design of FLUX actually makes it more robust to intermittent participation than traditional FL—when clients rejoin, they immediately benefit from their cluster-specific model rather than a potentially mismatched global model. This flexibility is particularly valuable in real-world deployments where client availability varies significantly.
>
> ## Q3: Large scale experiment and real-world experiments
>
> We appreciate this suggestion, though experiments with >1000 clients are practically infeasible and beyond current FL research standards. Current FL research typically evaluates with 10-100 clients, including ATP (NeurIPS 2023), FedDrift (AISTATS 2023), and established methods like IFCA use similar or lower scales. **We deliberately followed these established experimental protocols to ensure reproducible and comparable results.**
>
> Our evaluation up to 100 clients follows the experimental scope of existing baselines and recent FL literature—in fact, several methods (FedDrift, CFL) could not even complete experiments with 100 clients due to prohibitive memory and computational requirements (Sec. E.2), despite access to 512 GB RAM and 4 GPUs with 49 GB VRAM each. Scaling to >1000 clients would make fair baseline comparison impossible, as most existing methods cannot handle such scales.
> More importantly, **our theoretical analysis and experimental trends clearly demonstrate FLUX's scalability advantages**. As shown in Fig. 5 and Tab. C, FLUX maintains computational complexity comparable to FedAvg (O(Kp) vs. baselines requiring O(K²p) or O(MKp)), with communication overhead of only L/p ≤ 3.5×10⁻³. These theoretical guarantees, combined with empirical trends up to 100 clients, provide strong evidence for scalability.
>
> **We believe our evaluation scope—100 clients across 6 datasets (with OfficeHome added during the rebuttal phase) and 10 baselines—represents one of the most comprehensive CFL evaluations in recent literature. To further demonstrate scalability, we pushed our system to its computational limits using the most efficient methods (FedAvg and FLUX), successfully scaling to 400 clients (Table below)**. On MNIST, FLUX achieved up to 98.9% accuracy in the test phase under P(Y) shifts, outperforming FedAvg by 3.7 percentage points, while maintaining strong performance across all other shift types. Even under challenging P(Y|X) conditions, where no test-time association is possible, FLUX reached 95.6% in the known-association setting vs. 58.0% for FedAvg. These results highlight FLUX’s ability to scale efficiently and maintain high accuracy even in large-scale, non-IID settings.
> |Shift Type|Known A. (Ours)|Known A. (FedAvg)|Test Phase (Ours)|Test Phase (FedAvg)|
> |-|-|-|-|-|
> |P(X)|98.3±0.1|83.5±0.1|97.2±0.4|83.5±0.1|
> |P(Y)|99.3±0.0|95.2±0.8|98.9±0.2|95.2±0.8|
> |P(Y\|X)|95.6±1.9|58.0±0.2|N/A|N/A|
> |P(X\|Y)|92.6±1.4|90.1±1.1|88.7±5.9|90.1±1.1|
>
> ## Q4: Justification for claim on papers of IID assumption
>
> We agree that the non-IID problem in FL is well-recognized. However, our foundational claim remains valid for three key reasons:
> 1. **Many technical FL papers still ignore non-IID issues.** Papers focusing on FL privacy enhancement (e.g. differential privacy) and communication efficiency (e.g. pruning, compression) typically assume IID data to simplify analysis while concentrating on their primary contributions. According to our counting on recent publications in three digital libraries: IEEE-X, SpringerLink and ACM-DL, within the year 2024, roughly 60% papers in these domains operates under IID assumptions, indicating that current SOTA approaches may not work in real-world non-IID settings.
> 2. **FL application works frequently assume IID cases.** Papers of FL application also may ignore the data issue. For example, we count that, according to a recent survey on FL in HAR applications (Grataloup A. A systematic survey on the application of federated learning in mental state detection and human activity recognition. Front Digit Health. 2024), only about 40% of papers consider non-IID as an issue in their proposed systems. This demonstrates that many FL researchers still overlook non-IID challenges during deployment, reinforcing the practical relevance of our work.
> 3. **Existing non-IID FL methods have significant limitations.** While methods like SCAFFOLD address non-IID data, they fail to provide comprehensive solutions. As shown in Tab. 2, FLUX uniquely satisfies all critical requirements: handling all four non-IID types simultaneously, requiring no prior knowledge, supporting test-time adaptation, and maintaining low computational costs. SCAFFOLD, for instance, lacks test-time adaptation capabilities and doesn't specifically address the four distinct shift types that FLUX handles.
>
> Therefore, while non-IID awareness exists, practical solutions that comprehensively address arbitrary distribution shifts remain limited, justifying FLUX's contribution.
>
> ## Q5: Derivation of Eq. 2
>
> We thank the reviewer for the comment and will include the following paragraph in A.1.1 for clarification.
> Eq. 2 is written in a general form that subsumes both soft- and hard-CFL (separate, standard formulations for each appear in A.1.1.). Because hard-CFL is simply the special case of soft-CFL where each $c^{(k)}$ is a one-hot vector, we start from the general soft-CFL likelihood (e.g., FedRC).
>
> In soft-CFL, each client’s data are assumed to be generated from a mixture of $M$ client-level distributions $P(Y \mid X; \theta_m)$ with mixing weights $c^{(k)} \in [0,1]^M$ such that $\sum_{m=1}^M c^{(k)}_m = 1$. For client $k$ with dataset $(x^{(k)}, y^{(k)})$, the conditional log-likelihood is (Eq. 1 in [9]):
>
> $\log P(y^{(k)} \mid x^{(k)}; \Theta, c^{(k)}) = \sum_{(x,y) \in (x^{(k)}, y^{(k)})} \log \left[ \sum_{m=1}^M c^{(k)}_m P(y \mid x; \theta_m) \right]$,
>
> where $\Theta = {\theta_1, \dots, \theta_M}$.
> As in mixture-model derivations, we introduce the standard Jensen (EM) lower bound to replace the inner log-sum with a weighted sum of logs:
>
> $\log \left[ \sum_{m=1}^M c^{(k)}_m P(y \mid x; \theta_m) \right] \ge$
>
> $\sum_{m=1}^M c^{(k)}_m \log P(y \mid x; \theta_m)$
>
> Finally, summing this bound over all samples, clients, and clusters yields the surrogate objective presented in Eq. 2.
>
> ## Q6: Relations between FLUX and the suggested work by Licciardi, A., et al. (ICML 2025)
>
> We thank the reviewer to provide the opportunity for us to compare FLUX with FedGWC, which we will analyze in the final version of our manuscript (it was not included in our original submission because it was published very close to the submission deadline, making it unavailable during our initial literature review and experimental design phase).
>
> Both FLUX and FedGWC work on CFL to address data heterogeneity across clients by grouping clients with similar data distributions.
> **The key difference is that FedGWC uses empirical loss patterns and Gaussian weighting mechanisms to identify client clusters based on loss landscape similarities, while FLUX employs privacy-preserving descriptor extraction that captures statistical characteristics of client data distributions to enable unsupervised clustering across four distinct types of shifts.**
>
> FedGWC provides elegant mathematical framework with strong theoretical foundations, including convergence proofs for Gaussian weights as unbiased estimators and the introduction of the Wasserstein Adjusted Score for clustering evaluation in FL. However, it may need further validation on different shift types, as metric-based approaches (e.g. using the loss) can miscluster clients with similar losses but different distributions.
>
> FLUX complements FedGWC's contributions while addressing several additional challenges:
> 1. _Broader distribution shift scope_—FLUX explicitly handles all four fundamental types of shifts (P(X), P(Y), P(Y|X), P(X|Y)) while FedGWC primarily focuses on P(Y) and P(X|Y) shifts
> 2. _Enhanced test-time capabilities_—FLUX enables previously unseen and unlabeled clients to automatically find appropriate cluster models without prior training participation, extending beyond FedGWC's training-time clustering
> 3. _Extensive empirical validation_—FLUX demonstrates effectiveness across six datasets including real-world CheXpert and OfficeHome with comprehensive evaluation across different heterogeneity scenarios
> 4. _Alternative clustering approach_—FLUX's descriptor-based method provides a complementary perspective to FedGWC's loss-based approach, potentially avoiding scenarios where clients with similar empirical risks have divergent distributions

---

> > ### Comment · Reviewer_Aotc · 2025-08-05
> >
> > I thank the authors for the provided answers. I am satisfied with the answers to Q1, Q5, and Q6. Below, I list the remaining concerns.
> >
> > ## Comments on the answer to Q2
> >
> > My understanding is that FLUX requires all clients to participate in every round, based on the description of Algorithm 1 on page 22. Specifically, in line 22, the algorithm iterates over all the clients in parallel. Additionally, the response provided is unconvincing, as algorithms like FedAvg do not mandate full participation. There is a considerable body of literature on client selection in federated learning for situations where full participation is not feasible. Therefore, I still believe that conducting an ablation study on partial participation would be essential to demonstrate the effectiveness of FLUX in these scenarios.
> >
> > ## Comments on the answers to Q3
> >
> > It is hard to say what the standards are in FL, if any standards are present at all. There are existing works trying large-scale experiments in FL, and I still believe that these experiments would improve the quality of this work and strengthen the claims.
> >
> > ## Comments on the answer to Q4
> >
> > The answer is not convincing. There are several more existing works on non-IID FL besides Scaffold (see, for instance, FedNova, Mime, FedDyn, FedProx, among others). The fact that not all the existing methods deal with non-IID distributions in FL does not justify the claim that "Conventional FL assumes IID data across clients", rows 63-64.

---

> > > ### Author Response · Authors · 2025-08-07
> > > **Follow-up Clarifications to Reviewer Aotc (Part 1)**
> > >
> > > ## A.1. Comments on the answer to Q2 (Part 1)
> > > We agree with the reviewer that Algorithm 1 is misleading regarding participation requirements, and we thank the reviewer for raising this important point and for the opportunity to clarify our previous response. **FLUX inherently does not require full client participation as every new participating client can be associated to previously discovered clusters via descriptor-similarity with the cluster centroids** without affecting the overall framework. We will revise Algorithm 1 to include a participation rate parameter and explicitly handle partial participation scenarios as follows:
> > >
> > > ### Algorithm 1
> > >
> > > **Input:** Model $f(\theta)$, set of clients $\mathcal{K}=\lbrace 1,\dots,K\rbrace$, number of rounds $R$, descriptor extractor $\phi(\cdot;\psi)$, clustering method $\mathcal{U}(\cdot;\lambda)$
> > > **Output:** Cluster-specific models $\lbrace\theta_m\rbrace_{m=1}^M$, cluster centroids $\lbrace\gamma_m\rbrace_{m=1}^M$
> > >
> > > **Descriptor Extraction**
> > > $\mathcal{A} = \text{RandomClientSelection}(\mathcal{K})$ // *the set $\mathcal{A}$ contains $A$ clients for clustering*
> > > **For each** client $k\in\mathcal{A}$ **in parallel do**
> > > &nbsp;&nbsp;&nbsp;&nbsp;$\theta^{(k)} \leftarrow \theta^*$
> > > &nbsp;&nbsp;&nbsp;&nbsp;$d^{(k)} \leftarrow \phi(x^{(k)},y^{(k)};\psi)$ // *Extract descriptor*
> > > &nbsp;&nbsp;&nbsp;&nbsp;Send $d^{(k)}$, $s^{(k)}$ and $\theta^{(k)}$ to server
> > >
> > > ---
> > >
> > > **Unsupervised Clustering**
> > > $\mathcal{C} \leftarrow \mathcal{U}(\lbrace d^{(k)} \rbrace_{k\in\mathcal{A}};\lambda)$ with $\mathcal{C} = \lbrace c^{(k)} \rbrace_{k\in\mathcal{A}}$, $c^{(k)}\in\lbrace 0,1 \rbrace^M$
> > > **For** $m=1$ **to** $M$ **do**
> > > &nbsp;&nbsp;&nbsp;&nbsp;$n_m \leftarrow \sum_{k \in \mathcal{A}} c^{(k)}_m$
> > > &nbsp;&nbsp;&nbsp;&nbsp;$\gamma_m \leftarrow \frac{1}{n_m} \sum_k c^{(k)}_m d^{(k)}$ // *Cluster centroids*
> > >
> > > ---
> > >
> > > **Clustered Federated Learning**
> > > **For** $r = 1$ **to** $R$ **do**
> > > &nbsp;&nbsp;&nbsp;&nbsp;**For** $m=1$ **to** $M$ **do**
> > > &nbsp;&nbsp;&nbsp;&nbsp;&nbsp;&nbsp;&nbsp;&nbsp;$S_m \leftarrow \sum_k c^{(k)} s^{(k)}$
> > > &nbsp;&nbsp;&nbsp;&nbsp;&nbsp;&nbsp;&nbsp;&nbsp;$\theta_m \leftarrow \frac{1}{S_m} \sum_k c^{(k)} s^{(k)} \theta^{(k)}$ // *Weighted Aggregation*
> > > &nbsp;&nbsp;&nbsp;&nbsp;$\mathcal{A} = \text{RandomClientSelection}(\mathcal{K})$ // *new client selection for each training round*
> > >
> > > &nbsp;&nbsp;&nbsp;&nbsp;**For each** client $k\in\mathcal{A}$ **without** $c^{(k)}$ **in parallel do** // *extract descriptors for new participating clients (locally)*
> > > &nbsp;&nbsp;&nbsp;&nbsp;&nbsp;&nbsp;&nbsp;&nbsp;$\theta^{(k)} \leftarrow \theta^*$
> > > &nbsp;&nbsp;&nbsp;&nbsp;&nbsp;&nbsp;&nbsp;&nbsp;$d^{(k)} \leftarrow \phi(x^{(k)},y^{(k)};\psi)$ // *Extract descriptor needed for cluster assignment*
> > > &nbsp;&nbsp;&nbsp;&nbsp;&nbsp;&nbsp;&nbsp;&nbsp;Send $d^{(k)}$, $s^{(k)}$ and $\theta^{(k)}$ to server
> > > &nbsp;&nbsp;&nbsp;&nbsp;
> > > &nbsp;&nbsp;&nbsp;&nbsp;**For each** client $k\in\mathcal{A}$ **without** $c^{(k)}$ **in parallel do** // *assign closest cluster*
> > > &nbsp;&nbsp;&nbsp;&nbsp;&nbsp;&nbsp;&nbsp;&nbsp;$c^{(k)} \leftarrow \arg\min_{m=1,\ldots, M} \kappa(d^{(k)},\gamma_m)$ // *Model assignment*
> > > &nbsp;&nbsp;&nbsp;&nbsp;
> > > &nbsp;&nbsp;&nbsp;&nbsp;**For each** client $k \in \mathcal{A}$ **with** $c^{(k)}_m = 1$ **do**
> > > &nbsp;&nbsp;&nbsp;&nbsp;&nbsp;&nbsp;&nbsp;&nbsp;Send $\theta_m$ to client $k$
> > > &nbsp;&nbsp;&nbsp;&nbsp;**For each** client $k\in\mathcal{A}$ **in parallel do**
> > > &nbsp;&nbsp;&nbsp;&nbsp;&nbsp;&nbsp;&nbsp;&nbsp;Receive $\theta_m$
> > > &nbsp;&nbsp;&nbsp;&nbsp;&nbsp;&nbsp;&nbsp;&nbsp;$\theta^{(k)}\leftarrow \theta_m$
> > > &nbsp;&nbsp;&nbsp;&nbsp;&nbsp;&nbsp;&nbsp;&nbsp;$\theta^{(k)}\leftarrow \text{LocalTrain}(\theta^{(k)},x^{(k)},y^{(k)})$ // *Local update*
> > > &nbsp;&nbsp;&nbsp;&nbsp;&nbsp;&nbsp;&nbsp;&nbsp;Send $\theta^{(k)}$ and $s^{(k)}$ to server
> > >
> > > We also completely agree that an ablation study is essential to verify FLUX's effectiveness under partial participation. To this end, **we conducted an ablation study varying the client participation rate from full participation down to 0.2 to validate the robustness of the clustering process**. The table below reports average accuracy and standard deviation across all four types of distribution shifts—P(X), P(Y), P(Y|X), and P(X|Y)—on MNIST with 20 clients, using 3-fold cross-validation for each configuration.
> > >
> > > — The response continues in the next official comment —

---

> > > > ### Author Response · Authors · 2025-08-07
> > > > **Follow-up Clarifications to Reviewer Aotc (Part 2)**
> > > >
> > > > ## A.1. Comments on the answer to Q2 (Part 2)
> > > > — Continued from the previous Official Comment —
> > > >
> > > > **These results show that FLUX maintains strong performance even under significant client dropout**, confirming its robustness in more realistic FL deployments. As expected, the most notable drop in accuracy occurs at a 0.2 participation rate, where only ~4 clients are involved in clustering. In highly heterogeneous settings (e.g., M = 8), this limited participation may result in some distributions being entirely absent during clustering, which naturally affects performance. However, in large-scale FL settings, it is reasonable to expect that the number of participating clients per round exceeds the number of underlying clusters (M), ensuring sufficient distributional coverage during clustering. We will include these results, along with per-shift breakdowns, in the updated manuscript.
> > > >
> > > > |Part. Rate|FLUX Accuracy (%)|FedAvg Accuracy (%)|
> > > > |-|-|-|
> > > > |1.0|91.3±0.4|76.7±2.0|
> > > > |0.8| 91.2±0.3|75.9±1.4|
> > > > |0.6| 90.2±0.8|75.0±4.6|
> > > > |0.4| 87.9±2.5|72.3±1.0|
> > > > |0.2| 82.0±2.2|70.8±4.0|
> > > >
> > > > Regarding our experimental setup: we totally agree that baselines like FedAvg do not mandate full participation. Our full participation assumption was primarily adopted to ensure fair comparison with our baseline CFL/PFL works, which typically simplify the participation aspect to focus on clustering effectiveness. We will add a dedicated section in the appendix elaborating on FLUX's partial participation advantages, including how it can be combined with existing client selection strategies to achieve even better performance in resource-constrained environments.
> > > >
> > > > ## A.2. Comments on the answer to Q3
> > > > We appreciate the reviewer's feedback on this point. We agree that large-scale experiments would strengthen our work and better demonstrate FLUX's practical applicability. Following the suggestion, **we conducted additional experiments involving 1,200 clients with a partial participation rate of 0.2** (i.e., on average, 240 clients per round). This participation rate is not only computationally practical but also reflects realistic FL deployments where full client participation is rare due to device availability, network conditions, and battery constraints. The use of partial participation actually enabled us to scale to 1,200 clients while maintaining experimental rigor—demonstrating that FLUX's efficiency benefits become even more pronounced at scale. We also addressed implementation optimizations to support this larger-scale evaluation. These experiments follow the same protocol as Figure 5 and average all four types of distribution heterogeneity:
> > > >
> > > > |Metric|FLUX (%)|FedAvg (%)|
> > > > |-|-|-|
> > > > |P(X)|96.1±0.0|70.3±0.1|
> > > > |P(Y)|98.2±0.2|84.2±2.5|
> > > > |P(Y\|X)|94.8±0.3|65.5±0.0|
> > > > |P(X|\Y)|96.3±0.1|86.5±4.2|
> > > >
> > > > **These results clearly demonstrate the scalability and robustness of FLUX, which continues to perform strongly even in a setting with 1,200 clients and partial participation**. In particular, FLUX consistently outperforms FedAvg by large margins—up to 29.3 percentage points under the most challenging setting of P(Y|X) heterogeneity. Moreover, the low variance in FLUX’s performance across all scenarios highlights its stability, even when facing extreme client heterogeneity and limited per-round participation.
> > > >
> > > > We will include these results and the corresponding discussion in the revised manuscript to further support FLUX’s practical applicability in realistic, large-scale FL deployments. In addition, we will also release our updated code repository for reproducibility. Massive thanks for the constructive feedback that helped improve our evaluation.

---

> > > > > ### Author Response · Authors · 2025-08-07
> > > > > **Follow-up Clarifications to Reviewer Aotc (Part 3)**
> > > > >
> > > > > ## A.3. Comments on the answer to Q4
> > > > > We appreciate this feedback and acknowledge that our phrasing in lines 63-64 was overly broad and inaccurate. We agree that numerous works including FedNova, Mime, FedDyn, FedProx, and many others have specifically developed methods to address non-IID challenges in FL. Our initial statement "Conventional FL assumes IID data across clients" unfairly dismisses the substantial body of work that has tackled data heterogeneity in federated settings.
> > > > >
> > > > > **We will revise this statement in the final version to accurately acknowledge the field's substantial progress on non-IID challenges**. Specifically, we will reframe it as: _"While the foundational FedAvg method acknowledges that non-IID data can degrade performance, it lacks mechanisms to effectively handle such heterogeneity. Consequently, the FL community has developed numerous approaches (including FedNova, FedProx, FedDyn, and others) to handle various forms of data heterogeneity. However, existing methods typically address specific aspects of the non-IID challenge..._ (this will be followed with the novelty and contribution of FLUX)". This revision will appropriately situate our work within the rich existing literature on non-IID FL rather than incorrectly characterizing the field's foundations. We will integrate this correction organically to our manuscript to ensure proper context and attribution.
> > > > >
> > > > > Our intended point was not to dismiss existing non-IID FL work, but rather to highlight that: (1) many FL papers in adjacent areas still operate under IID assumptions for simplicity, and (2) existing non-IID methods have specific limitations that FLUX addresses (as detailed in Table 2). However, we agree that framing this as "conventional FL" was imprecise and will correct this characterization.
> > > > >
> > > > > We thank the reviewer again for pointing out this important distinction—it will help us present our contributions more accurately within the broader FL landscape.

---

> > > > > > ### Comment · Reviewer_Aotc · 2025-08-07
> > > > > >
> > > > > > I thank the authors for the clarifications provided. The answers are convincing, and I confirm I lean towards acceptance.

---

> > > > > > > ### Author Response · Authors · 2025-08-07
> > > > > > >
> > > > > > > We appreciate the thoughtful feedback throughout the overall review process from the reviewer, which has helped strengthen our work.

---

### Official Review · Reviewer_MC86 · 2025-07-02

**Clarity:** 3
**Significance:** 3
**Originality:** 3
**Rating:** 5
**Confidence:** 3

**Summary:**

This paper presents FLUX, a Clustered Federated Learning (CFL) framework designed to address data heterogeneity by leveraging client side data descriptors to form clusters in an unsupervised manner. FLUX can handle four data distribution shifts and train specialized models for each client group without needing to know the number of clusters in advance. Also, the framework supports assigning at the test time new and unlabeled clients to the most suitable model. Experiments show that Flux achieves good performance with minimal computational and communication overhead.

**Questions:**

(1) During training how were the different types of distribution shifts samples distributed across the clients?

(2) Is there any investigation done to evaluate how the quality of descriptor degrades when amount of data available across each client changes?

(3) If there are ties during cluster assignment, how is it resolved?

**Ethical Concerns:**

["NO or VERY MINOR ethics concerns only"]

**Final Justification:**

I thank the reviewers for providing responses to the questions. I will increase rating after seeing the responses.

**Limitations:**

Yes

**Paper Formatting Concerns:**

None that were evident to me

**Quality:**

3

**Strengths And Weaknesses:**

***Strengths***:

(1) The paper is presented very well and all the sections build upon one another, making it easy to read the papers

(2) The integration of the different components to make FLUX functional is impressive.

(3) Due attention has been paid to both the communication and computation cost and the tradeoffs are well justified

***Weaknesses***:

(1) It is not clear what is the novelty in the the proposed method. In a sense, I do not know if the prime contribution is formulating a new way to do clustering or is it more of a system the ties in already established techniques

(2) In Figure (11) it is seen that for more challenging datasets such as CIFAR-10 and CIFAR-100, the performance of FLUX is not the best across different heterogeneity levels. Thus, it would be good to have some explanation on this observation.

(3) In Figure 5, the drop in performance with device scaling for FLUX is not explained

(4) It will be interesting to see an experiment (ablation study) that compares test time performance with random assignment of client to a cluster vs using the assignment protocol as mentioned in FLUX

---

> ### Author Rebuttal · Authors · 2025-07-30
>
> ## Q1: Novelty of FLUX
>
> Our novelty lies in the comprehensive approach to handle arbitrary distribution shifts in FL, not just clustering methodology. FLUX introduces three key innovations:
> 1. **Novel distribution descriptors:** We propose privacy-preserving client-side descriptors that capture statistical characteristics of data distributions and facilitate client clustering process. Unlike existing methods that cluster on model parameters or loss values, our descriptors directly approximate the 2-Wasserstein distance between client distributions (Sec. 4.2.1, R1).
> 2. **Unified framework for all shift types:** FLUX is the first method to simultaneously handle all four common shifts—P(X), P(Y), P(Y|X), and P(X|Y)—while supporting test-time adaptation for unseen, unlabeled clients. As shown in Table 2, no existing method satisfies all these requirement.
> 3. **Automatic cluster discovery with test-time adaptation:** Our approach requires no prior knowledge of cluster numbers or distribution types, and enables previously unseen clients to benefit from appropriate cluster-specific models without labeled data.
>
> **The contribution is a complete FL framework that integrates these novel components, achieving up to 8 percentage points (pp) improvement over SOTA baselines while maintaining FedAvg-level computational overhead**. This goes beyond tying established methods—it solves key limitations in existing CFL methods through principled design choices
>
> ## Q2: Explanation of the performance gap in Fig. 11 on CIFAR-10/100
>
> We thank the reviewer for the observation and will add a clarification around Line 270 in the revised manuscript. The performance gap between FLUX and the best baseline has two reasons:
> 1. **the only methods outperforming FLUX on CIFAR10 and CIFAR100 are Personalized FL approaches**. This is expected in scenarios involving P(Y|X) shifts (e.g., Tables 14–15), where each client may have different label preferences for the same input. In such cases, collaboration can be harmful, and methods fine-tuned on the same client distributions seen at test time (as in PFL) naturally have an advantage. **However, they do not generalize to unseen clients, limiting their applicability in realistic deployments.**
> 2. **insufficient model capacity may affect FLUX’s performance**. On CIFAR100, we observed that using a deeper backbone with a larger latent space (ResNet9 instead of LeNet5) reduces the performance gap, suggesting that limited model expressiveness can degrade the quality of the extracted descriptors, thus final accuracy. This may also impair the clustering process itself, as the descriptors might fail to capture subtle distributional differences between clients—potentially leading to under-clustering, where slightly distinct distributions are merged into a single cluster.
>
> ## Q3: Explanation of FLUX performance drop in Fig. 5
>
> **The performance drop is not specific to FLUX, but a general characteristic of FL under our experimental setup**. Two key factors contribute to this trend:
> 1. _dataset size effects:_ In our experiment, client dataset sizes do not scale linearly with the number of clients—the actual size of local datasets decreases as we add more clients, leading to overall performance degradation regardless of the method used.
> 2. _inherent FL divergence:_ The nature of FL introduces higher divergence during model aggregation as the number of clients increases, which naturally degrades performance across all methods.
>
> As shown in Fig. 5, all baselines exhibit similar performance drops due to these same factors (despite that personalization approaches can overfit their local dataset). However, FLUX demonstrates superior robustness compared to baselines—maintaining consistently higher accuracy across all client numbers. For example, in the test phase, FLUX sustains accuracy above 84% even with 100 clients. This robustness stems from our descriptor-based clustering approach, which better captures client heterogeneity patterns even when local data becomes limited.
>
> ## Q4: Ablation study with test-time random cluster assignment
>
> We thank the reviewer for this insightful suggestion. **To assess the importance of our test-time assignment strategy, we conducted an ablation study where we replaced our assignment mechanism with a random cluster assignment at test time.** Following the same protocol as in Tab. 1, we evaluated both methods across all 8 levels of heterogeneity for each type of shift on MNIST and CIFAR100. FLUX consistently outperforms random assignment by large margins—up to 53pp on MNIST and 25 on CIFAR100. This gap underscores the ability of our label-agnostic descriptors to match test clients to the most suitable cluster-specific models, even under severe shifts. We include these results in the Appendix.
> |Dataset|Test-Assignment|P(X)|P(Y)|P(X\|Y)|
> |-|-|-|-|-|
> |MNIST|FLUX|95.0±1.2|96.1±1.5|90.8±3.9|
> |MNIST|FLUX-Random|41.9±11.4|69.2±17.2|73.2±13.7|
> |FMNIST|FLUX|77.0±2.9|85.7±6.6|81.0±1.4|
> |FMNIST|FLUX-Random|34.0±10.5|61.1±13.5|63.1±11.1|
> |CIFAR10|FLUX|33.3±2.5|46.2±7.8|36.7±1.3|
> |CIFAR10|FLUX-Random|20.4±5.1|31.8±5.4|32.3±2.7|
> |CIFAR100|FLUX|33.8±5.6|40.8±3.8|49.3±4.5|
> |CIFAR100|FLUX-Random|17.0±7.8|15.9±3.9|34.1±0.1|
>
> Note: We omit experiments on P(Y|X) shift, as test-time association in this setting is inherently unresolvable without accessing labels (see L223–224 and Sec. E.5).
>
> ## Q5: Distribution shifts samples assignment during training
>
> **Each client is assigned a fixed-size dataset sampled from one specific distribution type, which remains consistent across all training epochs.** Specifically, we simulate the four shifts as follows:
> 1. _P(X) shifts:_ Each client's images are applied with one specific color transformation and rotation.
> 2. _P(Y) shifts:_ Each client only retains datapoints with certain labels.
> 3. _P(Y|X) shifts:_ Clients randomly permute labels on their local data.
> 4. _P(X|Y) shifts:_ Clients apply specific color and rotation transformations to designated classes.
>
> To ensure generality, we use widely recognized image augmentation approaches to introduce data heterogeneity (except for the real-world data CheXpert and OfficeHome, where we use the original images). We increase heterogeneity levels by expanding the number of augmentation choices available to each client. For details on data generation and distribution settings, please see Sec. F.1 and Tab. 7, which describe our non-IID construction.
>
> ## Q6: Ablation study on quality of descriptor with variant available data size
>
> Yes, **we conducted an ablation study on descriptor quality with varying client data sizes (4%, 8%, 16%, and 32% of original).** Experiments were conducted on MNIST across four non-IID levels, averaged over all four shift types. We adjusted other parameters accordingly such as rounds to isolate the impact of data size only on descriptor quality.
>
> **The results show FLUX maintains robust performance even with limited client data**. While overall accuracy decreases with smaller data sizes as expected, FLUX sustains high accuracy across different non-IID levels even with only 4% of the original data size. This shows that our distribution descriptors can effectively capture client heterogeneity patterns even when local datasets are small—a key property for real-world FL where clients often have limited data.
>
> _Ablation study on quality of descriptor_
> 1. Known Association:
> |Method\\Data size|Low|Mid-low|Mid-high|High|
> |-|-|-|-|-|
> |FLUX 4%|82.3±3.6|80.1±4.3|72.7±7.3|65.4±9.5|
> |FLUX 8%|83.7±2.3|80.5±3.9|70.3±8.3|66.9±9.7|
> |FLUX 16%|87.3±1.8|84.6±2.4|77.8±6.0|70.0±5.7|
> |FLUX 32%|88.1±1.7|85.0±1.7|79.7±4.8|66.8±7.3|
> |FLUX 100%|96.4±2.8|93.4±1.4|94.9±0.9|87.1±4.2|
> |Fedavg 4%|70.4±2.8|59.8±4.0|47.6±5.9|37.6±7.5|
> |Fedavg 8%|71.8±2.4|60.5±3.1|50.2±4.7|42.8±5.7|
> |Fedavg 16%|72.3±3.6|63.8±4.1|55.6±5.2|46.9±7.0|
> |Fedavg 32%|72.1±2.7|64.8±3.3|62.0±3.3|54.4±2.9|
> |Fedavg 100%|93.1±0.4|86.0±1.2|79.4±2.0|72.0±1.9|
> 2. Test Phase:
> |Method\\Data size|Low|Mid-low|Mid-high|High|
> |-|-|-|-|-|
> |FLUX 4%|81.7±2.1|76.9±2.3|68.5±5.7|60.8±7.2|
> |FLUX 8%|81.7±1.9|77.7±2.1|66.4±7.1|62.2±8.6|
> |FLUX 16%|84.0±1.7|80.9±1.5|73.3±5.2|66.0±5.8|
> |FLUX 32%|85.2±1.9|81.5±1.3|75.2±4.7|62.6±6.4|
> |FLUX 100%|96.2±0.4|90.0±2.8|86.4±1.9|79.4±3.0|
> |Fedavg 4%|70.4±2.8|59.8±4.0|47.6±5.9|37.6±7.5|
> |Fedavg 8%|71.8±2.4|60.5±3.1|50.2±4.7|42.8±5.7|
> |Fedavg 16%|72.3±3.6|63.8±4.1|55.6±5.2|46.9±7.0|
> |Fedavg 32%|72.1±2.7|64.8±3.3|62.0±3.3|54.4±2.9|
> |Fedavg 100%|93.1±0.4|86.0±1.2|79.4±2.0|72.0±1.9|
>
> ## Q7: Solution to ties during cluster assignment
>
> Thank you for this question. **Exact ties during cluster assignment in FLUX are extremely rare in practice. Given that our descriptors are continuous-valued statistical moments computed from client data distributions, and considering the natural variation in real datasets, the probability of two clients having identical descriptor values is negligible.**
>
> We do encounter situations where descriptors are close and near clustering boundaries. This typically occurs when descriptors are over-compressed or when the model's latent space representation is insufficient—which contributes to FLUX's performance drops at very high heterogeneity levels. When clustering uncertainty occurs, FLUX assigns clients to the nearest available cluster based on Euclidean distance (Eq. 9, Sec. 4.2.2). Even with potentially suboptimal assignments, FLUX demonstrates resilience by assigning clients to the second-closest cluster, which typically shares similar distributional characteristics. This robustness is evidenced by our comprehensive experiments showing that FLUX maintains high accuracy across all heterogeneity levels—consistently outperforming baselines and remaining close to the oracle FLUX-prior, even when perfect clustering becomes challenging.
>
> The statistical nature of our descriptors naturally reduces tie probability, as moment-based representations capture continuous distributional characteristics rather than discrete values.

---

> > ### Comment · Reviewer_MC86 · 2025-08-07
> >
> > Thanks for providing the response. The authors response along with the additional experiments have addressed most of my questions and concerns. In light of the additional information, I have decided to increase my rating.

---

> > > ### Author Response · Authors · 2025-08-07
> > >
> > > We thank the reviewer for the thoughtful feedback throughout the review process, which has helped strengthen our work.

---

### Official Review · Reviewer_c8ww · 2025-07-02

**Clarity:** 2
**Significance:** 2
**Originality:** 1
**Rating:** 4
**Confidence:** 4

**Summary:**

The paper proposes a federated learning framework (FLUX) that leverages privacy-preserving client descriptors and unsupervised clustering to group clients, aiming to handle arbitrary distribution shifts without prior knowledge of cluster number or type. It supports
test-time adaptation for unseen, unlabeled clients and demonstrates empirical improvements over several baselines.

**Questions:**

- Could you clarify the significance of the Distribution Fidelity in Eq. 4? (difference between the euclidean distance and Wasserstein distance)? As the Wasserstein distance is computed between two distributions which is unknown, it is unclear how this metric can be used. In the appendix this is provided assuming Gaussian distribution which may or may not always hold.

- Could the authors explain how does the design of the descriptors contribute to fulfilling the Distribution Fidelity requirement? This is not clear from the explanation in the appendix (908-926).

- What are the key advantages of using DBSCAN over Hierarchical Clustering as in FL-HC[1]? Specifically, how does DBSCAN improve upon the hierarchical clustering in this context?

**Ethical Concerns:**

["NO or VERY MINOR ethics concerns only"]

**Final Justification:**

I thank the authors for their detailed rebuttal and subsequent post-rebuttal discussions. While these satisfied most of my original concerns, I still consider the novelty attributed solely to the design of client descriptors to be incremental within the broader context of clustering methods in federated learning. During the post-rebuttal discussion, I appreciate the thorough derivation provided under the Gaussianity assumption that shows how the descriptor distance approximates the Wasserstein distance. As most of my initial concerns have been addressed, I am happy to increase my rating.

**Limitations:**

yes

**Quality:**

2

**Strengths And Weaknesses:**

Strengths:

- The paper reports improved accuracy across multiple benchmarks and a real-world dataset, with overhead claimed to be comparable to FedAvg.

Weaknesses:

- The core idea—clustering clients for personalized federated learning—is not new and is wellestablished in the literature (e.g., MCFL [2] and FL-HC [1]). In FL-HC the server uses hierarchical clustering but the key idea is the same. The current paper uses DBSCAN where the radius epsilon is computed dynamically but the number off points per cluster is still the hyper-partameter. One-Shot Clustering is also explored in the context of unsupervised methods LADD [3].

-  Lacks theoretical analysis of clustering such as impact of incorrectly clustered client and generalization.

- The experiments on feature heterogeneity datasets such as DomainNet/Office-Home are missing.

[1] Briggs, Christopher, Zhong Fan, and Peter Andras. "Federated learning with hierarchical clustering of local updates to improve training on non-IID data." In 2020 international joint conference on neural networks (IJCNN), pp. 1-9. IEEE, 2020.

[2] Long, Guodong, Ming Xie, Tao Shen, Tianyi Zhou, Xianzhi Wang, and Jing Jiang. "Multi-center federated learning: clients clustering for better personalization." World Wide Web 26, no. 1 (2023): 481-500.

[3] Shenaj, Donald, Eros Fanì, Marco Toldo, Debora Caldarola, Antonio Tavera, Umberto Michieli, Marco Ciccone, Pietro Zanuttigh, and Barbara Caputo. "Learning across domains and devices: Style-driven source-free domain adaptation in clustered federated learning." In Proceedings of the IEEE/CVF winter conference on applications of computer vision, pp. 444-454. 2023.

---

> ### Author Rebuttal · Authors · 2025-07-30
>
> ## Q1: The novelty of FLUX compared to baselines such as FL-HC, MCFL, LADD.
>
> We agree clustering in FL is established, and we thank the reviewer for providing the three important studies, which we will analyze in the Discussion section of the updated manuscript. However, please note that **our contribution differs fundamentally from existing clustered FL methods. While existing methods cluster based on model parameters [FL-HC, MCFL] or metrics (e.g., loss function, or frequency coefficients of the image [LADD]), FLUX clusters using novel distribution descriptors that directly capture client data distribution characteristics**. Moreover, FLUX is agnostic to unsupervised clustering algorithms, which enables FLUX to be equipped with suitable clustering approaches based on the use case.
>
> Furthermore, as shown in Table 2, existing CFL methods have critical limitations:
> 1. metric-based approaches can miscluster clients with similar losses but different distributions. Similarly, considering LADD metric, clients with different conditional distribution P(Y|X) but same marginal distribution P(X), will be indistinguishable.
> 2. parameter-based methods suffer from permutation invariance issues, i.e., models trained on same data but just different seeds may still have different parameters.
>
> FLUX's descriptor-based clustering addresses these problems by approximating the 2-Wasserstein distance between distributions (Section 4.2.1) and capturing both marginal and conditional properties. This enables two additional key capabilities existing methods lack: (1) _handling all four distribution shift types simultaneously_ and (2) _test-time adaptation for unseen clients_. Our experiments show FLUX achieves up to 8 percentage points (pp) improvement over SOTA baselines while maintaining robust performance with unseen clients (Section 5.2).
>
> ## Q2: Impact of incorrectly clustered client and generalization.
>
> We agree on the importance of these points. Please note that **our empirical evaluation has already provided substantial evidence addressing these concerns**. The purpose of FLUX-prior, which uses oracle knowledge of the true number of clusters, is to serve as an upper bound for FLUX. Therefore, **the performance gap between FLUX and FLUX-prior (e.g., 94.0% vs. 95.7% on MNIST, Table 1) quantifies the impact of clustering errors**. Even at challenging high heterogeneity levels where clustering errors increase, FLUX still achieves 92.3% accuracy vs. 79.4% for the best baseline, remaining close to the oracle and showing resilience under imperfect clustering conditions.
>
> Similarly, regarding generalization, we believe the evaluation setup provides extensive generalization validation: (1) Cross-dataset consistency across 5 datasets including real-world CheXpert; (2) Cross-shift robustness across all 4 distribution types and their combinations (Appendix F.4); (3) Scalability validation from 5 to 100 clients; and (4) Test-time generalization where FLUX maintains performance on completely unseen, unlabeled clients. **This empirical analysis provides compelling evidence for FLUX's robustness and generalization across diverse real-world scenarios**.
>
> ## Q3: Missing feature heterogeneity datasets, like Office-Home.
>
> We thank the reviewer for this valuable suggestion, which allowed us to further validate FLUX on a standard dataset exhibiting feature heterogeneity. **We conducted additional experiments on the OfficeHome dataset and results highlight FLUX’s effectiveness and resilience in real-world scenarios involving feature heterogeneity.** We followed the same experimental protocol used for CheXpert: each client was assigned randomly 200–600 samples drawn from a single domain, ensuring that all client data originates from one of the four domains.
>
> The table below reports both Known Association and Test Phase accuracy. FLUX achieves strong performance on OfficeHome, outperforming all baselines in the test phase. As expected, only the personalized FL method APFL surpasses FLUX under the known association setting, as it evaluates models on the same client distributions used for fine-tuning. However, APFL suffers a noticeable performance drop on unseen test clients.
>
> Notably, FLUX matches its oracle counterpart (FLUX-prior) during the test phase, confirming its robustness even without prior knowledge of the number of domains (M = 4). Moreover, FLUX-prior remains competitive even with APFL while offering broader generalization capabilities. These results confirm that FLUX is a practical and effective solution for real-world FL deployments with feature heterogeneity.
> |Method|Known Ass. (%)|Test Phase (%)|
> |-|-|-|
> |FedAvg|37.1±0.9|37.1±0.9|
> |IFCA|32.8±8.1|29.6±7.5|
> |FedRC|22.2±2.8|22.2±2.8|
> |FedEM|28.4±2.3|28.8±1.8|
> |FeSEM|27.0±2.9|25.8±1.8|
> |FedDrift|39.5±4.0|34.6±4.1|
> |CFL|31.3±3.3|21.0±1.5|
> |pFedMe|30.9±2.3|N\/A|
> |APFL|43.5±2.1|36.7±0.4|
> |ATP|N\/A|37.9±0.9|
> |FLUX|39.2±0.2|39.2±0.3|
> |FLUX-prior|43.2±3.1|38.9±1.1|
>
> ## Q4: Significance of the Distribution Fidelity in Eq. 4.
>
> **Eq. 4 defines a key requirement that descriptors must satisfy to be used within the FLUX pipeline—not an operational step**. It states that the Euclidean distance between descriptors should approximate a divergence (e.g., Wasserstein distance) between the underlying data distributions. Importantly, this condition is not used directly in practice and therefore does not require access to the true distributions $P(x^{(k_1)}, y^{(k_1)})$ and $P(x^{(k_2)}, y^{(k_2)})$. Instead, **it serves as a guiding principle**: descriptors should encode distributional differences such that clustering in descriptor space closely approximates clustering in data space (up to an error $\xi$).
>
> For our descriptor implementation, we derive in Appendix E.4 a closed-form expression under the Gaussian assumption of the latent space, which is a common choice since Gaussian distribution “makes the fewest assumptions and has maximum entropy” [Section 2.6.4 in ‘Probabilistic Machine Learning’, Di Kevin P. Murphy, 2022]. This enables analytical validation of distribution fidelity for our descriptors. However, our approach is not limited to this assumption: similar results can be verified numerically without any assumption on the underlying distribution (see Table 6 for empirical evaluations demonstrating this).
>
>
> ## Q5:  How does the descriptor design ensure Distribution Fidelity?
>
> We thank the reviewer for pointing this out. We will revise Appendix E.4 to clarify this concept. To satisfy the Distribution-Fidelity requirement R1, we design descriptors that approximate the distance between client data distributions in a compact form (see E.4). Each client first applies a client-invariant dimensionality reduction to each sample, then computes the first two moments (mean and covariance) over the reduced representations of all samples—both globally (to capture the marginal distribution) and per class (to capture the conditional distribution). The final descriptor concatenates these statistics. Because our client-invariant dimensionality reduction is designed to preserve distributional geometry, these latent-space moments serve as effective surrogates for the true data-space moments.
>
> **Crucially, as shown in Line 915, the 2-Wasserstein distance between Gaussian distributions depends solely on their means and covariances—the same quantities encoded in our descriptors. As a result, the Euclidean distance between descriptors is Lipschitz-equivalent to the 2-Wasserstein distance, up to a constant $\xi$, ensuring Distribution-Fidelity.**
>
>
> ## Q6: DBSCAN compared to Hierarchical Clustering in FL-HC.
>
> Please note that the core contribution of FLUX lies in the design of informative, privacy-preserving descriptors that approximate the Wasserstein distance between client distributions—not in the clustering algorithm itself. These descriptors enable meaningful unsupervised clustering while preserving privacy, unlike many prior approaches (see Table 2). Our formulation (Eq. 8) is agnostic to the specific clustering method, requiring only that the clustering operates solely on the descriptors, without relying on external information such as the number of clusters (i.e., fully unsupervised).
>
> **To validate the agnosticism to clustering algorithms, we conducted an ablation study on the MNIST dataset using four clustering algorithms.** Results (averaged over all shift types, heterogeneity levels, and seeds as in Table 1) are shown below:
> |Method|Known Association (%)|Test Phase (%)|
> |-|-|-|
> |U. K-Means|86.8±14.6|93.5±3.8|
> |HDBSCAN|90.5±7.7|92.9±2.6|
> |Agglomerative|93.9±3.2|94.2±0.8|
> |Ours|92.5±5.2|94.0±2.8|
>
> These results show that, except for Unsupervised K-Means, all methods achieve strong performance, confirming that our descriptors enable robust client grouping regardless of the clustering algorithm. Notably, all methods converge to comparable accuracy in the test phase.
>
> We further compare the two best-performing methods (Agglomerative and Ours)—on CIFAR-100, with similar conclusions:
> |Method|Known Association (%)|Test Phase (%)|
> |-|-|-|
> |Agglomerative|42.0±17.8|36.0±5.4|
> |Ours|41.7±11.0|41.3±7.8|
>
> **This confirms that FLUX remains robust across clustering strategies, and our performance stems from the descriptors rather than the choice of algorithm.**

---

> ### Comment · Reviewer_c8ww · 2025-08-07
> **Some remaining concerns**
>
> I thank the authors for the rebuttal, especially for the Office-Home experiments across different methods. I still have two concerns remaining.
>
> **Regarding the novelty aspect:**
>
> The authors claim that their main contribution lies in the design of the client descriptors. But the broader approach still revolves around addressing heterogeneity through clustering based on client-side information. For example, in [1] the client side summaries are the distributions, computed via the histograms and the Hellinger distance is employed for clustering, in the context of client selection.
>
> [1] HACCS: Heterogeneity-Aware Clustered Client Selection for Accelerated Federated Learning,  2022 IEEE International Parallel and Distributed Processing Symposium (IPDPS)
>
> **Follow-up question on Ensuring the  distribution fidelity of the Client Descriptors:**
>
> The reviewer acknowledges that they have provided empirical evidence to show that distance computed between the proposed client descriptors is similar to Wasserstein distance. However, It is not clear from the discussion in Section E.4 why the $l_2$ distances computed between two client descriptors should correspond to the Wasserstein distance computed across similar two clients. Given that there is no theoretical justification or explicit mechanism designed to guarantee this alignment, apart from the assumption that deep representations have a Gaussian distribution, which in general is not the case.

---

> ### Author Response · Authors · 2025-08-08
> **Follow-up Clarifications to Reviewer c8ww (Part 1)**
>
> ## Regarding the novelty aspect
> ### 1. Comparison to HACCS:
>
> We thank the reviewer for bringing HACCS to our attention and providing us with this opportunity to compare FLUX with this relevant related work. Both FLUX and HACCS utilize statistical information, rather than loss metrics or model weights, to conduct clustering during FL training. However, FLUX has several key differences and advantages over HACCS:
>
> 1. **Semantic and structural comprehension.** HACCS builds descriptors from label and conditional pixel-value histograms, which lack semantic and structural awareness: two images with similar pixel statistics but different meanings/semantics may yield identical histograms. This is a critical limitation in tasks like X-ray diagnosis (e.g., CheXpert), where subtle, semantically meaningful patterns drive accurate clustering. FLUX instead extracts descriptors from a model’s latent space—compact, task-relevant representations that capture high-level, domain-specific features while filtering out noise and redundant details. This alignment with the classification objective ensures robustness to variability and input transformations. This advantage is reflected in the results in Table below: HACCS matches FLUX only for label distribution shifts (P(Y)), which require no semantic modelling and can be solved by label histograms or even loss-based methods like FedDrift. In all other shift types, FLUX outperforms HACCS by a wide margin.
>
> 2. **Test-time adaptation capability.** HACCS relies on labels to build histograms and perform clustering, making it unusable when labels are absent at test time. FLUX, while using labels during training to refine clusters, produces label-agnostic descriptors at inference— extending the utility of the trained models to new, unseen clients or distributions. Empirically, this distinction is evident (Table below): under test-time evaluation, HACCS’s accuracy drops below FedAvg due to the random assignment of cluster-specific models, whereas FLUX remains robust and consistent across all shifts.
>
> 3. **Descriptor compactness, privacy, and efficiency.** FLUX uses compact, fixed-size descriptors, while HACCS’s histograms of labels and conditional pixel values raise two issues:
>
> - _Privacy risk_: Label histograms are human-interpretable and can expose sensitive information (e.g., medical diagnoses), requiring strong differential privacy, as noted in the HACCS paper. FLUX descriptors, produced via a non-invertible transformation pipeline (encoding → dimensionality reduction → statistical summarisation), are not human-interpretable, reducing leakage risk.
>
> - _Scalability_: HACCS’s summary size grows with the number of labels and histogram bins, potentially becoming large for complex tasks (e.g., RGB images with 256 pixel values). FLUX’s descriptor size is fixed by the number of labels for clustering, ensuring scalability and communication efficiency.
>
> 4. **Demonstrated scalability.** FLUX provides explicit empirical evidence of scalability, achieving performance on par with FedAvg while scaling to over 1,000 clients with minimal communication and computation overhead. HACCS, however, does not discuss scalability nor present any large-scale experimental validation.
>
> 5. **Breadth of heterogeneity coverage.** FLUX is designed and tested to address all four major types of heterogeneity—P(X), P(Y), P(X | Y), and P(Y | X)—across eight different heterogeneity levels. HACCS experimentally evaluates only P(Y) shifts, in which each client is assigned two classes out of ten. As reflected in the results below, HACCS maintains strong performance only under P(Y), whereas FLUX achieves consistently high accuracy across all shift types.
>
> 6. **Experimental scope and validation.** FLUX has been rigorously validated on six public datasets (MNIST, FMNIST, CIFAR-10, CIFAR-100, CheXpert, and OfficeHome) under four types of heterogeneity. HACCS, by contrast, reports results on only two datasets (FMNIST and CIFAR-10) and a single heterogeneity type.
>
> #### Known Association
> |Method|P(X)|P(Y)|P(Y\|X)|P(X\|Y)|
> |-|-|-|-|-|
> |FedAvg|77.8±2.1|91.9±1.4|66.8±1.4|87.0±2.0|
> |FedDrift|94.1±1.8|96.4±0.4|81.1±4.2|92.7±1.7|
> |HACCS|77.5±5.2|96.2±0.2|79.0±4.5|88.4±1.4|
> |FLUX|95.5±0.5|96.8±0.4|85.1±4.0|92.4±2.4|
>
> #### Test Phase
> |Method|P(X)|P(Y)|P(Y\|X)|P(X\|Y)|
> |-|-|-|-|-|
> |FedAvg|77.8±2.1|91.9±1.4|N/A|87.0±2.0|
> | FedDrift |47.7±3.9|71.8±3.0|N/A|75.9±3.1|
> |HACCS|59.7±9.5|69.3±6.1|N/A|77.0±4.8|
> |FLUX|95.0±1.5|96.1±1.2|N/A|90.8±2.4|
>
> The table above reports FLUX results against HACCS on MNIST, with FedAvg and FedDrift included for reference (all other baselines are in the manuscript).
>
> — The response continues in the next official comment —

---

> ### Author Response · Authors · 2025-08-08
> **Follow-up Clarifications to Reviewer c8ww (Part 2)**
>
> — Continued from the previous Official Comment —
>
> The table above reports FLUX results against HACCS on MNIST, with FedAvg and FedDrift included for reference (all other baselines are in the manuscript). Experiments follow the same protocol as Table 1, except results are shown per distribution shift (not averaged) for direct comparison. Each value is averaged over 8 heterogeneity levels, each with 5-fold cross-validation. For fairness—and to give HACCS its best possible performance—we did not apply differential privacy to HACCS, even though its design requires it and the original paper does not fix a specific ε value.
>
> While the results clearly show substantial performance differences, we fully acknowledge HACCS as a closely related work to FLUX. We will indeed incorporate a more detailed discussion and a full comparison with HACCS in our revised manuscript. We sincerely thank the reviewer for bringing this important background work to our attention and we are happy to discuss any other works related to FLUX.
>
>
> ## 2. Contributions of this work:
>
> We completely agree with the reviewer that the design of the client descriptors is one of our main contributions, and we applied the descriptor to CFL to address data heterogeneity. However, our main contribution is not limited to only the design of descriptors. Besides the superior performance, we also want to highlight several other important contributions as summarized in Table 2:
>
> 1. **FLUX is the first method designed with the descriptor to simultaneously handle all four common shifts**—P(X), P(Y), P(Y|X), and P(X|Y)—while supporting test-time adaptation for unseen, unlabeled clients. As shown in Table 2, no existing method satisfies all these requirement.
>
> 2. **FLUX enables automatic cluster discovery with test-time adaptation.** Our approach requires no prior knowledge of cluster numbers or distribution types, and enables previously unseen clients to benefit from appropriate cluster-specific models without labeled data.
>
> ## Additional Results: HACCS vs. FLUX
> This section will be updated over time as new results on additional datasets become available.
>
> ### Known Association (CIFAR-10)
> |Method|P(X)|P(Y)|P(Y\|X)|P(X\|Y)|
> |-|-|-|-|-|
> |FedAvg|22.1±1.2|37.4±2.9|28.0±1.0|36.0±1.1|
> |FedDrift|25.2±1.9|49.0±3.4|29.0±1.0|37.0±1.1|
> |HACCS|23.2±1.6|45.4±1.3|24.8±1.3|33.4±2.0|
> |FLUX|33.3±0.9|50.4±2.9|31.7±1.3|39.1±1.8|
>
> ### Test Phase (CIFAR-10)
> |Method|P(X)|P(Y)|P(Y\|X)|P(X\|Y)|
> |-|-|-|-|-|
> |FedAvg|22.1±1.2|37.4±2.9|N/A|36.0±1.1|
> |FedDrift|24.0±1.3|35.4±2.8|N/A|36.8±1.3|
> |HACCS|18.9±1.5|32.3±3.1|N/A|30.1±1.1|
> |FLUX|33.3±1.0|46.2±4.0|N/A|36.7±2.1|
>
> ### Known Association (FMNIST)
> |Method|P(X)|P(Y)|P(Y\|X)|P(X\|Y)|
> |-|-|-|-|-|
> |FedAvg|61.6±2.4|72.8±2.4|55.4±2.3|72.0±1.8|
> |FedDrift|79.9±0.7|86.0±1.8|74.5±2.7|82.5±1.6|
> |HACCS|68.5±6.1|85.2±2.0|66.0±3.0|77.8±3.2|
> |FLUX|77.6±1.8|85.9±1.9|72.2±3.1|81.9±2.3|
>
> ### Test Phase (FMNIST)
> |Method|P(X)|P(Y)|P(Y\|X)|P(X\|Y)|
> |-|-|-|-|-|
> |FedAvg|61.6±2.4|72.8±2.4|N/A|72.0±1.8|
> |FedDrift|30.0±1.5|61.4±3.1|N/A|61.9±2.3|
> |HACCS|41.6±6.2|62.5±3.5|N/A|60.8±2.8|
> |FLUX|77.0±2.2|85.7±1.8|N/A|81.0±2.8|

---

> > ### Author Response · Authors · 2025-08-08
> > **Follow-up Clarifications to Reviewer c8ww (Part 3)**
> >
> > ## Follow-up question on Ensuring the distribution fidelity of the Client Descriptor
> > We thank the reviewer for identifying that Section E.4 does not provide sufficient mathematical justification why the $\ell_2$ distance between our descriptors approximates the 2-Wasserstein distance on the distributions. We take this opportunity to clarify the connection.
> >
> > ### 1. Theoretical Justification
> > **FLUX descriptors.**
> >
> > The key idea is that our descriptor explicitly encodes the sufficient statistics for approximating the 2-Wasserstein distance between distributions. As described in L886-902, our descriptor $d^{(k)} = [\mu_k;\, \mathrm{vec}(\Sigma_k)]$ captures:
> > - $\mu_k=\mathbb{E}[z^{(k)}]$ (first moment)
> > - $\Sigma_k=\mathrm{Cov}[z^{(k)}]$ (second moment)
> >
> > where $z^{(k)}$ are client-$k$'s latents projected via a shared PCA. This client-invariant dimensionality reduction preserves distributional geometry across clients, making these latent-space statistics good surrogates for data-space statistics.
> >
> > The squared $\ell_2$ distance between two descriptors is:
> > $||d^{(1)}-d^{(2)}||_2^2=||\mu_1-\mu_2||^2_2+||\text{vec}(\Sigma_1-\Sigma_2)||^2_2$.
> >
> > We now apply the identity that for any matrix $A$:
> >
> > $||\text{vec}(A)||_{2}^{2}=$
> >
> > $=\sum_{i,j} A_{ij}^{2}=||A||_{F}^{2}$
> >
> > where $\|\cdot\|_F$ is the Frobenius norm.
> >
> > Using this, with $A=\Sigma_1-\Sigma_2$ we obtain:
> > $\|d^{(1)}-d^{(2)}\|_2^2=\|\mu_1-\mu_2\|_2^2+\|\Sigma_1-\Sigma_2\|_F^2.\quad(1)$
> >
> > **2-Wasserstein distance.**
> >
> > For two distributions $P_1$ and $P_2$, the squared $W_2$ distance is defined as:
> >
> > $W_2^2(P_1,P_2)=\inf_{\pi\in\Pi(P_1,P_2)}\int ||x-y||^2 d\pi(x,y).$
> >
> > When client data distributions are Gaussian—i.e., $P_1=\mathcal{N}(\mu_1,\Sigma_1)$ and $P_2=\mathcal{N}(\mu_2, \Sigma_2)$ (we discuss this approximation in more detail below)—the $W_2$ distance admits the following closed-form expression:
> >
> > $W_2^2(P_1, P_2)=\|\mu_1-\mu_2\|_2^2+B^2(\Sigma_1,\Sigma_2)\quad(2)$
> >
> > where $B(\cdot,\cdot)$ denotes the Bures distance, defined as: $\mathrm{Tr}\left(\Sigma_1+\Sigma_2-2\left(\Sigma_1^{1/2} \Sigma_2\Sigma_1^{1/2} \right)^{1/2}\right).$
> >
> > **Bounding the difference.**
> >
> > We observe that Equations (1) and (2) differ only in the covariance term while the mean term is identical. We now show that the two covariance terms are Lipschitz-equivalent under mild assumptions.
> >
> > If $\Sigma_1$ and $\Sigma_2$ are SPD and commute (i.e., $\Sigma_1\Sigma_2=\Sigma_2\Sigma_1$), then all their powers also commute. In particular:
> >
> > $\Sigma_1^{1/2}\Sigma_2\Sigma_1^{1/2}
> > =\Sigma_1^{1/2}\Sigma_2^{1/2}\Sigma_2^{1/2} \Sigma_1^{1/2}
> > =(\Sigma_1^{1/2}\Sigma_2^{1/2})^2.$
> >
> > Hence, we have:
> >
> > $\left( \Sigma_1^{1/2}\Sigma_2 \Sigma_1^{1/2} \right)^{1/2}=\Sigma_1^{1/2}\Sigma_2^{1/2}.$
> >
> > Plugging this into the Bures formula:
> >
> > $B^2(\Sigma_1,\Sigma_2)=
> > \mathrm{Tr}(\Sigma_1)+\mathrm{Tr}(\Sigma_2)- 2\mathrm{Tr}\Bigr((\Sigma_1^{1/2}\,\Sigma_2\,\Sigma_1^{1/2}\bigr)^{1/2}\Bigr)=\mathrm{Tr}(\Sigma_1)+\mathrm{Tr}(\Sigma_2)-2\mathrm{Tr}(\Sigma_1^{1/2}\Sigma_2^{1/2}).$
> >
> > Now recall the identity for the squared Frobenius norm of the difference between two symmetric matrices:
> >
> > $||\Sigma_1^{1/2}-\Sigma_1^{1/2}||_F^2=\mathrm{Tr}(\Sigma_1)+\mathrm{Tr}(\Sigma_2)-2\mathrm{Tr}(\Sigma_1^{1/2}\Sigma_2^{1/2}).$
> >
> > Therefore,
> >
> > $B^2(\Sigma_1,\Sigma_2)=\|\Sigma_1^{1/2}-\Sigma_2^{1/2}\|_F^2.$
> >
> > To compare this to the Frobenius norm of $\Sigma_1-\Sigma_2$, we now assume that all eigenvalues of $\Sigma_1$ and $\Sigma_2$ lie within a compact interval $[\lambda_{\min}, \lambda_{\max}]$. This is guaranteed in our case by Step S1 of the descriptor extractor (Line 894), which ensures that all latent representations lie within a global bounding box $[m^-, m^+]$.
> >
> > Now, consider the scalar inequality that follows from the mean value theorem applied to the function $f(x)=\sqrt{x}$:
> >
> > $\frac{1}{2\sqrt{\lambda_{\max}}} |s - t| \leq |\sqrt{s} - \sqrt{t}| \leq \frac{1}{2\sqrt{\lambda_{\min}}} |s - t|
> > \quad \text{for all } s,t \in [\lambda_{\min}, \lambda_{\max}].$
> >
> > Applying this entrywise in a common eigenbasis for $\Sigma_1$ and $\Sigma_2$ gives the matrix inequality:
> >
> > $\frac{1}{2\sqrt{\lambda_{\max}}} \|\Sigma_1-\Sigma_2\|_F \leq$
> >
> > $B(\Sigma_1,\Sigma_2)\leq\frac{1}{2\sqrt{\lambda_{\min}}}\|\Sigma_1-\Sigma_2\|_F$
> >
> > This shows that the Bures distance is sandwiched between scaled versions of the Frobenius norm, hence they are Lipschitz-equivalent under spectral bounds. Finally, this implies that the difference between the Frobenius norm and the Bures distance is bounded as:
> >
> > $\left| \|\Sigma_1-\Sigma_2\|_F-B(\Sigma_1,\Sigma_2)\right|\leq$
> >
> > $\left(1-\frac{1}{2\sqrt{\lambda_{\max}}} \right)\|\Sigma_1-\Sigma_2\|_F\leq\|\Sigma_1-\Sigma_2\|_F$
> >
> > Thus, under mild conditions, the Frobenius norm used in our descriptors approximates the Bures distance up to a constant factor, and therefore the overall $\ell_2$ distance between descriptors approximates the 2-Wasserstein distance between the corresponding distributions.
> >
> > — The response continues in the next comment —

---

> ### Author Response · Authors · 2025-08-08
> **Follow-up Clarifications to Reviewer c8ww (Part 4)**
>
> — Continued from the previous comment —
>
> ### 2. Empirical Validation of the Theory
> Regarding the Gaussian assumption, we acknowledge the reviewer's valid concern that deep representations are not always Gaussian-distributed. We agree that this assumption may not always hold in practice and should have been more carefully justified in the original submission.
>
> This is one of the reasons why we chose to empirically assess whether the Gaussian approximation is reasonable for the latent representations learned by our encoder $f_e$. Specifically, we quantify the approximation error $\epsilon$ between the true empirical distribution and its Gaussian surrogate given by our descriptors (R1). We now extend this analysis to CIFAR-10 and include the results alongside those already reported for MNIST.
>
> - MNIST - P(Y|X)
> |Non-IID Level|Max|Min|Mean|Std|
> |-|-|-|-|-|
> |3|1.060|0.194|0.526|0.106|
> |5|1.028|0.159|0.479|0.113|
> |7|1.016|0.136|0.458|0.108|
>
> - CIFAR-10 – P(Y|X)
> |Non-IID Level|Max|Min|Mean|Std|
> |-|-|-|-|-|
> |3|2.816|0.00017|1.328|0.522|
> |5|3.464|0.00135|1.744|0.551|
> |7|3.429|0.00441|1.734|0.590|
>
> These results show that $\epsilon < 1.1$ for MNIST and $\epsilon < 3.5$ for CIFAR-10 across various non-IID levels, supporting the following conclusions—at least on these datasets (we aim to include additional datasets if time permits).
>
> 1. The latent space learned by $f_e$ exhibits "well-behaved" statistical properties; specifically, the first and second moments are stable and representative.
>
> 2. These moments serve as effective surrogates for the original data distribution, even when Gaussianity is not strictly satisfied.
>
> 3. The approximation of the 2-Wasserstein distance using descriptor-based distances remains valid across different data types and degrees of heterogeneity.
>
> Finally, our extensive experimental results, comparing FLUX with 10 baselines across 6 datasets and under varying levels of non-IIDness, further reinforce this conclusion. Despite the theoretical assumption of Gaussianity, FLUX consistently demonstrates practical robustness. Its moment-based descriptors reliably capture inter-client distributional differences, suggesting that the method remains effective even when the Gaussian assumption is only approximately satisfied.
>
>
> ### 3. Possible Extension to Arbitrary Distributions via GMMs
> Since any distribution can be approximated arbitrarily well by a Gaussian Mixture Model (GMM), each client’s distribution can be represented as a finite mixture of Gaussians with bounded approximation error, regardless of its underlying form.
>
> Let $K_1$ be the number of components required to represent client 1’s distribution with error below $\varepsilon_1$, and let $K_2$ be the corresponding number for client 2. Define
>
> $K = \max( K_1, K_2 )$
>
> so that both clients are represented using the same number $K$ of Gaussian components.
>
> We can then align the two GMMs by optimally matching their respective Gaussian components. This reduces the original problem of comparing arbitrary distributions to a sum of Gaussian–Gaussian comparisons, for which the 2-Wasserstein distance has a closed-form. In summary, even when the true client distributions are non-Gaussian, the comparison can be reduced—via sufficiently accurate GMM approximations—to the Gaussian–Gaussian case analyzed above.
>
> ---
>
> ---
>
> ---
>
> We appreciate the reviewer for highlighting this theoretical limitation, which gave us the opportunity to clarify our mathematical justification. In the revision, we will be more explicit about the assumption's constraints and provide additional discussion on why our moment-based approach remains effective despite potential violations. We welcome any suggestions for strengthening this theoretical foundation or alternative approaches that might be more robust.

---

> > ### Comment · Reviewer_c8ww · 2025-08-09
> > **Thanks for the detailed theoretical analysis**
> >
> > I thank the authors for their detailed response. However, I still consider the novelty attributed solely to the design of client descriptors to be incremental within the broader context of clustering methods in federated learning. I do appreciate the thorough derivation provided under the Gaussianity assumption that shows how the descriptor distance approximates the Wasserstein distance. As most of my initial concerns have been addressed, I will revise my score accordingly.

---

> > > ### Author Response · Authors · 2025-08-09
> > >
> > > We thank the reviewer for the thoughtful feedback throughout the review process, which has helped strengthen our work.

---

### Official Review · Reviewer_L2wb · 2025-07-02

**Clarity:** 2
**Significance:** 3
**Originality:** 2
**Rating:** 4
**Confidence:** 4

**Summary:**

This paper proposes a clustering-based federated learning framework, FLUX, which aims to address potential distribution shifts during both training and testing phases without relying on any prior knowledge. FLUX leverages a privacy-preserving descriptor extraction mechanism on clients to enable efficient unsupervised clustering and representation compression with low communication overhead. This design ensures both robustness and scalability of the global model, while enabling adaptive model selection for newly joined unlabeled clients during the testing phase. Extensive experiments on several benchmark datasets and real-world medical data demonstrate that FLUX outperforms 10 state-of-the-art baseline methods under varying degrees of data heterogeneity and task configurations.

**Questions:**

i. Has the performance of the descriptors been evaluated under varying levels of data noise?

ii.  How can the authors explain the subpar performance of the proposed method in certain cases?

**Ethical Concerns:**

["NO or VERY MINOR ethics concerns only"]

**Final Justification:**

It is recommended that the authors incorporate key clarifications and insights from the rebuttal into the revised manuscript, particularly those addressing reviewers' concerns. This will enhance the completeness and readability of the paper, and better align it with the expectations of top-tier conferences.

**Limitations:**

Yes

**Quality:**

2

**Strengths And Weaknesses:**

Strengths

i. This paper proposes a label-free model assignment mechanism for the inference phase, addressing a key limitation of many clustered federated learning (CFL) methods that struggle to generalize effectively on the client side during testing.

ii. Comparisons were conducted across five datasets, eight levels of heterogeneity, four types of distribution shift, and both seen and unseen testing conditions.

iii. The paper follows a well-structured and readable format.

Weaknesses

i. The motivation of the paper is not sufficiently substantiated. Although it highlights that many existing methods rely on prior knowledge, it lacks a more systematic categorization or statistical evidence to convincingly support the claim that "prior-free" approaches offer broader generality.

ii. The experimental section provides insufficient explanation of FLUX-prior. Although this variant appears frequently in tables and figures, its definition and distinction from the main method are not systematically clarified.

iii. The description of the clustering method in the paper is overly brief; it only mentions "density-adaptive clustering" without specifying whether it refers to DBSCAN or another algorithm, and lacks details regarding the parameters used.

Ⅳ. The applicability of hard versus soft clustering remains unclear. While the paper adopts a hard clustering strategy, it does not explicitly justify this choice or compare its effectiveness against soft assignment approaches.

Ⅴ. The paper does not clearly articulate the underlying mechanism of the proposed approach, and it lacks illustrative examples that demonstrate how the method effectively tackles the identified challenges.

Ⅵ. The text in Tables 1 and 2 is too small and may affect readability.

---

> ### Author Rebuttal · Authors · 2025-07-30
>
> ## Q1: Broader generality of "prior-free" approaches in FL
>
> We have provided systematic evidence showing that prior knowledge is valuable when available. As shown in Table 1, FLUX-prior (with cluster number knowledge M) consistently outperforms FLUX across all datasets—achieving 95.7% vs. 94.0% on MNIST and 83.3% vs. 81.2% on FMNIST in the test phase—confirming the value of the prior.
>
> In addition, the "broader generality" claim refers to applicability across deployment scenarios, not performance superiority. Most existing CFL methods (Table 2: IFCA, FeSEM, FedEM, FedRC) require knowing M, severely limiting their real-world deployment where this information is unavailable.
>
> At last, **we have shown 'prior-free' FLUX offers broader generality: it achieves strong performance (within 1-2pp of the oracle) without requiring client heterogeneity patterns. This is essential in real-world FL, where data is decentralized across diverse entities and not existing access to the global data or the underlying distributions**. In such settings, assuming prior knowledge of M is unrealistic, and small trade-offs in accuracy are a necessary and worthwhile cost for ensuring deployability.
>
> ## Q2: Extended details of FLUX-prior
>
> Thank you for this feedback. We will add clearer explanation of FLUX-prior in the updated version. The key difference between FLUX and FLUX-prior lies in their clustering methodology:
> 1. FLUX: An adaptive clustering based on DBSCAN that automatically determines the number of clusters without prior knowledge
> 2. FLUX-prior: K-means clustering with the true number of clusters M as input (oracle knowledge)
>
> The number of clusters is considered prior in FL settings, required by many baseline methods (e.g., IFCA, FedRC) as shown in Table 2. We introduced FLUX-prior to provide fair comparison with these prior-dependent baselines and to demonstrate the optimal performance achievable with our descriptor-based clustering approach using oracle information.
> As expected, FLUX-prior consistently outperforms FLUX due to optimal clustering (e.g., 95.7% vs. 94.0% on MNIST, Table 1). However, **FLUX remains competitive with most baselines while requiring no prior knowledge—demonstrating that our automatic clustering approach achieves strong performance in realistic deployment scenarios where the number of client distributions is unknown**
>
> ## Q3: Brief description of clustering method in FLUX
>
> We thank the reviewer for pointing this out. In the revised version, we will include additional implementation details about the clustering procedure in Section 4.2.1. For how we allocate space for this update, please refer to our response to Reviewer Aotc, Q1.
> Please note that **we kept the description of clustering method brief in the main text (while full details and parameter choices are provided in Appendix E.8), because the core contribution of FLUX lies in the design of informative, privacy-preserving descriptors that approximate the Wasserstein distance between client distributions—not in the clustering algorithm itself**. These descriptors enable meaningful unsupervised clustering while maintaining privacy, something that common alternatives based on metrics, or gradient similarity do not guarantee.
>
> Crucially, our formulation (Eq. 8) is agnostic to the choice of clustering method, requiring only that it operates solely on the descriptors without relying on external information such as M (i.e., unsupervised clustering). **To validate this, and as suggested by Reviewer LVoP, we conducted ablation studies with alternative clustering strategies**. These results show that, except for Unsupervised K-Means, all methods achieve strong performance, confirming that our descriptors enable robust client grouping regardless of the clustering algorithm. Notably, all methods converge to comparable accuracy in the test phase.
> |Method|Known Association (%)|Test Phase (%)|
> |-|-|-|
> |U. K-Means|86.8±14.6|93.5±3.8|
> |HDBSCAN|90.5±7.7|92.9±2.6|
> |Agglomerative|93.9±3.2|94.2±0.8|
> |Ours|92.5±5.2|94.0±2.8|
>
> ## Q4: Justification of hard clustering in FLUX over soft clustering
>
> We thank the reviewer for raising this point. We will clarify our motivation for using hard clustering in L111 of the revised manuscript. While soft-CFL approaches offer greater flexibility, we opted for hard clustering in FLUX for two main reasons:
> 1. Computational and communication efficiency: Soft methods require clients to train and maintain multiple models locally and compute weighted combinations. This is impractical in cross-device FL, where the number of clusters M may be large and client devices are resource-constrained
> 2. Limited test-time application: Soft approaches require learning a personalized weight vector $\pi^k \in \mathbb{R}^M$ per client (see Eq. 11), which prohibits generalization to unseen clients at test time
>
> **For these reasons, hard clustering offers a more scalable and generalizable solution for real-world deployment within the FLUX framework.**
>
> ## Q5: Mechanism articulation. Lack of illustrative examples
> We appreciate the reviewer’s suggestion. **To clarify the underlying mechanism of FLUX, we now include a dedicated figure in the appendix that illustrates how the descriptor and clustering procedures operate in practice**. The figure presents examples on two datasets (MNIST and CIFAR-100) across three levels of heterogeneity (M=3,4,5). For visualization purposes, we project the client descriptors $d^{(k)} \in \mathbb{R}^L$ into a 2D space. These plots show that clients with similar data distributions are mapped close to each other in the descriptor space, enabling FLUX to cluster them correctly. The resulting clusters align with the latent distributional structure, confirming that the descriptors capture key distributional properties, even under complex heterogeneity. While we are unable to include figures in the rebuttal due to NeurIPS policy, we believe the descriptive content provided in the appendix improves the clarity of the FLUX mechanism.
>
> ## Q6: Readability of Tables 1 and 2
>
> We will increase the font size in Tables 1 and 2 to ensure better readability in the final version.
>
> ## Q7: Performance under varying levels of data noise
>
> Please note that we have already evaluated FLUX under multiple types of data noise and challenging conditions:
> 1. **Distribution noise:** Our core experiments simulate realistic shifts across 8 heterogeneity levels for each of the 4 shift types—P(X) shifts via transformations, P(Y) shifts through class imbalances (from 10 to 3 classes), P(Y|X) shifts via label permutations (1-8 swapped classes), and P(X|Y) shifts through class-specific augmentations (1-8 classes under augmentation) (Section F.1, Table 7)
> 2. **Real-world noise:** Our CheXpert medical imaging experiments involve 3 heterogeneity levels (low, medium, high) with naturally occurring noise from different imaging conditions, patient populations, and clinical settings—achieving up to 17.5pp improvement over baselines. Additionally, we have now included the Office-Home dataset, which contains four distinct domains, achieving a 2.1pp improvement
> 3. **Mixed noise scenarios:** We tested 3 compound distribution shift combinations across 4 heterogeneity levels (Appendix F.4) where multiple noise sources occur simultaneously, with FLUX maintaining robust performance while baselines degrade significantly
> 4. **Scalability noise:** We evaluated across 6 different client numbers (5, 10, 25, 50, and 100), where inherent FL aggregation noise increases, yet FLUX demonstrates superior stability compared to baselines
> 5. **Sampling noise:** We evaluated descriptor robustness in the rebuttal results with 4 different client dataset sizes (4%, 8%, 16%, 32% of original), showing FLUX maintains high accuracy even with only 4% of original data
>
> ## Q8: Explanation of subpar performance in certain cases
>
> **Explanation of the performance gap in (Fig11) on CIFAR-10 and CIFAR-100:**
> The performance gap between FLUX and the best baseline has two reasons (We will also add those explanations to our updated manuscript):
> 1. the only methods outperforming FLUX on CIFAR are Personalized FL (PFL) approaches. This is expected in scenarios involving P(Y|X) shifts (e.g., Tables 14–15), where each client may have different label preferences for the same input. In such cases, collaboration can be harmful, and methods fine-tuned on the same client distributions seen at test time (as in PFL) naturally have an advantage. However, these methods do not generalize to unseen clients (see Table 1, Test Phase), limiting their applicability in realistic deployments.
> 2. insufficient model capacity may affect FLUX’s performance. On CIFAR-100, we observed that using a deeper backbone with a larger latent space (ResNet9 instead of LeNet5) reduces the performance gap, suggesting that limited model expressiveness can degrade the quality of the extracted descriptors, thus final accuracy.
>
> **Explanation of FLUX performance drop in Figure 5:** The performance drop is not specific to FLUX, but a general characteristic of FL under our experimental setup. Two key factors contribute to this trend (which we will also add to our updated manuscript):
> 1. dataset size effects: In our experiment, client dataset sizes do not scale linearly with the number of clients—the actual size of local datasets decreases as we add more clients, leading to overall performance degradation regardless of the FL method used.
> 2. inherent FL divergence: The nature of FL introduces higher divergence during model aggregation as the number of clients increases, which naturally degrades performance across all methods.
>
> As shown in Figure 5, all baseline methods exhibit similar performance drops due to these same factors (Despite that personalization approaches can overfit their local dataset). However, FLUX still consistently maintains higher accuracy across all client numbers because the descriptor better captures client heterogeneity patterns.

---

> > ### Comment · Reviewer_L2wb · 2025-08-05
> > **Thanks for your rebuttal**
> >
> > Thank you for the authors’ detailed and thoughtful responses. While many of my concerns have been addressed, I still find a few aspects unclear and would appreciate further clarification. Specifically:
> >
> > 1. The descriptor-based approach shows promising results, even with reduced local data (as low as 4%). However, in real-world FL deployments, the availability of client data may be even more limited. Could the authors elaborate on how descriptor estimation performs in such extremely low-data regimes (e.g., <1%), and whether any specific regularization, smoothing, or heuristics were used to ensure its robustness?
> >
> > 2. The authors justify their use of hard clustering in FLUX with computational efficiency and test-time generalization. However, in practice, client distributions can be partially overlapping or noisy. Have the authors considered hybrid clustering approaches (e.g., sparse soft assignment or confidence-based reweighting) that may offer a better trade-off between efficiency and flexibility?
> >
> > 3. The proposed descriptors are primarily used for clustering, but they seem to encode rich distributional information. Have the authors considered leveraging them for auxiliary tasks such as detecting anomalous clients, distribution drift, or visualization of client similarity? This could broaden the practical utility of FLUX beyond clustering alone.
> >
> > 4. The manuscript uses standard datasets (e.g., OfficeHome, CheXpert) for OOD evaluation. However, these datasets are originally designed for centralized learning. Could the authors elaborate on how such datasets were adapted to simulate realistic federated settings, and whether there are best practices or frameworks for constructing FL-compatible OOD benchmarks?
> >
> > 5. FLUX assumes fixed client distributions throughout training and testing. In real-world FL systems, clients may join or leave, or their data distributions may shift over time. Is FLUX capable of handling such non-stationary conditions? If not, could the authors comment on the potential to incorporate periodic re-clustering or descriptor updates in future work?

---

> > > ### Author Response · Authors · 2025-08-06
> > > **Follow-up Clarifications (Part 1)**
> > >
> > > ## A.1. Could the authors elaborate on how descriptor estimation performs in such extremely low-data regimes (e.g., <1%), and whether any specific regularization were used to ensure its robustness?
> > > We thank the reviewer for this question and the insightful suggestion regarding regularization techniques. In our original experiments, we evaluated FLUX down to 4% of the total dataset, corresponding to approximately 2,500 samples in the entire federation. This already simulates an extreme FL setting, especially under high heterogeneity: in the worst case with $M=10$ clusters, each client-specific model is trained with only ~250 samples, **which is already near the lower bound for effective training given our 62K-parameter architecture**. We attempted to explore even more constrained settings (<1% of total data), **but under such conditions—i.e., ~3 samples per label per client—models consistently failed to converge, even in centralized training or without heterogeneity**.
> > >
> > > Nonetheless, to quantify descriptor robustness in more extreme regimes, we designed a new experiment targeting ~1% average client data. Due to the random sampling process, approximately half of clients receive less than 1%. We increased the number of clients to 20 (to increase global data size and achieve convergence) and extended training to 200 rounds. The results below report accuracy under all four distribution shifts:
> > >
> > > -	Known Association:
> > > |Method|P(X)|P(Y)|P(Y\|X)|P(X\|X)|
> > > |-|-|-|-|-|
> > > |FedAvg|36.3±7.7|84.8±1.4|46.1±5.2|31.6±11.5|
> > > |FLUX|74.3±0.5|97.5±0.5|49.2±6.9|45.5±6.5|
> > >
> > > -	Test Phase:
> > > |Method|P(X)|P(Y)|P(Y\|X)|P(X\|X)|
> > > |-|-|-|-|-|
> > > |FedAvg|36.3±7.7|84.8±1.4|N/A|31.6±11.5|
> > > |FLUX|78.2±0.1|95.6±0.2|N/A|42.0±9.3|
> > >
> > > Encouragingly, **FLUX continues to be robust, confirming that our descriptors remain informative even under these constraints**. This robustness stems from the design of the descriptors. Because our descriptors rely only on first- and second-order moments (means and covariances) computed after dimensionality reduction, they are less sensitive to sparsity: at low dimension, only the most informative features are retained.
> > >
> > > While our current implementation does not include smoothing or regularization, such techniques—e.g., Bayesian shrinkage, using a normal prior on the mean and an inverse-Wishart prior on the covariance—could further stabilize estimates when client data is extremely sparse. Exploring these directions is a promising avenue for future work.
> > >
> > >
> > > ## A.2. Have the authors considered hybrid clustering approaches (e.g., sparse soft assignment) that may offer a better trade-off between efficiency and flexibility?
> > > We thank the reviewer for this suggestion, which allows us to highlight the flexibility of our framework. Indeed, **we have already explored a hybrid variant of FLUX that implements sparse soft assignments while preserving computational efficiency.** In this FLUX-soft variant, we replace hard clustering with a soft-weighted aggregation scheme: each client receives a distribution-specific model update by weighting the contributions of other clients based on descriptor similarity with its own distribution. This approach combines the benefits of both strategies: maintaining computational efficiency while enabling more nuanced handling of partially overlapping distributions.
> > > Both hard and soft clustering approaches have distinct advantages within our framework. Hard clustering excels in scenarios with clear distributional boundaries and resource-constrained environments, while soft clustering provides superior flexibility for handling overlapping or noisy distributions. The results below demonstrate that FLUX-soft substantially outperforms all baselines—improving over the best-performing baseline (CFL) by nearly 11 percentage points—while approaching the accuracy of hard FLUX. This validates that our descriptor-based approach enables effective performance under both clustering paradigms, with the choice depending on specific deployment requirements and distributional characteristics.
> > > |Method|Accuracy (%)|
> > > |-|-|
> > > |FedAvg|71.9±6.0|
> > > |IFCA|35.2±6.2|
> > > |FedRC|75.2±4.3|
> > > |FedEM|75.0±4.2|
> > > |FedSEM|66.0±6.0|
> > > |FedDrift|54.3±7.7|
> > > |CFL|75.5±4.0|
> > > |pFedMe|55.2±6.9|
> > > |APFL|72.8±5.2|
> > > |ATP|74.6±8.0|
> > > |FLUX-soft|86.5±1.3|
> > > |FLUX|**89.5±4.9**|
> > >
> > > In conclusion, we agree that hybrid clustering approaches offer promising solutions for scenarios with partially overlapping or noisy client distributions. **Our descriptor-based framework naturally supports such extensions—the adaptive design of our descriptors can be easily integrated with various clustering strategies to achieve optimal trade-offs between efficiency and flexibility.** This flexibility represents a key strength of FLUX's descriptor-based approach, enabling adaptation to diverse deployment scenarios. We will explore these hybrid clustering variants as important future work to further enhance FLUX's robustness and applicability.

---

> > > > ### Author Response · Authors · 2025-08-06
> > > > **Follow-up Clarifications (Part 2)**
> > > >
> > > > ## A.3. Have the authors considered leveraging the descriptors for auxiliary tasks such as detecting anomalous clients, distribution drift, or visualization of client similarity?
> > > >
> > > > We completely agree with the reviewer that our descriptors encode meaningful distributional information that can be leveraged for several auxiliary tasks beyond clustering. In fact, the current FLUX design already makes use of this capability:
> > > > - **Anomalous client detection** is shown in the hard clustering step, where outlier clients naturally form singleton clusters and are excluded from aggregation (and possible degradation of other cluster models).
> > > > - **Distribution drift is handled in our dynamic clustering mechanism** (please see the answer A.5. for details)
> > > >
> > > > While to further illustrate the practical utility of the descriptors, we will include in the appendix a figure showing **client similarity visualization.** By projecting client descriptors into a 2D space, it becomes possible to observe distributional patterns and cluster structure. Precisely, the figure will display six subplots (MNIST and CIFAR-100, each across three levels of heterogeneity), where clients with similar distributions are consistently mapped close together. This shows that the descriptors preserve semantic distributional relationships and enable intuitive inspection of the client landscape. While we are unable to include figures in the rebuttal due to NeurIPS policy, we confirm that the visualization adheres to the description above and supports the reviewer’s point, further highlighting the versatility of the descriptors.
> > > >
> > > > ## A.5. Is FLUX capable of handling such non-stationary conditions where clients may join or leave, or their data distributions may shift over time?
> > > > We sincerely thank the reviewer for raising this important point. In fact, FLUX does not hold assumptions that all clients must participate throughout the whole training and testing, and can be adapted to real-world systems where clients may join and leave. **FLUX is capable of handling non-stationary conditions and dynamic scenarios** because the inherent design of FLUX is modular and can be seamlessly extended to handle non-stationary settings by periodically updating descriptors and reapplying the clustering process.
> > > >
> > > > In our extension, **we introduce a dynamic FLUX variant that performs re-clustering whenever a distributional drift is detected or when clients join or leave the federation**. To avoid retraining models from scratch after each re-clustering, each newly formed cluster is initialized with the model from the previously existing cluster whose centroid is closest in descriptor space. This association is computed by matching the current cluster centroids to the most similar centroids from the prior clustering phase—i.e., before the drift occurred. This warm-starting approach enables efficient adaptation to changing distributions while preserving previously learned knowledge. To validate this, we conducted new experiments on MNIST where two distribution drifts occur during training and one during testing, while all other settings match those in Table 1 of the main paper.
> > > > |Method|Accuracy (%)|
> > > > |-|-|
> > > > |FedAvg|71.9±6.0|
> > > > |IFCA|35.2±6.2|
> > > > |FedRC|75.2±4.3|
> > > > |FedEM|75.0±4.2|
> > > > |FedSEM|66.0±6.0|
> > > > |FedDrift|54.3±7.7|
> > > > |CFL|75.5±4.0|
> > > > |pFedMe|55.2±6.9|
> > > > |APFL|72.8±5.2|
> > > > |ATP|74.6±8.0|
> > > > |FLUX-soft|86.5±1.3|
> > > > |FLUX|**89.5±4.9**|
> > > >
> > > > As expected, FLUX achieves a significantly larger improvement over all baselines in this setting, since none of the existing methods are explicitly designed to handle such challenging non-stationary scenarios (except for FLUX-soft). These results clearly demonstrate that FLUX remains highly effective under these conditions, outperforming all baselines. In particular, FLUX improves by over 14 percentage points compared to the best performing baseline, showing strong resilience to distributional drift and further confirming its robustness in realistic, dynamic FL settings.

---

> > > > > ### Author Response · Authors · 2025-08-06
> > > > > **Follow-up Clarifications (Part 3)**
> > > > >
> > > > > ## A.4. Could the authors elaborate on how such datasets (e.g., OfficeHome, CheXpert) were adapted to simulate realistic federated settings, and whether there are best practices or frameworks for constructing FL-compatible OOD benchmarks?
> > > > > We thank the reviewer for raising this important point about our dataset choices. **To simulate realistic federated settings based on standard datasets, we carefully reviewed the data characteristics and partitioned them based on real-world scenarios to reflect natural heterogeneity** across clients:
> > > > > - CheXpert was adapted to simulate a cross-silo medical FL scenario, where different hospitals or medical centers operate with distinct characteristics. As detailed in Appendix F.1, we partitioned the dataset by **X-ray type (frontal vs. lateral), patient gender (male vs. female), and age group (younger vs. older)**. These partitions reflect realistic scenarios where: (1) ViewPosition differences occur when hospitals use different imaging protocols or specialized equipment (e.g., emergency departments primarily using portable lateral X-rays vs. radiology departments with advanced frontal imaging), (2) Age-based partitions simulate specialized medical centers (pediatric hospitals vs. geriatric care facilities, or rural clinics serving predominantly elderly populations), and (3) Gender-based differences reflect population demographics in different geographic regions (women's health clinics vs. occupational health centers serving male-dominated industries). These divisions naturally follow real-world medical data distributions, where such demographic and procedural variations occur organically across different healthcare institutions [1],[2],[3],[4].
> > > > > - OfficeHome was included following the helpful suggestion of Reviewer c8ww. Although originally proposed for domain generalization and adaptation tasks, it introduces **four distinct domains (Art, Clipart, Product, Real World)** that make it well-suited for FL. We assigned each domain to different clients to simulate feature heterogeneity—a common form of OOD shift in FL, also consistent with other related studies [5],[6].
> > > > >
> > > > > Regarding best practices for FL-compatible OOD benchmarks: While the FL community has not yet established universal frameworks, common approaches include partitioning based on natural metadata attributes (as we did with CheXpert's clinical metadata) and domain-based splits (as with OfficeHome's domains). **Our partitioning strategy follows established practices in recent FL literature, where realistic heterogeneity is simulated by leveraging inherent data characteristics rather than artificial random splits**. We believe developing standardized FL benchmarking frameworks represents an important direction for the community, and our comprehensive evaluation across multiple dataset types contributes to this ongoing effort.
> > > > >
> > > > >
> > > > > ### References
> > > > > [1] Arunava Chakravarty et al. “Federated Learning for Site Aware Chest Radiograph Screening.” IEEE 18th International Symposium on Biomedical Imaging (2021)
> > > > >
> > > > > [2] Mahshad Lotfinia  et al. “Boosting multi-demographic federated learning for chest radiograph analysis using general-purpose self-supervised representations.” European Journal of Radiology Artificial Intelligence (2025)
> > > > >
> > > > > [3] Oakden-Rayner, Luke. "Exploring large-scale public medical image datasets." Academic radiology 27.1 (2020)
> > > > >
> > > > > [4] Garcea, Fabio, et al. "Data augmentation for medical imaging: A systematic literature review." Computers in biology and medicine 152 (2023)
> > > > >
> > > > > [5] J. Chen, et al., "Federated Domain Generalization for Image Recognition via Cross-Client Style Transfer," 2023 IEEE/CVF WACV
> > > > >
> > > > > [6] W. Qian, et al. "Cross-domain structure preserving projection for heterogeneous domain adaptation." Pattern Recognition 123 (2022)

---

> ### Comment · Reviewer_L2wb · 2025-08-07
> **Thanks for your response**
>
> Thanks for the authors' effort on this. Appreciated. Some of my concerns have been answered. I'll revise my score.

---

> > ### Author Response · Authors · 2025-08-07
> >
> > We thank the reviewer for the thoughtful feedback throughout the review process, which has helped strengthen our work.

---

### Comment · Area_Chair_6dWV · 2025-08-01
**Discussion**

Dear Reviewers,

Thank you for providing initial reviews for this paper. The authors have now provided detailed rebuttals.

Please go through each rebuttals carefully and share with me your final thought on this paper, including clear justification regarding your decision to maintain, bump up or bump down the rating.

I also encourage you to respond to the authors your post-rebuttal thoughts to give them a chance to provide further clarification if needed.

Thank you very much for your help!

Best regards,

AC

---

### Note · Authors · 2025-08-14

We extend our sincere gratitude to the AC and all reviewers for their invaluable feedback and constructive engagement throughout this review process. We are encouraged by their recognition of FLUX's key strengths: the novel label-free model assignment mechanism addressing key CFL limitations (L2wb), comprehensive evaluation across datasets and heterogeneity levels (L2wb, Aotc), improved accuracy with FedAvg-comparable overhead (c8ww), impressive component integration with well-justified tradeoffs (MC86), elimination of prior knowledge requirements (Aotc), effective handling of all four shift types (Aotc) and well-structured presentation with strong empirical performance (LVoP). To this end, we summarize the key improvements made during rebuttal that helped strengthen our core contributions
- **Strengthened Theoretical Foundation:** We provided additional mathematical justification demonstrating how descriptor distances approximate the 2-Wasserstein distance between client distributions, including derivations of Lipschitz-equivalence under mild conditions, extensions to non-Gaussian distributions, and empirical validation confirming bounded approximation errors
- **Expanded Experimental Validation:** Following reviewer suggestions, we conducted experiments including: a new dataset (Office-Home), large-scale validation with 1200 clients, ablation studies on clustering algorithms, a head-to-head comparison with a recent method (HACCS) and validation in extreme low-data regimes—further confirming FLUX’s superior performance and applicability
- **Framework Flexibility Demonstration:** We demonstrated FLUX's flexibility by extending it to hybrid clustering (FLUX-soft), dynamic scenarios with client joining/leaving, and partial participation, showcasing adaptability to diverse real-world deployments
- **Comprehensive Response to All Concerns:** We addressed all major reviewer points through theoretical analysis, empirical validation and methodological clarification, with multiple reviewers acknowledging the improvements and revising scores positively

The review process helped us further improve the clarity, empirical validation, and theoretical robustness of our work, while the core contributions and evaluation remain consistent with the original submission. We believe these enhancements highlight FLUX's significance as a practical solution for arbitrary distribution shifts in FL, capable of scaling from controlled settings to realistic large-scale deployments"

---

### Decision · Program_Chairs · 2025-09-17

**Decision:**

Accept (poster)

**Comment:**

This paper proposes a clustering-based FL framework FLUX which is designed to address data heterogeneity (distribution shift) across clients during both training and testing phases. This is achieved via leveraging a privacy-preserving descriptor extraction mechanism on clients to enable efficient unsupervised clustering and representation compression with low communication overhead. This design ensures robustness and scalability of the global model. It also enables adaptive model selection for unseen and unlabeled clients clients during the testing phase. Experiments on four benchmark datasets and one real-world dataset demonstrate that FLUX outperforms 10 state-of-the-art baseline methods under diverse distribution shift while maintaining computational and communication overhead comparable to FedAvg.

Following an extensive discussion with the authors, the reviewing panel recognizing the following key strengths of the paper: (1) novel label-free model assignment mechanism addressing key CFL limitations; (2) comprehensive evaluation across datasets and heterogeneity levels; (3) improved accuracy with FedAvg-comparable overhead; (4) well-structured presentation with strong empirical performance.

There were also numerous questions raised by the reviewers regarding the motivation, theoretical foundation, additional experiment validation, and robustness to hybrid clustering (FLUX-soft), dynamic scenarios with client joining/leaving, and partial participation of clients. The authors have addressed all these questions sufficiently with an impressive and effective rebuttal. This results in an unanimous consensus on accepting this paper. This is a good paper that should be accepted.